# Comparison of CMIP6 Historical Climate Simulations and Future Projected Warming to an Empirical Model of Global Climate

Laura A. McBride[1], Austin P. Hope[2], Timothy P. Canty[2], Brian F. Bennett[2], Walter R. Tribett[2], Ross J. Salawitch[1,2,3]

[1]Department of Chemistry and Biochemistry, University of Maryland College Park, College Park, 20740, USA
[2]Department of Atmospheric and Oceanic Science, University of Maryland College Park, College Park, 20740, USA
[3]Earth System Science Interdisciplinary Center, University of Maryland College Park, College Park, 20740, USA

*Correspondence to*: Laura McBride (mcbridel@umd.edu)

**Abstract.**

The sixth phase of the Coupled Model Intercomparison Project (CMIP6) is the latest modeling effort for general circulation models to simulate and project various aspects of climate change. Many of the general circulation models (GCMs) participating in CMIP6 provide archived output that can be used to calculate effective climate sensitivity (ECS) and forecast future temperature change based on emissions scenarios from several Shared Socioeconomic Pathways (SSPs). Here we use our multiple linear regression energy

balance model, the Empirical Model of Global Climate (EM-GC), to simulate and project changes in global mean surface temperature (GMST), calculate ECS, and compare to results from the CMIP6 multi-model ensemble. An important aspect of our study is comprehensive analysis of uncertainties due to radiative forcing of climate from tropospheric aerosols (AER RF) in the EM-GC framework. We quantify the attributable anthropogenic warming rate (AAWR) from the climate record using the EM-GC and use

AAWR as a metric to determine how well CMIP6 GCMs replicate human-driven global warming over the last forty years. The CMIP6 multi-model ensemble indicates a median value of AAWR over 1975-2014 of 0.221°C decade$^{-1}$ (range of 0.151 to 0.299°C decade$^{-1}$; all ranges given here are for 5[th] and 95[th] confidence intervals), which is notably faster warming than our median estimate for AAWR of 0.157°C decade$^{-1}$ (range of 0.120 to 0.195°C decade$^{-1}$) inferred from analysis of the Hadley Center Climatic

Research Unit Version 5 data record for GMST. Estimates of ECS found using the EM-GC assuming climate feedback does not vary over time (best estimate 2.33°C; range of 1.40 to 3.57°C) are generally consistent with the range of ECS of 1.5 to 4.5°C given by IPCC's Fifth Assessment Report. The CMIP6

multi-model ensemble exhibits considerably larger values of ECS (median 3.74°C; range of 2.19 to 5.65°C). Our best estimate of ECS increases to 3.08°C (range of 2.23 to 5.53°C) if we allow climate feedback to vary over time. The dominant factor in the uncertainty for our empirical determinations of AAWR and ECS is imprecise knowledge of AER RF for the contemporary atmosphere, though the uncertainty due to time dependent climate feedback is also important for estimates of ECS. We calculate the likelihood of achieving the Paris Agreement target (1.5°C) and upper limit (2.0°C) of global warming relative to pre-industrial for seven of the SSPs using both the EM-GC and the CMIP6 multi-model ensemble. In our model framework, SSP1-2.6 has a 53% probability of limiting warming at or below the Paris target by the end of century and SSP4-3.4 has a 64% probability of achieving the Paris upper limit. These estimates are based on the assumptions that climate feedback has been and will remain constant over time since the prior temperature record can be fit so well assuming constant climate feedback. In addition, we quantify the sensitivity of future warming to the curbing of the current rapid growth of atmospheric methane and show major near-term limits on the future growth of methane are especially important for achievement of the 1.5°C goal of future warming. We also quantify warming scenarios assuming climate feedback will rise over time, a feature common among many CMIP6 GCMs; under this assumption, it becomes more difficult to achieve any specific warming target. Finally, we assess warming projections in terms of future anthropogenic emissions of atmospheric carbon. In our model framework, humans can emit only another 150 ± 79 Gt C after 2019 to have a 66% likelihood of limiting warming to 1.5°C, and another 400 ± 104 Gt C to have the same probability of limiting warming to 2.0°C. Given the estimated emission of 11.7 Gt C per year for 2019 due to combustion of fossil fuels and deforestation, our EM-GC simulations suggest the 1.5°C warming target of the Paris Agreement will not be achieved unless carbon and methane emissions are severely curtailed in the next 10 years.

## 1 Introduction

The goals of the Paris Agreement, negotiated in December of 2015, are to keep global warming below 2.0°C relative to the start of the Industrial Era and pursue efforts to limit global warming to 1.5°C. General circulation models (GCMs) project future temperature change using various evolutions of greenhouse

gases and determine the likelihood of achieving the goals of the agreement. Many GCMs are participating
in the sixth phase of the Coupled Model Intercomparison Project (CMIP6) to quantify how the models
represent different aspects of climate change (Eyring et al., 2016). Accurate projections of future
temperature are critical for achieving the goals of the Paris Agreement. Chapter 11 of IPCC's Fifth
Assessment Report shows that some of the previous generations of these models participating in phase 5
of the Coupled Model Intercomparison Project (CMIP5) (Taylor et al., 2012) tended to overestimate the
increase in global mean surface temperature (GMST) for the 21$^{st}$ century (Kirtman et al., 2013). In this
analysis we use a multiple linear regression energy balance model to quantify the change in GMST from
1850-2019, project future changes in GMST, compare to the CMIP6 multi-model ensemble, and
determine the likelihood of achieving the goals of the Paris Agreement.

Several prior studies have used a multiple linear regression approach to model the GMST anomaly
in order to quantify the impact of anthropogenic and natural factors on climate (Foster and Rahmstorf,
2011; Lean and Rind, 2008, 2009; Zhou and Tung, 2013). Typically, total solar irradiance, volcanoes,
and El Niño southern oscillation (ENSO) are the natural components represented in the multiple linear
regression. Greenhouse gases and aerosols are the anthropogenic factors. We use multiple linear
regression, in connection with a dynamic ocean module that accounts for the export of heat from the
atmosphere to the ocean, to represent the natural and anthropogenic components of the climate system.
In addition to the typical natural factors listed above, we include the Atlantic meridional overturning
circulation (AMOC), Pacific decadal oscillation (PDO), and Indian Ocean dipole (IOD) to provide a
robust representation of the natural climate system (Canty et al., 2013; Hope et al., 2017). Our
anthropogenic components also include the effect of land-use change (i.e., deforestation) on Earth's
albedo and the export of heat from the atmosphere to the ocean as the atmosphere warms.

Our analysis builds on the work of Canty et al. (2013) and Hope et al. (2017) and includes several
key updates. One is the extension back in time of our analysis to 1850. The Hadley Center Climatic
Research Unit (Morice et al., 2012, 2021), Berkley Earth Group (Rohde and Hausfather, 2020), and
Cowtan and Way (2014) provide GMST records starting in 1850, which now allows for simulations of
GMST that cover 170 years. The second update is the use of the Shared Socioeconomic Pathways (SSPs)
(O'Neill et al., 2017) as our climate scenarios for greenhouse gas and aerosol abundances. The third is

the adoption of an upper ocean to our model, formulated in a manner that matches the equations of Bony et al. (2006) and Schwartz (2012). A description of the model, the various input parameters used, and the updates listed above is given in Sect. 2. Section 3 shows results of CMIP6 and EM-GC comparisons to the historical climate record, estimations of effective climate sensitivity (ECS), as well as comparisons of our model and CMIP6 projections of future GMST change. Discussion of these results is provided in Sect. 4, along with concluding remarks.

## 2 Data and Methodology

### 2.1 Empirical model of global climate

In this analysis we use the empirical model of global climate (EM-GC), which provides a multiple linear regression, energy balance simulation of GMST. As detailed in the following paragraphs, the EM-GC solves for ocean heat uptake efficiency ($\kappa$) and six regression coefficients to minimize the cost function in Eq. (1).

$$Cost\ Function = \sum_{i=1}^{N_{MONTHS}} \frac{1}{\sigma_{OBSi}^2} (\Delta T_{OBSi} - \Delta T_{MDLi})^2 \quad (1)$$

In this equation, $\Delta T_{OBS}$ represents a time series of observed monthly GMST anomalies, $\Delta T_{MDL}$ is the modeled monthly change in GMST, $\sigma_{OBS}$ is the 1-sigma uncertainty associated with each temperature observation, $i$ is the index for each month, and $N_{MONTHS}$ is the total number of months used in the analysis. For this analysis, we trained the model from 1850-2019. The observed GMST anomalies are blended near surface air and sea surface temperature differences relative to the GMST anomaly over 1850-1900, which is assumed to represent pre-industrial conditions.

We consider several anthropogenic and natural factors as components of $\Delta T_{MDL}$. The radiative forcing (RF) due to greenhouse gases (GHGs), anthropogenic aerosols (AER), land-use change (LUC), and the export of heat from the atmosphere to the world's oceans are the anthropogenic components of $\Delta T_{MDL}$. The influence on GMST from total solar irradiance (TSI), El Niño southern oscillation (ENSO), the Atlantic meridional overturning circulation (AMOC), volcanic eruptions that reach the stratosphere and enhance stratospheric aerosol optical depth (SAOD), the Pacific decadal oscillation, (PDO) and the Indian Ocean dipole (IOD) are the natural components of $\Delta T_{MDL}$. Equation (2) shows how we calculate $\Delta T_{MDL}$, the modeled monthly change in GMST.

$$\Delta T_{MDLi} = \frac{1 + \gamma}{\lambda_P} \{GHG\ \Delta RF_i + AER\ \Delta RF_i + LUC\ \Delta RF_i - Q_{OCEAN\ i}\} + C_0 + C_1 \times SAOD_{i-6} +$$

$$C_2 \times TSI_{i-1} + C_3 \times ENSO_{i-2} + C_4 \times AMOC_i + C_5 \times PDO_i + C_6 \times IOD_i \tag{2}$$

In Eq. (2), $GHG\ \Delta RF_i$, $AER\ \Delta RF_i$, and $LUC\ \Delta RF_i$ represent monthly time series of the increase in the stratospheric adjusted values of the RF of climate (Solomon, 2007) since 1750. The parameter $\lambda_P$ represents the response of a blackbody to a perturbation in the absence of climate feedback (3.2 W m$^{-2}$, (Bony et al., 2006)). The SAOD, TSI, and ENSO are lagged by 6, 1, and 2 months respectively. The lag of 6 months for SAOD is representative of the time needed for the surface temperature to respond to a change in the aerosol loading due to a volcanic eruption (Douglass and Knox, 2005). This lag is the same as used by Lean and Rind (2008) and Foster and Rahmstorf (2011). The 1 month delay for TSI yields the maximum value of $C_2$, the solar irradiance regression coefficient. Lean and Rind (2008) and Foster and Rahmstorf (2011) also use a 1 month lag for TSI in their analyses. The 2 month delay for the response of GMST to ENSO is the lag needed to obtain the largest value of the correlation coefficient of the Multivariate ENSO Index version 2 (MEI.v2) (Wolter and Timlin, 1993; Zhang et al., 2019) versus the value of $T_{ENSO}$ calculated by Thompson et al. (2009). In Thompson et al. (2009), $T_{ENSO}$ is the simulated response of GMST to variability induced by ENSO, taking into consideration the effective heat capacity of the atmospheric-ocean mixed layer. Lean and Rind (2008) used a 4-month lag for ENSO.

The term $AMOC_i$ represents the influence of the change in the strength of the thermohaline circulation on GMST (Knight et al., 2005; Medhaug and Furevik, 2011; Stouffer et al., 2006; Zhang and Delworth, 2007). We use the Atlantic multidecadal variability, based on the area weighted monthly mean sea surface temperature (SST) in the Atlantic Ocean between the equator and 60°N (Schlesinger and Ramankutty, 1994), as a proxy for the strength of AMOC. A strong AMOC is characterized by northward flow of energy that would otherwise be radiated to space, which occurs in both the ocean and atmosphere and leads to particularly warm summers in Europe (Kavvada et al., 2013) as well as a number of other well documented influences in other climatic regions (Nigam et al., 2011). The total anthropogenic RF is used to detrend the AMOC signal. This method provides a more realistic approach to infer the changes in the strength of AMOC and its effect on GMST than other detrending options (Canty et al., 2013).

The dimensionless parameter $\gamma$ represents the sensitivity of the global climate to feedbacks that occur due to a change in the RF of GHGs, AER, and LUC. We relate $\gamma$ to the climate feedback parameter, $\lambda_\Sigma$, as shown in Eq. (3).

$$1 + \gamma = \frac{1}{1 - \left(\frac{\lambda_\Sigma}{\lambda_P}\right)}$$

$$\text{where } \lambda_\Sigma = \Sigma \text{ all climate feedbacks} \tag{3}$$

$$\text{i.e., } \lambda_\Sigma = \lambda_{\text{Water Vapor}} + \lambda_{\text{Lapse Rate}} + \lambda_{\text{Clouds}} + \lambda_{\text{Surface Albedo}}$$

The relation between $\lambda_\Sigma$ and $\gamma$ in Eq. (3) is commonly used in the climate modeling community (Sect. 8.6 of Solomon (2007)). Our value of $\lambda_\Sigma$ is related to the IPCC's Fifth Assessment Report ((Stocker et al., 2013), hereafter IPCC 2013) definition of $\lambda$ via $\lambda_\Sigma = \lambda_P - \lambda$.

   Our model explicitly accounts for the export of heat from the atmosphere to the world's oceans (i.e., ocean heat export or OHE). The quantity $Q_{\text{OCEAN}}$ in Eq. (2) represents OHE. In our previous analyses (Canty et al., 2013; Hope et al., 2017), $Q_{\text{OCEAN}}$ was subtracted outside of the climate feedback multiplicative term $(1+\gamma)/\lambda_P$. We have rewritten Eq. (2) to be comparable to the formulation for this term used by Bony et al. (2006) and Schwartz (2012). Due to this update, our model fits the historical climate

   record with higher values of climate feedback, especially for strong aerosol cooling (see Fig. S1 and supplement for more information). We calculate $Q_{\text{OCEAN}}$ by simulating the long-term trend in observed ocean heat content (OHC) as shown in Eq. (4) and Eq. (5).

$$Q_{OCEANi} = \kappa\left(\Delta T_{ATM,HUMANi} - \Delta T_{OCEAN,HUMANi}\right) \tag{4}$$

$$\kappa = \frac{OHE \times \Delta t}{\int_{t_{START}}^{t_{END}}\left(\left[\frac{1+\gamma}{\lambda_P}\{GHG\ RF_{i-72} + AER\ RF_{i-72} + LUC\ RF_{i-72}\}\right] - [f_0 \sum_0^{i-72} Q_{OCEAN}]\right)dt} \tag{5}$$

 The $\kappa$ term is the ocean heat uptake efficiency (W m$^{-2}$ °C$^{-1}$) and is based on the definition used in Raper et al. (2002), where $\kappa$ is the ratio between the atmosphere and ocean temperature difference that best fits observed OHC data (Sect. 2.2.8 describes the OHC data records used in our analysis). The value of $\kappa$ is determined based on the best fit (described below) between $Q_{\text{OCEAN}}$ and the observed OHC record. The term $\Delta T_{\text{OCEAN,HUMAN}}$ represents the temperature response of the well-mixed, top 100 m of the ocean due

   to the total anthropogenically driven rise in OHC. This formulation of $\Delta T_{\text{OCEAN,HUMAN}}$ allows the model ocean to warm in response to an atmospheric warming. We use a 6 year lag (72 months) for $Q_{\text{OCEAN}}$ to

account for the time needed for the energy leaving the atmosphere to heat the upper ocean and penetrate to depth, based on Schwartz (2012). Our analysis of modeled GMST is insensitive to whether this 6 year lag or the 10 year lag from Lean and Rind (2009) is used. The $t_{START}$ and $t_{END}$ limits on the integral in Eq. (5) are the start and end years associated with each OHC record. The start and end years vary between the 5 OHC records (see supplement for the different start and end years). The constant $f_0$ term in Eq. (5) is a combination of the heat capacity of ocean water, the fraction of total ocean volume in the surface layer, and the fraction of total $Q_{OCEAN}$ that warms the surface layer and is equal to $8.76 \times 10^{-5}$ °C m$^2$ W$^{-1}$. We represent the global ocean as being 1 km deep for 10% of the ocean area (representing the continental shelves) and 4 km deep for the remaining area, which approximates the average depth of the actual world's oceans to within 3%; 3.7 km compared to 3.682-3.814 km from Charette and Smith (2010). Based on our analysis of decadal ocean warming as a function of depth extracted from CMIP5 GCMs, we have determined that 13.7% of the rise in total OHC occurs in the well mixed, upper 100 m of the ocean, the term represented by $\Delta T_{OCEAN,HUMAN}$ in equation (4). The bottom panel of Fig. 1 compares our modeled OHC to the observed OHC record based on the average of five data sets; the value of κ resulting in the best simulation of observed OHC is shown.

We use the reduced chi-squared ($\chi^2$) metric to define the goodness of fit between the modeled and measured GMST anomaly for the atmosphere and also between simulated and observed OHC. Equation (6) and Eq. (7) show the calculations for $\chi^2$ for the atmosphere, and Eq. (8) shows the calculation for $\chi^2$ for the ocean. Minimization of the difference between the measured and modeled GMST anomaly results in the EM-GC being able to replicate the observed rise in temperature over the past 170 years quite well, as shown in Fig. 1. We have added two additional new features to the model to assure accurate representation of the rise in OHC as well as the rise in GMST since 1940. The first new feature, Eq. (7), was added to ensure all simulations matched the past 80 years of observations well. Without the $\chi^2_{RECENT}$ constraint, some solutions with a value of $\chi^2_{ATM}$ less than or equal to 2 have visually poor simulations of the rise in GMST over the past 4 to 5 decades. The second new feature, Eq. (8), was added because in the original model formulation some selections of the radiative forcing due to tropospheric aerosols (AER $\Delta RF_i$ in Eq. (2)) converged in a way that produced simulations of OHC that seemed physically improper

based on visual inspection of observed and modeled OHC. As a result of these two issues, all calculations

shown here are subject to three goodness-of-fit constraints, described by Eq. (6) to (8):

$$\chi^2_{ATM} = \frac{1}{N_{YEARS} - N_{FITTING\ PARAMETERS} - 1} * \sum_{j=1}^{N_{YEARS}} \frac{1}{\langle \sigma_{OBSj} \rangle^2} \left( \langle \Delta T_{OBSj} \rangle - \langle \Delta T_{MDLj} \rangle \right)^2 \qquad (6)$$

$$\chi^2_{RECENT} = \frac{1}{N_{YEARS,REC} - N_{FITTING\ PARAMETERS} - 1} * \sum_{j=1}^{N_{YEARS,REC}} \frac{1}{\langle \sigma_{OBSj} \rangle^2} \left( \langle \Delta T_{OBSj} \rangle - \langle \Delta T_{MDLj} \rangle \right)^2 \qquad (7)$$

$$\chi^2_{OCEAN} = \frac{1}{N_{YEARS} - N_{FITTING\ PARAMETERS} - 1} * \sum_{j=1}^{N_{YEARS,OHC}} \frac{1}{\langle \sigma_{OBSj} \rangle^2} \left( \langle OHC_{OBSj} \rangle - \langle OHC_{MDLj} \rangle \right)^2 \qquad (8)$$

Here, $\langle \Delta T_{OBS} \rangle$, $\langle \Delta T_{MDL} \rangle$, and $\langle \sigma_{OBS} \rangle$ in Eq. (6) and Eq. (7) represent the annually averaged observed,

modeled, and uncertainty in the GMST anomaly, respectively. The variable $N_{FITTING\ PARAMETERS}$ is equal

to 9 for typical simulations, the sum of 7 (the number of regression coefficients) plus 2 (model output

parameters $\gamma$ and $\kappa$). In Eq. (8), $\langle OHC_{OBS} \rangle$ and $\langle OHC_{MDL} \rangle$ represent the annual averaged observed and

modeled OHC. The $\sigma_{OBS}$ term in Eq. (8) is the uncertainty in the OHC record (see Sect. 2.2.8 for more

information). The equation for all three formulations of $\chi^2$ is based on annual averages, rather than

monthly time series. We calculate $\chi^2$ with annual values because the autocorrelation functions of $\Delta T_{OBS}$

and $\Delta T_{MDL}$ display similar shapes using annual averages, and do not match utilizing monthly averages

(see supplement of Canty et al. (2013) for further explanation). The Hadley Center Climate Research Unit

(HadCRUT) version 4 uncertainties for GMST are used for the $\sigma_{OBS}$ in Eq. (6) to (8) for all of the GMST

records analyzed here (see Sect. 2.2.1 and the supplement for more information). For Eq. (6) to (8), we

define an acceptable fit to the climate record as $\chi^2 \leq 2$. The number of years ($N_{YEARS}$) varies across the

three equations. Equation (6) uses the total number of years in the GMST record, which for HadCRUT5

is 170 years. The number of years in Eq. (8), $N_{YEARS,OHC}$, depends on the OHC data set used, as each data

set spans a different range. The average of five OHC data sets, which we use as our primary OHC series,

extends from 1955-2017, a total of 63 years. The value of $\chi^2_{OCEAN}$ found using Eq. (8) is displayed on the

bottom panel of Fig. 1. All model simulations shown throughout this paper have $\chi^2_{OCEAN} \leq 2$, representing

a good fit to the observed rise in OHC over the time of the data record.

The calculation of $\chi^2_{RECENT}$ shown in Eq. (7) is used to constrain the model to match the observed

changes in GMST over the time frame 1940-2019, a total of 80 years ($N_{YEARS,REC}$ equals 80). This time

frame was chosen to include a full cycle of AMOC, as the strength of the thermohaline circulation tends

to vary on a period of 60-80 years (Chen and Tung, 2018; Kushnir, 1994; Schlesinger and Ramankutty, 1994). As noted above, the $\chi^2_{RECENT}$ constraint was added to our model framework because without this constraint the model is able to provide numerically good but poor visual fits to the GMST anomaly under certain conditions (i.e., the red line in the top panel of Fig. 1 starts to strongly deviate from the black line beginning in about 2000 under certain conditions). All model simulations shown below have $\chi^2_{RECENT} \leq$

2 representing a good fit to the observed rise in GMST over the past 80 years, which results in modeled GMST that replicates observed GMST for the entire time series.

Figure 1 shows the observed (HadCRUT5) and modeled GMST anomaly from 1850-2019, and the various anthropogenic and natural components that constitute modeled GMST. Figure 1a shows the value of climate feedback, 1.62 W m$^{-2}$ °C$^{-1}$, that is needed to achieve a best fit to the climate record for

this simulation, resulting in values of $\chi^2_{ATM} = 0.80$ and $\chi^2_{OCEAN} = 0.31$. Figure 1b is the total contribution of human activity to variations in GMST, which includes GHGs, AER, LUC, and the export of heat from the atmosphere to the ocean. For the simulation shown, the aerosol radiative forcing is −0.9 W m$^{-2}$, the best estimate given by IPCC 2013 (Myhre et al., 2013). This panel also notes the best estimate of the time rate of change of GMST attributed to humans from 1975-2014, or the attributable anthropogenic warming

rate (AAWR (see Sect. 2.3)). Figure 1c illustrates the contribution to the GMST anomaly from TSI and SAOD over the 170-year period. The influences of ENSO and AMOC are indicated in Figs. 1d and 1e, respectively. Furthermore, the contribution of AMOC to the rise in GMST over 1975-2014 (the same time period used to define AAWR) is also specified on Fig. 1e (dotted black line). Figure 1f indicates the small effect of IOD and PDO on GMST in our model framework. The last panel, Fig. 1g, shows the time series

of observed OHC based on the average of five data sets for the upper 700 m of the ocean (black points and blue error bars; see Sect. 2.2.8) and the modeled value of OHC (red line). For this simulation, a value of κ equal to 1.17 W m$^{-2}$ °C$^{-1}$ fits the OHC data best. This value of κ falls within the range of empirical estimates for this parameter given by Raper et al. (2002). The sum of the contributions of human activity, TSI, SAOD, ENSO, AMOC, PDO and IOD to the GMST anomaly shown in Fig. 1b to 1f plus the value

of $C_0$ equals the modeled GMST anomaly, shown by the red line in Fig. 1a.

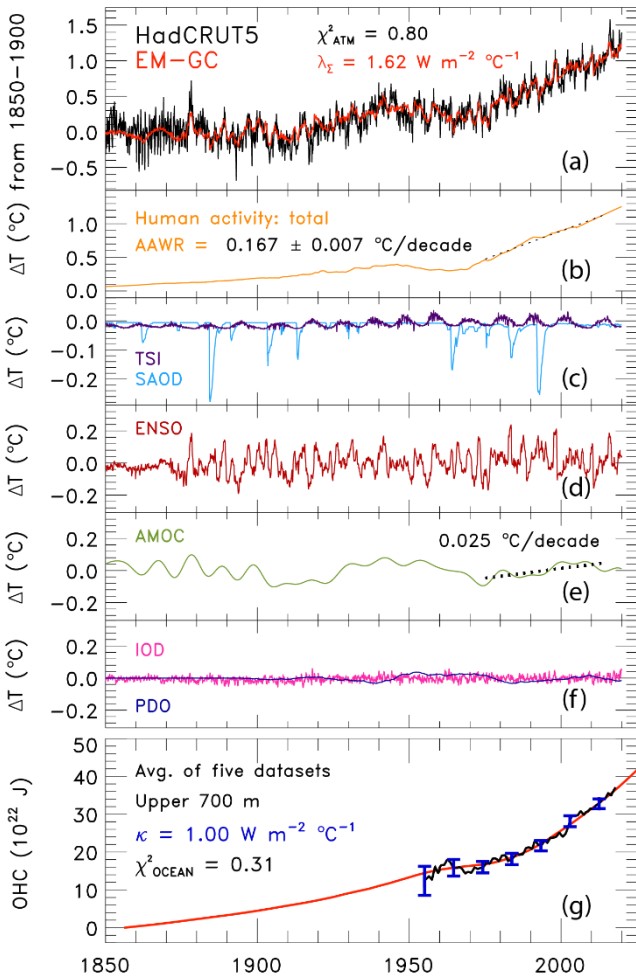

**Figure 1.** Measured and modeled GMST anomaly (ΔT) relative to a pre-industrial (1850-1900) baseline. (a) Observed (black) HadCRUT5 and modeled (red) ΔT from 1850-2019. This panel also displays the values of $\lambda_\Sigma$ and $\chi^2_{ATM}$ (see text) for this best-fit simulation. (b) Contributions from total human activity. This panel also denotes the best estimate value of the attributable anthropogenic warming rate from 1975-2014 (black dashed) as well as the 2σ uncertainty in the slope for a model run that uses the best estimate of AER $RF_{2011}$ of −0.9 W m⁻². (c) TSI (purple) and SAOD (light blue). (d) Influences from ENSO on ΔT. (e) Contributions from AMOC to ΔT and to observed warming from 1975-2014. (f) Influences from PDO (blue) and IOD (pink) on ΔT. (g) Measured (black) and modeled (red) ocean heat content (OHC) as a function of time for the average of five data sets (see text), the value of $\chi^2_{OCEAN}$ for this run, as well as the ocean heat uptake efficiency, κ, needed to provide the best-fit to the OHC record. The error bars (blue) denote the uncertainty in OHC used in this analysis (see Sect. 2.2.8).

Altering the training period of our model has a slight effect on our results (see Fig. S2, S3, and the supplement for information on various training periods). We project relatively similar results for end of century warming for training periods that start in 1850 and end in either 2009 or 1999, compared to results shown throughout the paper for a training period of 1850 to 2019, indicating the stability of our

approach. As detailed in the supplement, we do find some differences from the results shown in the paper upon use of a training period of 1850 to 1989 due to the reduction in the number of years considered from the available OHC records.

## 2.2 Model Inputs

### 2.2.1 Temperature data

We use seven global mean surface temperature anomaly records. These records include the Hadley Centre Climatic Research Unit version 4 (HadCRUT4, (Morice et al., 2012)) and version 5 (HadCRUT5 (Morice et al., 2021)) from 1850-2019, National Centers for Environmental Information NOAAGlobalTemp v5 (NOAAGT, (Smith et al., 2008; Zhang et al., 2019)) from 1880-2019, NASA Goddard Institute of Space Studies Surface Temperature Analysis v4 (GISTEMP, (Hansen et al., 2010)) from 1880-2019, Berkeley Earth Group (BEG, (Rohde and Hausfather, 2020)) from 1850-2019, Cowtan and Way (2014) (CW14) from 1850-2019, and the Japanese Meteorological Agency (JMA (Ishihara, 2006)) from 1891-2019. We use the uncertainty time series from HadCRUT4 for all GMST records because the HadCRUT4 uncertainty provides a realistic description of the variation in GMST among the seven records (see the supplement, Figs. S4 and S5, and Table S1 for more information). Our analysis primarily uses the HadCRUT5 GMST data set, but in some sections, results are shown for the other data sets. All temperature anomalies are with respect to a pre-industrial baseline (1850-1900). To alter each data record so that the temperature anomaly is relative to the same pre-industrial baseline, we adjust all data sets relative to the HadCRUT5 baseline of 1961-1990. We then adjust each data set by the same amount to the HadCRUT5 pre-industrial baseline as described in the supplement.

### 2.2.2 Shared Socioeconomic Pathways

For this analysis, we use the estimates of the future abundances of greenhouse gases and aerosols provided by the SSPs. There are twenty-six scenarios, five baseline pathways and twenty-one mitigation scenarios. The baseline pathways follow specific narratives for factors such as population, education, economic growth, and technological developments of sources of renewable energy (Calvin et al., 2017; Fricko et al., 2017; Fujimori et al., 2017; Kriegler et al., 2017; van Vuuren et al., 2017) to represent several possible

futures spanning different challenges for adaptation and mitigation to climate change as illustrated in Fig. 1 of O'Neill et al. (2014). The twenty-one mitigation scenarios follow one of the baseline pathways but
include specific climate policy to reach a designated radiative forcing at the end of the century.

As part of CMIP6, the ScenarioMIP experiment (O'Neill et al., 2016) includes eight SSPs (SSP1-1.9, SSP1-2.6, SSP4-3.4, SSP2-4.5, SSP4-6.0, SSP3-7.0, SSP5-8.5, and SSP5-3.4-OS) that GCMs use to project future GMST. The first number is the reference pathway that the scenario follows (i.e., SSP1 follows the first SSP narrative) and the numbers after the dash are the target radiative forcing at the end
of the century (i.e., SSP1-2.6 reaches around 2.6 W m$^{-2}$ in 2100). The ScenarioMIP experiment designates Tier 1 and Tier 2 scenarios. The Tier 1 scenarios are SSP1-2.6, SSP2-4.5, SSP3-7.0, and SSP5-8.5 , and the Tier 2 scenarios are SSP1-1.9, SSP4-3.4, SSP4-6.0, and SSP5-3.4-OS (an overshoot pathway that follows SSP5-8.5 until around 2040, where carbon dioxide emissions drastically decrease and become negative in 2065). Our analysis includes seven of the eight ScenarioMIP SSPs: all but the overshoot
pathway. We highlight four in the main paper: two Tier 1 (SSP1-2.6 and SSP2-4.5) and two Tier 2 (SSP1-1.9 and SSP4-3.4) scenarios. Analysis of the other three SSPs is included in the supplement. Figure 2 shows the atmospheric abundance of the three major anthropogenic GHGs (carbon dioxide, methane, and nitrous oxide) for each of the seven SSPs we consider as well as observations of the global mean atmospheric abundance for these gases to the end of 2019 (Dlugokencky, 2020; Dlugokencky and Tans,
2020).

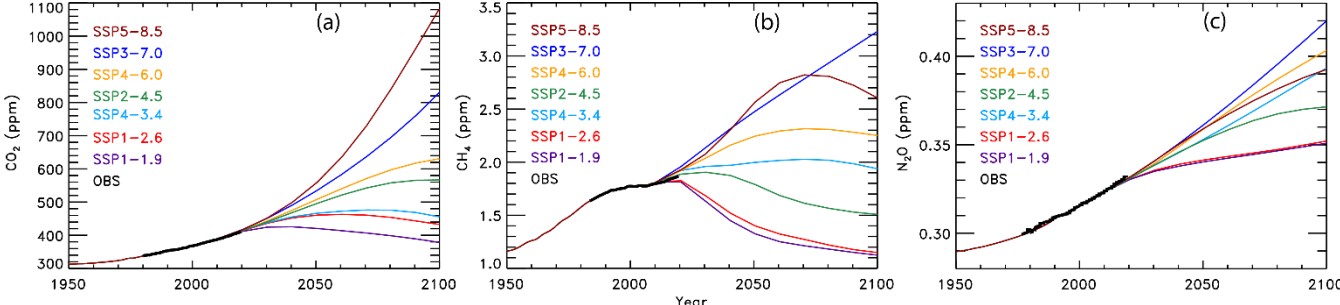

**Figure 2.** Observed and projected greenhouse gas mixing ratios. (a) Carbon dioxide abundances from observations (black) and seven of the ScenarioMIP SSPs (colors, as indicated). (b) Methane abundances from observations and ScenarioMIP SSPs. (c) Nitrous oxide abundances from observations and ScenarioMIP SSPs.

### 2.2.3 Greenhouse gases

The historical values of GHG mixing ratios were provided by Meinshausen et al. (2017) from 1850-2014. We used the equations from Myhre (1998) to calculate the change in RF due to carbon dioxide ($CO_2$), methane ($CH_4$), nitrous oxide ($N_2O$), ozone depleting substances (ODS), hydrofluorocarbons, perfluorocarbons, and sulfur hexafluoride relative to RF in year 1850. We also used the updated preindustrial values of $CH_4$ and $N_2O$ from IPCC 2013 and the radiative efficiencies from WMO (2018). The radiative forcing of $CH_4$ also includes the 15% enhancement from the increase in stratospheric water vapor due to rising atmospheric $CH_4$ (Myhre et al., 2007). Values of GHG mixing ratios, other than ODSs, from 2015-2100 are from the SSP Database (Calvin et al., 2017; Fricko et al., 2017; Fujimori et al., 2017; Kriegler et al., 2017; Rogelj et al., 2018; van Vuuren et al., 2017) and are provided on a decadal basis. These mixing ratios were interpolated onto a monthly time scale. We used the estimates of future ODS abundances provided in Table 6-4 of the 2018 Ozone Assessment Report (Carpenter et al., 2018), because the SSP database did not provide these estimates. We also include tropospheric ozone ($O_3^{TROP}$) as a GHG, because tropospheric ozone rivals $N_2O$ as the third most important anthropogenic GHG (Fig 8.15 of Myhre et al. (2013)). The RF due to $O_3^{TROP}$ from the RCPs provided by the Potsdam Institute for Climate Impact Research (Meinshausen et al., 2011) is used, because the SSP database does not provide estimates. Values of RF due to $O_3^{TROP}$ from RCP2.6, RCP4.5, RCP6.0, and RCP8.5 are substituted in for SSP1-2.6, SSP2-4.5, SSP4-6.0, and SSP5-8.5, respectively. We created new time series for the RF due to $O_3^{TROP}$ for SSP4-3.4 and SSP3-7.0 using linear combinations of RF time series from RCP2.6 and RCP8.5, with weights based on the end of century total RF value due to all GHGs of the respective time series. Finally, the RF time series for $O_3^{TROP}$ from RCP2.6 was also used for SSP1-1.9. Figure S6 shows the ozone RF time series used in this analysis and the supplement provides more information about the creation of the time series for the RF due to $O_3^{TROP}$.

### 2.2.4 Aerosol radiative forcing

The value of the change in total aerosol radiative forcing (direct and indirect) in 2011 relative to pre-industrial (AER $RF_{2011}$) is highly uncertain. Chapter 8 of the IPCC 2013 report gives a best estimate of AER $RF_{2011}$ as $-0.9$ W m$^{-2}$, a likely range between $-0.4$ and $-1.5$ W m$^{-2}$, and a 5[th] to 95[th] percent confidence interval between $-0.1$ and $-1.9$ W m$^{-2}$ (Myhre et al., 2013). This substantial range in AER

$RF_{2011}$ results in a large spread in future projections of global GMST. Figure 3 shows the effect of varying the value of AER $RF_{2011}$ on projections of GMST in our EM-GC framework, for the same SSP4-3.4 GHG scenario. The middle panel on Figs. 3a, 3b, and 3c shows the contribution to GMST of GHGs, LUC, AER, as well as net human activities. As the value of AER $RF_{2011}$ decreases and aerosols cool more strongly, the value of climate feedback (model parameter $\lambda_\Sigma$) rises, and the net contribution of human impact on GMST by the end of the century increases. Depending on which value of AER $RF_{2011}$ is used, the rise in GMST by year 2100 for the SSP4-3.4 pathway could range from 1.5°C (Fig. 3a) to 2.8°C (Fig. 3c) relative to pre-industrial. Strong aerosol cooling offsets a substantial fraction of GHG-induced warming, and a large value of climate feedback ($\lambda_\Sigma = 2.41$ W m$^{-2}$ °C$^{-1}$) is needed to fit the historical climate record (Fig. 3c). In this case, future warming is large, well above the goals of the Paris Agreement by the end of the century. Conversely, weak aerosol cooling offsets only a small fraction of GHG-induced warming, resulting in a small value of climate feedback ($\lambda_\Sigma = 1.08$ W m$^{-2}$ °C$^{-1}$) needed to fit the observed GMST record (Fig. 3a). The use of any of the values of AER $RF_{2011}$ in Fig. 3 can result in a very good fit to the climate record (i.e., $\chi^2_{ATM} \leq 2$, $\chi^2_{RECENT} \leq 2$, and $\chi^2_{OCEAN} \leq 2$).

We use the total aerosol RF time series provided by the SSP database for each SSP scenario. The database provides AER RF from 2005-2100, with values for all SSPs nearly identical until about 2010 (Riahi et al., 2017; Rogelj et al., 2018). In the EM-GC, we calculate temperature projections over the entire observational period, beginning in 1850. We create AER RF time series that begin in 1850 and span the range of uncertainty given by Chapter 8 of IPCC 2013. We use historical estimates of AER RF from 1850-2014 for the four RCPs provided by the Potsdam Institute for Climate Research (Meinshausen et al., 2011). The AER RF value in 2014 from the appropriate historical estimate (i.e., RCP 4.5 is used for SSP2-4.5) is scaled by a constant factor, such that the historical RCP value at the end of 2014 matches the SSP time series at the start of 2015. This scaling yields a continuous time series for the RF of climate due to tropospheric aerosols. This scaled time series has AER $RF_{2011}$ nearly equal to −1.0 W m$^{-2}$, which we take as the SSP-based best estimate of the change in total aerosol radiative forcing in 2011 relative to pre-industrial. Next, the single continuous time series is scaled, again by a constant multiplicative factor, to match the IPCC 2013 best estimate and range of uncertainty for AER $RF_{2011}$ (Myhre et al., 2013). This procedure results in five additional time series of AER RF. Six time series of AER RF are created for

each SSP, having values of AER $RF_{2011}$ equal to −0.1, −0.4, −0.9, −1.0, −1.5, and −1.9 W m$^{-2}$. Figure S7 shows these six AER RF time series for SSP1-2.6 and SSP4-3.4. In the EM-GC framework, we further scale these six time series to create a total of 400 AER RF time series to fully analyze the range of AER $RF_{2011}$ given by Myhre et al. (2013).

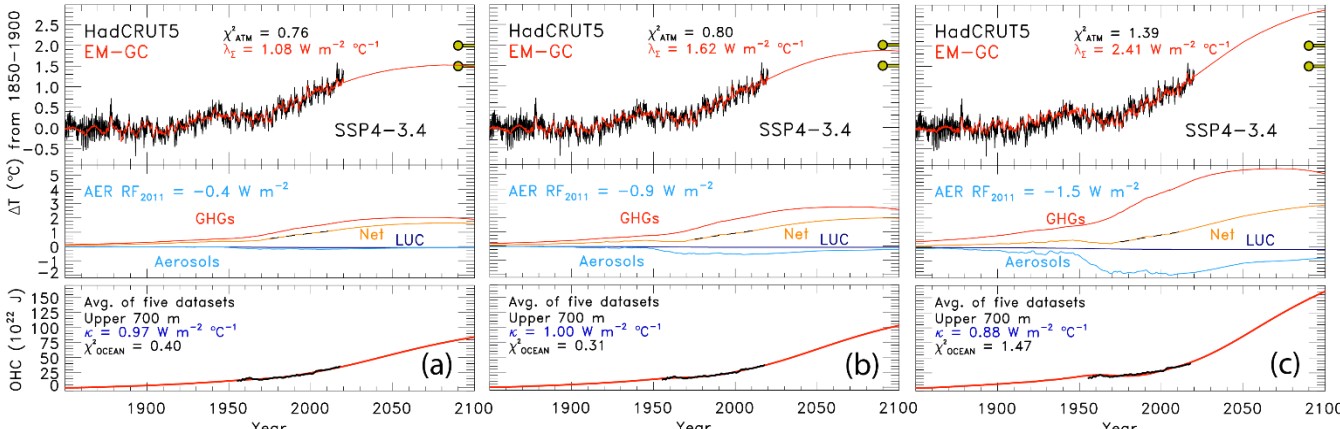

**Figure 3.** Measured (HadCRUT5) and EM-GC simulated GMST anomaly ($\Delta$T) relative to a pre-industrial (1850-1900) baseline, as well as projected $\Delta$T to end of century for SSP4-3.4. Top panel of each plot displays observed (black) and simulated (red) $\Delta$T, as well as the values of $\lambda_\Sigma$ and $\chi^2_{ATM}$ for each model run. The Paris Agreement target (1.5°C) and upper limit (2.0°C) are shown (gold circles). The second panel shows the contribution of GHGs, aerosols, and land-use change to $\Delta$T, as well as the net human component. The bottom panel compares observed (black) and modeled (red) values of OHC for simulations constrained by the average of five data sets (see text) and also provides the numerical values of $\kappa$ needed to obtain best-fits to the OHC record as well as best-fit values of $\chi^2_{OCEAN}$. The only difference between (a), (b), and (c) is the time series for RF due to tropospheric aerosols used to constrain the EM-GC; values of AER $RF_{2011}$ for each time series are (a) −0.4 W m$^{-2}$, (b) −0.9 W m$^{-2}$, (c) −1.5 W m$^{-2}$.

### 2.2.5 Total solar irradiance and stratospheric aerosol optical depth

We use the TSI time series provided for the CMIP6 models from 1850-2014 (Matthes et al., 2017) and append values from the Solar Radiation and Climate Experiment (SORCE) (Dudok de Wit et al., 2017) for 2015 to the end of 2019. The values of $TSI_i$ used in Eq. (2) are differences of monthly mean values minus the long-term average (i.e., TSI anomalies). Consistent with prior studies (e.g., Lean and Rind (2008) and Foster and Rahmstorf (2011)) variations in solar irradiance due to the 11-year solar cycle have a small but noticeable effect on the EM-GC simulation of the GMST anomaly (Fig. 1c). For projections of future warming, we set the term $TSI_i$ in Eq. (2) equal to zero from the start of 2020 until 2100.

The time series for SAOD is a combination of values computed from extinction coefficients for the CMIP6 GCMs (Arfeuille et al., 2014) from 1850-1978 and the Global Space-based Stratospheric Aerosol Climatology (GloSSAC v2.0) (Thomason et al., 2018) from 1979-2018. Extinction coefficients at 550 nm were integrated from the tropopause to 39.5 km and averaged over the globe using a cosine of latitude weighting. The CMIP6 and GloSSAC extinction coefficients span 80°S to 80°N. To extend the SAOD time series to the end of 2019, we use the level 3, gridded SAOD product from the Cloud-Aerosol Lidar and Infrared Pathfinder Satellite Observations (CALIPSO) (Vaughan et al., 2004). Time series of globally averaged SAOD from CALIPSO have a very similar shape to the GloSSAC time series over the period of overlap (2006-2018) with a slight offset because GloSSAC uses estimates of CALIPSO data for SAOD. To append the SAOD after 2018, we took the average difference between the two time series for the overlapping months and then adjusted the CALIPSO time series by this offset. This slight adjustment to the CALIPSO record has no bearing on our results, since the effect of volcanic activity on GMST has been small over the past 2 decades (Fig. 1c). We set the term $SAOD_i$ in Eq. (2) equal to the value in December 2019 from the start of 2020 until 2100.

### 2.2.6 El Niño southern oscillation, Pacific decadal oscillation, and Indian Ocean dipole

We use the MEI.v2 (Wolter and Timlin, 1993; Zhang et al., 2019) to characterize the influence of ENSO on GMST. In order to obtain a time series that spans the entire training period of our model, 1850-2019, we append three time series to create an MEI.v2 index over the full extent of our model training period. The MEI.v2 provides two month averages of empirical orthogonal functions of five different climatic variables from 1979 to present (Zhang et al., 2019). To have the ENSO index extend back to 1850, we compute differences in SST anomalies over the tropical Pacific basin as defined by the MEI.v2 from 1850-1870 using HadSST3 (Kennedy et al., 2011). Our internal computation of this surrogate for the MEI index is then appended to the MEI.ext of Wolter and Timlin (2011), which extends from 1871-1978, and the MEI.v2 index of (Zhang et al., 2019) (1979-2019). This full time series provides a representation of ENSO that covers from 1850 to present. Consistent with prior regression-based approaches (Foster and Rahmstorf, 2011; Lean and Rind, 2008), we find a significant portion of the monthly and at times annual

variation in GMST is well explained by ENSO (Fig. 1d). As for the other natural terms, we assume $ENSO_i$ in Eq. (2) is zero for 2020-2100.

The Pacific decadal oscillation is the leading principal component of North Pacific monthly SST variability poleward of 20°N (Barnett et al., 1999). The PDO index maintained by the University of Washington provides monthly values from 1900-2018. The PDO varies on a multidecadal time scale and affects climate in the North Pacific and North America, and has secondary effects in the tropics (Barnett et al., 1999). In our model framework, the expression of PDO on GMST is dependent on the model specification of the AER RF time series, as shown in Fig. S8. At low values of AER $RF_{2011}$, such as −0.1 W m$^{-2}$, the effect of PDO on GMST is negligible and the contribution from AMOC dominates. At high values of AER $RF_{2011}$ (−1.5 W m$^{-2}$), the effect of PDO on GMST is equal to the contribution from AMOC. At high values of AER $RF_{2011}$, we obtain results similar to findings from England et al. (2014) and Trenberth and Fasullo (2013) that shows the PDO exhibits an appreciable influence on GMST, especially for the 2000-2010 time period.

The Indian Ocean dipole is based on the difference in the anomalous sea surface temperatures (SST) between the western equatorial Indian Ocean (50°-70° E and 10° S-10° N) and the south eastern equatorial Indian Ocean (90° E-110° E & 10° S-0° N) as defined in Saji et al. (1999). We use 1° × 1° SSTs from the Centennial in situ Observation-Based Estimate (COBE) (Ishii et al., 2005) to create an IOD index from 1850-2019. As noted above and shown on Fig. 1f, the regression coefficients for PDO and IOD are quite small. We find little influence of either PDO or IOD in the HadCRUT5 time series of GMST, but these terms are retained for completeness. We assume $PDO_i$ and $IOD_i$ in Eq. (2) are zero after the start of 2019 and 2020, respectively.

### 2.2.7 Atlantic meridional overturning circulation

We use the Atlantic multidecadal variability (AMV) index as the area weighted, monthly mean SST from HadSST4 (Kennedy et al., 2019), between the equator and 60° N in the Atlantic Ocean (Schlesinger and Ramankutty, 1994) to characterize the influence of the AMOC on GMST. The AMV index is detrended using the RF anomaly due to anthropogenic activity over the historical time frame of the analysis, as discussed in Sect. 3.2.3 of Canty et al. (2013), because this detrending option removes the influence of

long-term global warming on the AMV index. The detrended AMV index serves as a proxy for variations in the strength of the AMOC (Knight et al., 2005; Medhaug and Furevik, 2011; Zhang and Delworth, 2007), which has particularly noticeable effects on climate in the Northern Hemisphere (Jackson et al., 2015; Kavvada et al., 2013; Nigam et al., 2011). For this analysis, the index has been Fourier filtered to remove frequencies above 9 $yr^{-1}$ to retain only the low frequency, high amplitude component of the thermohaline circulation (Canty et al., 2013). As noted above and shown in Fig. 1, a considerable portion of the long-term variability in GMST is attributed to variations in the strength of AMOC, including about 0.025°C $decade^{-1}$ over the 1975-2014 time period. There is considerable debate about the validity of the use of a proxy such as the AMV index as a surrogate for the climatic effects of AMOC that is centered mainly around how much of the variability of the index is either internal or externally forced (Haustein et al., 2019; Knight et al., 2005; Medhaug and Furevik, 2011; Stouffer et al., 2006). We stress, as explained in Sect. 2.3, none of our major scientific conclusions are altered if we neglect AMV as a regression variable.

### 2.2.8 Ocean heat content records

Ocean heat content data records from five recent and independent papers are used in this study. We utilize OHC data from Balmaseda et al. (2013), Carton et al. (2018), Cheng et al. (2017), Ishii et al. (2017), and Levitus et al. (2012), as well as the average of the records to model the export of heat (OHE) from the atmosphere to the ocean. Figure S9 shows these five OHC records as well as the multi-measurement average. While most of these data sets have a common origin, they differ in how extensive temporal and spatial gaps in the coverage of ocean temperatures have been handled, ranging from data assimilation (Carton et al., 2018) to an iterative radius of influence mapping method (Cheng et al., 2017). The five data sets are all set to zero in 1986, which is the midpoint of the multi-measurement time series, by applying an offset for visual comparison. Since OHE, in units of W $m^{-2}$, is based on the slope of each OHC data set, this offset has no impact on the computation of OHE from OHC that is central to our study. For the computation of OHE from OHC, we use a value of the surface area of the world's oceans equal to $3.3 \times 10^{14}$ $m^2$ (Domingues et al., 2008). The OHC records we analyze are for the upper 700 m of the ocean. To calculate the OHE for the whole ocean, we multiply the OHE by 1/0.7 to account for the fact

that the upper 700 m of the ocean holds 70% of the heat (Sect. 5.2.2.1 (Solomon, 2007)). When we

subtract the amount of heat going into the ocean in Eq. 2 ($Q_{OCEAN}$), we also must account for the difference

in surface area between the global atmosphere and the world's oceans. Since the $Q_{OCEAN}$ term is computed

for the surface area of the ocean, but the forcing is applied to the whole atmosphere, we multiply the

$Q_{OCEAN}$ term by the ratio of the surface area of the ocean to the surface area of the atmosphere, which is

0.67.

As noted above, the calculation of $\chi^2_{OCEAN}$ shown in Eq. (8) is used to constrain our model

representation of the rise in OHC. Only model runs that provide a good fit to the observed OHC record

are shown below. For these five OHC data sets, uncertainty estimates are not always provided.

Furthermore, some studies that do provide uncertainties give estimates that seem unreasonably small (see

Fig. S10 and the supplement). Because of the discrepancy in uncertainties between OHC records, we

create a new uncertainty time series using both the 1$\sigma$ standard deviation of the average of the five OHC

records and the uncertainties from the Cheng et al. (2017) (hereafter Cheng 2017) OHC record. We create

this new uncertainty from 1955-2019 by a monthly time step and use either the 1$\sigma$ standard deviation of

the average of the five OHC records or the uncertainties from the Cheng 2017 OHC record, whichever is

larger, for that month. We use the Cheng 2017 OHC uncertainties because these estimates are the largest

of the five data sets. Additionally, the standard deviation from the mean of the five OHC records is very

low in the 1980s, which is an artifact of our normalization treatment, not inherent to any of the records.

This combined uncertainty estimate is substituted in for each individual data set and the average, resulting

in our use of the same time varying uncertainty in OHC for all data sets. Figure S10 and the supplement

provide more detail on the creation of this time dependent uncertainty estimate for OHC.

The choice of OHC record has only a small effect on future projections of GMST using the EM-

GC. Figure 4 illustrates the effect of varying OHC record on future temperature. The bottom panels show

the observed and modeled OHC, the value of $\kappa$ needed to best fit the OHC data record, and the resulting

value of $\chi^2_{OCEAN}$. Of the two OHC records shown, Balmaseda et al. (2013) (Fig. 4a) yields the lowest

value of $\kappa$ and Ishii et al. (2017) (Fig. 4b) results in the highest estimate of $\kappa$. For the same value of AER

$RF_{2011}$ (i.e., −0.9 W m$^{-2}$) and GHG scenario (SSP4-3.4), we find a difference of 0.25°C in the modeled

rise in GMST in year 2100 for these two simulations (red lines on top panels). For most of the remaining

analysis, we use the multi-measurement average of the five OHC data records. In Sects. 3.1 and 3.2 we quantify the effect of OHC data record on both attributable anthropogenic warming rate and effective
climate sensitivity.

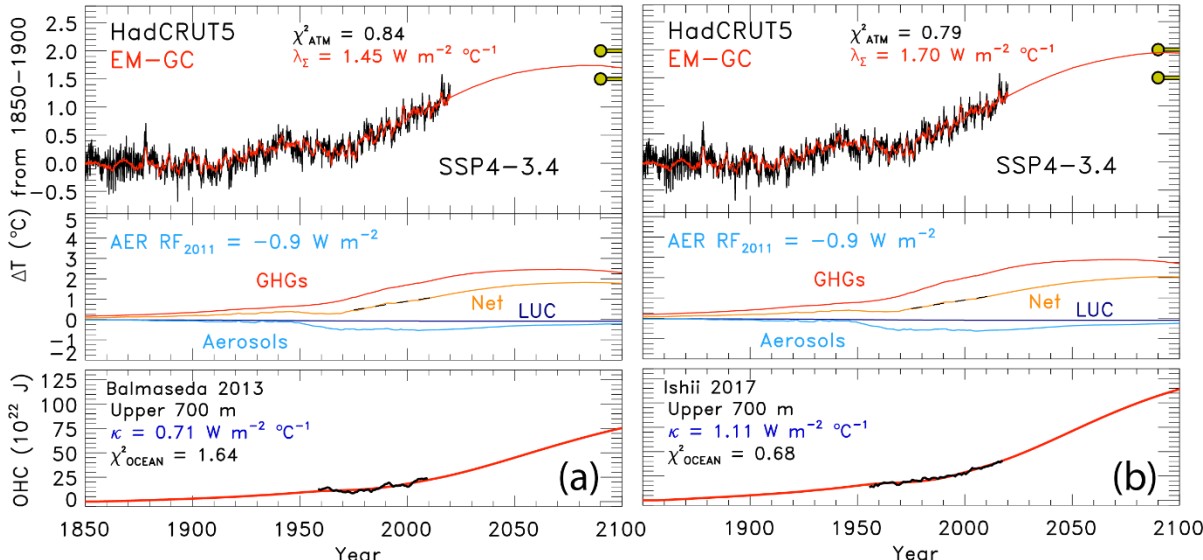

**Figure 4.** Measured (HadCRUT5) and EM-GC simulated GMST change (ΔT) from 1850-2019, as well as projected ΔT to year 2100 for SSP4-3.4. Top panel of each plot shows observed (black) and simulated (red) ΔT, the $\lambda_\Sigma$ and $\chi^2_{ATM}$ values, and the Paris Agreement target and upper limit. The second panel displays the contribution of GHGs, aerosols, and land-use change on ΔT. The bottom panel compares the observed (black) and modeled (red) OHC for two different OHC records and displays the value of κ needed to provide best-fits to the OHC record, as well as best-fit values of $\chi^2_{OCEAN}$. Both use an aerosol RF in 2011 of −0.9 W m$^{-2}$. (a) OHC record from Balmaseda et al. (2013). (b) OHC record from Ishii et al. (2017).

## 2.3 Attributable anthropogenic warming rate

The attributable anthropogenic warming rate, or AAWR, is the time rate of change of GMST due to
humans from 1975-2014. We use AAWR as a metric in the EM-GC framework to quantify the human influence on global warming over the past few decades, and most importantly to also assess how well the CMIP6 GCMs can replicate this quantity. This analysis is motivated by the study of Foster and Rahmstorf (2011), who examined the human influence on the time rate of change of GMST from 1979-2010 using a residual method. We extend the end year of our analysis to 2014 because this is the last year of the
CMIP6 Historical simulation. We pushed the start year back to 1975 so that our analysis covers a forty-

year period, over which the effect of human activity on GMST rose nearly linear with respect to time (Fig. 1b and Fig. S10c).

We calculate AAWR utilizing the EM-GC by computing a linear fit to the $\Delta T_{HUMAN,ATM}$ term:

$$\Delta T_{ATM,HUMANi} = \frac{1+\gamma}{\lambda_p} \{GHG\ \Delta RF_i + AER\ \Delta RF_i + LUC\ \Delta RF_i - Q_{OCEAN}\} \tag{9}$$

for a regression that spans 1850-2019. The $\Delta T_{HUMAN,ATM}$ term represents the net impact of the change in GMST due to RF of climate by anthropogenic GHGs, tropospheric aerosols, as well as the variation in surface reflectivity due to land-use change (deforestation), taking into account that for each model time step, a portion of the human-induced climate forcing is exported to the world's oceans. For each simulation, the slope of the linear least squares fit to the 480 monthly values of $\Delta T_{HUMAN,ATM}$ is used to

determine AAWR. For the time period 1975-2014, a value for AAWR of 0.144 ± 0.005 °C decade$^{-1}$ is found using a value of AER RF$_{2011}$ equal to −0.9 W m$^{-2}$, where the uncertainty corresponds to the 2σ standard error of a linear least squares fit. The computation of AAWR found by fitting monthly values of $\Delta T_{HUMAN,ATM}$ is insensitive to modest changes in start and end year for the AAWR calculation (see Table S1). The value of $\lambda_\Sigma$, and therefore AAWR, is also insensitive whether or not the AMOC, PDO, or IOD

terms are included in the regression framework (Canty et al., 2013; Hope et al., 2017). We are able to fit the climate record better (i.e., smaller values of $\chi^2$ in Eqs. (6), (7), and (8)) by including the AMOC term. However, computed values of AAWR are insensitive to whether AMOC is used in the regression because whatever contributions the variation in the strength of the thermohaline circulation may have had on GMST are not considered in Eq. (9) (see Fig. S11 for further explanation).

The determination of AAWR from historical CMIP6 near surface air temperature output involves conducting a regression of deseasonalized, globally averaged, monthly $\Delta T$ ($\Delta T^{DES,GLB}$) from each GCM (Hope et al., 2017), termed the REG method. The archived CMIP6 Historical runs are constrained by observed variations in SAOD and influenced by other factors such as internal model generated ENSOs. The $\Delta T^{DES,GLB}$ time series for all of the runs from each CMIP6 GCM are averaged together to obtain one

time series of $\Delta T^{DES,GLB}$ for each GCM. This average $\Delta T^{DES,GLB}$ time series is used to compute AAWR. The regression approach is used to compute the influence of SAOD on GMST from CMIP6 GCMs. The time needed for GMST to respond to a change in the aerosol loading in the stratosphere due to a volcanic

eruption in each GCM can exhibit a significant difference compared to the empirically determined response time of 6 months discussed in Sect. 2.1. A lag was determined for each GCM by calculating the value of the monthly delay between volcanic eruptions and the surface temperature response that resulted in the largest regression coefficient for SAOD. We regress the $\Delta T^{DES,GLB}$ against SAOD and the anthropogenic effect on temperature, which is approximated as a linear function from 1975-2014. The value of AAWR is the slope of the anthropogenic effect on temperature. Figure S12 illustrates the REG method used to determine AAWR from the CMIP6 GCMs. Table S3 depicts the slight effect on values of AAWR for the CMIP6 GCMs of changing the start or end year for the regression. At the time of analysis, there are 50 CMIP6 GCMs with the necessary archived output to calculate AAWR, with the values of AAWR found using REG shown in Table S3. Figure S13 and the supplement compare values of AAWR found using the REG method applied to EM-GC output with values of AAWR found using Eq. (9), as support for the validity of using the REG method to determine AAWR from CMIP6 output.

We also use a second method to extract the value of AAWR from the CMIP6 multi-model ensemble. This method, termed LIN, involves a linear regression of global, annual average values of GMST from the CMIP6 multi-model ensemble (Hope et al., 2017). For LIN, we exclude the years of obvious volcanic influence on the rise in GMST from the CMIP6 multi-model ensemble Historical simulations: i.e., data for 1982 and 1983 (following the eruption of El Chichón) and 1991 and 1992 (following the eruption of Mount Pinatubo) are excluded. Archived global, annual average values of GMST covering 1975-2014, excluding these four years, are fit using linear regression, with the AAWR set equal to the slope of the fit. Values of AAWR for 1975-2014 found using LIN are also shown in Table S4 for each GCM. Analysis of AAWR for these 50 GCMs of LIN versus REG (see Fig. S14) results in a correlation coefficient ($r^2$) of 0.995 and a mean ratio of $1.009 \pm 0.015$, with LIN-based AAWR exceeding REG-based AAWR by about 1%. The close agreement of AAWR found using both methods provides strong evidence for the accurate determination of AAWR from the CMIP6 GCMs. We use the REG method in this analysis because it provides a more rigorous technique to remove the influence of SAOD on GMST from the CMIP6 multi-model ensemble compared to the LIN method.

The CMIP6 multi-model ensemble provides simulations of near surface air temperature (TAS), which we use to calculate AAWR. The EM-GC uses blended near surface air temperature to determine

values of AAWR. Cowtan et al. (2015) provide a method to create blended near surface air temperature output from the GCMs. The CMIP6 multi-model ensemble contains archived fields of TAS and the temperature at the interface of the atmosphere and the upper boundary of the ocean (TOS) (Griffies et al., 2016), whereas only a subset of GCM groups provide the archived land fraction needed to calculate blended near surface air temperature using the Cowtan et al. (2015) method. Cowtan et al. (2015) compare the modeled and measured trend in global temperature over 1975-2014 and found a 4.0% difference in the trend upon the use of blended temperature from CMIP5 GCMs, rather than global modeled TAS. Their analysis focused on a comparison of modeled and measured temperature, not just the anthropogenic component. We have used the method of Cowtan et al. (2015) to create blended CMIP6 temperature output, for the CMIP6 GCMs that provide TAS, TOS, and the land fraction. Upon our use of blended CMIP6 temperature output for these GCMs, and calculation of AAWR for 1975-2014, we find that AAWR based on blended CMIP6 temperature is 3.5% lower than AAWR found when using only TAS. Tokarska et al. (2020b) estimate an effect of 0.013°C decade$^{-1}$ in the trend of CMIP6 temperature output upon the use of blended CMIP6 temperature instead of TAS, while Cowtan et al. (2015) report a difference of 0.030°C decade$^{-1}$ between the trend in observations and modeled output. Since the difference between values of AAWR found using blended CMIP6 temperature output and TAS is so small and does not affect any of our conclusions, we use TAS output from the CMIP6 multi-model archive because this choice allows many more GCMs to be examined.

## 2.4 Effective climate sensitivity

The equilibrium climate sensitivity represents the warming that would occur after climate has equilibrated with atmospheric $CO_2$ at the 2×pre-industrial level (Kiehl, 2007; Otto et al., 2013; Schwartz, 2012). In our model framework, we infer the climate sensitivity based on an estimate of climate feedback from the historical record, resulting in the effective climate sensitivity (ECS) (Tokarska et al., 2020a). Effective climate sensitivity is defined by IPCC 2013 as "an estimate of the global mean surface temperature response to doubled carbon dioxide concentration evaluated from model output or observations for evolving non-equilibrium conditions". To calculate ECS from the EM-GC, we use:

$$ECS = \frac{1+\gamma}{\lambda_P} \times 5.35 \, \text{W} \, \text{m}^{-2} \times \ln(2) \tag{10}$$

which represents the rise in GMST for a doubling of $CO_2$, assuming no other perturbations as well as equilibrium in other components of the climate system (i.e., $Q_{OCEAN} = 0$) (Mascioli et al., 2013). The expression for the radiative forcing of $CO_2$ is from Myhre (1998). The quantity $\gamma$ in Eq. (10), which represents the sensitivity of the GMST to feedbacks within the climate system, is the only variable component of ECS. We only use values of $\gamma$ that result in good fits ($\chi^2 \leq 2$ for Eq. (6) to (8)) between modeled and observed GMST and modeled and observed OHC. We refer to the quantity in Eq. (10) as effective climate sensitivity, rather than equilibrium climate sensitivity, because for most of our analysis we assume a constant value of climate feedback inferred from prior observations.

For the estimate of climate sensitivity from the CMIP6 multi-model ensemble, we use the method described by Gregory et al. (2004) (See the supplement and Fig. S15 for more information). The Gregory et al. (2004) method also estimates effective climate sensitivity from the CMIP6 GCMs (Gregory et al., 2004; Sherwood et al., 2020; Zelinka et al., 2020) because it assumes the feedbacks inferred from the first 150 years of the abrupt $4\times CO_2$ CMIP6 GCM simulations persist until equilibrium. At the time of this analysis, 28 models released the necessary output to the CMIP6 archive (see Table S5 for the list of models and individual values of ECS). Several recent analyses suggest the Gregory method underestimates the true value of equilibrium climate sensitivity from the CMIP6 multi-model output (Rugenstein et al., 2020; Sherwood et al., 2020; Zelinka et al., 2020). However, effective climate sensitivity is strongly correlated with the amount of warming simulated by GCMs for high carbon emission scenarios and is more relevant for warming over the time scale of interest (rest of this century) due to the long time needed to achieve equilibrium (Sherwood et al., 2020). We use the Gregory method to calculate ECS from the CMIP6 GCMs because this procedure is preferred by Eyring et al. (2016) for the use within the CMIP6 community.

The estimates of climate sensitivity from Eq. (10) and those found using the Gregory et al. (2004) method are termed "effective" because they assume climate feedback inferred from either the historical climate record or the abrupt $4\times CO_2$ experiment persists until equilibrium. However, these estimates of ECS differ in that the perturbation to the RF of climate over the historical record is considerably smaller than the RF of climate that underlies the $4\times CO_2$ experiment of the Gregory et al. (2004) method. We quantify the impact of time variable climate feedback on climate sensitivity in Sect. 3.3.6.

## 2.5 Aerosol weighting method

Probabilistic forecasts of the future rise in GMST for various SSPs are an important part of our analysis. Probabilities of AAWR and ECS are computed by considering the uncertainty in AER $RF_{2011}$. We also provide probabilistic estimates of AAWR and ECS. All of these quantities are computed by incorporating the uncertainty in the radiative forcing of climate due to tropospheric aerosols within results of our EM-GC simulations. We use an asymmetric Gaussian to assign weights to the value of GMST, AAWR or ECS found for various time series of radiative forcing by aerosols associated with particular values of AER $RF_{2011}$. Figure 5a shows the asymmetric Gaussian function we use to maximize the values of AAWR or ECS at the best estimate of AER $RF_{2011}$ of $-0.9$ W m$^{-2}$, accomplished by giving these values the highest weighting. The IPCC 2013 "likely" range limits of AER $RF_{2011}$ of $-0.4$ and $-1.5$ W m$^{-2}$ (Myhre et al., 2013) are assigned to the $1\sigma$ values of the Gaussian, and the AAWR or ECS estimates occurring at the "likely" range AER $RF_{2011}$ limits are given the same weighting. The $-0.1$ and $-1.9$ W m$^{-2}$ limits of the AER $RF_{2011}$ range are assigned as the $2\sigma$ values of the asymmetric Gaussian, based on the IPCC 2013 description of these two values as being 5 and 95% uncertainty limits (Myhre et al., 2013). The Gaussian we use is asymmetric due to the fact that the distribution of the likely range and 5[th] and 95[th] percentiles of the values of AER $RF_{2011}$ are not distributed symmetrically from the best estimate of $-0.9$ W m$^{-2}$. For example, the likely ranges of AER $RF_{2011}$ are given as $-0.4$ W m$^{-2}$ and $-1.5$ W m$^{-2}$; the $-0.4$ W m$^{-2}$ value is 0.5 W m$^{-2}$ from the best estimate whereas $-1.5$ W m$^{-2}$ is 0.6 W m$^{-2}$ from the best estimate. We fit a Gaussian to the likely range and 5[th] and 95[th] percentiles that has slightly different shape on either side of the best estimate, as shown in Fig. 5a.

Figure 5b shows the value of AAWR in °C decade$^{-1}$ as a function of the climate feedback parameter, $\lambda_\Sigma$, and AER $RF_{2011}$. We are able to find more good fits to the observed GMST for small values of AER $RF_{2011}$ than at larger values of AER $RF_{2011}$. Therefore, we bin values of AAWR (Fig. 5b), ECS (Fig. 5c), or future GMST (described in Sect. 3.3) by AER $RF_{2011}$ and find the probability distribution for values of AAWR, ECS, or future GMST within each bin. The resulting probability distributions are assigned the weights associated with each value of AER $RF_{2011}$ in the bins to arrive at the probabilistic estimates of AAWR or ECS shown in Sect. 3. If we did not use this procedure and instead simply averaged

all of the values for AAWR and ECS shown in Fig. 5, undue emphasis would be given to model results that occur at small AER RF$_{2011}$ (see Fig. S16 for unweighted ECS values). This aerosol weighting method allows the expert assessment of the likely range of RF due to tropospheric aerosols given in Chapter 8 of IPCC 2013 (Myhre et al., 2013) to be quantitatively incorporated into our computations of AAWR, ECS, and GMST.

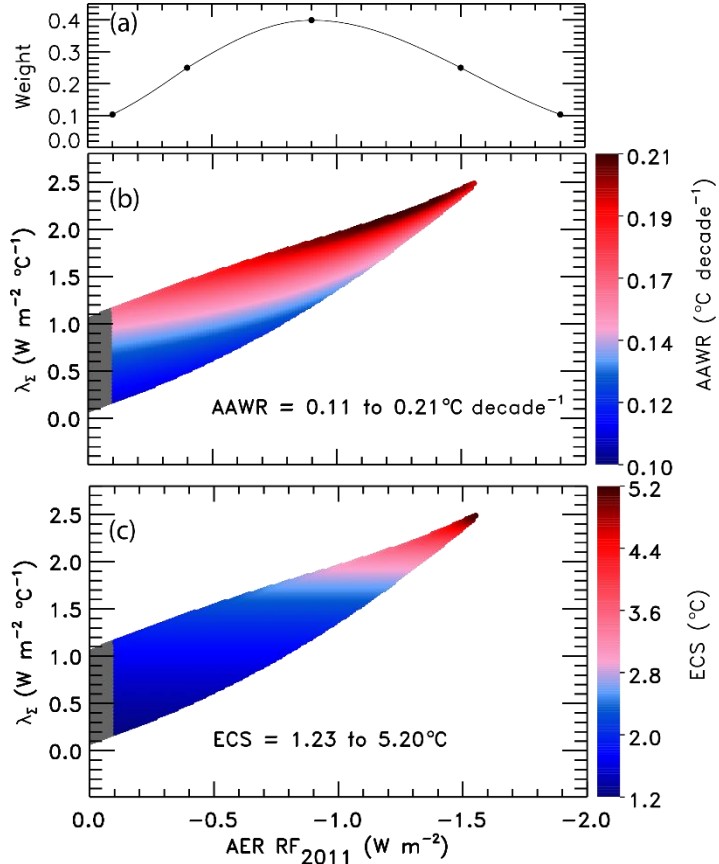

**Figure 5.** Aerosol weighting method. (a) The weights assigned to an asymmetric Gaussian distribution of AER RF$_{2011}$ based on values provided by chapter 8 of IPCC 2013. The five black circles indicate the assigned weights for the AER RF$_{2011}$ best estimate of $-0.9$ W m$^{-2}$, likely range of $-0.4$ and $-1.5$ W m$^{-2}$, and the 5[th] and 95[th] confidence intervals of $-0.1$ and $-1.9$ W m$^{-2}$. (b) Values of AAWR in °C decade$^{-1}$ as a function of climate feedback parameter, $\lambda_\Sigma$, and the value of AER RF$_{2011}$ associated with various time series for the RF of climate due to tropospheric aerosols. The colors denote the values of AAWR calculated from 1975-2014 using the EM-GC trained with the HadCRUT5 $\Delta$T record. (c) ECS in °C as a function of $\lambda_\Sigma$ and the value of AER RF$_{2011}$. The colors denote values of ECS found using the EM-GC. For panels (b) and (c), model results are shown only for combinations of $\lambda_\Sigma$ and RF due to tropospheric aerosols for which good fits to the climate record could be achieved.

### 3 Results

### 3.1 AAWR, comparison to CMIP6 multi-model ensemble

An important measure of any climate model is the ability to accurately simulate the human influence on the global mean surface temperature (GMST) anomaly. We use the attributable anthropogenic warming rate (AAWR) found by our highly constrained Empirical Model of Global Climate (EM-GC) to quantify how well the CMIP6 multi-model ensemble (see Table S7 for a list of CMIP6 GCMs analyzed in this study) is able to simulate the human influence on global warming over the past several decades.

Figure 6 compares values of AAWR from 1975-2014 computed using our EM-GC with AAWR found utilizing archived output from the CMIP6 multi-model ensemble. Seven GMST data sets and five OHC records can be used to estimate AAWR with the EM-GC. For each choice, AAWR exhibits sensitivity to the variation of the time series of radiative forcing due to tropospheric aerosols. Each box and whisker plot found using our EM-GC shows, for a particular choice of GMST and OHC data record, the 25th, 50th, and 75th percentiles of AAWR (box), and 5th and 95th percentiles (whiskers) found using the aerosol weighting method described in Sect. 2.5. The star symbol indicates the minimum and maximum values of AAWR for each value of GMST data set and OHC record. The choice of OHC record and GMST data set has a slight effect on AAWR, as shown by the colored EM-GC symbols in Fig. 6. The averages of the five 25th, 50th, and 75th percentiles of AAWR found using the HadCRUT5 data set for GMST are 0.138, 0.157, and 0.176°C decade$^{-1}$, respectively. The 5th and 95th percentile values of AAWR from HadCRUT5 are 0.120 and 0.195°C decade$^{-1}$.

The box and whisker symbol labeled CMIP6 in Fig. 6 shows the 5th, 25th, 50th, 75th, and 95th percentiles of AAWR calculated from 50 GCMs, also from 1975-2014, as described in Sect. 2.3. The stars denote the minimum and maximum values of AAWR from the GCMs. Two CMIP6 models exhibit values of AAWR similar to the median values we infer from the HadCRUT4, CW14, NOAAGT, BEG, GISTEMP, and HadCRUT5 data records using the EM-GC. In particular INM-CM5-0 (Volodin and Gritsun, 2018) yields 0.147°C decade$^{-1}$ and MIROC6 (Tatebe et al., 2019) results in 0.157°C decade$^{-1}$ (Table S4 provides values of AAWR for all individual CMIP6 GCMs). The median value of AAWR from the CMIP6 multi-model ensemble is 0.221°C decade$^{-1}$, about 40% larger than the 50th percentile value of AAWR found using the HadCRUT5 data set for GMST. The 5th, 25th, 75th, and 95th percentiles of AAWR

from the CMIP6 multi-model ensemble are 0.151, 0.192, 0.245, and 0.299°C decade$^{-1}$, respectively. Some

CMIP6 GCMs exhibit values of AAWR that are 0.14°C decade$^{-1}$ larger than our largest empirical

estimates for 1975-2014; the maximum value of AAWR from the GCMs is 0.354 °C decade$^{-1}$. The

maximum value of AAWR based off the historical climate record using the EM-GC is 0.213°C decade$^{-1}$

(HadCRUT5 data set using the Ishii et al. (2017) OHC record and a time series for RF due to tropospheric

aerosols consistent with AER RF$_{2011}$ equal to −1.5 W m$^{-2}$). All of the EM-GC based values of AAWR in

Fig. 6 are below the 50$^{th}$ percentile of AAWR from the CMIP6 multi-model ensemble of 0.221°C

decade$^{-1}$, supporting the notion that CMIP6 GCMs tend to exhibit a faster rate of anthropogenic warming

over the past four decades than the actual atmosphere.

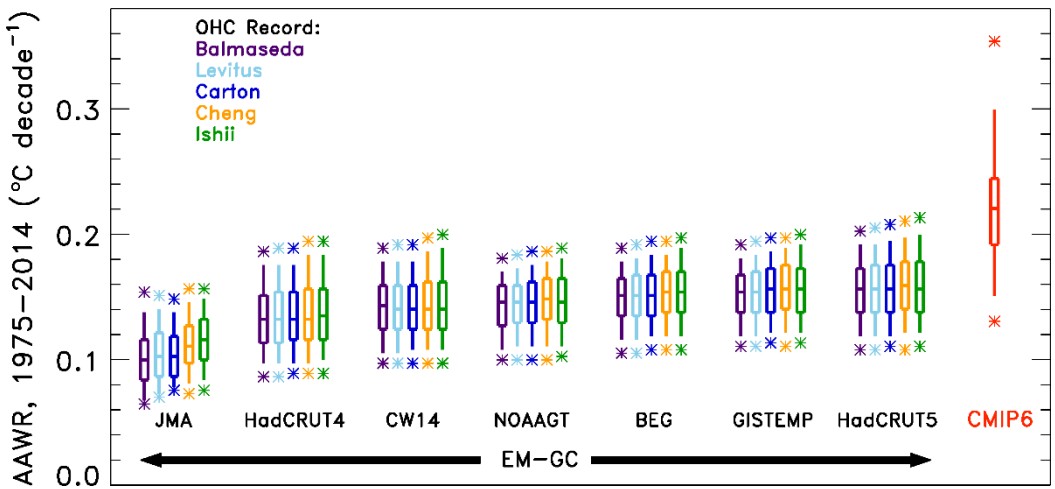

**Figure 6.** AAWR from the EM-GC and CMIP6 multi-model ensemble for 1975-2014. Seven temperature data sets and five ocean heat content records are used to compare values of AAWR computed from the EM-GC. The box represents the 25$^{th}$, 50$^{th}$, and 75$^{th}$ percentiles, the whiskers denote the 5$^{th}$ and 95$^{th}$ percentiles, and the stars show the minimum and maximum values of AAWR from the EM-GC based on the aerosol weighting method described in Sect. 2.5. The red box labeled "CMIP6" shows the 25$^{th}$, 50$^{th}$, and 75$^{th}$ percentiles, the whiskers represent the 5$^{th}$ and 95$^{th}$ percentiles, and the stars denote the minimum and maximum values of AAWR from the 50 member CMIP6 multi-model ensemble.

Our determination that the rate of global warming from the CMIP6 multi-model ensemble over

the time period 1975-2014 significantly exceeds the rise in GMST attributed to human activity is aligned

with a similar finding highlighted in Figure 11.25b of chapter 11 of the IPCC 2013 report that CMIP5

models tend to warm too quickly compared to the actual climate system over the time period 1975-2014

(Kirtman et al., 2013). The values of AAWR from the CMIP6 multi-model ensemble from our analysis

present a similar finding as Tokarska et al. (2020b) and CONSTRAIN (2020), that some of the CMIP6 models over estimate recent warming trends. Tokarska et al. (2020b) examine the trend in the human component of GMST from 1981-2014. We arrive at a similar conclusion as these studies that CMIP6 GCMs overestimate the rate of global warming for the 1982-2014 time period of AAWR as shown in Table S2 and S3. Our results, the finding by the IPCC 2013 report, Tokarska et al. (2020b), and CONSTRAIN (2020) appear to be quite different than the conclusion of Hausfather et al. (2020) that past climate models have matched recent temperature observations quite well. The Hausfather et al. (2020) study does not examine CMIP5 GCMs, let alone CMIP6 GCMs, and the last two rows of their Table 1 indicate that the skill of climate models forecasting the change in GMST over time decreased considerably between the Third Assessment Report (TAR) and the Fourth Assessment Report (AR4). The change in temperature over time for the TAR and AR4 only span 17 and 10 years, respectively (Hausfather et al., 2020). In Fig. 6, we examine the ability of the GCMs to simulate the rise in GMST attributed to humans over a 40 year time period, which provides a better measure of how well the models simulate the observations than the shorter time period. The temperature change over time for the TAR and AR4 examined by Hausfather et al. (2020) ends in 2017, which was right after a very strong ENSO, so their analysis may be influenced by the 2015 to 2016 ENSO event. In contrast, our analysis of AAWR is not influenced by natural variability such as ENSO because we examine the human component of global warming after explicitly accounting for and removing the influence of ENSO on GMST. Consequently, our determination of AAWR from observations (Table S2) and GCMs (Table S3) depends only to a small extent on the specification of start year (for values ranging from 1970 to 1984) and end year (2004 to 2018). Our analysis shows that upon quantification of the human driver to global warming within both the data record and climate models, the CMIP6 GCMs warm faster than observed GMST over the past four decades, regardless of precise specification of start and end year.

## 3.2 ECS

Climate sensitivity is a metric often used to compare the sensitivity of warming among GCMs, as well as with warming inferred from the historical climate record. Figure 7 shows values of effective climate sensitivity (ECS) inferred from the climate record using our EM-GC, seven GMST data sets, and five

OHC records. As for AAWR, the largest variation in ECS is driven by uncertainty in AER $RF_{2011}$. The colored circles represent the ECS values found using the IPCC 2013 best estimate of AER $RF_{2011}$ of $-0.9$ W m$^{-2}$ (Myhre et al., 2013). The ECS values found utilizing the EM-GC are displayed using a box and whisker symbol. The middle line represents the median values of ECS, and the box is bounded by the 25[th] and 75[th] percentiles. The whiskers connect to the 5[th] and 95[th] percentiles, and the stars denote the minimum and maximum values. We use the aerosol weighting method described in Sect. 2.5 to calculate the percentiles for ECS; values of ECS found without aerosol weighting are shown in Fig. S16. Varying the choice of GMST data record has a slight effect on the value of ECS, whereas the choice of OHC record has a larger effect, as indicated by the various heights of the box and whiskers and the maximum values of ECS. In the EM-GC framework, the ocean heat export term ($Q_{OCEAN}$) represents disequilibrium in the climate system. We compute values of $Q_{OCEAN}$ from various records of OHC. If the current value of $Q_{OCEAN}$ is as large as suggested by the Cheng 2017 and Ishii et al. (2017) OHC records, then Earth's climate will exhibit a larger rise in GMST to reach equilibrium than if the value of $Q_{OCEAN}$ inferred from the OHC record of Balmaseda et al. (2013) is correct. The averages of the 25[th], 50[th], and 75[th] percentiles of ECS found using the HadCRUT5 data set for GMST are 1.74, 2.12, and 2.67°C, respectively. The average best estimate of ECS using the HadCRUT5 data set and an AER $RF_{2011}$ value of $-0.9$ W m$^{-2}$ is 2.33°C.

The box and whisker symbol labeled CMIP6 in Fig. 7 shows the 25[th], 50[th], 75[th], and 5[th] and 95[th] percentiles of ECS calculated from output of 28 CMIP6 models, as described in Sect. 2.4. Minimum and maximum values are represented by the stars. The values of ECS from the CMIP6 multi-model ensemble are larger than the majority of values inferred from the climate record using the EM-GC. The height of the box for the CMIP6 multi-model ensemble estimate of ECS is larger than the height of the boxes for ECS inferred from the climate record using the EM-GC, indicating that the GCMs exhibit a wide range of ECS values. The 25[th] and 75[th] percentiles of ECS from the CMIP6 multi-model ensemble are 2.84°C and 4.93°C, respectively. The 5[th] percentile of ECS from the CMIP6 multi-model ensemble is 2.19°C, and the 95[th] percentile is 5.65°C (see Table S4 for ECS values for specific models). In contrast, the average 5[th] and 95[th] percentiles from the EM-GC are 1.40°C and 3.57°C, respectively. The median value of ECS from the CMIP6 multi-model ensemble is 3.74°C, 1.6 times the best estimate of 2.33°C found using the

HadCRUT5 temperature record. All estimates of ECS described above are found assuming constant
climate feedback over time. If climate feedback changes over time, then our estimates of ECS will
increase as discussed in Sect. 3.3.6.

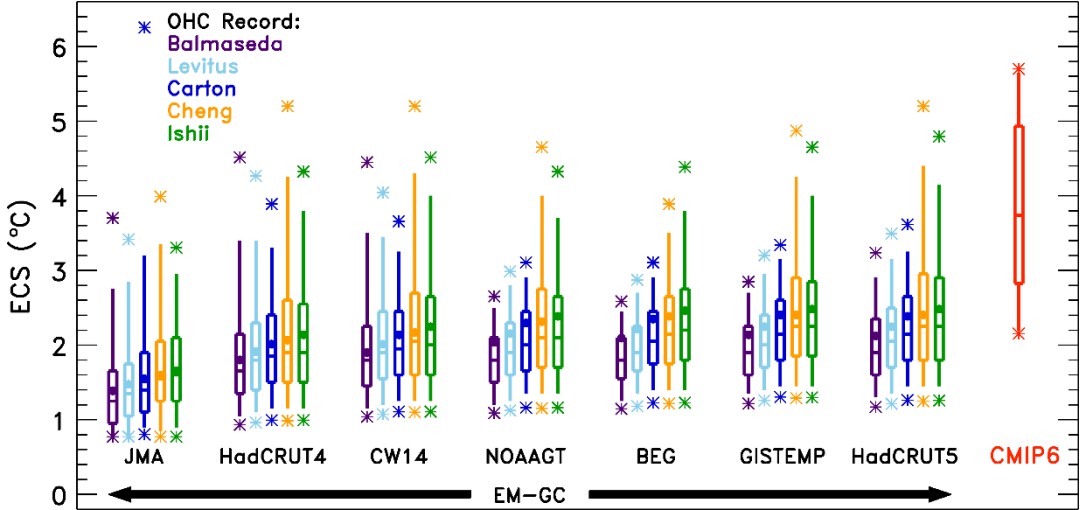

**Figure 7.** ECS from the EM-GC and the CMIP6 multi-model ensemble. Seven GMST data sets and five ocean
heat content records are used to compare values of ECS computed from the EM-GC. The box represents the 25[th],
50[th], and 75[th] percentiles, the whiskers denote the 5[th] and 95[th] percentiles, and the stars indicate the minimum and
maximum values of ECS using the EM-GC based on the weighting method described in Sect. 2.5. The circles
denote the value of ECS associated with the best estimate of AER $RF_{2011}$ of −0.9 W m$^{-2}$. The red box labeled
"CMIP6" represents the 25[th], 50[th], and 75[th] percentiles, the whiskers denote the 5[th] and 95[th] percentiles, and the
stars indicate the minimum and maximum values of ECS from the 28 member CMIP6 multi-model ensemble.

Figure 8 summarizes values of ECS found utilizing the analysis of the century and a half long
climate record using our EM-GC, our examination of a 28 member CMIP6 GCM ensemble, and 13 other
recent studies. The studies are divided into three categories: those that estimated ECS based on
observations (Historical Analysis), others that used GCM output but constrained the output in some way
(Constrained GCM Output), and studies that examined raw GCM output (GCM Output). We obtain a best
estimate for ECS of 2.33°C using the HadCRUT5 data record and a value of AER $RF_{2011}$ = −0.9 W m$^{-2}$
with a range of ECS of 1.40-3.57°C (5[th] and 95[th] percent confidence interval). The use of HadCRUT5
rather than HadCRUT4 imparts a significant rise to our best estimate of ECS, which is 1.99°C (range of
740 1.12-3.63°C) upon use of HadCRUT4. Both of these estimates of ECS largely fall within the range
provided by IPCC 2013 of 1.5°C to 4.5°C for ECS and is supported by three other derivations of ECS

from the empirical climate record: 2.0°C (range of 1.2-3.9°C) given by Otto et al. (2013), 1.87°C (range of 1.1-4.05°C) given by Lewis and Grünwald (2018), and 2.0°C (range of 1.2-3.1°C) given by Skeie et al. (2018) (all range values are for the 5$^{th}$ and 95$^{th}$ percent confidence interval). All of these studies proceeded the release of HadCRUT5. Our estimate of ECS covers a similar range of values given by Cox et al. (2018), Dessler et al. (2018), and Nijsse et al. (2020), as illustrated in Fig. 8. Our determination of ECS from the CMIP6 GCMs resembles that from Proistosescu and Huybers (2017) and Zelinka et al. (2020) as indicated in the GCM Output category of Fig. 8.

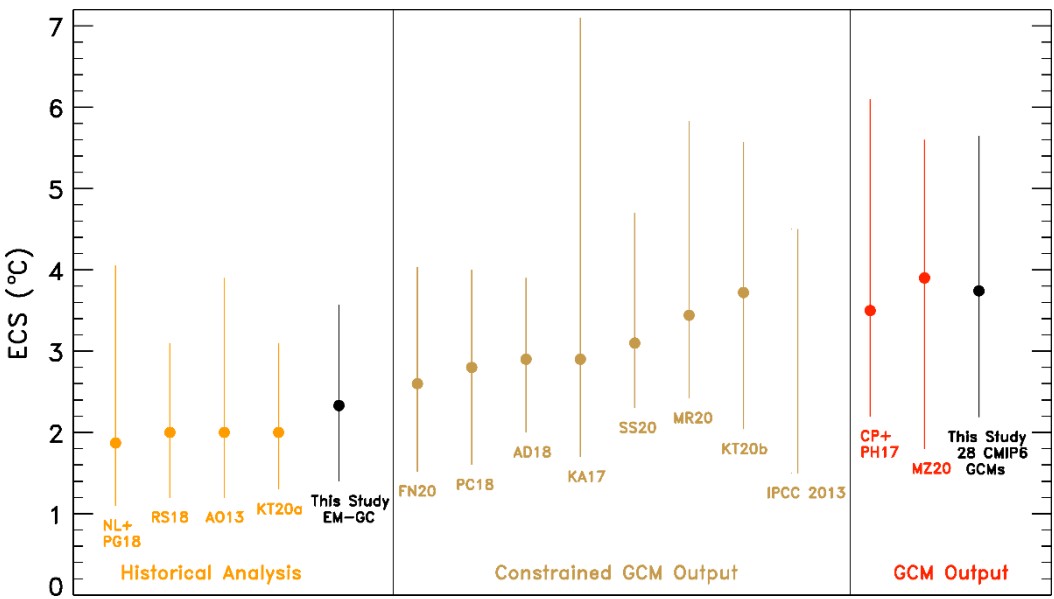

**Figure 8.** Values of ECS from the EM-GC (black) trained using the HadCRUT5 GMST record, our analysis of the CMIP6 multi-model ensemble (black), and 13 other studies grouped by type of analysis. The studies are listed by lead author (first initial of their first name and first initial of their last name) and the year of publication, unless there are only two authors, in which case initials of both authors are listed. Historical analysis includes Lewis and Grünwald (2018) NL+PG18, Otto et al. (2013) AO13, Skeie et al. (2018) RS18, and Tokarska et al. (2020a) KT20a. Constrained GCM output includes Armour (2017) KA17, Cox et al. (2018) PC18, Dessler et al. (2018) AD18, Nijsse et al. (2020) FN20, Rugenstein et al. (2020) MR20, Sherwood et al. (2020) SS20, Stocker et al. (2013) IPCC 2013, and Tokarska et al. (2020b) KT20b. GCM output includes Proistosescu and Huybers (2017) CP+PH17 and Zelinka et al. (2020) MZ20. The studies estimating effective climate sensitivity are AO13, NL+PG18, RS18, FN20, SS20 KT20a, KT20b, and MZ20. The studies estimating equilibrium climate sensitivity are KA17, AD18, PC18, MR20, and CP+PH17. See the supplement for the confidence intervals shown for each study and more information about which studies are estimating effective or equilibrium climate sensitivity.

Recent studies have shown that the CMIP6 multi-model ensemble exhibits higher values of ECS than the CMIP5 models because of larger, positive cloud feedbacks within the latest models (Gettelman

et al., 2019; Meehl et al., 2020; Sherwood et al., 2020; Zelinka et al., 2020). The IPCC 2013 report gives a likely range of 1.5°C to 4.5°C for climate sensitivity (Stocker et al., 2013), and some of the CMIP6 GCMs analyzed in this study have values of ECS more than 1°C above this range. However, some in the climate community seem to currently doubt whether the very large values of ECS are representative of the real world (CONSTRAIN, 2020; Forster et al., 2020; Lewis and Curry, 2018; Tokarska et al., 2020b). Gettelman et al. (2019) found that the newest version of the Community Earth System Model (CESM2) has a higher value of ECS than CESM1 (5.3°C versus 4.0°C) and urge the climate community to work together to determine the plausibility of such high values of ECS. Zhu et al. (2020) found that the high values of ECS in CESM2 and other GCMs is not supported by the paleoclimate record and are biased too warm. An analysis by Nijsse et al. (2020) coupled the CMIP6 multi-model ensemble to a two-box energy balance model and the climate record and obtained a median value of ECS of 2.6°C and range of 1.52-4.03°C (5[th] and 95[th] percentiles). Similarly, Sherwood et al. (2020) conclude cooling during the Last Glacial Maximum provides strong evidence against ECS being greater than 4.5°C and conclude ECS lies within the range of 2.3 to 4.7°C at the 5[th] to 95[th] percent confidence intervals.

We obtain a wide range of ECS values from our EM-GC simulations of the climate record due to consideration of the uncertainty in the radiative forcing of climate from tropospheric aerosols (Figs. 5c and 7). However, under one circumstance, we find values of ECS using the EM-GC that are similar to the maximum value of ECS from the CMIP6 multi-model ensemble. Our large estimate of ECS occurs if we assume that anthropogenic aerosols have exhibited strong cooling and offset a large amount of greenhouse gas warming, such that the observed GMST record can only be well simulated under the condition of large climate feedback (i.e., values of $\lambda_\Sigma$ in Eq. (3) greater than or equal to 2.45 W m$^{-2}$ °C$^{-1}$). If aerosols have truly strongly cooled the climate, offsetting the vast majority of the rise in RF due to greenhouse gases as suggested by Shen et al. (2020), the actual value of ECS may lie close to 5°C or larger. Under the scenario that aerosols have not cooled this strongly (Bond et al., 2013)), then it is feasible that ECS lies well below 5°C. The highest values of ECS found using our analysis (red portion of Fig 5c) are assigned low weights due to the assessment by Myhre et al. (2013) that the large AER RF$_{2011}$ associated with these ECS values is unlikely.

Four empirical determinations of ECS (our study plus Lewis and Grünwald (2018), Otto et al. (2013), and Skeie et al. (2018)) and the CMIP5 or CMIP6-constrained estimates of Cox et al. (2018), Dessler et al. (2018), and Nijsse et al. (2020) are in slight contrast with the 2.3-4.7°C range for ECS (5[th] and 95[th] confidence interval) published recently by Sherwood et al. (2020) (Fig 8). As noted above, Sherwood et al. (2020) use paleoclimate data to rule out the high range of ECS. They rely on a determination that the feedback due to clouds is moderately to strongly positive to rule out the low range of ECS found by our analysis and the studies noted above. We caution that knowledge of the cloud feedback from observations is generally limited to databases such as the International Satellite Cloud Climatology Project (ISCCP) (Schiffer and Rossow, 1983) and Pathfinder Atmospheres Extended (PATMOS-x) (Foster and Heidinger, 2013). While these databases are monumental in terms of complexity and scope, they cover only a fairly short (i.e., about 36 years) part of the century and a half climate record (Klein et al., 2017; Sherwood et al., 2020). Most assessments of total cloud feedback rely on some combination of observations such as ISCCP, PATMOS-x, or other satellite records together with the results of regression analysis, GCM projections, and large eddy simulations that are able to resolve some of the convective processes involved in cloud formation (Klein et al., 2017; Sherwood et al., 2020). The most important component of the global cloud feedback is tropical low clouds, which Sherwood et al. (2020) consider to exert a positive feedback on climate based largely on the results of Klein et al. (2017). The determination by Klein et al. (2017) of a likely positive feedback for tropical low altitude clouds is based on the mean and standard deviation of the central value of this feedback determined by five studies, even though four of these studies exhibit uncertainties that encompass zero feedback and the fifth nearly reaches zero (their Fig. 3). This fact, combined with the recent study by Weaver et al. (2020) who report no long term statistically significant trend in global cloud reflectivity at 340 nm averaged between 45° S and 45° N based on analysis of data collected by a variety of NOAA and NASA satellite instruments, causes us to suggest the true value of ECS may lie below the 2.3°C lower limit given by Sherwood et al. (2020).

In our model framework, the largest uncertainty in ECS is driven by imprecise knowledge of the radiative forcing of climate by tropospheric aerosols. As shown in Fig. 5c, a wide range of ECS values can be inferred from the century and a half long climate record. We stress that each value of ECS shown

in Fig. 5c is based on a simulation for which $\chi^2_{ATM}$, $\chi^2_{RECENT}$, and $\chi^2_{OCEAN}$ are all less than or equal to 2. Better knowledge of AER RF for the contemporary atmosphere would lead to a reduction in the uncertainty of ECS. Numerous studies of the climate record, including our century and a half simulations, infer the possibility of lower values of ECS than was given by a recent analysis of studies that involve examination of data from compendiums such as ISCCP and PATMOS-x (Sherwood et al., 2020). However, the analysis by Sherwood et al. (2020) did not examine consistency of the inferred value of ECS with the ability of models to accurately simulate the GMST anomaly between 1850 and present or over the past 40 years.

We conclude this section by commenting on the relationship between ECS and AAWR in our model framework. Eight of the CMIP6 GCMs (GFDL-ESM4, GISS-E2-1-G, INM-CM5-0, INM-CM4-8, MIROC6, MIROC-ES2L, NorESM2-LM, and NorESM2-MM) exhibit values of ECS and AAWR consistent with the minimum and maximum estimates based on our EM-GC constrained by the HadCRUT5 GMST record (Table S5 and Fig. S17). An analysis of the relationship between AAWR and ECS from the CMIP6 GCMs illustrates that 78% of the variance in ECS among the 28 CMIP6 GCMs that provide both quantities is explained by AAWR (see Fig. S17). This result indicates CMIP6 GCMs that accurately simulate the rise in observed $\Delta T$ over the past few decades exhibit values of ECS that are in line with our empirically based estimate.

## 3.3 Future projections

### 3.3.1 CMIP6

The CMIP6 multi-model archive provides future projections of the GMST anomaly relative to pre-industrial ($\Delta T$) using the ScenarioMIP Shared Socioeconomic Pathways (SSPs). Figure 9 shows the CMIP6 multi-model ensemble projections of $\Delta T$ for the four SSPs (SSP1-1.9, SSP1-2.6, SSP4-3.4, and SSP2-4.5) highlighted in our analysis. Each SSP scenario has varying amounts of gridded, monthly mean TAS projections submitted to the CMIP6 archive by GCMs. The global, monthly $\Delta T$ time series for all of the runs for each CMIP6 GCM were averaged together to obtain one time series of $\Delta T$. The varying amount of GCM output available for each SSP scenario is due to the fact that: a) SSP1-2.6 and SSP2-4.5 are Tier 1 scenarios (O'Neill et al., 2016) and are designated as priority over the other SSPs (as described

in Sect. 2.2.2), and b) not all GCMs have provided results to the CMIP6 archive at the time of the analysis. More CMIP6 multi-model output will likely become available as modeling groups who have not submitted output to the CMIP6 archive finalize their results. However, we do not expect additional GCM simulations will affect our conclusions unless the GCM output is significantly different than that currently available.

The red trapezoid in Fig. 9 labeled as the IPCC 2013 likely range is the same trapezoid as that displayed on Figure 11.25b from chapter 11 of the IPCC 2013 report (Kirtman et al., 2013). The recent observations of $\Delta T$ from HadCRUT5 lie towards the top of the likely range of warming designated by this trapezoid. Many of the projections of the rise in $\Delta T$ from the CMIP6 multi-model ensemble lie above the IPCC 2013 likely range of warming. The Paris Agreement target of 1.5°C and upper limit of 2.0°C are shown as yellow circles, included to allow for comparison of the future projections of $\Delta T$ from the CMIP6 multi-model ensemble with the goals of the agreement. The thick blue line on each plot is the CMIP6 multi-model mean of $\Delta T$, and the dashed blue lines are the minimum and maximum $\Delta T$ projections from the CMIP6 multi-model ensemble. For SSP1-1.9, the multi-model mean projection of $\Delta T$ in 2100 from the CMIP6 GCMs lies just above the Paris Agreement target at 1.6°C, whereas for SSP1-2.6 the CMIP6 multi-model mean reaches the Paris Agreement upper limit of 2.0°C at the end of this century. For both SSP4-3.4 and SSP2-4.5, the end of century CMIP6 multi-model mean lies above the Paris Agreement upper limit at 3.0°C and 3.1°C, respectively.

Figure 9 illustrates there are two groups of CMIP6 multi-model projections of $\Delta T$, with a few GCMs having future values of $\Delta T$ that are considerably higher than others. This divergence for GCM projections of $\Delta T$ is especially evident in Fig. 9a, c, and d. The two CMIP6 GCMs that have the highest values of $\Delta T$ across the four SSPs are CanESM5 (Swart et al., 2019) and UKESM1 (Sellar et al., 2020). The CanESM5 and UKESM1 GCMs have the highest values of AAWR (0.354°C decade$^{-1}$ and 0.299°C decade$^{-1}$, respectively), large values of ECS (5.70°C and 5.40°C, respectively), and exceed observed $\Delta T$ reported by HadCRUT5 for the past few decades.

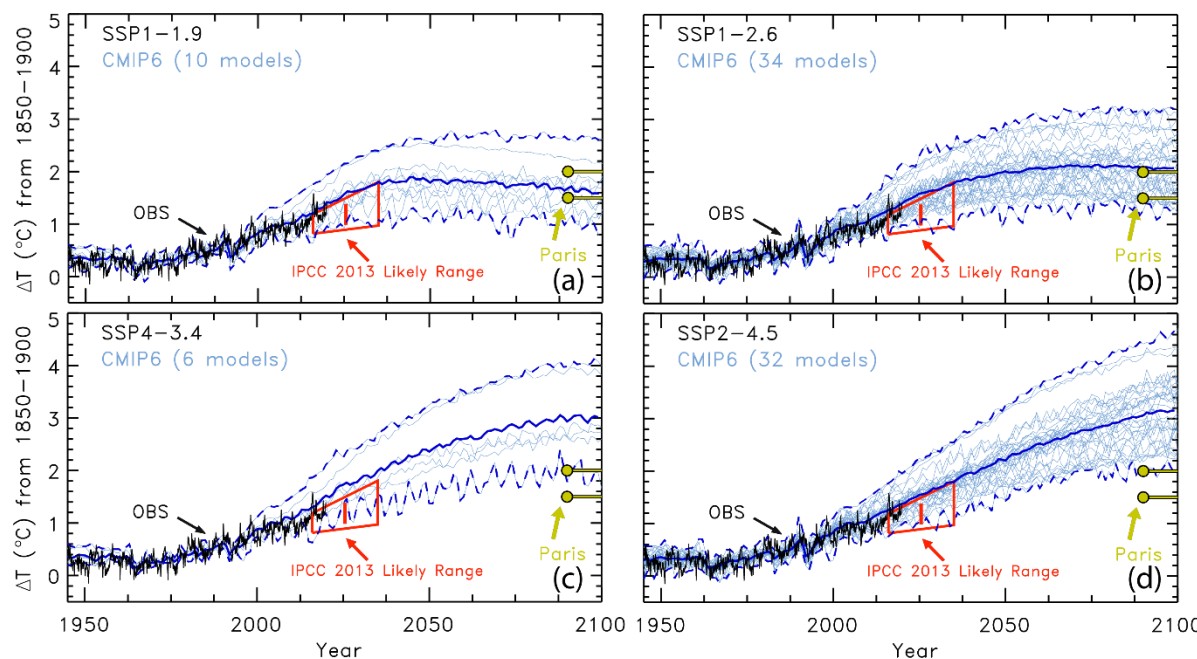

**Figure 9.** Historical simulations and future projections of GMST from the CMIP6 multi-model ensemble for several SSP scenarios. (a) GCM simulations from the Historical experiment, and future model projections from SSP1-1.9. Observations (black) are from HadCRUT5 to the end of 2019. The IPCC 2013 likely range of warming (red) is from Figure 11.25b from chapter 11 of the IPCC 2013 report. The CMIP6 multi-model mean (thick, blue) and minimum and maximum (dashed, blue) lines are shown. Global, monthly $\Delta T$ was created by averaging the TAS output over the globe with a cosine latitude weighting. The Paris Agreement target of 1.5°C and upper limit (yellow) of 2.0°C are included to demonstrate how the GCM projections compare. (b) Future GMST projections from SSP1-2.6. (c) Future GMST projections from SSP4-3.4. (d) Future GMST projections from SSP2-4.5.

### 3.3.2 EM-GC

The EM-GC is also used to project future changes in $\Delta T$ using the SSPs. Figure 10 shows the GMST anomaly in 2100 from pre-industrial ($\Delta T_{2100}$) as a function of the climate feedback parameter and AER $RF_{2011}$, for the four SSPs highlighted throughout. Only model runs from the EM-GC that achieved a good fit to the climate record ($\chi^2_{ATM} \leq 2$, $\chi^2_{RECENT} \leq 2$, $\chi^2_{OCEAN} \leq 2$) are shown. The EM-GC runs that satisfy these three $\chi^2$ constraints but fall outside of the IPCC 2013 range for AER $RF_{2011}$ (Myhre et al., 2013) are shaded grey (left hand side of each panel). We do not consider the EM-GC projections that lie outside of the IPCC 2013 range for AER $RF_{2011}$ in our projections of $\Delta T$, yet these results are shown to illustrate that the EM-GC can fit the climate record with estimates of the RF due to tropospheric aerosols that lie below (i.e., less cooling) of the 5[th] confidence interval of −0.1 W m$^{-2}$ for AER $RF_{2011}$ given by IPCC 2013. We cannot establish any good fits of the HadCRUT5 GMST record for AER $RF_{2011}$ with a cooling

stronger than about $-1.55$ W m$^{-2}$. The range of $\Delta T_{2100}$ we compute using the EM-GC for SSP1-1.9, SSP1-2.6, SSP4-3.4, and SSP2-4.5 are 0.75-2.06°C, 0.96-2.58°C, 1.18-3.01°C, and 1.45-3.47°C, respectively. Results for SSP4-6.0, SSP3-7.0, and SSP5-8.5 are shown in Fig. S18: $\Delta T_{2100}$ ranges are 1.70-4.02°C, 2.26-4.93°C, and 2.62-6.02°C for these three scenarios.

875   The large range of $\Delta T_{2100}$ found for any given SSP scenario is caused by the fact that the climate record can be fit nearly equally well by a considerably large combination of the climate feedback parameter (our $\lambda_\Sigma$) and scenarios for radiative forcing due to tropospheric aerosols. The more aerosols have cooled, offsetting the relatively well-known warming due to GHGs, the larger $\lambda_\Sigma$ must be to fit the climate record. Since the RF of aerosols is set to diminish in the future due largely to public health

880 concerns (Lelieveld et al., 2015; Shindell et al., 2016; Smith and Bond, 2014), the part of our model ensemble requiring relatively large values of $\lambda_\Sigma$ to achieve a good fit to the climate record will result in higher values of $\Delta T_{2100}$ than other members of our model ensemble with small values of $\lambda_\Sigma$. Most GCMs sample only a small portion of the possible combinations of $\lambda_\Sigma$ and AER RF$_{2011}$ shown in Figs. 10 and S18.

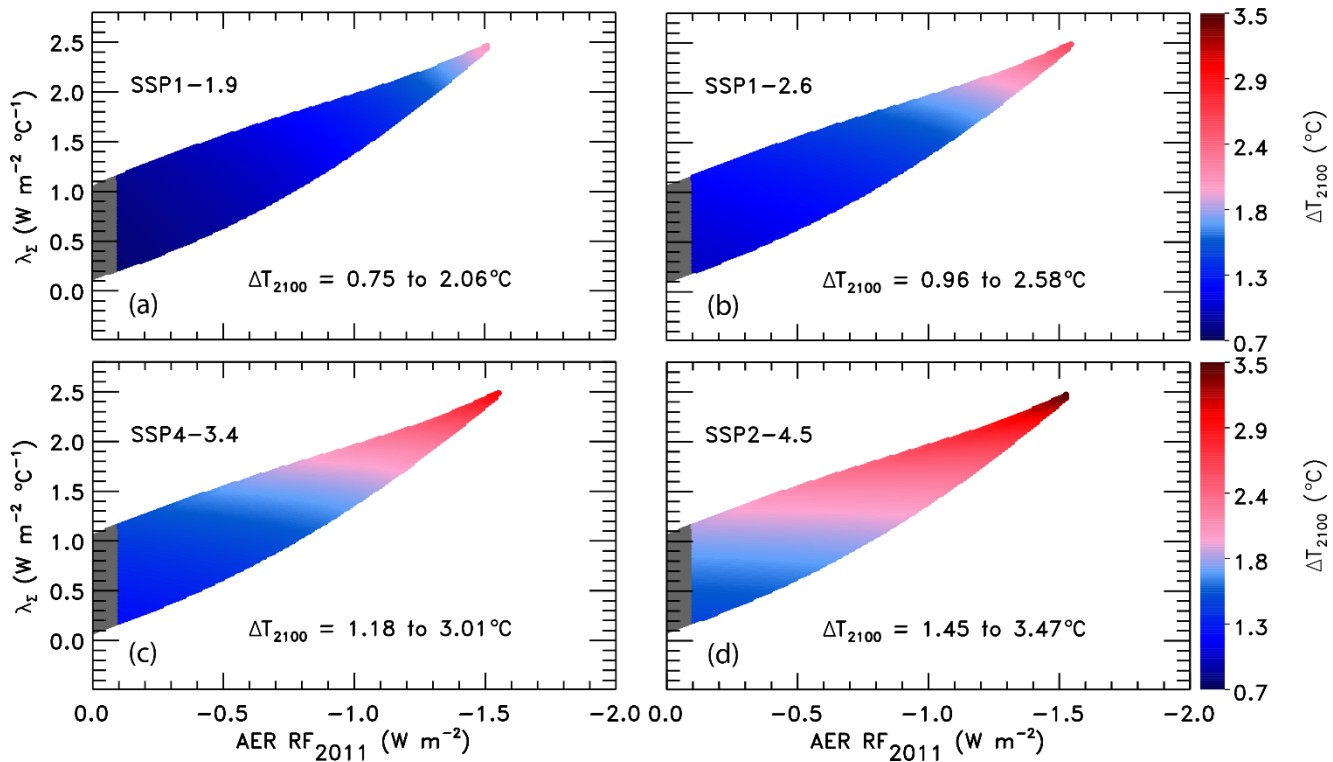

**Figure 10.** $\Delta T_{2100}$ as a function of climate feedback parameter and tropospheric aerosol radiative forcing in 2011 using the EM-GC trained with the HadCRUT5 $\Delta T$ record. (a) Future GMST change for SSP1-1.9. The region outside of the AER $RF_{2011}$ range provided by IPCC 2013 is shaded (grey). Colors denote the GMST change in year 2100 relative to pre-industrial. The color bar is the same across all four panels for comparison. (b) GMST anomaly for SSP1-2.6. (c) Future temperature change for SSP4-3.4. (d) GMST anomaly for SSP2-4.5.

### 3.3.3 Comparing CMIP6 and EM-GC

Time series of future projections of $\Delta T$ from the EM-GC can be illustrated as probabilistic forecasts. Figure 11 shows the change in future $\Delta T$ for SSP1-1.9, SSP1-2.6, SSP4-3.4, and SSP2-4.5 colored by the probability of reaching at least that rise in $\Delta T$ by the end of the century. The EM-GC

probabilities are computed from ensemble members for model runs constrained by the HadCRUT5 data records for GMST and the average of 5 OHC data records (Fig. S9) based on the aerosol weighting method, described in Sect. 2.5. The trapezoid from chapter 11 of IPCC 2013 (Kirtman et al., 2013) is shown on Fig. 11 in black to highlight that the EM-GC projections of the future rise in $\Delta T$ lie within the IPCC 2013 likely range of warming. The Paris Agreement target and upper limit are included to compare

the EM-GC projections of future $\Delta T$ to the Paris Agreement goals. The white shaded region is the EM-

GC's median estimate of future ΔT for each SSP scenario. The median estimate for $\Delta T_{2100}$ for simulations using SSP1-1.9 and SSP1-2.6 falls below the Paris Agreement target at 1.1°C and 1.4°C, respectively. The median estimate of $\Delta T_{2100}$ from the EM-GC for SSP4-3.4 is between the Paris Agreement target and upper limit at 1.8°C. For SSP2-4.5 the median estimate of $\Delta T_{2100}$ is 2.1°C, which is just above the Paris Agreement upper limit. The CMIP6 minimum, multi-model mean, and maximum projections of ΔT, based on the ensembles in Fig. 9, are also shown in Fig. 11. The CMIP6 minimum projection of the rise in ΔT falls near the EM-GC median estimate of ΔT for each SSP scenario. The CMIP6 multi-model mean value of the future change in ΔT falls below the EM-GC maximum value of ΔT, while the CMIP6 maximum value is far above the maximum projections of the future rise in ΔT using the EM-GC. Results for SSP4-6.0, SSP3-7.0, and SSP5-8.5 are provided in Fig. S19.

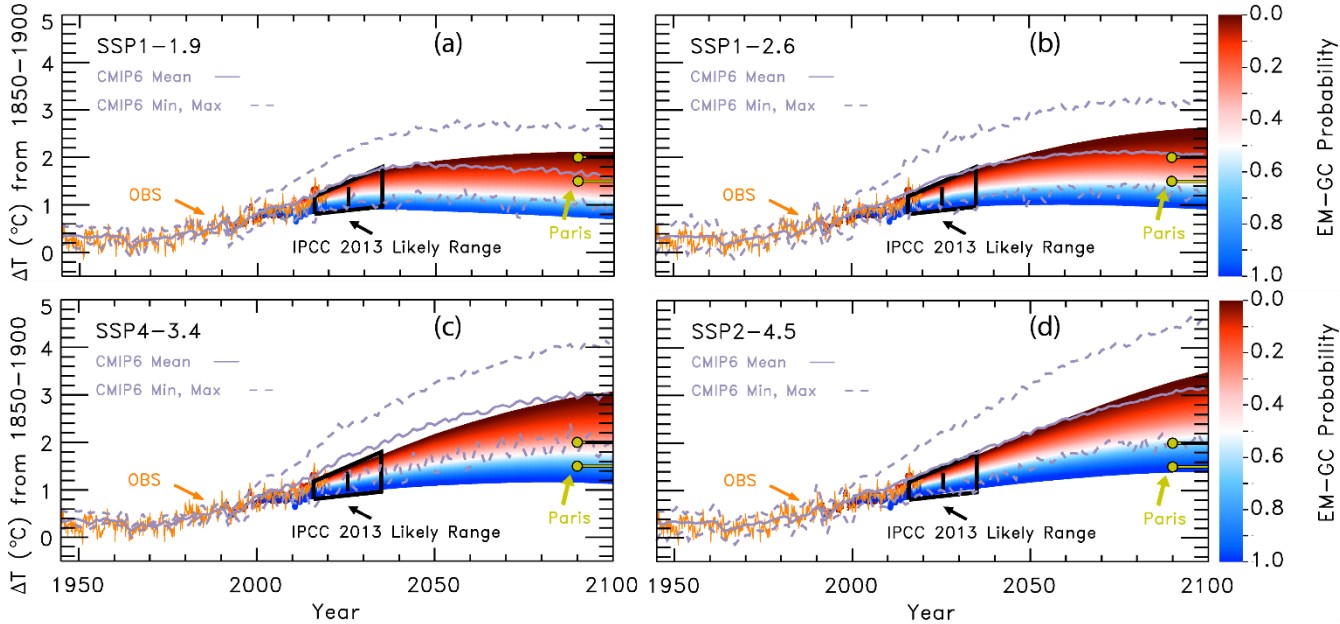

**Figure 11.** Probabilistic forecasts of the future rise in ΔT from the EM-GC trained using the HadCRUT5 ΔT record for several SSPs. (a) Future projections of ΔT for SSP1-1.9. Observations (orange) are from HadCRUT5. The IPCC 2013 likely range of warming (black) is from Figure 11.25b of chapter 11 of IPCC 2013. The Paris Agreement target and upper limit (yellow) are shown for comparison to EM-GC projections. The CMIP6 minimum, multi-model mean, and maximum values of ΔT are shown to compare to EM-GC projections. Colors denote the probability of reaching at least that temperature by the end of the century. (b) Future projections of ΔT for SSP1-2.6. (c) Future projections of ΔT for SSP4-3.4. (d) Future projections of ΔT for SSP2-4.5.

Figure 12 compares probability distribution functions (PDFs) for the projection of $\Delta T_{2100}$ utilizing the EM-GC with the HadCRUT5 GMST record and average of the five OHC data sets and the CMIP6

multi-model ensemble. For the CMIP6 multi-model results, we compute the probabilities of achieving the Paris Agreement target of 1.5°C and upper limit of 2.0°C (at the end of the century) by calculating how many of the GCMs participating in each scenario have projections of $\Delta T_{2100}$ below the target or upper limit. The probabilities for the projections of $\Delta T_{2100}$ using the EM-GC are computed using the aerosol weighting method, described in Sect. 2.5. The height of each histogram represents the probability that a particular range of $\Delta T_{2100}$, defined by the width of each line segment, will occur. The left-hand y-axis displays the probability of $\Delta T_{2100}$ using the EM-GC, while the right-hand y-axis represents the probability of $\Delta T_{2100}$ using the CMIP6 multi-model simulations. The values on the CMIP6 multi-model ensemble y-axis are double the values on the EM-GC y-axis, for visual comparison. The solid black line denotes the Paris Agreement target and the dotted black line signifies the upper limit on each panel. The PDFs for SSP4-6.0, SSP3-7.0, and SSP5-8.5 are shown in Fig. S20.

Numerical values of probabilities for staying at or below the Paris Agreement target for SSP1-2.6 or upper limit for SSP4-3.4 are given for the seven GMST records using the EM-GC and CMIP6 multi-model ensemble in Table 1. Projections of $\Delta T_{2100}$ based on the EM-GC provide more optimism for achieving the Paris Agreement goals than the CMIP6 multi-model ensemble, regardless of which GMST data record is used. For simulations constrained using the HadCRUT5 record, the SSP1-2.6 scenario provides a 53% (Table 1) likelihood of $\Delta T_{2100}$ staying at or below 1.5°C and SSP4-3.4 results in a 64% likelihood of limiting warming to 2.0°C by end of century. The probability of achieving the Paris Agreement target or upper limit increases upon using HadCRUT4 rather than HadCRUT5 in the EM-GC framework. The probability of achieving the 1.5°C target for SSP1-2.6 and 2.0°C upper limit for SSP4-3.4 using the HadCRUT4 GMST record are 64% and 74%, respectively (Table 1). This decline in attainment of the goals of the Paris Agreement upon use of HadCRUT5 reflects more rapid warming of this data record compared to HadCRUT4 (Fig. S4e versus S4c). The rapid warming in HadCRUT5 is driven by more accurate buoy records for SST and a statistical gap filling procedure to attain global coverage (Morice et al., 2021). The impact on the likelihood of achieving the Paris Agreement goals of for the other SSP scenarios upon using the HadCRUT4 or HadCRUT5 data records is detailed in Table S6.

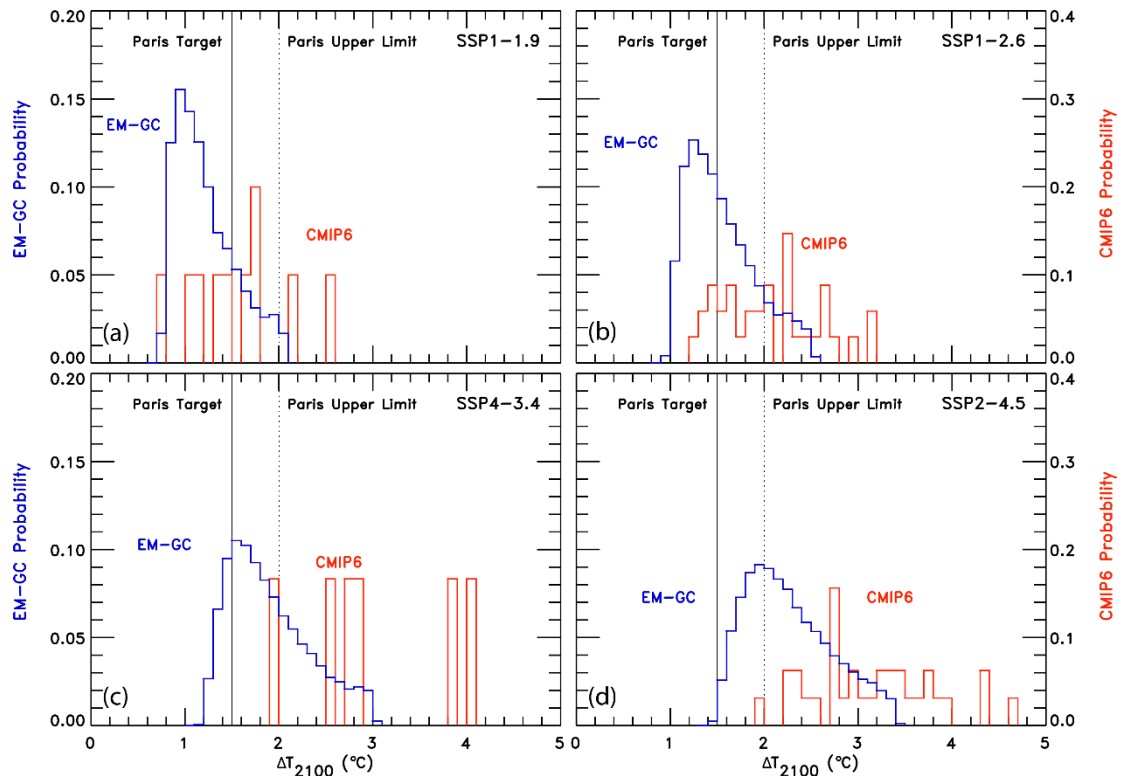

**Figure 12.** Probability density functions (PDF) for $\Delta T_{2100}$ found using the EM-GC trained with the HadCRUT5 temperature record (dark blue) and CMIP6 multi-model results (red). (a) PDF for EM-GC results and CMIP6 multi-model results for SSP1-1.9. The left-hand y-axis is for EM-GC probabilities and the righthand y-axis is for the CMIP6 multi-model ensemble probabilities. (b) PDF for SSP1-2.6. (c) PDF for SSP4-3.4. (d) PDF for SSP2-4.5.

An analysis by Tokarska et al. (2020b) supports our finding of a higher likelihood of attaining the goals of the Paris Agreement than suggested by the CMIP6 multi-model ensemble. Tokarska et al. (2020b) filter the CMIP6 multi-model output on the level of agreement with observations to show that the SSP1-2.6 scenario has a likely range of warming at 1.33-1.99°C above preindustrial by end of century. Previous studies suggested that a 2.6 W m$^{-2}$ scenario was in line with the 2.0°C goal (Kriegler et al., 2014, 2015;

O'Neill et al., 2016; Riahi et al., 2015). Our analysis suggests the 2.6 W m$^{-2}$ scenario provides between a 86-98% probability of limiting warming to 2.0°C and a 53-78% probability of achieving the more stringent 1.5°C target, depending on the GMST record (Table 1). If GHGs were to follow SSP4-3.4, we find a 19-58% probability of limiting warming to 1.5°C and a 64-87% probability of limiting warming to

2.0°C. Significant climate mitigation efforts will be required to keep the growth of $CO_2$, $CH_4$, and $N_2O$

below the trajectories shown for SSP1-2.6 and SSP4-3.4 (Fig. 2).

**Table 1.** Probability of achieving the Paris Agreement target (SSP1-2.6) or upper limit (SSP4-3.4) for seven GMST records using the EM-GC and the CMIP6 multi-model ensemble. The probabilities using the EM-GC are computed using the aerosol weighting method. The probabilities using the CMIP6 models are computed by calculating how many of the models for that scenario are below the temperature limits compared to the total number of models.

|  | Probability of Staying at or Below 1.5°C | | Probability of Staying at or Below 2.0°C | |
| --- | --- | --- | --- | --- |
|  | SSP1-2.6 | SSP4-3.4 | SSP1-2.6 | SSP4-3.4 |
| CMIP6 | 18% | 0% | 47% | 17% |
| HadCRUT5 | 53% | 19% | 86% | 64% |
| GISTEMP | 55% | 20% | 88% | 69% |
| CW14 | 60% | 29% | 89% | 71% |
| NOAAGT | 61% | 27% | 90% | 74% |
| BEG | 62% | 26% | 98% | 76% |
| HadCRUT4 | 64% | 35% | 90% | 74% |
| JMA | 78% | 58% | 95% | 87% |

### 3.3.4 Carbon budgets

The transient climate response to cumulative emissions (TCRE) relates the rise in ΔT to the cumulative amount of carbon released into the atmosphere by human activities. We illustrate TCRE from the EM-GC as probabilistic forecasts, as shown in Fig. S21, to analyze future projections of ΔT. We use the

955 probabilistic forecasts in Fig. S21 to determine the carbon budgets in Table 2. Table 2 contains estimated carbon budgets in the form of the total $CO_2$ emissions (Gt C) since 1870 that result in a 95%, 66%, and 50% probability of the future rise in ΔT staying below the Paris Agreement target of 1.5°C and upper limit of 2.0°C and the future $CO_2$ emissions since 2019. The largest variation in our carbon budget estimates is driven by the uncertainty in AER RF, which is incorporated into the probability of achieving

the Paris Agreement target and upper limit (see Fig. S21 and the supplement). We include a 10% uncertainty, determined from examination of CMIP5 coupled atmospheric / carbon cycle models from Friedlingstein et al. (2014) and Murphy et al. (2014) (see the supplement for more information), within each probability of attaining the Paris goals to represent how atmospheric $CO_2$ will respond to the prescribed carbon emissions.

To obtain a 66% likelihood of limiting the rise in future $\Delta T$ below 1.5°C, only 790 ± 79 Gt C can be released. For a 66% likelihood of the rise in $\Delta T$ staying below the 2.0 °C upper limit, 1,040 ± 104 Gt C can be emitted. To place these numbers in their proper perspective, about 640 Gt C have been released from 1870 through the end of 2019 due to land-use change, fossil fuel emissions, gas flaring, and cement production according to the Global Carbon Budget project (Friedlingstein et al., 2019). In our model framework, after 2019 society can therefore only emit 150 ± 79 Gt C to have a 66% chance of limiting warming to 1.5°C. This future emissions estimate rises to 400 ± 104 Gt C to have a 66% chance of limiting warming to 2.0°C.

**Table 2.** Total cumulative and future carbon emissions that will lead to crossing the Paris temperature thresholds based on the EM-GC trained using the HadCRUT5 $\Delta T$ record. Estimates of $\Sigma CO_2^{EMISSIONS}$ that would cause global warming to stay below indicated thresholds for 95%, 66%, and 50% probabilities and are rounded to the nearest 10 Gt C. The values in the top half of the table are the estimates of total cumulative carbon emissions that will lead to crossing the Paris Agreement thresholds with the 10% uncertainty for how atmospheric CO2 responds to prescribed carbon emissions (see text) included. The values in the bottom half of the table are the estimates of future cumulative carbon emissions after 2019 that will lead to crossing the Paris Agreement thresholds, with the same 10% uncertainty. The range of years given represents when the Paris Agreement thresholds will be passed based on the rate of emissions from SSP5-8.5 or continuing the 2019 rate of emissions of 11.7 Gt C yr$^{-1}$ (Friedlingstein et al., 2019).

| Total $\Sigma CO_2^{EMISSIONS}$ since 1870 from the EM-GC | | | |
| --- | --- | --- | --- |
| | **95%** | **66%** | **50%** |
| **1.5°C** | 730 ± 73 Gt C | 790 ± 79 Gt C | 830 ± 83 Gt C |
| **2.0°C** | 920 ± 92 Gt C | 1040 ± 104 Gt C | 1110 ± 111 Gt C |
| Future $\Sigma CO_2^{EMISSIONS}$ (assuming 640 Gt C released between 1870-2019) | | | |
| | **95%** | **66%** | **50%** |
| **1.5°C** | 90 ± 73 Gt C (2021[a]-2031[a]) (2021[b]-2033[b]) | 150 ± 79 Gt C (2025-2035) (2026-2039) | 190 ± 83 Gt C (2027-2038) (2029-2043) |
| **2.0°C** | 280 ± 92 Gt C (2033[a]-2043[a]) (2036[b]-2051[b]) | 400 ± 104 Gt C (2039-2049) (2045-2063) | 470 ± 111 Gt C (2047-2052) (2050-2069) |

[a] Year the 1.5°C target or 2.0°C upper limit will be exceeded assuming the rate of emission inferred from SSP5-8.5 and the 1σ uncertainty. Applies to the 66% and 50% probabilities.
[b] Year the 1.5°C target or 2.0°C upper limit will be exceeded assuming the 2019 rate of emission of 11.7 Gt C yr$^{-1}$ and the 1σ uncertainty Applies to the 66% and 50% probabilities.

An analysis by van Vuuren et al. (2020) assesses remaining carbon budgets based on cumulative emissions after 2010. Their analysis indicates only 228 Gt C can be released after 2010 to have a 66% probability of achieving the Paris Agreement target of limiting the rise in ΔT below 1.5°C in 2100. They base this estimate on an analysis of climate sensitivity and carbon cycle components, including an adjustment to TCRE for the tendency of CMIP5 GCMs to warm too quickly that had been suggested by Millar et al. (2017). We find a 66% probability of limiting warming to 1.5°C upon the release of 250 ± 79 Gt C between 2010 and 2100. Our results are similar to the findings in van Vuuren et al. (2020). Between 2010 and 2019, about 100 Gt C has been released to the atmosphere (Friedlingstein et al., 2019), so the remaining budget after 2019 for limiting warming to 1.5°C is about 128 Gt C according to van Vuuren et al. (2020). The remaining budget from our analysis is 150 ± 79 Gt C. Our analysis and that by van Vuuren et al. (2020) suggest at the pace of emissions in 2019 of 11.7 Gt C yr$^{-1}$ (Friedlingstein et al., 2019), society will cross this threshold in the next 10 years.

### 3.3.5 Blended methane

Atmospheric abundances of methane will likely continue to increase as society expands natural gas production and agriculture, making it important to analyze the impact of various methane scenarios on the rise of GMST. It is unlikely future atmospheric methane abundances will progress as indicated by SSP1-2.6 (see Fig. 2), a low radiative forcing scenario. Current observations shown in Fig. 2 illustrate that the methane mixing ratio is following SSP2-4.5 and has missed the initial decline needed to follow the SSP1-2.6 pathway. To analyze the effect varying future methane abundance pathways will have on GMST, we have generated linear interpolations of the SSP1-2.6 and SSP3-7.0 methane abundances and created four alternate scenarios (see Fig. S22), which we call blended methane scenarios. We can substitute one of the blended methane scenarios into the EM-GC instead of using the projection of methane specified by the SSP database to quantify the sensitivity of future warming to various evolutions of methane on the rise in GMST.

Figure 13 shows the probability of staying at or below the Paris Agreement target (gold colors) or upper limit (purple colors) for SSP1-2.6 (solid) and SSP4-3.4 (dotted) as a function of the methane mixing

ratio in 2100. The lowest atmospheric methane mixing ratio value in 2100 of 1.15 ppm is from the SSP1-2.6 methane pathway, the highest mixing ratio in 2100 of 3.20 ppm is from the SSP3-7.0 methane pathway. The four in between are the blended methane scenarios. As the atmospheric methane abundance increases, the likelihood of achieving the goals in the Paris Agreement decreases. For SSP1-2.6, the probability of limiting the rise in GMST below the 1.5°C target begins at 53% for HadCRUT5 using the SSP1-2.6 designated methane pathway and decreases as the blended scenarios are considered. The probability of achieving the Paris Agreement target declines to 30% if methane reaches 2.4 ppm in 2100 and to 16% if methane increases to 3.2 ppm in 2100. Even though SSP1-2.6 can have a 53% probability of limiting warming to 1.5°C, achieving this goal can likely only be attained by strict limits on both emissions of carbon dioxide and methane.

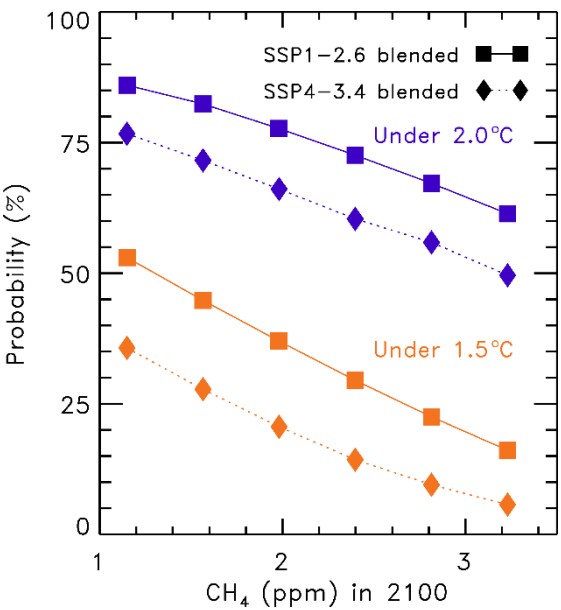

**Figure 13.** Probability of staying at or below the Paris Agreement target and upper limit for SSP1-2.6 and SSP4-3.4 as a function of varying methane scenarios using the EM-GC trained with the HadCRUT5 ΔT record. The atmospheric methane scenarios are calculated using linear combinations of methane abundances from SSP1-2.6 and SSP3-7.0 to span the range of future methane abundances.

In Sect. 3.3.3, we showed that if all GHGs follow the SSP4-3.4 scenario there would be a 64% probability of limiting warming to 2.0°C. If the methane pathway instead follows SSP1-2.6, which has an end of century mixing ratio of only 1.15 ppm, then the probability of achieving the Paris Agreement goal

rises to 77%. If the methane pathway follows SSP3-7.0 and the end of century mixing ratio increases to 3.2 ppm, then the probability of achieving the Paris Agreement goal declines to 50%.

Reducing the future anthropogenic emissions of methane might be more challenging than controlling future emissions of carbon dioxide, because methane has such a wide variety of sources related to energy, agriculture, and ruminants (Kirschke et al., 2013). Given the current widespread use of methane as a source of energy in the United States and parts of Europe (Saunois et al., 2020), combined with the continued growth in the global number of ruminants (Wolf et al., 2017), it seems unrealistic for atmospheric methane to follow the peak and sharp decline starting in 2025 of the SSP1-2.6 pathway (Fig. 3b). Our analysis suggests failure to limit methane to the SSP1-2.6 trajectory will have a larger impact on the achievement of the 1.5°C Paris goal compared to the 2.0°C upper limit. Figure 13 is designed to provide some perspective on the importance of limiting the growth of methane in the atmosphere.

### 3.3.6 Climate feedback

In our analysis above, we have assumed the value of $\lambda_\Sigma$ (and thus $\lambda$, see Eq (3) and corresponding text in Sect. 2.1) is constant over time. Time-constant $\lambda_\Sigma$ is the simplest assumption one can make. The climate record can be fit very well based on this conjecture, as shown in Fig. 1a. However, many GCMs suggest that climate feedback may vary over time (Dong et al., 2020; Marvel et al., 2018; Rugenstein et al., 2020). An analysis by Goodwin (2018) finds there is a delay in the response of climate feedback to a change in radiative forcing, on the order of a few days to several decades. In our EM-GC framework, we are able to conduct calculations allowing the value of $\lambda_\Sigma$ to vary over time with a delay between the change in radiative forcing and the response of $\lambda_\Sigma$, and to project future temperature with such an assumption. Up until this point, our simulations have used time-invariant $\lambda_\Sigma$ to be consistent with how our model results had been presented in prior publications (Canty et al., 2013; Hope et al., 2017) as well as several other empirically-based approaches (Chylek et al., 2014; Lean and Rind, 2008, 2009; Zhou and Tung, 2013). Recall from Sect. 2.1 that $\lambda_\Sigma = \lambda_P - \lambda$. To assess the effect of time varying climate feedback on our projections of global warming, we examine the sensitivity in terms of $\lambda^{-1}$, because this quantity scales proportionally with $\Delta T$ and our use of the inverse $\lambda$ allows for direct comparison to the results of Dong et al (2020), Marvel et al. (2018), and Rugenstein et al. (2020).

Figure 14 shows the change in observed and modeled GMST for an EM-GC simulation training to the HadCRUT5 GMST record and using an AER RF time series with a value of AER RF$_{2011}$ = −0.9 W m$^{-2}$ under four assumptions for $\lambda^{-1}$, all for SSP2-4.5 (solid lines). First, the value of $\lambda^{-1}$ is constant over time (Figs. 14a, e). Second, the value of $\lambda^{-1}$ rises by 50% between 1850-2100 (Figs. 14b, f: further discussion of Fig. 14b and f will occur at the end of this section). The third assumption allows $\lambda^{-1}$ to vary over time while $\chi^2_{RECENT}$ is always less than or equal to two (Figs 14c, g). Fourth, $\lambda^{-1}$ varies over time while $\chi^2_{ATM}$ is always less than or equal to two (Figs. 14d, h). We also assume the new time series for $\lambda^{-1}$ maintains an average value over the observational record identical to the constant value of 0.64 °C W$^{-1}$ m$^2$. We use a lag of 32.5 years to represent the mean value of the slowest response of the climate system to a RF perturbation reported by Goodwin (2018), which is associated with clouds and spatial adjustments of SST (32.5 years is the average of 20 and 45 years, the minimum and maximum values of the slowest response given in his Table 1). Figure S23 is identical to Fig. 14, except for the use of no delay between the RF perturbations and the response of climate feedback. The use of the response delays shorter than 32.5 years will result in projections between those shown in Fig. S23 and Fig. 14.

In Figs. 14 and S23 we also analyze a RF scenario termed SSP2-4.5′ that serves as a doubled CO$_2$ scenario (dotted lines). For SSP2-4.5′, the RFs due to all GHGs other than CO$_2$ as well as tropospheric aerosols from the start of 2020 onwards are held constant at end of 2019 values. The only component of RF allowed to vary after the start of 2020 is CO$_2$. The RF of climate due to all GHGs and tropospheric aerosols for SSP2-4.5′ is identical to that in SSP2-4.5 from the start of the simulation until the end of 2019. Since the mixing ratio of CO$_2$ at the end of century is 566 ppm, the warming found at the end of century for SSP2-4.5′ serves as the transient response of $\Delta T$ to rising CO$_2$ in our model framework. The fact that projections of $\Delta T$ found allowing only for future increases in CO$_2$ (dotted lines) agree so closely with those found assuming changes in RF due to all GHGs and tropospheric aerosols (solid lines) means that under the AER RF$_{2011}$ = −0.9 W m$^{-2}$ scaling assumption, the future change in RF due to all agents other than CO$_2$ nearly cancel. Projections found using the original SSP2-4.5 scenario may serve as a useful surrogate for a double CO$_2$ simulation. Figures S24 and S25 are the same as Fig. 14, except for the use of AER RF$_{2011}$ values of −0.4 and −1.5 W m$^{-2}$, respectively. There are slight departures between the SSP2-4.5 and SSP2-4.5′ projections of $\Delta T$ for these alternate aerosol scaling assumptions. Nonetheless,

these projections are quite similar because the future decline in RF due to the assumption of declining $CH_4$ within SSP2-4.5 nearly balances the future increase in RF due to $N_2O$ and all of the minor GHGs.

We fit the climate record over the past 170 years ($\chi^2_{ATM}$) and past 80 years ($\chi^2_{RECENT}$) extremely well for constant $\lambda^{-1}$ (Fig. 14a, e). If we allow the value of $\lambda^{-1}$ to scale with anthropogenic forcing by as much as possible such that the maximum value of $\chi^2_{RECENT}$ is always less than or equal to two, we obtain

the result shown in Figs. 14c, f. This simulation results in an increase in $\lambda^{-1}$ by nearly a factor of two in 2100 and a value of $\Delta T_{2100}$ of 3.4°C, about 1.0°C higher than when a constant value of $\lambda^{-1}$ is used. If we allow the value of $\lambda^{-1}$ to scale with anthropogenic forcing as much as possible such that the maximum value of $\chi^2_{ATM}$ is less than or equal to two, we arrive at the result shown in Figs. 14d, h. This simulation yields a rise in $\lambda^{-1}$ over two and a half centuries by a factor of 2.9 and a value of $\Delta T_{2100}$ of 4.2°C that is

nearly double the estimate of $\Delta T_{2100}$ for the time invariant $\lambda^{-1}$ (Fig 14a). The modeled change in $\Delta T$ starts to deviate from observations around year 1960 and persistently disagrees with the data out to present time (Fig. 14h). This simulation results in a $\chi^2_{RECENT}$ value of 3.89, which indicates poor agreement with observations.

Several other studies have investigated the amount of change in $\lambda^{-1}$. Marvel et al. (2018) suggest

that the increase in the median value of ECS from the CMIP5 GCMs between the historical and abrupt $4\times CO_2$ simulations may be driven by an increase in $\lambda^{-1}$ of 28 to 72%. Rugenstein et al. (2020) estimates a median increase of 17% for values of ECS from CMIP5 GCMs when examining millennial length simulations compared to the 150-year Gregory et al. (2004) method, which is consistent with about an 11% rise in $\lambda^{-1}$ (Fig. 2b of Rugenstein et al. (2020)). An analysis by Dong et al. (2020) estimates a median

increase in $\lambda$ of +0.4 W m$^{-2}$ K$^{-1}$, which corresponds to a 50% increase in $\lambda^{-1}$ (Fig. 1c, d of Dong et al. (2020)). Consequently, a doubling (Fig. 14c) or almost tripling of $\lambda^{-1}$ (Fig. 14d) over two and a half centuries is larger than the increase indicated by Marvel et al. (2018) and Dong et al. (2020) and the millennia order timescale in Sect. 12.5.3 of IPCC 2013 and Rugenstein et al. (2020). An increase in $\lambda^{-1}$ of 50% or lower (Fig. 14b) is in line with the estimate of the change in ECS due to time-variant $\lambda^{-1}$

indicated by Dong et al. (2020), Marvel et al. (2018), and Rugenstein et al. (2020).

Allowing $\lambda^{-1}$ to increase over time introduces important, additional uncertainty to our estimate of ECS. We denote values of ECS found using time-variant $\lambda^{-1}$ as $ECS_{\lambda(t)}$. Our best estimate of $ECS_{\lambda(t)}$ is 3.08°C (range of 2.23 to 5.53°C), which is derived from model results shown in Fig. 14b (AER $RF_{2011}$ = −0.9 W m$^{-2}$), Fig. S24b (AER $RF_{2011}$ = −0.4 W m$^{-2}$), and Fig. S25c (AER $RF_{2011}$ = −1.5 W m$^{-2}$). For this new estimate of $ECS_{\lambda(t)}$, we allow $\lambda^{-1}$ to increase by 50% or rise as much as possible and still achieve a value of $\chi^2_{RECENT}$ less than or equal to two. Simulations with strong aerosol cooling (AER $RF_{2011}$ = −1.5 W m$^{-2}$) cannot have $\lambda^{-1}$ rise by 50% and maintain a good fit to the climate record over the past 80 years (Fig. S25b, f). Our best estimate of $ECS_{\lambda(t)}$ of 3.08°C (range of 2.23 to 5.53°C) for a 32.5-year delay is similar to the value of ECS reported by Goodwin (2018) for a 100-year response time (2.9°C; range of 2.3 to 3.6°C).

The assumption of constant feedback within the EM-GC framework used in the rest of the manuscript is reasonable because there is no strong evidence from the climate record for a noticeable increase in $\lambda^{-1}$ on the multidecadal time scale associated with the simulations in Fig. 14. Assuming climate feedback is constant over time results in the best fits to the climate record for both of our $\chi^2$ constraints. If the true value of $\lambda^{-1}$ actually rises over time as suggested by the CMIP6 (Dong et al., 2020) and CMIP5 GCMs (Marvel et al., 2018; Rugenstein et al., 2020), our projections of global warming would be a few tenths of a degree warmer than our current best estimates assuming constant $\lambda^{-1}$, as shown in Fig. 14b. If $\lambda^{-1}$ is allowed to increase by 50%, our best estimate of ECS would rise from 2.33 to 3.08°C, which is a 32% increase. Time variant $\lambda^{-1}$ introduces additional uncertainty into our estimates of ECS; however, the largest uncertainty is still due to the imprecise knowledge of the RF due to tropospheric aerosols.

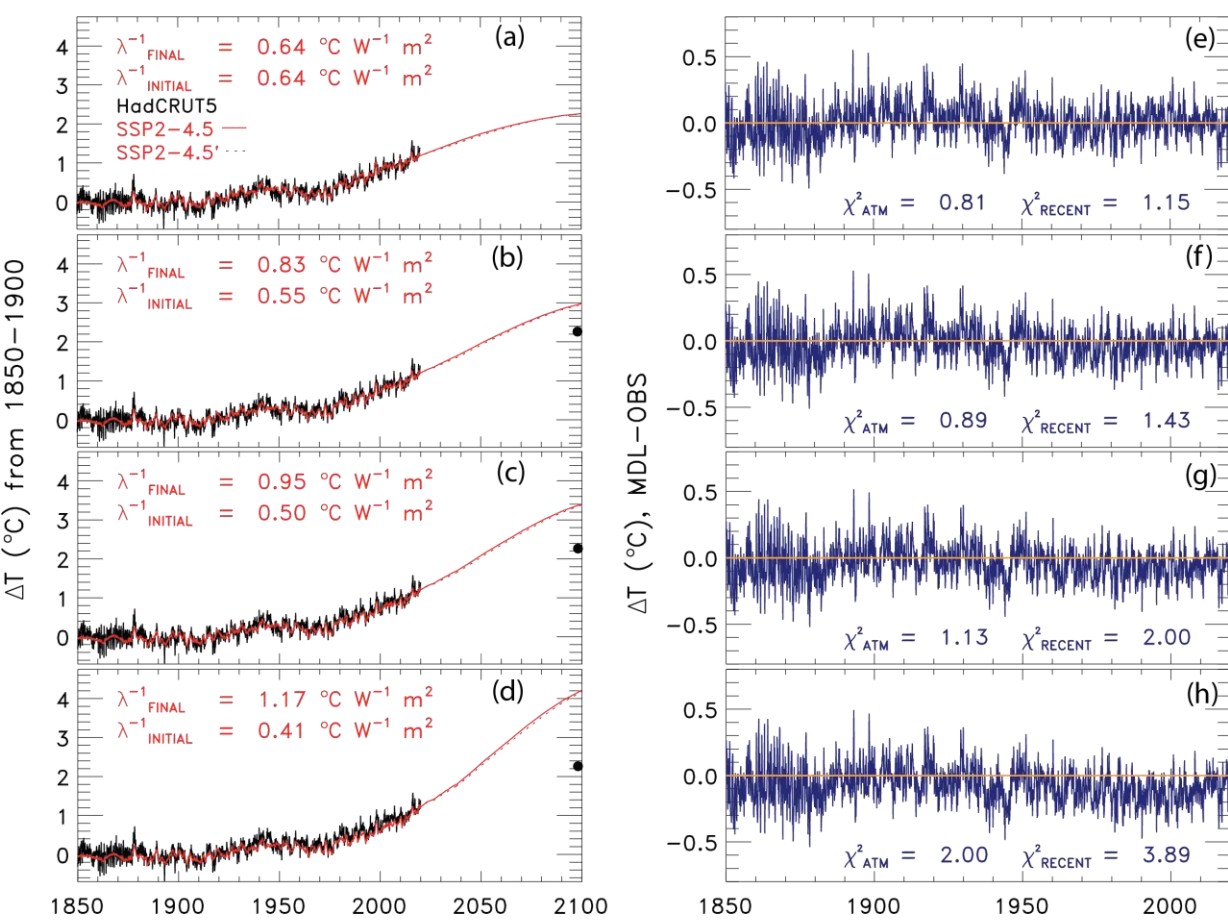

**Figure 14.** Change in GMST from 1850-2019 for observations from HadCRUT5 (black) and 1850-2100 for modeled (red) using SSP2-4.5 and a value of AER $RF_{2011}$ = −0.9 W m$^{-2}$ and the residual between modeled and observations incorporating a 32.5-year delay between $\lambda^{-1}$ and a change in RF. The solid line denotes a simulation for the original SSP2-4.5 scenario and the dashed line indicates the SSP2-4.5′ simulation (see text). (a) Rise in GMST assuming a constant value of $\lambda^{-1}$. (b) Rise in GMST allowing $\lambda^{-1}$ to increase by 50%. (c) Rise in GMST allowing $\lambda^{-1}$ to vary while the value of $\chi^2_{RECENT}$ is kept below 2. (d)  Rise in GMST allowing $\lambda^{-1}$ to vary while the value of $\chi^2_{ATM}$ is kept below 2. (e) Residual between modeled and observed rise in GMST from 1850-2019 for constant $\lambda^{-1}$. (f) Same as (e) but for increasing $\lambda^{-1}$ by 50%. (g) Same as (f) but for varying $\lambda^{-1}$ while the value of $\chi^2_{RECENT}$ is kept below 2. (h) same as (g) but for varying $\lambda^{-1}$ while the value of $\chi^2_{ATM}$ is kept below 2.

## 4 Conclusions

In this paper we use a multiple linear regression energy balance model (EM-GC), to analyze and project changes in the future rise in global mean surface temperature (GMST), calculate the attributable anthropogenic warming rate (AAWR, the component of the rise in GMST caused by human activities) over the past four decades, and compute the effective climate sensitivity (ECS, the rise in GMST that

would occur with atmospheric $CO_2$ at the 2×pre-industrial level assuming constant climate feedback). Projections of the rise in GMST (ΔT) are conducted for seven of the Shared Socioeconomic Pathway (SSP) projections of GHGs (O'Neill et al., 2017). We compare computations of AAWR, ECS, and projections of ΔT to values for each quantity computed from archived output provided by GCMs as part of CMIP6 (Eyring et al., 2016). A critical component of our study is comprehensive analysis of uncertainties in AAWR, ECS, and projections of ΔT in our EM-GC framework, due to the rather large uncertainty in radiative forcing of climate from tropospheric aerosols (AER RF).

The median value of AAWR from 1975-2014 computed using our EM-GC constrained by the century and a half long record for GMST provided by HadCRUT5 is 0.157°C decade$^{-1}$ and the 5th, and 95th percentiles are 0.120 and 0.195°C decade$^{-1}$, respectively. The median value of AAWR from the CMIP6 multi-model ensemble is 0.221°C decade$^{-1}$ and the 5th, and 95th percentiles are 0.151 and 0.299°C decade$^{-1}$, respectively. We show that the component of GMST attributed to human activity within the CMIP6 multi-model ensemble warms considerably faster than observations over the past four decades, a result that is consistent with recent analyses of output from the CMIP6 multi-model ensemble (CONSTRAIN, 2020; Tokarska et al., 2020b) as well as output from CMIP5 GCMs assessed in AR5 (i.e, Fig. 11.25b of Kirtman et al. (2013)). This finding differs from the conclusion of Hausfather et al. (2020), who showed fairly good agreement between projections of global warming from GCMs and observed ΔT. As detailed in Sect. 3.1, this paper examined GCMs that proceeded CMIP5 and examined ΔT for a time period that ends in 2017, a time when global temperature was influenced by a strong ENSO event that ended in 2016. The majority of the uncertainty in our EM-GC based estimate of AAWR is due to imprecise knowledge of the true value of AER RF.

In our model framework, the best estimate of ECS is 2.33°C and the 5th and 95th percentiles are 1.40 and 3.57°C, respectively. The median value of ECS from the CMIP6 multi-model ensemble is 3.74°C, which is around 1.6 times the best estimate value of ECS inferred from the observed climate record. The 5th and 95th percentiles of ECS from the CMIP6 multi-model ensemble are 2.19 and 5.65°C, respectively. We obtain a wide range of ECS values using the EM-GC because of the uncertainty in AER RF. With an AER RF$_{2011}$ equal to −1.6 W m$^{-2}$, the EM-GC calculates a value of ECS similar to the

maximum value of ECS from the CMIP6 multi-model mean. We cannot rule out the very high value of ECS, but we assign a low probability based on the IPCC 2013 low likelihood for the needed value of AER $RF_{2011}$. Our empirically based determination of ECS is in good overall agreement with the recent empirical determinations of Lewis and Grünwald (2018) (1.87°C, range of 1.1-4.05°C) and Skeie et al. (2018) (2.0°C, range of 1.2-3.1°C) and the slightly older empirically determination reported by Otto et al. (2013) (2.0°C, range of 1.2-3.9°C) (all range values are for the 5[th] and 95[th] percent confidence interval). A recent review of climate feedback and climate sensitivity published by Sherwood et al. (2020) reported ECS lies within the range of 2.3 to 4.7°C at the 5[th] to 95[th] percent confidence intervals; their lower bound for ECS is quite a bit higher than the lower bound found in our analysis, as well as by Cox et al. (2018), Dessler et al. (2018), Lewis and Grünwald (2018), Nijsse et al. (2020), Otto et al. (2013), Skeie et al. (2018), and Tokarska et al. (2020b). Our best estimate of ECS increases to 3.08°C (range of 2.23 to 5.53°C) if we allow climate feedback to rise over time, with the largest uncertainty in ECS still driven by the imprecise knowledge of the RF due to tropospheric aerosols.

We also examined the probability of limiting the future rise in GMST below the Paris Agreement target of 1.5°C and upper limit of 2.0°C. Our probabilistic forecasts of projections of ΔT include a comprehensive treatment of the uncertainty in AER RF, a capability outside the scope of the GCM intercomparisons conducted for CMIP6. Our analysis indicates that if GHGs were to follow the SSP1-2.6 pathway, there would be a 53% likelihood that the rise in ΔT would remain below the Paris Agreement target of 1.5°C (relative to pre-industrial) by the end of century based on HadCRUT5. We find that the SSP4-3.4 scenario provides a 64% likelihood of limiting global warming to below the Paris Agreement upper limit of 2.0°C by end of century. These probabilities have declined upon our use of HadCRUT5 compared to the GMST record of HadCRUT4 to 64% and 74% for the SSP1-2.6 and SSP4-3.4 scenarios, respectively. In contrast, the CMIP6 multi-model mean only suggests a 18% probability of achieving the Paris Agreement target for SSP1-2.6 and a 17% probability of attaining the Paris Agreement goal for SSP4-3.4. The lower probabilities suggested by the CMIP6 multi-model ensemble is not surprising, given the tendency of most CMIP6 GCMs to warm faster observations over the past four decades. Our projections of ΔT using a physically based model tied to observations of ocean heat content, quantification

of natural as well as anthropogenic drivers of variations in GMST, and consideration of uncertainty in AER RF are shown to be remarkably similar to the expert assessment of the future rise in GMST that was sketched out in Fig. 11.25b of AR5 (Kirtman et al., 2013), and the empirically-based filtering of CMIP6 model output recently published by Tokarska et al. (2020b). Finally and most importantly, our estimates are based on the assumption that climate feedback has been and will continue to remain constant over time, since the prior temperature record can be fit so well under this assumption. As described in Sect. 3.3.6, if climate feedback rises over time, larger warming will be realized than that found under this assumption of temporally invariant feedback.

We also quantify the sensitivity of the probability of achieving the Paris Agreement target (1.5°C) or upper limit (2.0°C) to future atmospheric abundances of methane. The end of century mixing ratio of methane in the SSP1-2.6 scenario is 1.15 ppm, considerably less than the contemporary abundance of 1.88 ppm. The likelihood of attaining the 1.5°C target for SSP1-2.6 decreases as future methane emissions increase, declines to 30% if methane reaches 2.4 ppm in 2100 and to 16% if methane increases to 3.2 ppm at end of century. Our analysis described in Sect. 3.3.5 demonstrates that major near-term limits on the future growth of methane are especially important for achievement of the 1.5°C limit to future warming that constitutes the goal of the Paris Agreement.

Finally, we have also quantified in the EM-GC framework the remaining budgets of carbon (i.e., $CO_2$) emissions that can occur while attaining either the goal or upper limit of the Paris Agreement. We find that after 2019, society can only emit another 150 ± 79 Gt C to have a 66% likelihood of limiting warming to 1.5°C. This future emissions estimate rises to 400 ± 104 Gt C to have a 66% probability of limiting warming to 2.0°C. Given the anthropogenic emissions of carbon due to combustion of fossil fuels, cement production, gas flaring, and land-use change were about 11.7 Gt C per year in 2019 (Friedlingstein et al., 2019), our study indicates that the target (1.5°C warming) of the Paris Agreement will not be achieved unless carbon emissions are severely curtailed in the next 10 years.

We conclude by noting that the CMIP6 multi-model ensemble provides many useful parameters such as sea level rise, sea ice decline, and precipitation changes, that provide a great societal understanding of the impact of climate change. We do not mean to undermine the importance of the CMIP6 GCMs by this analysis. Rather, we hope that studies such as this, along with other recent

evaluations of CMIP6 multi-model output such as Nijsse et al. (2020) and Tokarska et al. (2020b) will provide improved use of the CMIP6 multi-model ensemble for policy decisions. Our EM-GC was built to specifically simulate and project changes in GMST; we do not examine numerous other components of the climate system that affect society. We emphasize that our projections show that unless society can implement steep reductions in the emissions of carbon ($CO_2$) and methane ($CH_4$) in the next 10 years, 1.5°C global warming goal of the Paris Agreement will not be achieved.

## 5 Acronyms

AAWR – Attributable anthropogenic warming rate

AR4 – Fourth Assessment Report

AER – Anthropogenic aerosols

AER $RF_{2011}$ – Radiative forcing due to anthropogenic aerosols in 2011

AMOC – Atlantic meridional overturning circulation

AMV – Atlantic multidecadal variability

BEG – Berkley Earth Group

CALIPSO – Cloud-Aerosol Lidar and Infrared Pathfinder Satellite Observations

CMIP5 – Coupled Model Intercomparison Project Phase 5

CMIP6 – Coupled Model Intercomparison Project Phase 6

COBE - Centennial in situ Observation-Based Estimate

CW14 – Cowtan and Way (2014) temperature record

ECS – Effective climate sensitivity

$ECS_{\lambda(t)}$ – Effective climate sensitivity, time dependent feedback

EM-GC – Empirical Model of Global Climate

ENSO – El Niño Southern Oscillation

GCM – General Circulation Model

GHG – Greenhouse gas

GISTEMP – Goddard Institute for Space Studies Surface Temperature Analysis v4

GloSSAC – Global Space-based Stratospheric Aerosol Climatology

GMST – Global mean surface temperature

HadCRUT – Hadley Center Climatic Research Unit

IPCC – Intergovernmental Panel on Climate Change

ISCCP – International Satellite Cloud Climatology Project

IOD – Indian Ocean dipole

LIN – Linear method

LUC – Land-use change

MEI – Multivariate ENSO index

NOAAGT – National Center for Environmental Information NOAAGlobalTemp v5

ODS – Ozone depleting substances

OHC – Ocean heat content

OHE – Ocean heat export

PATMOS-X - Pathfinder Atmospheres Extended

PDO – Pacific decadal oscillation

RCP – Representative concentration pathway

REG – Regression method

RF – Radiative forcing

SAOD – Stratospheric aerosol optical depth

SORCE – Solar Radiation and Climate Experiment

SSP – Shared Socioeconomic Pathway

SST – Sea surface temperature

TAR – Third Assessment Report

TAS – Near surface air temperature

TCRE – Transient climate response to cumulative emissions

TOS – Temperature at the interface of the atmosphere and the upper boundary of the ocean

TSI – Total solar irradiance

**6 Data availability**

All data used as inputs into the EM-GC are available from resources on the web. We have provided the links to the resources below. The data are also available along with the EM-GC output used in this analysis at https://doi.org/10.5281/zenodo.4300780 (McBride et al., 2021) on Zenodo.org.

IOD: The COBE SST data is provided by the NOAA ESRL physical sciences division from their web site https://www.esrl.noaa.gov/psd/.

Tropospheric ozone RF: http://www.pik-potsdam.de/~mmalte/rcps/ .

MEI.v2 and MEI.ext: https://psl.noaa.gov/enso/mei/data/meiv2.data and https://psl.noaa.gov/enso/mei.ext/table.ext.html

PDO:  http://research.jisao.washington.edu/pdo/PDO.latest.txt

SAOD: https://eosweb.larc.nasa.gov/project/glossac/glossac

TSI: http://lasp.colorado.edu/home/sorce/data/tsi-data/

OHC Records:
    Balmaseda: http://www.cgd.ucar.edu/cas/catalog/ocean/OHC700m.tar.gz
    Carton: https://www.atmos.umd.edu/~ocean/index_files/soda3_readme.htm
    Cheng: http://159.226.119.60/cheng/
    Ishii: http://159.226.119.60/cheng/
    Levitus: https://www.nodc.noaa.gov/OC5/3M_HEAT_CONTENT/

SSP Database: All information for the SSPs obtained from the SSP database is at https://tntcat.iiasa.ac.at/SspDb/dsd?Action=htmlpage&page=about .

CMIP6 Input Data:
https://docs.google.com/document/d/1pU9IiJvPJwRvIgVaSDdJ4O0Jeorv_2ekEtted34K9cA/edit#heading=h.jdoykiw7tpen

CMIP6 Model Output Archive: https://esgf-node.llnl.gov/search/cmip6/

# 7 Author Contribution

LAM, APH, and TPC developed the model code used in this analysis, LAM, APH, and BFB collected data, RJS supervised, administrated, and developed the project, LAM wrote the original draft, and RJS, APH, BFB, TPC, and WRT participated in the review and editing of the manuscript.

## 8 Competing Interests

The authors declare that they have no conflict of interest.

## 9 Acknowledgements

We would like to acknowledge the World Climate Research Programme for coordinating and promoting CMIP6 through its Working Group on Coupled Modelling. We thank the climate modeling groups participating in CMIP6 for producing and making their model results available, the Earth System Grid Federation (ESGF) for archiving the data and providing access, and the several funding agencies who support ESGF and CMIP6. This project could not occur without the results from CMIP6. We appreciate very much financial support from the NASA Climate Indicators and Data Products for Future National Climate Assessments (INCA) program (award NNX16AG34G). This study was partially supported by NOAA grants NA14NES4320003 and NA19NES4320002 (Cooperative Institute for Satellite Earth System Studies - CISESS) at the University of Maryland/ESSIC. We thank University of Maryland Undergraduate Lauren Borgia for participating in extensive, in-depth discussions of recent papers on cloud feedback and climate sensitivity. Finally, we thank both reviewers for very careful reads of the original paper that led to substantial improvements in the manuscript, as well as Martin Stolpe for contacting us privately, while the paper was in discussion, regarding an erroneous description of the effect of creating blended near surface air temperature that had appeared in the submitted paper.

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
