# Peer review of "Comparison of CMIP6 Historical Climate Simulations and Future Projected Warming to an Empirical Model of Global Climate"

_Earth System Dynamics, 2020_

## Referee Comment (RC1) · Anonymous Referee #1 · 30 Sep 2020

McBride etal presents a physics informed statistical model of temperature change induced by anthropogenic and natural forcing. They use a comprehensive set of observational datasets to inform the nine parameters of their model. They conclude that the observed attributable anthropogenic warming rate, and climate sensitivity, is not only significantly lower than the model mean of CMIP6, but also, for climate sensitivity, significantly lower than our recent assessment of multiple lines of evidence on climate sensitivity.

I believe the stark contrast between the current study and the recent assessment by Sherwood et al. (2020) is partially an artefact of data choices, most significantly the

choice of the HadCRUT dataset for global temperature, which has poor coverage of quickly rising Artic temperatures. The inclusion of older estimates of ocean heat uptake have a secondary effect. I think that the paper will be a useful contribution to debate around climate sensitivity after major revisions.

Major comments:

1) The HadCRUT dataset is known to underestimate recent warming. The model mean is computed without compensating for the fact that Artic temperatures are underrepresented, leading to an underestimate of warming (Cowtan and Way, 2014, Cowtan, 2017). Even with corrected sea surface temperature, this still leads to under-reported warming compared to a record with global coverage (Cowtan, 2017). The HadCRUT data should be replaced by the Cowtan record.

2) According to the IPCC's SROCC report, older estimates of ocean heat uptake have biases that may lead to an underestimate of ocean heat uptake (Bindoff, 2019, p.457). Carton et al (2018), whose record was derived with data assimilation, indicated that previous estimates with data assimilation (possibly like Balmaseda's record), may have contained errors that have prevented them from being sufficiently. Similarly, Cheng et al (2017) and Ishii et al (2017) can be considered superior to the old standard of Levitus (2012).

3) Armour (2017) showed that climate sensitivity estimates from energy budgets can be reconciled with climate models by treating models as observations: if you estimate climate sensitivity of models using only data up to the present, your climate model sensitivity will be underestimated. The reason is that most models show increased sensitivity over time. The climate feedback parameter in McBride et al is assumed to be time-constant without justification. On timescales longer than the 150 years of the Gregory method, positive feedbacks are set to increase even further (Rugenstein, 2020).

4) Considering the previous comment, I would like to be convinced the simple method

can be used to estimate climate sensitivity when applied to climate models. Is this method able to give accurate predictions of climate sensitivity of climate models in contrast to previous energy balance methods?

5) I could not quite understand the computations behind the TCRE: how are uncertainties in the carbon cycle taken into account? This is important for the 66% and 95% likelihood estimates.

6) The paper is quite long and I think that it will become more convincing after a good look at the prose. My minor comments will give further suggestions.

Citations:

Armour, K. Energy budget constraints on climate sensitivity in light of inconstant climate feedbacks. Nature Clim Change 7, 331–335 (2017). https://doi.org/10.1038/nclimate3278

Cowtan, K. and Way, R.G. (2014), Coverage bias in the HadCRUT4 temperature series and its impact on recent temperature trends. Q.J.R. Meteorol. Soc., 140: 1935-1944. doi:10.1002/qj.2297

Coverage bias in the HadCRUT4 temperature series and its impact on recent temperature trends UPDATE (2017) https://www-users.york.ac.uk/∼kdc3/papers/coverage2013/update.171107.pdf

Bindoff, N.L., W.W.L. Cheung, J.G. Kairo, J. Arístegui, V.A. Guinder, R. Hallberg, N. Hilmi, N. Jiao, M.S. Karim, L. Levin, S. O'Donoghue, S.R. Purca Cuicapusa, B. Rinkevich, T. Suga, A. Tagliabue, and P. Williamson, 2019: Changing Ocean, Marine Ecosystems, and Dependent Communities. In: IPCC Special Report on the Ocean and Cryosphere in a Changing Climate [H.-O. Pörtner, D.C. Roberts, V. Masson-Delmotte, P. Zhai, M. Tignor, E. Poloczanska, K. Mintenbeck, A. Alegría, M. Nicolai, A. Okem, J. Petzold, B. Rama, N.M. Weyer (eds.)]. In press

Rugenstein, M., Bloch‐Johnson, J., Gregory, J., Andrews, T., Mauritsen,

T., Li, C., et al. (2020). Equilibrium climate sensitivity estimated by equilibrating climate models. Geophysical Research Letters, 47, e2019GL083898. https://doi.org/10.1029/2019GL083898

Minor comments: 76. Remove the word "active", as that implies a dynamic ocean, which is not what the model has

82. The paper uses many capitalised abbreviations, which is inevitable. However, words like months and obs can be written in lowercase to make reading more pleasant.

101. Maybe repeat what delta TMDL means

132. I'm not sure whether your definition of lambda can also be called a climate feedback parameter. It would be confusing to have two different parameters with the same name

136. This sentence can be removed, as it's not providing any information relevant to the study.

188. Normally the reduced chi-squared parameter is denoted $\chi\nu2$ to differentiate from normal chi-squared.

Figure 1: The AAWR in panel b is different from the lead, which one is correct?

239-243. In this study, the datasets are referred to by the name of the institutions responsible for them, but they have specific names. Could you replace CRU with Had-CRUT, GISS with GISTEMP and so forth.

242: Typo. Berkley=Berkeley

247. Transformation usually means adjusting the mean and variance, where you're only adjusting the mean

257. The baseline is defined as no mitigation, so this sentence would be corrected if you remove that word

263-269. Description of the tiers is unnecessary for this study, consider dropping it.

272. Add "the" ("in the supplement")

279-281. Which equation comes from which source?

289. Remove brackets around Myhre

295. Upon -> on?

320. Are you sure it's not perfectly identical?

323. Remove "described above", it's unnecessary

363-369. If I understand it correctly, three different time series are appended. Would it not be easier to derive the entire time series yourself? That would also be more easy to describe.

415-416. Normalization involves both the mean and standard deviation, offsets are always additive. Maybe rewrite as "the five datasets are all set to zero in 1986 by applying an offset"

433. I didn't understand which standard deviation of the mean was taken

463. Replace "based upon" by "using", remove "shown below"?

477-480. I didn't understand why AAWR is not affected at all, as regression variables, such as lambda, are surely influenced by the inclusion of AMOC.

519. if I understand it correctly, these equations assume there is no uncertainty at all in the radiative forcing at the doubling of CO2, which is inconsistent with definitions of radiative forcing and with CMIP6 models.

550-552. I did not understand what an asymmetric Gaussian was, could you explain?

649. Remove "as indicated"

675. The value of 1.85 contradicts the value in the next paragraph of 2.01. Which one

is correct?

Figure 8: This figure only uses studies with low climate sensitivity and compares them to assessments of climate sensitivity (Sherwood/IPCC). Either explain the selection criteria, or add some studies to make this figure more balanced (the carbon brief provides an excellent overview: https://www.carbonbrief.org/explainer-how-scientists-estimate-climate-sensitivity)

689. The word "yet" implies a contradiction. However, with the very wide uncertainty specified by the IPCC, these probably overheating models are still within range.

698. rm the word "actual"?

704-705. Consider deleting "ninety-five…multi-model ensemble" as I think it is an unnecessary detail. Presenter 713. Remove "then"?

739. Remove information between brackets, repetition of information within paragraph.

782. Bifurcation has a specific meaning within mathematics, consider replacing by bimodality. If more models are added, check whether it's still true.

Figure 10: Use different colour scheme. The rainbow colour scheme has false perceptual thresholds or hides real ones: https://www.nature.com/articles/519291d.

811. Replace "will" with "is set", we don't know the future.

931. Replace "since" with "after".

669. Insert dioxide after carbon

1002. Unnecessary to show all these percentiles, remove 25 and 75.

1009. 2017 was not an El Niño-year and non-El Ninõ-years 2018 and 2019 were comparable in temperature.

1012. Similar, summarise, so do not show all percentiles

1056. Similar, summarise.

1061. Replace "will" with "will not"

1071. Repetition of the information in 1061

1073. What is a literal interpretation? The model democracy interpretation?

1074. Modeling is not the only source of information on warming of 1.5 degrees, many studies extrapolate current trends.

Figure S1: Replace the rainbow colour scheme.

Figure S7: Caption should indicate that it's the unweighted one.

---

## Referee Comment (RC2) · Anonymous Referee #2 · 10 Oct 2020

This manuscript used a multiple linear regression energy balance model, EM-GC, to estimate the attributable anthropogenic warming rate (AAWR), the equilibrium climate sensitivity (ECS), and the future projections. The authors compared the results from EM-GC with those obtained from CMIP6. They found that the CMIP6 GCMs tend to exhibit a faster rate of warming, which induced larger AAWR, larger ECS, and smaller remaining budgets of carbon emissions. One highlight of this work is the use of Aerosol Weighting Method, which allowed a probabilistic estimation. This work is very interesting and the authors have done many detailed analyses. However, before I can recommend accepting this manuscript, I have several concerns that need to be addressed.

[Figure]

1. To run the EM-GC model, it seems that one needs to determine nine regression coefficients and parameters. Constrained by the observed GMST and the OHC, one can obtain a set of the nine coefficients/parameters to ensure a good fit to the historical observations. However, I am not sure if the selected set of coefficients/parameters is unique, or one can use a totally different set of coefficients/parameters to achieve a similar fitting skill? I also have concerns that whether the coefficients/parameters are still useful for the future projections? I would like to suggest the author to perform a test to prove the validity of the model and the stability of the coefficients/parameters. For example, the authors may consider to divide the historical period into two halves, use the first half to determine the coefficients/parameters, and use the second half to test the stability.

2. From Fig. 1f, the authors found that the PDO has very limited contributions to the GMST. I don't understand this finding, as to my knowledge, the different phases of the PDO play an important role in modulating the GMST. For example, the recently well discussed warming hiatus in the beginning of this century has been found to be closely related to the PDO. An explanation about the findings in Fig. 1f is needed.

3. Another concern is about the comparison of the AAWR that obtained from EM-GC and CMIP6 models. Since different methods are used to calculate the AAWR, I am not sure if the results are comparable. Especially for the CMIP6 models, the REG method seems to be too simple to calculate the AAWR. I am not sure if the AAWR values obtained from the CMIP6 models are as pure as those obtained from EM-GC.

4. In line 228, ". . .also specified on Fig. 1f", "Fig. 1f" should be "Fig. 1e".

5. In line 975, "then" should be "than".

6. In line 1061, ". . .of the Paris Agreement will be achieved", "will be" should be "will not be".

---

## Author Response (AR1)

Reviewer comments are in black and our reply is in blue.

McBride et al. present a physics informed statistical model of temperature change induced by anthropogenic and natural forcing. They use a comprehensive set of observational datasets to inform the nine parameters of their model. They conclude that the observed attributable anthropogenic warming rate, and climate sensitivity, is not only significantly lower than the model mean of CMIP6, but also, for climate sensitivity, significantly lower than our recent assessment of multiple lines of evidence on climate sensitivity.

I believe the stark contrast between the current study and the recent assessment by Sherwood et al. (2020) is partially an artefact of data choices, most significantly the choice of the HadCRUT dataset for global temperature, which has poor coverage of quickly rising Arctic temperatures. The inclusion of older estimates of ocean heat uptake has a secondary effect. I think that the paper will be a useful contribution to debate around climate sensitivity after major revisions.

> We thank the reviewer for taking the time to carefully read our manuscript and suggest useful changes. Upon revision, we will make changes to address all of these comments, as detailed below each comment.

1) The HadCRUT dataset is known to underestimate recent warming. The model mean is computed without compensating for the fact that Artic temperatures are underrepresented, leading to an underestimate of warming (Cowtan and Way, 2014, Cowtan, 2017). Even with corrected sea surface temperature, this still leads to under-reported warming compared to a record with global coverage (Cowtan, 2017). The HadCRUT data should be replaced by the Cowtan record.

> We will address comment 1 and comment 2 together just below, since they are both related to the data records used in the EM-GC framework.

2) According to the IPCC's SROCC report, older estimates of ocean heat uptake have biases that may lead to an underestimate of ocean heat uptake (Bindoff, 2019, p.457). Carton et al (2018), whose record was derived with data assimilation, indicated that previous estimates with data assimilation (possibly like Balmaseda's record), may have contained errors that have prevented them from being sufficiently. Similarly, Cheng et al (2017) and Ishii et al (2017) can be considered superior to the old standard of Levitus (2012).

> We plan, upon revision, to show model results for the use of Cowtan and Way (2014) (hereafter CW14) data record for global mean near-surface temperature together with the Cheng et al. (2017) record for ocean heat content (OHC).

> When we ran the EM-GC using the CW14 temperature record, one issue that arose is the relatively small published uncertainties between 1850-1900 associated with this record. The EM-GC was not able to calculate any good fits as defined by the computation of a value of the reduced chi-squared parameter for GMST, $\chi^2_{ATM,}$ being less than 2 upon using the CW14 temperature time series and the published uncertainties. The New Fig. S4 below compares the uncertainties associated with the Hadley Centre Climate Research Unit (HadCRUT), Berkeley Earth Group (BEG), and CW14.

[Figure]

**New Figure S4.** GMST anomaly relative to pre-industrial over time. (a) HadCRUT with the HadCRUT uncertainties. (b) CW14 with the CW14 uncertainties. (c) BEG with the BEG uncertainties. (d) CW14 with the HadCRUT uncertainties.

The uncertainties for CW14 (New Fig. S4b) are much smaller than those for the HadCRUT (New Fig. S4a) and BEG (New Fig. S4c) temperature records. The small values of CW14 uncertainties, especially from 1850-1900, cause the EM-GC to not be able to achieve good fits to this temperature record. We have two choices for use of the CW14 record; either relax the constraint for $\chi^2_{ATM}$ (i.e., run with $\chi^2_{ATM} \leq 4$), or modify the CW14 uncertainties.

Upon revision, we propose to show model results for which we combine the uncertainties from HadCRUT with the data values for GMST from the CW14 record (New Fig. S4d), since CW14 is based upon the HadCRUT temperature record. Upon use of this combination of data and uncertainty, we are able to find good fits to the CW14 temperature record that look reasonable.

When the CW14 temperature record is used instead of HadCRUT, we see modest changes in the climate feedback parameter, $\lambda_\Sigma$, the attributable anthropogenic warming rate, AAWR, equilibrium climate sensitivity, ECS, and our projected global mean surface temperature (GMST) in 2100, $\Delta T_{2100}$, as detailed below. The New Fig. S3 below shows a plot similar to Fig. 1 of the submitted paper, for which we use the CW14 temperature record along with the Cheng et al. (2017) record for OHC.

[Figure]

**New Figure S3.** Measured and modeled GMST anomaly ($\Delta T$) relative to a pre-industrial (1850-1900) baseline. (a) Observed (black) and modeled (red) $\Delta T$ from 1850-2019. This panel also displays the values of $\lambda_\Sigma$ and $\chi^2_{ATM}$ (see text) for this best-fit simulation. (b) Contributions from total human activity. This panel also denotes the numerical value of the attributable anthropogenic warming rate from 1975-2014 (black dashed) as well as the $2\sigma$ uncertainty in the slope. (c) Solar irradiance (light blue) and major volcanoes (purple). (d) Influences from ENSO on $\Delta T$. (e) Contributions from AMOC to $\Delta T$ and to observed warming from 1975-2014. (f) Influences from PDO (blue) and IOD (pink) on $\Delta T$. (g) Measured (black) and modeled (red) ocean heat content (OHC) as a function of time for the Cheng 2017 OHC record, the value of $\chi^2_{OCEAN}$ for this run, as well as the ocean heat uptake efficiency, $\kappa$, needed to provide the best-fit to the OHC record. The error bars (blue) denote the uncertainty in OHC used in this analysis (see Sect. 2.2.8).

Our estimate of $\lambda_\Sigma$ upon use of the best estimate of aerosol radiative forcing in 2011 (AER RF$_{2011}$ of $-0.9$ W m$^{-2}$) increases from 1.38 W m$^{-2}$ °C$^{-1}$ (submitted paper) to 1.51 W m$^{-2}$ °C$^{-1}$ (New Fig S3) upon use of CW14 for GMST and the Cheng et al. (2017) record for OHC. The estimate of the value of AAWR increases from 0.144 °C/decade to 0.153°C/decade. This sensitivity of $\lambda_\Sigma$ and AAWR to data choice for GMST and OHC will be highlighted in the revised paper. We propose to add New Figure S3 to the supplement to document the effect on $\lambda_\Sigma$ of data choice, and to add model results for the CW14/Cheng et al. (2017) data combination to three additional figures in the Main paper, as detailed below.

Proposed update to Fig. 6 contains the estimates of AAWR for the four original GMST data records and the CW14 record. The use of the CW14 GMST results in slightly higher values of AAWR than HadCRUT, more in line with the values of AAWR from BEG. This result shows that utilizing a global temperature record does result in a modest increase in AAWR.

[Figure]

**Proposed update to Figure 6.** AAWR from the EM-GC and CMIP6 multi-model ensemble for 1975-2014. Five temperature data sets and five ocean heat content records are used to compare values of AAWR computed from the EM-GC. The box represents the 25th, 50th, and 75th percentiles, the whiskers denote the 5th and 95th percentiles, and the stars show the minimum and maximum values of AAWR from the EM-GC based upon the aerosol weighting method described in Sect. 2.5. The red box labeled "CMIP6" shows the 25th, 50th, and 75th percentiles, the whiskers represent the 5th and 95th percentiles, and the stars denote the minimum and maximum values of AAWR from the 50 member CMIP6 multi-model ensemble.

Proposed update to Fig. 7 contains the values of ECS utilizing the EM-GC for the four original GMST data sets as well as the CW14 record. The estimates of ECS using the CW14 GMST data set are larger than the estimates of ECS using the HadCRUT record. Similar to the values of AAWR, the values of ECS using the CW14 record are more in line with the BEG record than the HadCRUT record.

[Figure]

**Proposed update to Figure 7.** ECS from the EM-GC and the CMIP6 multi-model ensemble. Five GMST data sets and five ocean heat content records are used to compare values of ECS computed from the EM-GC. The box represents the $25^{th}$, $50^{th}$, and $75^{th}$ percentiles, the whiskers denote the $5^{th}$ and $95^{th}$ percentiles, and the stars indicate the minimum and maximum values of ECS using the EM-GC based upon the weighting method described in Sect. 2.5. The circles denote the value of ECS associated with the best estimate of AER $RF_{2011}$ of $-0.9$ W m$^{-2}$. The red box labeled "CMIP6" represents the $25^{th}$, $50^{th}$, and $75^{th}$ percentiles, the whiskers denote the $5^{th}$ and $95^{th}$ percentiles, and the stars indicate the minimum and maximum values of ECS from the 28 member CMIP6 multi-model ensemble.

Proposed update to Fig. 12 contains the probability density function (PDF) for estimates of $\Delta T_{2100}$ using both the HadCRUT/Average OHC record and the CW14/Cheng record. The use of the CW14/Cheng record results in a PDF that is shifted towards higher values of $\Delta T_{2100}$, which results in lower probabilities of achieving the Paris Agreement target (1.5°C) and upper limit (2.0°C), as shown in Proposed Update to Table 1.

The addition of the CW14 GMST data set combined with the Cheng 2017 OHC record makes our results more robust and shows the modest changes in AAWR, ECS, and $\Delta T_{2100}$ that occur upon changing the GMST record.

To address the second comment made by the reviewer, current Fig. S4 shows the five OHC records we use in our analysis normalized to 1986. This figure shows how the various OHC records relate to each other. All of the records follow the same overall trend, of increasing OHC from the beginning of the data record to the end. We would like to include all five data records for completeness. Our Proposed update to Fig. 6 and Fig. 7 illustrates that the choice of OHC record does have a slight impact on the values of AAWR and ECS but are not as important as the uncertainty in AER $RF_{2011}$. Upon revision, we include the combination of the CW14 GMST data set and the Cheng OHC record to show the effect of the OHC record that results in one of our highest kappa values, which leads to the most warming of all the OHC data records.

[Figure]

**Proposed update to Figure 12.** Probability density functions (PDF) for $\Delta T_{2100}$ found using the EM-GC with the HadCRUT temperature record (dark blue), the EM-GC with the CW14 temperature record (light blue) and Cheng 2017 OHC data set, and CMIP6 multi-model results. (a) PDF for EM-GC (blue) results and CMIP6 multi-model (red) results for SSP1-1.9. The left-hand y-axis is for EM-GC probabilities and the righthand y-axis is for the CMIP6 multi-model ensemble probabilities. (b) PDF for SSP1-2.6. (c) PDF for SSP4-3.4. (d) PDF for SSP2-4.5.

**Proposed Update to Table 1.** List of SSP scenarios analyzed in this study and the probabilities of achieving the Paris Agreement target or upper limit based on the EM-GC using the HadCRUT4 temperature record and average of the five OHC records and the CMIP6 multi-model ensemble. The second half of the table shows the probabilities of achieving the Paris Agreement target or upper limit based on the EM-GC using the CW14 temperature record and Cheng 2017 OHC record. The probabilities using the EM-GC are computed using the aerosol weighting method. The probabilities using the CMIP6 models are computed by calculating how many of the models for that scenario are below the temperature limits compared to the total number of models.

| | Probability of Staying at or Below 1.5°C | | Probability of Staying at or Below 2.0°C | |
|---|---|---|---|---|
| | **EM-GC** | **CMIP6** | **EM-GC** | **CMIP6** |
| **SSP1-1.9** | 84.1% | 50.0% | 96.7% | 80.0% |
| **SSP1-2.6** | 64.8% | 15.2% | 88.4% | 51.5% |
| **SSP4-3.4** | 37.6% | 0.0% | 74.0% | 16.7% |
| **SSP2-4.5** | 10.5% | 0.0% | 53.1% | 3.1% |
| **SSP4-6.0** | 0.6% | 0.0% | 26.6% | 0.0% |
| **SSP3-7.0** | 0.0% | 0.0% | 1.3% | 0.0% |
| **SSP5-8.5** | 0.0% | 0.0% | 0.0% | 0.0% |
| **Using CW14 and Cheng OHC Record** | | | | |
| **SSP1-1.9** | 82.4% | | 97.5% | |

| | | |
|---|---|---|
| **SSP1-2.6** | 57.0% | 85.5% |
| **SSP4-3.4** | 28.1% | 69.6% |
| **SSP2-4.5** | 4.2% | 43.2% |
| **SSP4-6.0** | 0.0% | 17.4% |
| **SSP3-7.0** | 0.0% | 0.0% |
| **SSP5-8.5** | 0.0% | 0.0% |

We conclude this section by addressing the statement made by the reviewer, in the second (introductory) paragraph, that the stark contrast between our results and the recent assessment by Sherwood et al. (2020) may partially be an artefact of our data choices. The figure below, Response Fig. 1, shows the probability density function (PDF) for the rise in GMST in year 2100 ($\Delta T_{2100}$) found using our model trained by GMST from HadCRUT (dark blue) and CW14 and OHC from Cheng et al. (2017) (light blue), and from the CMIP6 GCMs (red lines). All three of these lines are for SSP2-4.5. We also show a PDF for RCP-4.5 from Fig. 23 of Sherwood et al. (2020) (brown). Since projections found using our EM-GC are based on SSP2-4.5 and the Sherwood projection is based on RCP4.5, the comparison is not exactly "like to like", but all of these $\Delta T_{2100}$ projections are for GHG scenarios designated to reach a rise of about 4.5 W m$^{-2}$ by end of century (relative to pre-industrial). Note the large region of overlap between our projections of $\Delta T_{2100}$ and those of Sherwood et al. shown in this figure. The results in Response Fig. 1 show that differences between our projections of $\Delta T_{2100}$ and those from Sherwood et al. (2020) are not, in fact, ***primarily*** due to our use of data for GMST from HadCRUT rather than CW14. However, our use of data from CW14 does move our projections closer to those of Sherwood et al. (2020), so the reviewer is correct that data choice does in fact contribute ***partially*** to this difference. Projections of $\Delta T_{2100}$ found using our approach, and those given by Sherwood et al. (2020), both fall far short of the CMIP6 projections. The Sherwood et al. (2020) analysis lies in between our projections and those of the CMIP6 GCMs because, as noted starting on line 721 of our submitted paper, "they rely on a determination that the feedback due to clouds is moderately to strongly positive …". Text on lines 721 to 740 of our submitted paper provides further explanation of the fundamental difference between results found using our approach and that of Sherwood et al., which is primarily due to uncertainty in cloud feedback.

We will be happy to include Response Fig. 1 as a figure in the supplement of our revised paper and add the text in the preceding paragraph to the paper (Main and Supplement) to highlight the comparison between our results and those from Sherwood et al. (2020), if so directed either by the reviewer or editor.

[Figure]

**Response Figure 1.** Probability density functions (PDF) for $\Delta T_{2100}$ found using the EM-GC with the HadCRUT temperature record (dark blue), the EM-GC with the CW14 temperature record (light blue), the CMIP6 multi-model results (red), and results for RCP4.5 from Fig. 23 from Sherwood et al. (2020).

3) Armour (2017) showed that climate sensitivity estimates from energy budgets can be reconciled with climate models by treating models as observations: if you estimate climate sensitivity of models using only data up to the present, your climate model sensitivity will be underestimated. The reason is that most models show increased sensitivity over time. The climate feedback parameter in McBride et al is assumed to be time-constant without justification. On timescales longer than the 150 years of the Gregory method, positive feedbacks are set to increase even further (Rugenstein, 2020).

Excellent point that we plan to address upon revision, as detailed below.

The assumption of the time-constant $\lambda_\Sigma$ is the simplest assumption one can make. The fact the climate record can be fit so well based upon this conjecture does provide support for the validity of this assumption. Nonetheless, as stated by the reviewer, many GCMs indicate that climate feedback varies over time. In our EM-GC framework, we are able to conduct calculations allowing the value of $\lambda_\Sigma$ to vary over time, and to project future temperature with such an assumption. We therefore propose to add a new figure, which would be New Fig. 15, to address this concern raised and to add a new Section 3.3.6 where we describe model results for both constant climate feedback, and time varying climate feedback. In this new section, we will assess the effect of time varying climate feedback on our projections of global warming in terms of $\lambda^{-1}$, because this quantity scales proportionally with $\Delta T$ and the use of the inverse $\lambda$ allows for direct comparison to other studies. Recall from Sect. 2.1 that $\lambda_\Sigma = \lambda_P - \lambda$.

Our proposed New Fig. 15 shows the change in observed and modeled GMST under several assumptions. The first assumption is that the value of $\lambda^{-1}$ is constant over time. Of course, as noted in the submitted paper, we are able to fit the climate record over the past 170 years ($\chi^2_{ATM}$) and past 80 years ($\chi^2_{RECENT}$) quite well using this assumption. If we allow the value of $\lambda^{-1}$ to rise over time so that the value of $\chi^2_{ATM}$ is always less than or equal to 2, we obtain the result shown in New Fig. 15b. The modeled change in GMST starts to deviate from the observations around year 2000. This deviation is seen in the residual between modeled and observed GMST in New Fig. 15f. If we allow the value of $\lambda^{-1}$ to vary over time so that the value of $\chi^2_{RECENT}$ is less than or equal to 2, we get the result shown in New Fig. 15c. The modeled change in GMST starts to deviate dramatically from observations around year 1990. This stark deviation is seen in the residual between modeled and observed GMST in New Fig. 15g. The $\chi^2_{ATM}$ value in New Fig. 15g is 3.63, which does not satisfy our reduced chi-squared constraints, and interestingly appears to resemble the behavior of some CMIP6 GCMs. New Figure 15d has $\lambda^{-1}$ vary by 50% over two and a half centuries, which is comparable to estimates from Marvel et al. (2018) and Rugenstein et al. (2020). Upon revision, we would show New Fig. 15 and highlight the sensitivity of our projections of the rise in GMST by year 2100, for the SSP4-3.4 scenario, to the assumption of whether or not climate feedback is constant over time, as well as compare to estimates of $\lambda^{-1}$ from previous studies. Appropriate words would be added to the abstract and conclusions, such that the reader will be well aware that our baseline projections are based on time-invariant climate feedback, and that if the true climate feedback actually rises over time as suggested by some of the CMIP6 GCMs, our projections of global warming would be strongly affected.

[Figure]

**New Figure 15.** Change in GMST from 1850-2019 for observations from HadCRUT (black) and 1850-2100 for modeled (red) using SSP4-3.4 and the residual between modeled and observations. (a) Rise in GMST assuming a constant value of $\lambda^{-1}$. (b) Rise in GMST allowing $\lambda^{-1}$ to vary while the value of $\chi^2_{ATM}$ is kept below 2. (c) Rise in GMST allowing $\lambda^{-1}$ to vary while the value of $\chi^2_{RECENT}$ is kept below 2. (d) Rise in GMST allowing $\lambda^{-1}$ to increase by 50%. (e) Residual between modeled and observed rise in GMST from 1850-2019 for constant $\lambda^{-1}$. (f) Same as (e) but for varying $\lambda^{-1}$ while the value of $\chi^2_{ATM}$ is kept below 2. (g) Same as (f) but for varying $\lambda^{-1}$ while the value of $\chi^2_{RECENT}$ is kept below 2. (h) same as (g) but for increasing $\lambda^{-1}$ by 50%.

To address the reviewer's second point, we utilize the Gregory et al. (2004) method to calculate ECS from the CMIP6 GCMs because this procedure is preferred by Eyring et al. (2016) for the use in CMIP6. Our use of this method results in our calculated values of ECS being consistent with what GCM groups utilize for ECS, as has been published in numerous recent papers and will almost certainly be shown in the upcoming IPCC report. We agree with the reviewer that the Gregory et al. (2004) method may underestimate ECS, as shown in Rugenstein et al. (2020). Nonetheless, our paper will have greater value to the community if we use calculations for ECS that are consistent with the primary method employed by CMIP6. Upon revision, we will add more clarification to Sect. 2.4 about why we have chosen to use the Gregory et al. (2004) method, and we will note that this method may underestimate the value of ECS as suggested by the reviewer, with a citation to the Rugenstein et al. (2020) paper.

4) Considering the previous comment, I would like to be convinced the simple method can be used to estimate climate sensitivity when applied to climate models. Is this method able to give accurate predictions of climate sensitivity of climate models in contrast to previous energy balance methods?

Thanks for another excellent point that we plan to address upon revision, as detailed below.

Based on this comment, we have for the first time used our approach in the EM-GC framework to calculate ECS for the CMIP6 models. To use the EM-GC with the CMIP6 output, we calculated the CMIP6 multi-model mean change in GMST from 1850-2100 using the SSP2-4.5 scenario, because it reaches a doubling of preindustrial $CO_2$ by the end of the century. We used the standard deviation of the CMIP6-multi model mean to represent the uncertainty in the rise in GMST for our reduced chi-squared calculations. We trained the EM-GC from 1850-2100, included the CMIP6 prescribed values of SAOD and TSI, and did not include any natural variability. Our results are shown below in the proposed New Figure S13.

[Figure]

**New Figure S13.** GMST anomaly in 2100 relative to pre-industrial ($\Delta T_{2100}$) as a function of climate feedback parameter and AER RF$_{2011}$ and the values of ECS from the CMIP6 GCMs using three methods. (a) $\Delta T_{2100}$ for SSP2-4.5 using the CMIP6 multi-model mean and the average of the five OHC records. (b) $\Delta T_{2100}$ for SSP2-4.5 using the CMIP6 multi-model mean and the Cheng 2017 OHC record. (c) Values of ECS found using the Gregory et al. (2004) method (red), CMIP6 multi-model mean using the Cheng 2017 OHC (orange), and the CMIP6 multi-model mean using the average of the five OHC records. The box represents the 25$^{th}$ 50$^{th}$, and 75$^{th}$ percentiles of the values of ECS and the whiskers denote the 5$^{th}$ and 95$^{th}$ percentiles. The stars indicate the minimum and maximum values of ECS.

The New Fig. S13 shows the values of $\lambda_\Sigma$ and AER RF$_{2011}$ for which the EM-GC calculates good fits to the CMIP6 multi-model mean for SSP2-4.5 from 1850-2100, using the average of five OHC records to represent the amount of heat going into the ocean (New Fig. S13a) or the Cheng 2017 OHC record (New Fig. S13b). Use of either the average of five OHC records or the Cheng 2017 OHC record results in similar values of $\lambda_\Sigma$, AER RF$_{2011}$, and $\Delta T_{2011}$. We use the aerosol weighting method described in Sect. 2.5 of the submitted paper to calculate the box and whiskers shown in New Fig. S13c.

The box and whisker plots in New Fig. S13c using the EM-GC approach are similar to the box and whisker plot using the Gregory et al. (2004) method. The EM-GC approach does not yield values of ECS as low as some of the CMIP6 GCMs exhibit upon use of the Gregory et al. (2004) method, but the median values are relatively similar. The EM-GC approach is able to obtain values of ECS around the maximum value reported using the Gregory et al. (2004) method. The reviewer has raised an excellent point, and we contend

that the inclusion of New Fig. S13 in supplement, complemented by a new paragraph in Section 3.2 of the Main paper, constitutes an important revision to our paper that will be of great interest to the community.

5) I could not quite understand the computations behind the TCRE: how are uncertainties in the carbon cycle taken into account? This is important for the 66% and 95% likelihood estimates.

Thanks for yet another excellent point that we also plan to address upon revision, as detailed below.

We had not taken the carbon cycle into account. To do so, we examined Friedlingstein et al. (2014) and Murphy et al. (2014) and found that the uncertainty in their estimates of atmospheric $CO_2$ concentration from emissions driven runs from the CMIP5 coupled carbon cycle models is about 10% (1-sigma). Upon revision, we propose to use this 10% value that relates uncertainty in $CO_2$ to carbon emissions, to represent uncertainty in the global carbon cycle for our estimates of the remaining carbon budget, before certain temperature thresholds are passed. We propose to update Table 2 as shown below and update the corresponding text in Sect. 3.3.4 and the conclusions to reflect this addition of uncertainty in the global carbon cycle to our estimates of total cumulative carbon emissions.

**Proposed Update to Table 2.** Total cumulative and future carbon emissions that will lead to crossing the Paris temperature thresholds based on the EM-GC. Estimates of $\Sigma CO_2^{EMISSIONS}$ that would cause global warming to stay below indicated thresholds for 95%, 66%, and 50% probabilities. The values in the top half of the table are the estimates of total cumulative carbon emissions that will lead to crossing the Paris Agreement thresholds with the 10% uncertainty. The values in the bottom half of the table are the estimates of future cumulative carbon emissions after 2019 that will lead to crossing the Paris Agreement thresholds. The range of years given represents when the Paris Agreement thresholds will be passed based upon the rate of emissions from SSP5-8.5 or continuing the current rate of emissions of 11.7 Gt C yr$^{-1}$.

| Total $\Sigma CO_2^{EMISSIONS}$ since 1870 from the EM-GC | | | |
|---|---|---|---|
| | **95%** | **66%** | **50%** |
| **1.5°C** | 746 ± 75 Gt C | 906 ± 91 Gt C | 974 ± 97 Gt C |
| **2.0°C** | 933 ± 93 Gt C | 1203 ± 120 Gt C | 1323 ± 132 Gt C |
| Future $\Sigma CO_2^{EMISSIONS}$ (assuming 638 Gt C released between 1870-2019) | | | |
| | **95%** | **66%** | **50%** |
| **1.5°C** | 108 ± 75 Gt C (2022[a]-2032[a]) (2021[b]-2034[b]) | 268 ± 91 Gt C (2032-2042) (2034-2049) | 336 ± 97 Gt C (2036-2045) (2039-2056) |
| **2.0°C** | 295 ± 93 Gt C (2033[a]-2043[a]) (2036[b]-2052[b]) | 565 ± 120 Gt C (2046-2056) (2057-2077) | 685 ± 132 Gt C (2051-2061) (2066-2088) |

[a] Year the 1.5°C target or 2.0°C upper limit will be exceeded assuming the rate of emission inferred from SSP5-8.5 and the 1-sigma uncertainty

 Year the 1.5°C target or 2.0°C upper limit will be exceeded assuming current rate of emission of 11.7 Gt C yr$^{-1}$ and the 1-sigma uncertainty

6) The paper is quite long, and I think that it will become more convincing after a good look at the prose. My minor comments will give further suggestions.

We thank the reviewer for this helpful suggestion, and we will take a careful look at the prose upon revision.

Minor comments:
76. Remove the word "active", as that implies a dynamic ocean, which is not what the model has

Thank you, we will fix

82. The paper uses many capitalised abbreviations, which is inevitable. However, words like months and obs can be written in lowercase to make reading more pleasant.

For the sake of conforming to past precedent (Canty et al., 2013; Hope et al., 2017), we would like to keep the use of capitalized abbreviations. We will be happy to make the change, if directed by the reviewer or editor. If directed to make this change, we will need to redo several figures that use the capitalized abbreviations in subscripts.

101. Maybe repeat what delta TMDL means

Thank you, we will add

132. I'm not sure whether your definition of lambda can also be called a climate feedback parameter. It would be confusing to have two different parameters with the same name

The literature is littered with numerous definitions of "climate feedback", and various names for our lambda. We have called this quantity the "climate feedback parameter" in our prior papers (Canty et al., 2013; Hope et al., 2017), so for the sake of conforming to past precedent, we prefer to retain this description of lambda in the current paper. The only other time we talk about another climate feedback parameter is when we describe that our quantity uses a different formalism from Bony et al., 2006 and Gregory, 2000, on page six of the submitted paper, and in our new Sect. 3.3.6. We had hoped this text would illuminate, rather than confuse. This is a minor point and we will be happy to remove the sentence in question, if so directed.

136. This sentence can be removed, as it's not providing any information relevant to the study.

We would like to keep this sentence because it gives support to the relationship of $\lambda_\Sigma$ and $\gamma$ used in our analysis. Again, this is a minor point and we will be happy to remove the sentence in question, if so directed.

188. Normally the reduced chi-squared parameter is denoted $\chi\nu2$ to differentiate from normal chi-squared.

The reviewer is correct; a superscript of "nu" is used in formal mathematics literature, for reduced chi-squared. Again, we have omitted this subscript in our prior papers (Canty et al., 2013; Hope et al., 2017) and for the sake of conforming to past precedent as well as for readability (i.e., less clutter), we prefer to not add this subscript. However, we will be delighted to add the subscript, if so directed by either the reviewer or editor. We will need to change several figures that currently do not have the superscript "nu".

Figure 1: The AAWR in panel b is different from the lead, which one is correct?

Thanks for this great catch! The value of AAWR shown in panel b is specifically for an aerosol radiative forcing pathway with a value of radiative forcing equal to $-0.9$ W m$^{-2}$. The value of AAWR reported in the abstract is the median value. We had used the wrong wording in the abstract and referred to the median value as the best estimate; upon revision, we will fix this mistake.

239-243. In this study, the datasets are referred to by the name of the institutions responsible for them, but they have specific names. Could you replace CRU with Had-CRUT, GISS with GISTEMP and so forth.

Great suggestion. We will replace CRU with HadCRUT, GISS with GISTEMP, NCEI with NOAAGT, and add CW14 for Cowtan and Way, 2014.

242: Typo. Berkley=Berkeley

Great catch; much thanks.  We will fix.

247. Transformation usually means adjusting the mean and variance, where you're only adjusting the mean

Thank you we will replace "transformation" with "adjust".

257. The baseline is defined as no mitigation, so this sentence would be corrected if you remove that word

We are referring to Figure 1 of O'Neill et al. (2014) which describes the "challenges" space spanned by the SSPs. These are socioeconomic challenges for adaptation and socioeconomic challenges for mitigation. O'Neill et al. (2014) defines the challenges to mitigation as "socioeconomic factors that would make the mitigation task easier or harder for any given target and mitigation policy". We would like to edit the sentence to read "The baseline pathways follow specific narratives for factors such as population, education, economic growth, and technological developments of sources of renewable energy (Calvin et al., 2017; Fricko et al., 2017; Fujimori et al., 2017; Kriegler et al., 2017; van Vuuren et al., 2017) to represent several possible futures spanning different challenges for adaptation and mitigation to climate change as illustrated in Fig. 1 of O'Neill et al. (2014)." to draw the reader to this important paper that describes the creation of the SSPs.

263-269. Description of the tiers is unnecessary for this study, consider dropping it.

We would like to keep a short description of the tiers in this study, because the two different tiers were the basis for choosing which SSPs we highlighted in the main paper.

We propose eliminating the detailed description of the two tiers and will just describe which SSP scenarios are part of tier 1 and tier 2.

272. Add "the" ("in the supplement")

Thank you, we will fix.

279-281. Which equation comes from which source?

All of the equations used to calculate radiative forcing come from Myhre et al. (1998). We use the updated values of preindustrial concentrations from Myhre et al. (2013) and the updated radiative efficiencies from WMO (2018). Upon revision, we will update the text to better describe from which source we obtain information for the radiative forcing calculations.

289. Remove brackets around Myhre

Thank you, we will fix.

295. Upon -> on?

Thank you, we will fix.

320. Are you sure it's not perfectly identical?

The values of aerosol radiative forcing provided by the SSP Database are the same in 2005 and are slightly different in 2010, hence our wording of nearly identical. We would like to change the text has written to, "The database provides AER RF from 2005-2100, with values for all SSPs nearly identical until about 2010". Upon examination of the aerosol radiative forcing time series on the SSP database, the time series start to deviate around 2015.

323. Remove "described above", it's unnecessary

Thank you we will fix.

363-369. If I understand it correctly, three different time series are appended. Would it not be easier to derive the entire time series yourself? That would also be easier to describe.

Yes, we did append three time series to create the MEI.v2 time series for ENSO in our model framework. While it may be easier to derive the entire time series ourselves, we want to train to data whenever possible. Hence our decision to append the two MEI time series together. We created the MEI time series ourselves from 1860-1871 because the MEI.ext did not extend past 1871. We propose to rework the text in the first paragraph of Sect. 2.2.6 to clarify our method for arriving at the full MEI.v2 time series.

415-416. Normalization involves both the mean and standard deviation, offsets are always additive. Maybe rewrite as "the five datasets are all set to zero in 1986 by applying an offset"

Thank you, we will change as suggested.

433. I didn't understand which standard deviation of the mean was taken

We are referring to the 1 sigma standard deviation of the average of the five OHC records. We will clarify this in the text.

463. Replace "based upon" by "using", remove "shown below"?

Thank you, we will fix.

477-480. I didn't understand why AAWR is not affected at all, as regression variables, such as lambda, are surely influenced by the inclusion of AMOC.

The value of AAWR and $\lambda_\Sigma$ are only slightly affected by not including AMOC. We get a worse fit to the climate record when not including AMOC. Response Fig. 2 shows a plot similar to Fig. 1 from the paper, found upon not including AMOC, IOD, or PDO in the regression. As shown in Response Fig. 2a, the value of $\lambda_\Sigma$ has decreased from 1.38 W m$^{-2}$ °C$^{-1}$ (submitted paper) to 1.34 W m$^{-2}$ °C$^{-1}$. The value of AAWR declines from 0.144°C/decade to 0.141°C/decade. The decrease in the value of AAWR is well within the 2σ uncertainty of AAWR. Our estimate of $\lambda_\Sigma$ and AAWR are much more sensitive to the value of AER RF$_{2011}$, as shown in Fig. 3 and Fig. 6. Our value of $\chi^2_{ATM}$ increases from 0.71 to 1.20. The inclusion of AMOC, as well as PDO and IOD, is important to achieve a better fit to the climate record, allowing more of the parameter space for climate feedback and aerosol RF of climate to be considered upon use reduced of our chi-squared ≤ 2 filtering.

[Figure]

**Response Figure 2.** Measured and modeled GMST (ΔT) relative to pre-industrial (1850-1900) baseline. (a) Observed (black) and modeled (red) ΔT from 1850-2019. (b) Contributions from total human activity. This panel also denotes the numerical value of AAWR from 195-2014 (black dashed) as well as the 2σ uncertainty in the slope. (c) Solar irradiance (light blue) and major volcanoes (purple). (d) Influences from ENSO on ΔT. (e) Influence from AMOC is set to zero. (f) Influences from PDO (blue) and IOD (pink) are set to zero. (g) Measured (black) and modeled (red) OHC as a function of time for the average of five data sets.

519. if I understand it correctly, these equations assume there is no uncertainty at all in the radiative forcing at the doubling of CO2, which is inconsistent with definitions of radiative forcing and with CMIP6 models.

> We do not include uncertainty in the RF of climate due to a doubling of $CO_2$, and prefer to retain this formality, because this uncertainty is so small compared to so many other uncertainties. We are co-authors of a paper entitled "Reduced Complexity Model Intercomparison Project Phase 2: Synthesising Earth system knowledge for probabilistic climate projections", led by Zebedee Nicholls of the University of Melbourne, that was

submitted to the AGU journal Earth's Future on 12 Nov 2020. About half of the Reduced Complexity Climate Models used in this paper considered this uncertainty for projections of global warming out to 2300; our model along with about the other half did not consider this uncertainty. The author team of this Nicholls et al. paper decided to proceed without restricting uniformity for the treatment of this uncertainty, since it has such a small impact. That said, if so directed by either the reviewer or editor, we will gladly consider this uncertainty, but we note it will add a considerable computational burden for completion of our proposed revision, and we are certain it will not change our results in any meaningful manner.

550-552. I did not understand what an asymmetric Gaussian was, could you explain?

In our study, we use the term asymmetric Gaussian to refer to the fact that the distribution of the likely range and 5[th] and 95[th] percentiles of the values of AER $RF_{2011}$ from Myhre et al. (2013) are not distributed symmetrically from the best estimate of AER $RF_{2011} = -0.9$ W m$^{-2}$. The likely range of AER $RF_{2011}$ is given as $-0.4$ W m$^{-2}$ and $-1.5$ W m$^{-2}$. The $-0.4$ W m$^{-2}$ value is 0.5 W m$^{-2}$ from the best estimate, while the value of $-1.5$ W m$^{-2}$ is 0.6 W m$^{-2}$ from the best estimate. The 5[th] and 95[th] percentiles of AER $RF_{2011}$ are given as $-0.1$ and $-1.9$ W m$^{-2}$, which are 0.8 W m$^{-2}$ and 1.0 W m$^{-2}$ from the best estimate, respectively. We fit a Gaussian to the best estimate, likely range, and 5[th] and 95[th] percentiles. Because the likely range and 5[th] and 95[th] percentiles are not arranged symmetrically around the best estimate, the Gaussian is asymmetric.

Upon revision, we propose to add clarifying text to help readers understand this concept, to our method Section 2.5.

649. Remove "as indicated"

Thank you we will fix.

675. The value of 1.85 contradicts the value in the next paragraph of 2.01. Which one is correct?

Thanks for noting this contradiction. Both values are correct. The value of 1.85°C is the median value of ECS found using the HadCRUT temperature record. The value of 2.01°C is the best estimate value of ECS found using the HadCRUT temperature record. The best estimate corresponds to an AER $RF_{2011}$ value of $-0.9$ W m$^{-2}$, whereas the median does not. We will add a sentence that describes the best estimate of ECS refers to the specific value of AER $RF_{2011}$ of $-0.9$ W m$^{-2}$.

Figure 8: This figure only uses studies with low climate sensitivity and compares them to assessments of climate sensitivity (Sherwood/IPCC). Either explain the selection criteria, or add some studies to make this figure more balanced (the carbon brief provides an excellent overview: https://www.carbonbrief.org/explainer-how-scientists-estimateclimate-sensitivity)

We propose to update this figure to include more estimates of ECS, as shown below. The new figure is now divided into three categories: studies that used a historical analysis,

studies that examined output from the GCMs and constrained it in some way, and studies that examined output from the GCMs and did not use any constraints. The studies that are included are from manuscripts published in the last few years. We will update the modified text, accordingly.

[Figure]

**Proposed Update to Figure 8.** Values of ECS from the EM-GC (black), CMIP6 multi-model ensemble (black), and 13 other studies grouped by type of study. The studies are listed by first author first initial of their first name and first initial of their last name and the year of publication, unless there are only two others, in which case both authors are listed. Historical analysis includes Lewis and Grünwald (2018) as NL+PG18, Otto et al. (2013) as AO13, and Skeie et al. (2018) as RS18. Constrained GCM output includes Armour (2017) as KA17, Cox et al. (2018) as PC18, Dessler et al. (2018) as AD18, Nijsse et al. (2020)as FN20, Rugenstein et al. (2020) as MR20, Sherwood et al. (2020) as SS20, Stocker et al. (2013) as IPCC 2013, Tokarska et al. (2020) as KT20. GCM output includes Proistosescu and Huybers (2017) as CP+PH17 and Zelinka et al. (2020) as MZ20.

689. The word "yet" implies a contradiction. However, with the very wide uncertainty specified by the IPCC, these probably overheating models are still within range.

Thank you for pointing out this apparent contradiction. We will replace the word "yet" with "and" and add "analyzed in this study" after CMIP6 GCMs to remove the implied contradiction.

698. rm the word "actual"?

Thank you, we will remove "actual".

704-705. Consider deleting "ninety-five…multi-model ensemble" as I think it is an unnecessary detail. Presenter 713. Remove "then"?

Thank you we will remove this sentence and the "then" later on in the paragraph.

739. Remove information between brackets, repetition of information within paragraph.

Thank you we will remove the 5$^{th}$ percent confidence interval that is written between the brackets.

782. Bifurcation has a specific meaning within mathematics, consider replacing by bimodality. If more models are added, check whether it's still true.

Thank you, we will change "bifurcation" to "bimodality".

Figure 10: Use different colour scheme. The rainbow colour scheme has false perceptual thresholds or hides real ones: https://www.nature.com/articles/519291d.

Thank you for pointing out this issue with the rainbow color scheme. We will replace the color scheme in Fig. 5, Fig. 10, Fig. S1, and Fig. S8 with a blue-red color scheme. Below is an example of the revised color scheme for Fig. 10.

[Figure]

811. Replace "will" with "is set", we don't know the future.

Excellent point, we will change as suggested.

931. Replace "since" with "after".

Thank you, we will make this change.

669. Insert dioxide after carbon

We think the reviewer is referring to line 969; "carbon" will be changed to "carbon dioxide".

1002. Unnecessary to show all these percentiles, remove 25 and 75.

Thank you, we will change as suggested.

1009. 2017 was not an El Niño-year and non-El Niño-years 2018 and 2019 were comparable in temperature.

Thank you we will fix by clarifying the El Niño event ended in 2016.

1012. Similar, summarise, so do not show all percentiles

Thank you, we will fix.

1056. Similar, summarise.

Thanks again, we will fix.

1061. Replace "will" with "will not"

Great catch; thanks! We'll of course fix.

1071. Repetition of the information in 1061

We prefer to retain this sentence, to emphasize a key result from our analysis, but upon revision will gladly remove if so directed by the reviewer or editor.

1073. What is a literal interpretation? The model democracy interpretation?

We would like to remove the phrase "a literal interpretation of" so the sentence reads "We suggest there is slightly more time to achieve these steep reductions than indicated by the CMIP6 multi-model mean".

1074. Modeling is not the only source of information on warming of 1.5 degrees, many studies extrapolate current trends.

The purpose of this concluding paragraph is to mention the value of the CMIP6 GCMs. We are unsure exactly what studies the reviewer is referring to in their comment and would ask the reviewer to provide some examples if they believe it is important for us to mention these studies in this paragraph.

We would like to propose rewording the sentence to "The incredibly valuable output of the CMIP6 GCMs is important for determining the consequences for society of 1.5°C, 2.0°C, or even larger rises in GMST". This rewording removes the phrase "rely entirely on" that may have been the cause of the issue for the reviewer.

Figure S1: Replace the rainbow colour scheme.

Thank you, we will address by no longer using the rainbow color scheme, using instead the color scheme shown below:

[Figure]

Figure S7: Caption should indicate that it's the unweighted one.

Thank you, we will fix.

Reviewer comments are in black and our reply is in blue.

This manuscript used a multiple linear regression energy balance model, EM-GC, to estimate the attributable anthropogenic warming rate (AAWR), the equilibrium climate sensitivity (ECS), and the future projections. The authors compared the results from EM-GC with those obtained from CMIP6. They found that the CMIP6 GCMs tend to exhibit a faster rate of warming, which induced larger AAWR, larger ECS, and smaller remaining budgets of carbon emissions. One highlight of this work is the use of Aerosol Weighting Method, which allowed a probabilistic estimation. This work is very interesting and the authors have done many detailed analyses. However, before I can recommend accepting this manuscript, I have several concerns that need to be addressed.

> We thank the reviewer for taking the time to carefully read our manuscript and suggest many useful changes. Upon revision, we will make changes to the paper to address all of these comments, as detailed below.

1. To run the EM-GC model, it seems that one needs to determine nine regression coefficients and parameters. Constrained by the observed GMST and the OHC, one can obtain a set of the nine coefficients/parameters to ensure a good fit to the historical observations. However, I am not sure if the selected set of coefficients/parameters is unique, or one can use a totally different set of coefficients/parameters to achieve a similar fitting skill? I also have concerns that whether the coefficients/parameters are still useful for the future projections? I would like to suggest the author to perform a test to prove the validity of the model and the stability of the coefficients/parameters. For example, the authors may consider to divide the historical period into two halves, use the first half to determine the coefficients/parameters, and use the second half to test the stability.

> Great suggestion, which we plan to address upon revision.
>
> The only parameters important for the future projections of GMST are the climate sensitivity parameter, $\gamma$, and the ocean heat uptake efficiency, $\kappa$. The regression coefficients $C_0$-$C_6$ in Eq. (2), which modify natural drivers of climate variability, are only used to simulate the observed change in GMST from 1850-2019. However, future temperature projections consider only the anthropogenic components governed by RF due to GHGs, aerosols, as well as climate feedback (related to $\gamma$) and ocean heat uptake (related to $\kappa$). We are able to obtain much better fits to the actual climate record upon consideration of the full range of natural drivers of climate variability; hence, the inclusion of $C_0$-$C_6$ allows for a more realistic evaluation of the range of model parameter space for $\gamma$ and $\kappa$ under which "good fits" to the prior climate record can be obtained.
>
> We have taken the suggestion from the reviewer to alter the training period of our model to test the stability and propose to show these results in a new Supplemental Figure. New Fig. S2 shows the projections of the change in GMST in 2100, $\Delta T_{2100}$, as a function of the climate feedback parameter, $\lambda_\Sigma$, and the value of aerosol radiative forcing in 2011, AER $RF_{2011}$, for 4 different training periods: 1850-1989 (New Fig. S2a), 1850-1999 (New Fig. S2b), 1850-2009 (New Fig. S2c), and 1850-2019 (New Fig. S2d), which is the normal training period used in our analysis. Values of $\Delta T_{2100}$ are shown only for combinations of

$\lambda_{\Sigma}$ and AER RF$_{2011}$ (value of aerosol radiative forcing in 2011) that lead to good fits to the climate record, which means values of the three reduced chi-squared $(\chi^2)$ parameters are all less than or equal to 2. We project relatively similar results for end of century warming for the training periods that end in 2019, 2009, and 1999. The training period that ends in 1989 (New Fig. S2a) yields a different "shape" of model parameter space for which good fits to the climate record can be obtained, compared to the other training periods. The different shape for this shorter training period is due to the formulation of the ocean component of our model. In training to 1989, we are only considering 35 years of the observed OHC record. We are able to calculate good fits to the OHC record over this shorter time period that diverge from the OHC record after 1989. The highest values of $\Delta T_{2100}$ in New Fig. S2a are associated with the largest values of $\lambda_{\Sigma}$, which in our model corresponds to excessively high values of $\kappa$ that we can rule out, based on OHC data collected during 1990 to 2020.

We propose upon revision to add a paragraph to Sect. 2.1 to the paper noting the stability of the forecasts of end-of-century warming for the training periods of 1850-1999, 1850-2009, and 1850-2019, with most of the words supporting this finding appearing in the revised Supplement along with New Fig. S2.

[Figure]

**New Figure S2.** $\Delta T_{2100}$ as a function of climate feedback parameter and tropospheric aerosol radiative forcing in 2011 using the EM-GC for SSP4-3.4. (a) Training period of 1850-1989. The region outside of the AER $RF_{2011}$ range provided by IPCC 2013 is shaded (grey). Colors denote the GMST change in year 2100 relative to pre-industrial. The color bar is the same across all four panels for comparison. (b) Training period of 1850-1999. (c) Training period of 1850-2009. (d) Training period of 1850-2019, which is the normal training period used in our analysis.

2. From Fig. 1f, the authors found that the PDO has very limited contributions to the GMST. I don't understand this finding, as to my knowledge, the different phases of the PDO play an important role in modulating the GMST. For example, the recently well discussed warming hiatus in the beginning of this century has been found to be closely related to the PDO. An explanation about the findings in Fig. 1f is needed.

Another great suggestion, which we also plan to address upon revision.

In Fig. 1f of the submitted paper, we had shown the model run for the best estimate of AER $RF_{2011} = -0.9$ W m$^{-2}$. In this case, the PDO exhibits less influence on GMST than we find for AMOC. If we vary the value of AER $RF_{2011}$, which impacts the time series of aerosol RF of climate over the entire simulation, we find there are some model runs for which the PDO has the same or even larger influence on GMST compared to AMOC.

Upon revision, we propose to include the figure shown below as New Fig. S6, and modify the text in the paper to make clear that the expression of the PDO on GMST in our model framework is dependent on model specification of the aerosol RF of climate time series. At low values of AER $RF_{2011}$, the effect of PDO on GMST (New Fig. S6f) is negligible

and the contribution from AMOC dominates over PDO or IOD. At high values of AER RF$_{2011}$, the effect of PDO on GMST (New Fig. S6m) is equal to the contribution from AMOC. Upon revision, we will add a new paragraph to Sect. 2.2.6 discussing the importance of PDO at higher values of AER RF$_{2011}$ and include New Fig. S6 in the supplement with citations to England et al. (2014) and Trenberth and Fasullo (2015). The figure below is a robust result: the larger the scaling factor for aerosol RF, the greater the influence of PDO.

[Figure]

**New Figure S6.** Measured and modeled GMST anomaly (ΔT) relative to a pre-industrial (1850-1900) baseline for an AER RF$_{2011}$ = −0.1 W m$^{-2}$ and −1.5 W m$^{-2}$. (a) Observed (black) and modeled (red) ΔT from 1850-2019. This panel also displays the values of λ$_Σ$ and χ$^2$$_{ATM}$ (see text) for this best-fit simulation. (b) Contributions from total human activity. This panel also denotes the numerical value of the attributable anthropogenic warming rate from 1975-2014 (black dashed) as well as the 2σ uncertainty in the slope. (c) Solar irradiance (light blue) and major volcanoes (purple). (d) Influences from ENSO on ΔT. (e) Contributions from AMOC to ΔT and to observed warming from 1975-2014. (f) Influences from PDO (blue) and IOD (pink) on ΔT. (g) Measured (black) and modeled (red) ocean heat content (OHC) as a function of time for the average of five data sets (see text), the value of χ$^2$$_{OCEAN}$ for this run, as well as the ocean heat uptake efficiency, κ, needed to provide the best-fit to the OHC record. The error bars (blue) denote the uncertainty in OHC used in this analysis (see Sect. 2.2.8). (h)-(n) Same as (a)-(g), except for AER RF$_{2011}$ = −1.5 W m$^{-2}$.

3. Another concern is about the comparison of the AAWR that obtained from EM-GC and CMIP6 models. Since different methods are used to calculate the AAWR, I am not sure if the results are comparable. Especially for the CMIP6 models, the REG method seems to be too simple to calculate the AAWR. I am not sure if the AAWR values obtained from the CMIP6 models are as pure as those obtained from EM-GC.

We plan, upon revision, to add much more detail regarding how the attributable anthropogenic warming rate, AAWR, is estimated from CMIP6 GCM output.

In Sect. 2.3 of the submitted paper, we discuss two methods to determine the AAWR from the CMIP6 models, REG and LIN. REG is a regression-based approach and LIN is a linear fit method. For the GCM-based estimates of AAWR that appeared in the submitted paper, the LIN method tends to result in very slightly higher values than REG, as shown in Response Fig. 1.

[Figure]

**Response Figure 1.** Values of AAWR for 50 CMIP6 GCMs using the LIN and REG methods. The solid black line is the 1:1 line and the vertical and horizontal dashed lines are the maximum value of AAWR determined using the EM-GC and the HadCRUT temperature record. The CMIP6 GCMs that have values of AAWR less than the maximum value from the EM-GC are blue, and the CMIP6 GCMs that have values of AAWR greater than the maximum value from the EM-GC are red. The slope, 1σ standard deviation, and $R^2$ of the values of AAWR from the CMIP6 GCMs are shown.

The values of AAWR determined by the LIN method are about 4% higher than the values of AAWR determined by the REG method. The close agreement of AAWR found using

both methods provides strong evidence that we have correctly extracted this important quantity from the CMIP6 archive.

We have further examined our calculation of AAWR using the REG method in response to the reviewer's comment and have a few proposed changes that lead to a more robust estimate that we will implement upon revision.

As detailed below, close examination of the CMIP6 GCM output, shows that the representation of the effect of variations in total solar irradiance, TSI, on global mean surface temperature (GMST) in the GCMs leads to a regression coefficient that seems to be randomly distributed (see Response Fig. 2).

[Figure]

**Response Figure 2.** The change in GMST relative to 1961-1990 from the CMIP6 GCMs and the contribution from TSI and SAOD from 1960-2014. (a) The change in GMST from the 50 CMIP6 GCMs. (b) The residual in the change of GMST from the 50 CMIP6 GCMs after subtracting the contribution of TSI and SAOD determined by the REG method. The median value of AAWR is written on this panel and plotted in red. (c) The contribution of TSI in the 50 CMIP6 GCMs. (d) The contribution of SAOD in the 50 CMIP6 GCMs.

Response Fig. 2c shows the random representation of TSI in the CMIP6 GCMs. Upon the implementation of the REG method, some CMIP6 GCMs obtained negative coefficients for TSI, and others obtained positive coefficients. For some reason, many GCMs do not

represent the impact of variations in solar output on GMST in a manner that mimics the actual, observed relation. There is extensive literature on possible reasons TSI affects GMST, implicating causal factors such as cosmic-ray influence on cloud nucleation, that is nicely summarized at https://skepticalscience.com/cosmic-rays-and-global-warming-advanced.htm. If the true causal factor involves something like cosmic rays, this process will likely not be present in most GCMs. Because of the varying nature of TSI in the GCMs, we propose to update our calculation of REG to not include TSI in the regression.

We propose to alter the REG method in the following way. We will conduct one regression from 1975-2014, instead of two regressions as had been explained in Sect. 2.3 of the submitted paper. We will exclude TSI as a regressor and only include stratospheric aerosol optical depth (SAOD) and a linear function to represent the contribution of humans to the change in GMST. For SAOD, we will determine the appropriate lag for each model that results in the largest coefficient, to accurately represent how long it takes for the effect of enhanced SAOD to have on GMST within each model. Using this new REG method results in very slightly different values of AAWR compared to those in the submitted paper, as shown in New Fig. S10 and New Fig. S11 below.

[Figure]

**New Figure S10.** The change in GMST relative to 1961-1990 from the CMIP6 GCMs and the contribution and SAOD from 1975-2014. (a) The change in GMST from the 50 CMIP6 GCMs. (b) The residual in the change of GMST from the 50 CMIP6 GCMs after subtracting the contribution of SAOD determined by the updated REG method. The median value of AAWR is written on this panel and plotted in red. (c) The human component of global warming, $\Delta T_{ATM,HUMAN}$, from the EM-GC. A linear fit (black) and quadratic fit (red) are plotted on top to show that $\Delta T_{ATM,HUMAN}$ is almost exactly linear. (d) The contribution of SAOD in the 50 CMIP6 GCMs using a lag month calculated for each model.

A comparison of New Figure S10 to Response Figure 2 shows that AAWR found using the REG method is not much affected by removing TSI as a regressor. The values of AAWR determined from the CMIP6 GCMs are more similar to the values determined by the LIN method, under this new approach. New Figure S11 shows that there is now a 0.9% difference between the values of REG and LIN. New Figure S11c shows the human component of global warming, $\Delta T_{ATM,HUMAN}$, from the EM-GC. A linear fit and quadratic fit were taken of $\Delta T_{ATM,HUMAN}$. The linear fit and quadratic fit are very similar, indicating that $\Delta T_{ATM,HUMAN}$ is in fact nearly linear over this period of time. This result justifies our approach of approximating a linear function to represent $\Delta T_{ATM,HUMAN}$ in the AAWR calculation.

Upon revision, these figures will be noted in Main, and detail will be added to the Supplement to document our procedure for finding AAWR from the GCMs, allowing the reader to better assess the procedure and, in our view, accurate rendering of this quantity from the CMIP6 archive.

[Figure]

**New Figure S11.** Values of AAWR for 50 CMIP6 GCMs using the LIN and REG methods. The solid black line is the 1:1 line and the vertical and horizontal dashed lines are the maximum value of AAWR determined using the EM-GC and the HadCRUT temperature record. The CMIP6 GCMs that have values of AAWR less than the maximum value from the EM-GC are blue, and the CMIP6 GCMs that have values of AAWR greater than the maximum value from the EM-GC are red. The slope, 1σ standard deviation, and $R^2$ of the values of AAWR from the CMIP6 GCMs are shown.

4. In line 228, "…also specified on Fig. 1f", "Fig. 1f" should be "Fig. 1e".

    Thank you, we will fix.

5. In line 975, "then" should be "than".

    Thank you, we will fix.

6. In line 1061, "…of the Paris Agreement will be achieved", "will be" should be "will not be".

    Thank you, we will fix.

[revised manuscript text omitted]
)". Figure S1 illustrates the impact of updating Eq. (2) in our model to be comparable to the formulation in Bony et al. (2006) and Schwartz (2012). This figure displays the change in GMST anomaly in 2100 relative to pre-industrial ($\Delta T_{2100}$) as a function of $\lambda_\Sigma$ and AER $RF_{2011}$ for the two formulations of Eq. (2). Figure S1a uses the previous version of the EM-GC, where $Q_{OCEAN}$ was subtracted outside of the climate feedback multiplicative term, and Fig. 1b uses the new version of the EM-GC where $Q_{OCEAN}$ is subtracted within the climate feedback multiplicative term.

In the EM-GC framework, we calculate our value of $Q_{OCEAN}$ by finding the $\kappa$ needed to multiply the temperature difference between the atmosphere and the ocean to fit the observed OHC record. The model iterates over the ocean module, specifically the value of $\Delta T_{OCEAN,HUMAN}$ in Eq. (4), until the EM-GC converges on an estimate of $\kappa$ for a single OHC record and value of AER $RF_{2011}$. Figure S1 illustrates that the effect of changing Eq. (2) in the EM-GC impacts our estimates of the rise in $\Delta T_{2100}$ at high values of AER $RF_{2011}$. Strong aerosol cooling results in the ocean taking up more heat from the atmosphere than in the previous version of the EM-GC. The larger value of $Q_{OCEAN}$ results in a higher value of climate feedback needed to fit the historical climate record, because both AER $RF_{2011}$ and $Q_{OCEAN}$ are acting to cool the climate system. The higher values of climate feedback increase our maximum value of $\Delta T_{2100}$. This change brings some of the projections of $\Delta T_{2100}$ from the EM-GC closer to values of $\Delta T_{2100}$ from the CMIP6 multi-model ensemble.

Section 2.1 states "Altering the training period of our model has a slight effect on our results (see Fig. S2 and the supplement for information on various training periods)." Figure S2 shows the end of century projected warming as a function of $\lambda_\Sigma$ and AER $RF_{2011}$, for four different training periods: 1850-1989 (Fig. S2a), 1850-1999 (Fig. S2b), 1850-2009 (Fig. S2c) and 1850-2019 (Fig. S2d), which is the normal training period used in our analysis. Values of $\Delta T_{2100}$ are shown only for combinations of $\lambda_\Sigma$ and AER $RF_{2011}$ that lead to good fits to the climate record. We project relatively similar results for end of century warming for the training periods that end in 2019, 2009, and 1999. Our results using the training period from 1850-1999 are similar to observations and other reduced complexity models (Nicholls et al., 2020). The training period that ends in 1989 (Fig. S2a) yields a different "shape" of model parameter space for which good fits to the climate record can be obtained, compared to the other training periods. The different shape for this shorter training period is due to the formulation of the ocean component of our model. In training to 1989, we are only considering 35 years of the observed OHC record. We are able to calculate good fits to the OHC record over this shorter

forcing of climate cools in a manner that nearly mirrors the warming due to rising GHGs, resulting in a wider range of model parameters that lead to a "good fit" of the climate record, compared to model simulations constrained by data that extend closer to present-day. The highest values of $\Delta T_{2100}$ in Fig. S2a are associated with the largest values of $\lambda_\Sigma$, which in our model corresponds to excessively high values of $\kappa$ that we can rule out, based on OHC data collected during 1990 to 2020.

Section 2.2.1 states "see Fig. S4 and the supplement for information on CW14 GMST record". Figure S4 shows the GMST anomaly relative to pre-industrial over time for (a) HadCRUT record with the HadCRUT uncertainties, (b) CW14 record with the CW14 uncertainties, (c) BEG record with the BEG uncertainties, and (d) CW14 record with HadCRUT uncertainties. The uncertainties for CW14 are much smaller than those for the HadCRUT and BEG records. The small values of CW14 uncertainties, especially from 1850-1900, cause the EM-GC to not be able to achieve good fits to this temperature record. We have two choices for use of the CW14 GMST record; either relax the constraint for $\chi^2_{ATM}$ (i.e., run with $\chi^2_{ATM} \leq 4$), or modify the CW14 uncertainties. We chose to combine the uncertainties from HadCRUT with the values of GMST from the CW14 record since CW14 is based upon the HadCRUT GMST record. Upon use of this combination of data and uncertainty, we are able to find good fits to the CW14 temperature record that look similar to the fits obtained using the other GMST data sets.

Section 2.2.3 states "Figure  S5 shows the ozone RF time series used in this analysis and the supplement provides more information about the creation of the time series for the RF due to $O_3^{TROP}$". Figure  S5 displays the time series of tropospheric ozone RF used in our analysis for the various SSPs. Tropospheric ozone is an important GHG that rivals nitrous oxide as the third most important anthropogenic GHG. We include the RF due to tropospheric ozone ($O_3^{TROP}$) in our model for completion, even though the SSP database does not provide RF estimates for the various SSPs. We use values from the RCP scenarios provided by the Potsdam Institute for Climate Impact Research (Meinshausen et al., 2011). The values of the RF due to $O_3^{TROP}$ for SSP1-1.9 and SSP1-2.6 are from the RCP2.6 pathway. The RCP4.5 time series of $O_3^{TROP}$ is used for SSP2-4.5, the RCP6.0 time series is used for SSP4-6.0, and the RCP8.5 time series is used for SSP5-8.5. We create linear combinations of RCP2.6 and RCP8.5 to generate two new time series of the RF due to $O_3^{TROP}$ for SSP4-3.4 and SSP3-7.0. There is a large gap between the time series of the RF due to $O_3^{TROP}$ for RCP6.0 (shown as SSP4-6.0) and RCP8.5 (shown as SSP5-8.5) in Fig. S5. We created a time series that would split the difference between the two RCPs to represent the RF due to $O_3^{TROP}$ for SSP3-7.0. The SSP4-3.4 time series of the RF due to $O_3^{TROP}$ that was created lies in between the RCP2.6 (shown as SSP1-2.6) and RCP4.5 (shown as SSP2-4.5) time series in Fig. S5.

Section 2.2.8 states "Figure  S8 shows the five OHC records as well as the multi-measurement average".

Figure  S8 displays the five OHC content data sets, as well as the multi-measurement average, plotted as a function of time and normalized to year 1986. This figure illustrates how the shapes of the different OHC records compare. Each of the time series represents the amount of heat stored in the top 700 m of the world's oceans for that specific data set. Carton et al. (2018) is the shortest data set, and only spans 36 years (1982-2017). The second shortest record is Balmaseda et al. (2013a), which spans 52 years (1958.5-2009.5). Ishii et al. (2017) is the record in the middle with a range of 63 years (1955-2017). Both Cheng et al. (2017) and Levitus et al. (2012) have records that span 65 years (1955-2019). The length of the data set and the shape of the curve affect the estimate of ocean heat export (OHE), because we calculate OHE by taking a linear fit to the full OHC time series. Balmaseda et al. (2013a) has the lowest estimate of OHE because the slope of the curve is relatively shallow, due to the fact that it slightly rises, then decreases at the start of the record. Carton et al. (2018) has the highest estimate of OHE because the slope of the curve is the steepest of the five records.

Section 2.2.8 also says "For these five OHC data sets, uncertainty estimates are not always provided. Furthermore, some studies that do provide uncertainties give estimates that seem unreasonably small (see Fig  S9 and  supplement)" and "Figure  S9 and the supplement provide more detail on the creation of this time dependent uncertainty estimate for OHC". Figure  S9 shows the multi-measurement average as well as the five OHC data records as a function of time, the uncertainty for each corresponding data set, and the combined uncertainty used in this analysis. Panel (a) shows the multi-measurement OHC average with the standard deviation of the mean plotted around the average time series. The standard deviation is large at the beginning of the time series, due to the spread in the estimates of OHC between the different records (illustrated in Fig.  S8). The standard deviation decreases as the various OHC records converge near a similar estimate. The standard deviation is zero in 1986 because we normalized all of the time series to zero in this year to create the multi-measurement average. Because of this normalization, the standard deviation of the mean is not a realistic measure of uncertainty for the five OHC time series.

Panels (b), (c), (d), (e), and (f) display the uncertainty estimates for the five OHC data records. We use the standard deviation of the mean of five ensemble members of the European Centre for Medium-Range Weather Forecasts Ocean ReAnalysis System 4 (ORSA) (Balmaseda et al., 2013b) for the Balmaseda et al. (2013a) record. The standard deviation is plotted in panel (b) as the dotted blue line. The standard deviation is small at the beginning of the record, because the five ensemble members started at similar values of OHC in 1958 and diverged over time. The combined uncertainty of the standard deviation of the  average of the five OHC records and the Cheng et al. (2017) estimate is plotted as a dashed blue line. Panel (c) shows the Levitus et al. (2012) time series for the top 700 m updated to the end of 2019. The Levitus time series

utilizes the standard error over the whole ocean for their uncertainty estimate and is plotted as the dotted light blue line. The standard error is a very small uncertainty estimate compared to the other OHC data records, which is unreasonable considering the large variations in OHC between the different records. We use the standard deviation of eight reanalysis experiments to represent the uncertainty associated with the Carton et al. (2018) OHC record and is plotted as a dotted orange line in panel (e). The standard deviation of the  eight reanalysis experiments is rather small, which also is unrealistic. Panel (f) displays the Cheng et al. (2017) OHC record updated through the end of 2019 with the  1-sigma uncertainty. This uncertainty does not vary much throughout the data record, making it more realistic as an estimate for such an uncertain quantity as OHC. We created the combined uncertainty estimate of the standard deviation of the  average of the five OHC records and the Cheng et al. (2017)  1-sigma uncertainty to have the largest uncertainty possible due to the fact that OHC varies between the different records. The EM-GC cannot achieve $\chi^2_{OCEAN} \leq 2$ for Balmaseda et al. (2013a), Levitus et al (2012), and Carton et al. (2018) using their own respective estimates of uncertainty. Creating one uncertainty estimate to be used for all of the OHC records provides consistency and allows the EM-GC to achieve good fits between the observed and modeled OHC.

Section 2.3 states "Figure S10 illustrates the REG method used to determine AAWR from the CMIP6 GCMs" and "Analysis of AAWR for these 50 GCMs of LIN versus REG (see Fig. S11)...". Figure S10 shows the change in GMST from 1975-2014 from the CMIP6 GCMs and the contribution of SAOD from 1975-2014. There was about a 6 month lag between the response of GMST and enhancements of SAOD following the eruption of Mount Pinatubo in June 1991 (Douglass and Knox, 2005; Thompson et al., 2009); a 6 month delay for the response of GMST to SAOD is commonly used in regression analyses of the actual temperature record (Foster and Rahmstorf, 2011; Lean and Rind, 2008). The time needed for GMST to respond to a change in the aerosol loading in the stratosphere due to a volcanic eruption in each GCM can exhibit a significant difference compared to this empirically determined response time. Therefore, a lag was determined for each GCM by calculating the value of the monthly delay that resulted in the largest regression coefficient for SAOD (versus GMST). Due to the difference in model physics between the various GCMs, the value of the delay between the volcanic forcing and surface temperature response ranged from 0 to 11 months. The effect of SAOD on GMST for the 50 GCMs is shown in Fig. S10d. Figure S10b shows the residual in GMST after removing the influence of SOAD, and the median value of AAWR from the CMIP6 multi-model ensemble is plotted as a linear line. Figure S10c shows the human component of global warming, $\Delta T_{ATM,HUMAN}$, from the EM-GC. A linear fit and quadratic fit are shown to illustrate that $\Delta T_{ATM,HUMAN}$ is almost nearly linear from 1975-2014, supporting the approximation of $\Delta T_{ATM,HUMAN}$ as a linear function from 1975-2014 for the REG calculation.

Figure S11 shows the similarity between the values of AAWR determined using the LIN and REG methods. The ratio between the values of AAWR determined utilizing LIN and REG is 1.009, indicating there is only about a 0.9% difference in the values of AAWR using the two methods. Figure S11 also shows the values of AAWR that are below the maximum value of AAWR determined by the EM-GC utilizing the HadCRUT temperature record (blue) and the values that are above the maximum (red). About half of the GCMs result in values of AAWR less than the maximum value from the EM-GC and half of the GCMs result in values of AAWR greater than the maximum value from the EM-GC utilizing the HadCRUT GMST record.

Section 2.5 states "See Fig. S7 S14 for unweighted ECS values and Section 3.2 states "See Fig S7 S14 for results without aerosol weighting". Figure S7 S14 displays the values of ECS using the EM-GC and the CMIP6 multi-model ensemble. The EM-GC box contains the 25th, 50th, and 75th percentiles, the whiskers denote the 5th and 95th percentiles, and the stars represent the minimum and maximum values of ECS. The box labeled CMIP6 is unchanged from Fig. 7 8. The values of ECS are not treated with the aerosol weighting described in Sect. 2.5. This figure shows that most of the estimates of ECS found using the EM-GC are concentrated towards small values of ECS, due to the fact that the majority of the EM-GC model runs with good fits to the climate record ($\chi^2_{ATM}$, $\chi^2_{RECENT}$, and $\chi^2_{OCEAN}$) have weak aerosol cooling and low values of $\lambda_\Sigma$ (Fig. 5b). We use the aerosol weighting method to assign the same weights for the IPCC 2013 "likely" range limits of AER RF$_{2011}$ of $-0.4$ and $-1.5$ W m$^{-2}$ at the one sigma values of a Gaussian, and the $-0.1$ and $-1.9$ W m$^{-2}$ are at the two sigma values of a Gaussian. Using the aerosol weighting method adjusts our estimates of ECS so that the calculated percentiles occur at higher values.

Section 3.2 states "We tested our approach of calculating ECS utilizing the EM-GC to CMIP6 GCMs by altering the EM-GC framework to include CMIP6 output (see the supplement for details)". To use the EM-GC framework with the CMIP6 output, we calculated the CMIP6 multi-model mean change in GMST from 1850-2100 using the SSP2-4.5 scenario. We used the standard deviation of the CMIP6 multi-model mean to represent the uncertainty in the rise in GMST for our reduced chi-squared calculations. We trained the EM-GC from 1850-2100, included the CMIP6 prescribed values of SAOD and TSI, and did not include any natural variability, since effects on GMST due to factors such as ENSO should be randomly distributed within the various CMIP6 GCM runs that constitute the CMIP6 multi-model mean. We used the average of the five (Fig. S13a) and the Cheng OHC (Fig. S13b) records to calculate the amount of heat exported to the world's oceans. Our results in Fig. S13a,b show $\Delta T_{2100}$ for the values of AER RF$_{2011}$ and $\lambda_\Sigma$ that the EM-GC finds good fits to the CMIP6 multi-model GMST output for SSP2-4.5. Figure S13c shows the 5th, 25th, 50th, 75th, and 95th percentile values of ECS for the Gregory method and the altered EM-GC framework with the Cheng

OHC record and the average of the five OHC records. The comparison of ECS found using quite different approaches, illustrated in Figure S13, provides strong support for the veracity of ECS inferred from GCMs and from the climate record throughout our analysis.

170

[Figure]

**Figure S1.** GMST anomaly in 2100 relative to pre-industrial ($\Delta T_{2100}$) as a function of climate feedback parameter and AER $RF_{2011}$ for two versions of the EM-GC. (a) The change in $\Delta T_{2100}$ for SSP4-3.4 using the original formulation of Eq. (2) where $Q_{OCEAN}$ is subtracted outside of the feedback multiplicative term. (b) The change in $\Delta T_{2100}$ for SSP4-3.4 using the updated formulation of Eq. (2) where $Q_{OCEAN}$ is subtracted within the feedback multiplicative term similar to Bony et al. (2006) and Schwartz (2012). The EM-GC is able to fit higher values of $\lambda_\Sigma$ at strong aerosol cooling (around $-1.5$ W m$^{-2}$) for the new Eq. (2) compared to the original formulation in Canty et al. (2013) and Hope et al. (2017). The maximum value of future warming has increased due to the higher $\lambda_\Sigma$ values.

180

[Figure]

**Figure S2.** $\Delta T_{2100}$ as a function of climate feedback parameter and tropospheric aerosol radiative forcing in 2011 using the EM-GC for SSP4-3.4. (a) Training period of 1850-1989. The region outside of the AER RF$_{2011}$ range provided by IPCC 2013 is shaded (grey). Colors denote the GMST change in year 2100 relative to pre-industrial. The color bar is the same across all four panels for comparison. (b) Training period of 1850-1999. (c) Training period of 1850-2009. (d) Training period of 1850-2019, which is the normal training period used in our analysis.

[Figure]

**Figure S3.** Measured and modeled GMST anomaly (ΔT) relative to a pre-industrial (1850-1900) baseline. (a) Observed from CW14 (black) and modeled (red) ΔT from 1850-2019. This panel also displays the values of $\lambda_\Sigma$ and $\chi^2_{ATM}$ (see text) for this best-fit simulation. (b) Contributions from total human activity. This panel also denotes the numerical value of the attributable anthropogenic warming rate from 1975-2014 (black dashed) as well as the $2\sigma$ uncertainty in the slope for the best estimate of AER $RF_{2011}$ of $-0.9$ W m⁻². (c) Solar irradiance (light blue) and major volcanoes (purple). (d) Influences from ENSO on ΔT. (e) Contributions from AMOC to ΔT and to observed warming from 1975-2014. (f) Influences from PDO (blue) and IOD (pink) on ΔT. (g) Measured (black) and modeled (red) ocean heat content (OHC) as a function of time for the Cheng et at (2017) OHC record, the value of $\chi^2_{OCEAN}$ for this run, as well as the ocean heat uptake efficiency, κ, needed to provide the best-fit to the OHC record. The error bars (blue) denote the uncertainty in OHC used in this analysis (see Sect. 2.2.8).

[Figure]

**Figure S4.** GMST anomaly relative to pre-industrial over time. (a) HadCRUT with the HadCRUT uncertainties. (b) CW14 with the CW14 uncertainties. (c) BEG with the BEG uncertainties. (d) CW14 with the HadCRUT uncertainties.

[Figure]

Figure S2S5. Radiative forcing of tropospheric ozone for the various SSPs analyzed in our study. The time series labeled SSP1-2.6, SSP2-4.5, SSP4-6.0, and SSP5-8.5 are from the corresponding RCP scenarios. We created the time series from SSP4-3.4 and SSP3-7.0 using linear combinations of the SSP1-2.6 and SSP5-8.5 time series.

190

[Figure]

**Figure S6.** Measured and modeled GMST anomaly (ΔT) relative to a pre-industrial (1850-1900) baseline for an AER $RF_{2011}$ = −0.1 W m$^{-2}$ and −1.5 W m$^{-2}$. (a) Observed (black) and modeled (red) ΔT from 1850-2019. This panel also displays the values of $\lambda_\Sigma$ and $\chi^2_{ATM}$ (see text) for this best-fit simulation. (b) Contributions from total human activity. This panel also denotes the numerical value of the attributable anthropogenic warming rate from 1975-2014 (black dashed) as well as the 2σ uncertainty in the slope. (c) Solar irradiance (light blue) and major volcanoes (purple). (d) Influences from ENSO on ΔT. (e) Contributions from AMOC to ΔT and to observed warming from 1975-2014. (f) Influences from PDO (blue) and IOD (pink) on ΔT. (g) Measured (black) and modeled (red) ocean heat content (OHC) as a function of time for the average of five data sets (see text), the value of $\chi^2_{OCEAN}$ for this run, as well as the ocean heat uptake efficiency, κ, needed to provide the best-fit to the OHC record. The error bars (blue) denote the uncertainty in OHC used in this analysis (see Sect. 2.2.8). (h)-(n) Same as (a)-(g), except for AER $RF_{2011}$ = −1.5 W m$^{-2}$.

[Figure]

**Figure S7.** Radiative forcing time series due to tropospheric aerosols. (a) The RF time series due to tropospheric aerosols for SSP1-2.6. The solid grey circle denotes the value of AER $RF_{2011}$ given by the SSP database. The solid grey lined labeled the $-1.0$ W m$^{-2}$ time series is the AER RF time series given by the SSP database for SSP1-2.6. We appended a historical AER RF time series from the RCP scenarios and created five additional AER RF time series as described in Sect. 2.2.4. (b) Anthropogenic aerosol radiative forcing time series for SSP4-3.4.

[Figure]

200    **Figure** S8**.** Ocean heat content time series. The five ocean heat content data records used in this analysis, normalized to the year 1986 because this year is in the middle of the average time series. The grey shaded region is the combined uncertainty estimate used in this analysis, centered around the average of the five data sets. The average of the ocean heat content records (1955 – 2017) is computed when there are three or more data sets available for a given year.

[Figure]

205

**Figure** S9. The ocean heat content records and uncertainty estimates analyzed in this study. (a) The average OHC record along with the standard deviation of the mean represented by the dotted black line, and the combined uncertainty of the 1-sigma standard deviation of the  average of the five OHC records and the Cheng et al. (2017) estimates shown as the dashed black line. (b) Balmaseda OHC record with the standard deviation of the five ORSA ensemble members as the dotted line, and the combined uncertainty as the dashed line. (c) Levitus OHC record with the standard error as the native uncertainty, and the combined uncertainty. (d) Carton OHC record with the standard deviation of the mean of multiple ensemble members, and the combined uncertainty. (e) Cheng OHC record with the 1σ native uncertainty and the combined uncertainty. (f) Ishii OHC record with the combined uncertainty as the dashed line.

[Figure]

215

**Figure S10.** The change in GMST relative to 1961-1990 from the CMIP6 GCMs and the contribution from SAOD from 1975-2014. (a) The change in GMST from the 50 CMIP6 GCMs. (b) The residual in the change of GMST from the 50 CMIP6 GCMs after subtracting the contribution of SAOD determined by the updated REG method. The median value of AAWR is written on this panel and plotted in red. (c) The human component of global warming, $\Delta T_{ATM,HUMAN}$, from the EM-GC. A linear fit (black) and quadratic fit (red) are plotted on top to show that $\Delta T_{ATM,HUMAN}$ is almost exactly linear. (d) The contribution of SAOD in the 50 CMIP6 GCMs using a lag month calculated for each model.

[Figure]

**Figure S11.** Values of AAWR for 50 CMIP6 GCMs using the LIN and REG methods. The solid black line is the 1:1 line and the vertical and horizontal dashed lines are the maximum value of AAWR determined using the EM-GC and the HadCRUT temperature record. The CMIP6 GCMs that have values of AAWR less than the maximum value from the EM-GC are blue, and the CMIP6 GCMs that have values of AAWR greater than the maximum value from the EM-GC are red. The slope, $1\sigma$ standard deviation, and $R^2$ of the values of AAWR from the CMIP6 GCMs are shown.

**Table S1.** Values of AAWR calculated using the EM-GC as a function of start and end year. The value of AAWR from 1975-2014 is shown in red. Each model run uses the best estimate of AER RF$_{2011}$ (−0.9 W m$^{-2}$), and the average of five OHC records, and the HadCRUT GMST record. The impact on varying the start and end year on AAWR is slight, except when a short record is used (i.e. 1984-2004, a 21 year span). A two-decade time span is not long enough to calculate an accurate estimate of AAWR. The value of AAWR is more sensitive to the choice of OHC or temperature record used than the chosen time span.

220

**Start Year**

| AAWR (°C/decade) | 1970 | 1973 | 1975 | 1979 | 1982 | 1984 |
|---|---|---|---|---|---|---|
| **2004** | 0.154 ± 0.006 | 0.153 ± 0.007 | 0.153 ± 0.008 | 0.145 ± 0.009 | 0.138 ± 0.010 | 0.130 ± 0.010 |
| **2006** | 0.150 ± 0.006 | 0.149 ± 0.007 | 0.149 ± 0.008 | 0.141 ± 0.009 | 0.134 ± 0.009 | 0.126 ± 0.009 |
| **2008** | 0.148 ± 0.006 | 0.146 ± 0.006 | 0.146 ± 0.007 | 0.138 ± 0.008 | 0.131 ± 0.008 | 0.124 ± 0.007 |
| **2010** | 0.147 ± 0.005 | 0.145 ± 0.006 | 0.144 ± 0.007 | 0.137 ± 0.007 | 0.131 ± 0.007 | 0.125 ± 0.006 |
| **2012** | 0.146 ± 0.005 | 0.144 ± 0.005 | 0.144 ± 0.006 | 0.137 ± 0.006 | 0.132 ± 0.006 | 0.128 ± 0.006 |
| **2014** | 0.146 ± 0.004 | 0.145 ± 0.005 | 0.144 ± 0.005 | 0.139 ± 0.005 | 0.134 ± 0.006 | 0.130 ± 0.005 |
| **2016** | 0.147 ± 0.004 | 0.145 ± 0.004 | 0.145 ± 0.005 | 0.140 ± 0.005 | 0.137 ± 0.005 | 0.134 ± 0.005 |
| **2018** | 0.147 ± 0.003 | 0.146 ± 0.004 | 0.146 ± 0.004 | 0.142 ± 0.005 | 0.139 ± 0.005 | 0.137 ± 0.005 |

(**End Year** labels the leftmost column of the table)

225

**Table S2.** Average values of AAWR calculated from the CMIP6 multi-model results using the regression method as a function of start and end year. The uncertainty corresponds to the 1σ standard deviation of AAWR found from the 50 GCMs. The value of AAWR from 1975-2014 is shown in red. The values of AAWR from the CMIP6 multi-model ensemble is more sensitive to the choice of start and end year than the EM-GC due to the small number of models. We use the same start and end year, 1975-2014, for the determination of AAWR for both the EM-GC and the CMIP6 multi-model ensemble for consistency.

| | AAWR (°C/decade) | Start Year | | | | | |
|---|---|---|---|---|---|---|---|
| | | **1970** | **1973** | **1975** | **1979** | **1982** | **1984** |
| **End Year** | **2004** | 0.185 | 0.196 | 0.200 | 0.208 | 0.224 | 0.230 |
| | **2006** | 0.192 | 0.203 | 0.207 | 0.216 | 0.232 | 0.238 |
| | **2008** | 0.196 | 0.207 | 0.211 | 0.220 | 0.234 | 0.241 |
| | **2010** | 0.200 | 0.209 | 0.214 | 0.222 | 0.236 | 0.241 |
| | **2012** | 0.204 | 0.213 | 0.218 | 0.226 | 0.239 | 0.244 |
| | **2014** | 0.208 | 0.217 | 0.222 | 0.230 | 0.242 | 0.247 |

230

[Figure]

**Figure S12.** Steps for the calculation of ECS using the Gregory et al. (2004) method,  using GISS-E2-1-H (Kelley et al., 2020) as an example. (a) The change in Abrupt 4×CO₂ GMST (variable: tas) from the piControl experiment for 150 years. (b) Abrupt 4×CO₂ net downward radiative flux (variable: rtmt) versus the Abrupt 4×CO₂ GMST change from the piControl experiment for 150 years. The x-intercept of the orthogonal linear least squares fit of the GCM output shown in panel (b), divided by two yields the equilibrium climate sensitivity, which in this case is 3.09°C.

240  **Table S3.** Values of AAWR from 1975-2014 for the 50 CMIP6 multi-model Historical simulations available at time of the analysis (April 2020) for both the REG and LIN methods. The asterisk symbol (*) indicates there is only one run used to compute the value of AAWR for that GCM. No asterisk indicates the AAWR value shown in the table is the average of the values of AAWR for all runs of that model. The average ratio of LIN to REG for all 50 models is 1.009 ± 0.015, shown at the bottom of the table and in Fig. S11. The

245  correlation coefficient (r²) of 0.995 is also shown. We conclude our determination of AAWR from the CMIP6 multi-model ensemble is accurate to ±1%, which is much smaller than the difference between the CMIP6 multi-model ensemble values of AAWR and those found using the EM-GC framework.

| Model | AAWR, REG (°C/decade) | AAWR, LIN (°C/decade) | Model | AAWR, REG (°C/decade) | AAWR, LIN (°C/decade) |
|---|---|---|---|---|---|
| ACCESS-CM2 | 0.211 | 0.216 | GFDL-CM4* | 0.243 | 0.250 |
| ACCESS-ESM1-5 | 0.238 | 0.246 | GFDL-ESM4 | 0.203 | 0.224 |
| AWI-CM-1-1-MR | 0.215 | 0.220 | GISS-E2-1-G | 0.194 | 0.198 |
| BCC-CSM2-MR | 0.217 | 0.228 | GISS-E2-1-G-CC | 0.204 | 0.213 |
| BCC-ESM1 | 0.241 | 0.249 | GISS-E2-1-H | 0.237 | 0.244 |
| CAMS-CSM1-0 | 0.131 | 0.138 | HadGEM3-GC31-LL | 0.283 | 0.292 |
| CanESM5 | 0.354 | 0.361 | HadGEM3-GC31-MM | 0.227 | 0.234 |
| CanESM5-CanOE | 0.323 | 0.334 | INM-CM4-8* | 0.173 | 0.181 |
| CAS-ESM2-0 | 0.196 | 0.204 | INM-CM5-0 | 0.146 | 0.156 |
| CESM2 | 0.240 | 0.243 | IPSL-CM6A-LR | 0.230 | 0.236 |
| CESM2-FV2 | 0.221 | 0.224 | KACE-1-0-G | 0.254 | 0.260 |
| CESM2-WACCM | 0.273 | 0.291 | MCM-UA-1-0 | 0.225 | 0.231 |
| CESM2-WACCM-FV2 | 0.231 | 0.235 | MIROC6 | 0.157 | 0.168 |
| CIESM | 0.245 | 0.251 | MIROC-ES2L | 0.163 | 0.167 |
| CNRM-CM6-1 | 0.202 | 0.196 | MPI-ESM1-2-HAM | 0.180 | 0.186 |
| CNRM-CM6-1-HR* | 0.172 | 0.178 | MPI-ESM1-2-HR | 0.195 | 0.203 |
| CNRM-ESM2-1 | 0.170 | 0.172 | MPI-ESM1-2-LR | 0.192 | 0.197 |
| E3SM-1-0 | 0.267 | 0.278 | MRI-ESM2-0 | 0.203 | 0.210 |
| E3SM-1-1* | 0.283 | 0.285 | NESM3 | 0.242 | 0.253 |
| E3SM-1-1-ECA* | 0.275 | 0.274 | NorCPM1 | 0.180 | 0.185 |
| EC-Earth3* | 0.299 | 0.310 | NorESM2-LM | 0.167 | 0.182 |
| EC-Earth3-Veg* | 0.214 | 0.223 | NorESM2-MM* | 0.151 | 0.154 |
| FGOALS-f3-L | 0.218 | 0.226 | SAM0-UNICON* | 0.245 | 0.250 |
| FGOALS-g3 | 0.176 | 0.191 | TaiESM1* | 0.273 | 0.283 |

| FIO-ESM-2-0 | 0.229 | 0.237 | UKESM1-0-LL | 0.299 | 0.312 |

Ratio = 1.009 ± 0.015          $R^2 = 0.$995

[Figure]

**Figure S13.** GMST anomaly in 2100 relative to pre-industrial ($\Delta T_{2100}$) as a function of climate feedback parameter and AER RF$_{2011}$ and the values of ECS from the CMIP6 GCMs using three methods. (a) $\Delta T_{2100}$ for SSP2-4.5 using the CMIP6 multi-model mean and the average of the five OHC records. (b) $\Delta T_{2100}$ for SSP2-4.5 using the CMIP6 multi-model mean and the Cheng et al. (2017) OHC record. (c) Values of ECS found using the Gregory et al. (2004) method (red), CMIP6 multi-model mean using the Cheng et al. (2017) OHC (orange), and the CMIP6 multi-model mean using the average of the five OHC records. The box represents the 25th 50th, and 75th percentiles of the values of ECS and the whiskers denote the 5th and 95th percentiles. The stars indicate the minimum and maximum values of ECS.

250

[Figure]

**Figure S7S14.** Values of ECS found using the EM-GC and the CMIP6 multi-model ensemble without the aerosol weighting method. Values of ECS utilizing the EM-GC are calculated using four temperature data sets and five ocean heat content records (as indicated). The box represents the 25th, 50th, and 75th percentiles of the values of ECS and the whiskers denote the 5th and 95th percentiles for the different OHC records and each temperature record without using the aerosol weighting method (unweighted). The stars indicate the minimum and maximum values of ECS. The circles are the values of ECS associated with the best estimate of AER RF$_{2011}$ of −0.9 W m$^{-2}$. The box labeled CMIP6 is the 25th, 50th, and 75th percentiles of the values of ECS from the CMIP6 multi-model ensemble, the whiskers indicate the 5th and 95th percentiles, and the stars represent the minimum and maximum values of ECS from the CMIP6 multi-model ensemble.

**Table S4.** Equilibrium climate sensitivity (ECS) from 28 CMIP6 GCMs. We can only calculate ECS for GCMs that provide Abrupt 4×CO$_2$ near surface air temperature (output variable: tas), net downward radiative flux (output variable: rtmt), and piControl near surface air temperature (output variable: tas) to the CMIP6 archive at time of the analysis (April 2020). All estimates are for one model run except for CanESM5, which is the average of two runs.

| Model | ECS (K) |
| --- | --- |
| ACCESS-CM2 | 4.93 |
| ACCESS-ESM1-5 | 3.63 |
| BCC-CSM2-MR | 3.16 |
| BCC-ESM1 | 3.74 |
| CanESM5 | 5.70 |
| CESM2 | 5.32 |
| CESM2-FV2 | 5.06 |
| CESM2-WACCM | 4.73 |
| CESM2-WACCM-FV2 | 4.56 |
| E3SM-1-0 | 5.28 |
| EC-Earth3-Veg | 4.34 |
| GFDL-CM4 | 3.78 |
| GFDL-ESM4 | 2.61 |
| GISS-E2-1-G | 2.71 |
| GISS-E2-2-G | 2.25 |
| GISS-E2-1-H | 3.09 |
| HadGEM3-GC31-LL | 5.65 |
| INM-CM4-8 | 2.32 |
| INM-CM5-0 | 2.39 |
| IPSL-CM6A-LR | 4.97 |
| MCM-UA-1-0 | 3.68 |
| MIROC6 | 2.84 |
| MIROC-ES2L | 2.83 |
| NorESM2-LM | 2.19 |
| NorESM2-MM | 2.15 |
| SAM0-UNICON | 3.53 |
| TaiESM1 | 4.33 |
| UKESM1-0-LL | 5.40 |

[Figure]

**Figure S15.** GMST anomaly in 2100 from pre-industrial ($\Delta T_{2100}$) as a function of climate feedback parameter and AER $RF_{2011}$. (a) $\Delta T_{2100}$ for SSP4-6.0. The region outside of the tropospheric aerosol radiative forcing rage provided by IPCC 2013 (Myhre et al., 2013) is shaded grey. Colors denote the change in $\Delta T_{2100}$. (b) $\Delta T_{2100}$ for SSP3-7.0. (c) $\Delta T_{2100}$ for SSP5-8.5.

270

[Figure]

275

**Figure S16.** Probabilistic forecasts of future projections of ΔT using the EM-GC for the SSP4-6.0, SSP3-7.0, and SSP5-8.5 scenarios. (a) Future projections of ΔT for SSP4-6.0. Observations (orange) are from CRU. The IPCC 2013 likely range of warming (black) is from Figure 11.25b of chapter 11 of the IPCC 2013 report. The Paris Agreement target and upper limit (yellow) are shown for comparison to projections of ΔT using the EM-GC. The CMIP6 minimum, multi-model mean, and maximum values of the rise in ΔT are shown to compare to projections from the EM-GC. Colors denote the probability of reaching at least that temperature by the end of the century and are computed using the aerosol weighting method (see Sect. 2.5). (b) Future projections of ΔT for SSP3-7.0. (c) Future projections of ΔT for SSP5-8.5.

285

[Figure]

**Figure S17.** Probability density functions (PDF) for the increase in $\Delta T_{2100}$ using the EM-GC and the CMIP6 multi-model ensemble. (a) PDF for EM-GC (blue) results and CMIP6 multi-model results (red) for SSP4-6.0. The left-hand y-axis is for EM-GC probabilities and the righthand y-axis is for GCM probabilities. (b) PDF for SSP3-7.0. (c) PDF for SSP5-8.5.

[Figure]

290

**Figure** S11S18. Blended methane mixing ratios. The dotted lines are linear combinations of the time series of methane abundances using SSP1-2.6 and SSP3-7.0 to span the range of values of future methane. The solid lines are the SSP1-2.6 and SSP3-7.0 methane mixing ratio time series.

 **Table S5.** Details of the CMIP6 GCMs used in this study.

| Institution | Model | Model Output |
| --- | --- | --- |
| AS-RCEC | TaiESM1 | No reference provided |
| AWI | AWI-CM-1-1-MR | (Semmler et al., 2018a, 2018b, 2018c, 2019a, 2019b) |
| BCC | BCC-CSM2-MR | (Wu et al., 2018a, 2018b, 2018c; Xin et al., 2019a, 2019b, 2019c, 2019d) |
| | BCC-ESM1 | (Zhang et al., 2018a, 2018b, 2019) |
| CAMS | CAMS-CSM1-0 | (Rong, 2019a, 2019b, 2019c, 2019d, 2019e, 2019f) |
| CAS | CAS-ESM2-0 | (Chai, 2019) |
| | FGOALS-f3-L | (YU, 2019a, 2019b, 2019c, 2019d, 2019e) |
| | FGOALS-g3 | (Li, 2019a, 2019b, 2019c, 2019d, 2019e) |
| CCCma | CanESM5 | (Swart et al., 2019f, 2019g, 2019h, 2019i, 2019j, 2019k, 2019l, 2019m, 2019n, 2019o) |
| | CanESM5-CanOE | (Swart et al., 2019a, 2019b, 2019c, 2019d, 2019e) |
| CNRM-CERFACS | CNRM-CM6-1 | (Voldoire, 2018, 2019c, 2019d, 2019e, 2019f) |
| | CNRM-CM6-1-HR | (Voldoire, 2019a, 2019b, 2020a, 2020b) |
| | CNRM-ESM2-1 | (Seferian, 2018; Voldoire, 2019g, 2019h, 2019i, 2019j, 2019k, 2019l) |
| CSIRO | ACCESS-ESM1-5 | (Ziehn et al., 2019a, 2019b, 2019c, 2019d, 2019e, 2019f, 2019g) |
| CSIRO-ARCCSS | ACCESS-CM2 | (Dix et al., 2019a, 2019b, 2019c, 2019d, 2019e, 2019f, 2019g) |
| E3SM-Project | E3SM-1-0 | (Bader et al., 2018, 2019a, 2019b) |
| | E3SM-1-1-ECA | (Bader et al., 2020) |
| E3SM-Project RUBISCO | E3SM-1-1 | (Bader et al., 2019c) |
| EC-Earth-Consortium | EC-Earth3 | (EC-Earth Consortium (EC-Earth), 2019i, 2019j, 2019k, 2019l, 2019m) |
| | EC-Earth3-Veg | (EC-Earth Consortium (EC-Earth), 2019a, 2019b, 2019c, 2019d, 2019e, 2019f, 2019g, 2019h) |

| | | |
|---|---|---|
| FIO-QLNM | FIO-ESM-2-0 | (Song et al., 2019a, 2019b, 2019c, 2019d) |
| HAMMOZ-Consortium | MPI-ESM1-2-HAM | (Neubauer et al., 2019) |
| INM | INM-CM4-8 | (Volodin et al., 2019a, 2019b, 2019c, 2019d, 2019e, 2019f, 2019g) |
| | INM-CM5-0 | (Volodin et al., 2019m, 2019h, 2019n, 2019i, 2019j, 2019k, 2019l) |
| IPSL | IPSL-CM6A-LR | (Boucher et al., 2018a, 2018b, 2018c, 2019a, 2019b, 2019c, 2019d, 2019e, 2019f, 2019g) |
| MIROC | MIROC6 | (Shiogama et al., 2019a, 2019b, 2019c, 2019d, 2019e, 2019f, 2019g; Tatebe and Watanabe, 2018a, 2018b, 2018c) |
| | MIROC-ES2L | (Hajima et al., 2019; Tachiiri et al., 2019a, 2019b, 2019c, 2019d, 2019e) |
| MOHC | HadGEM3-GC31-MM | (Ridley et al., 2019c) |
| MOHC NERC | HadGEM3-GC31-LL | (Good, 2019, 2020a, 2020b; Ridley et al., 2018, 2019a, 2019b) |
| MOHC, NERC, NIMS-KMA, NIWA | UKESM1-0-LL | (Byun, 2020; Good et al., 2019a, 2019b, 2019c, 2019d, 2019e, 2019f; Tang et al., 2019a, 2019b, 2019c) |
| MPI-M AWI | MPI-ESM1-2-LR | (Wieners et al., 2019a, 2019b, 2019c, 2019d, 2019e) |
| MPI-M DWD DKRZ | MPI-ESM1-2-HR | (Jungclaus et al., 2019; Schupfner et al., 2019a, 2019b, 2019c, 2019d; Steger et al., 2019) |
| MRI | MRI-ESM2-0 | (Yukimoto et al., 2019a, 2019b, 2019c, 2019d, 2019e, 2019f, 2019g, 2019h) |
| NASA-GISS | GISS-E2-1-G | (NASA Goddard Institute for Space Studies (NASA/GISS), 2018a, 2018b, 2018c, 2020a, 2020b, 2020c, 2020d) |
| | GISS-E2-1-G-CC | No reference provided |
| | GISS-E2-2-G | (NASA Goddard Institute for Space Studies (NASA/GISS), 2019a) |
| | GISS-E2-1-H | (NASA Goddard Institute for Space Studies (NASA/GISS), 2018d, 2019b, 2019c) |
| NCAR | CESM2-WACCM-FV2 | (Danabasoglu, 2019d, 2019e, 2020a) |

| | CESM2 | (Danabasoglu, 2019c, 2019d, 2019e, 2019f, 2019g, 2019h; Danabasoglu et al., 2019) |
|---|---|---|
| | CESM2-FV2 | (Danabasoglu, 2019b, 2019c, 2020b) |
| | CESM2-WACCM | (Danabasoglu, 2019f, 2019g, 2019h, 2019a, 2019i, 2019j, 2019k) |
| | NorCPM1 | (Bethke et al., 2019a, 2019b, 2019c) |
| NCC | NorESM2-LM | (Seland et al., 2019a, 2019b, 2019c, 2019d, 2019e, 2019f, 2019g) |
| | NorESM2-MM | (Bentsen et al., 2019a, 2019b, 2019c, 2019d, 2019e, 2019f, 2019g) |
| NIMS-KMA | KACE-1-0-G | (Byun et al., 2019a, 2019b, 2019c, 2019d, 2019e) |
| NOAA-GFDL | GFDL-CM4 | (Guo et al., 2018a, 2018b, 2018c, 2018d, 2018e) |
| | GFDL-ESM4 | (John et al., 2018a, 2018b, 2018c, 2018d, 2018e; Krasting et al., 2018a, 2018b, 2018c) |
| NUIST | NESM3 | (Cao, 2019a, 2019b, 2019c; Cao and Wang, 2019) |
| SNU | SAM0-UNICON | (Park and Shin, 2019a, 2019b, 2019c) |
| THU | CIESM | (Huang, 2019a, 2019b, 2020a, 2020b) |
| UA | MCM-UA-1-0 | (Stouffer, 2019a, 2019b, 2019c, 2019d, 2019e, 2019f, 2019g) |

300

---

## Referee Report (RR1)

The major comments from my previous review were about the data used, the assumption of a constant feedback parameter and the length/prose of the paper. While the authors have tried to address all, I am not yet satisfied with the  solutions and I recommend another round of major revisions. I still believe the manuscript will become a valuable piece is the discussion of climate sensitivity, as it provides a comprehensive overview of modes of internal variability.

Major comments:

1. The authors still use HadCRUT4 as if it were a global average. They justified this by citing other papers that have done the same. The Nicholls paper seems to make the same mistake as the authors. However, the other paper does not. Liang et al. (2020) use HadCRUT4, but they do take into account that it is not a global average and that there is missing data, especially around the poles.  They use a mask of the CMIP output so that the spatial coverage of the datasets is the same. This takes some work to implement, so I suggest the authors choose any of the four datasets with global coverage. Instead of comparing HadCRUT4 with CW14 (f.i. in Table 1, Figure 12), the authors can compare CW14 with NOAAGlobalTemp v5. Dropping HadCRUT4 from the manuscript completely also helps making the paper shorter.

   HadCRUT4 uses HadSST3 for sea surface temperatures, which further shows slower warming due to biases in ship measurements in comparison with HadSST4. As I understand it, even the incomplete (not infilled) provisional version of HadCRUT5 shows more warming than HadCRUT4. CW14 uses HadSST3 as well, potentially explaining why it warms more slowly compared to some other global averages.

2. The authors have developed an application of EM-GC with blended observations, but temperature output of CMIP6 to test whether EM-GC has predictive power for future temperatures. This is not quite what I intended with my comment, but I admit I wasn't clear before. I had hoped the authors would develop a pure model-based test of predictive power. The outcome of the blended result shows that EM-CG often underestimates ECS, but the authors claim in the body of the text that it is a very good predictor.

3. The authors now examine a time-varying feedback parameter, which varies with radiative forcing. They do not give justification for why they integrate a time-varying feedback like that. Global feedback is thought to change because of cloud feedbacks above a slow-changing ocean. A delay of a couple of decades between radiative forcing and the change of feedback is therefore expected. Disregarding the physics lead to a biased outcome, as the model is trying to fit the rise in the feedbacks too early, and it is only natural that would fail. Scaling with RF would mean that there is barely any feedback in the first half of the twentieth century, which is also unphysical.

   Different formulations for time-varying global feedback exist for simple models, such as (Armour et al., 2013; Geoffroy et al., 2013; Goodwin, 2018). All of these formulations have in common that the feedback only changes some time after radiative forcing, with different lags. I think the Goodwin approach is most suitable for adjustment into EM-GC. Goodwin, also using a data-driven model, shows that the upper range of climate sensitivity is extremely sensitive to the time-scale.

   I further believe that getting an optimal global constant by fitting, and then adjusting the

model to include time-varying feedbacks will tend to favour the former. Ideally, the fitting is done simultaneously.

4. I don't see how the authors determined the uncertainty around the carbon cycle. I cannot find a mention of 10% of Friedlingstein (which concluded that emission-driven simulations warm a tad more than concentration-driven simulations in CMIP5). Ten percent seems low, but this is not my expertise.

5. The paper is still quite long. In the minor comments I will make another set of suggestions to make the paper easier to understand. This will not be an exhaustive list. There are good guides on the internet for writing concisely, that have helped me become a better writer. For instance: https://writingcenter.gmu.edu/guides/writing-concisely.
The EM-GC model does not model the carbon cycle explicitly, and discussion of the carbon cycle may also be an option to remove. I don't see the value of showing all SSPs in f.i. Figure 9. Consider dropping those with few CMIP6 models.

Geoffroy, O., Saint-Martin, D., Bellon, G., Voldoire, A., Olivié, D. J. L., & Tytéca, S. (2013). Transient climate response in a two-layer energy-balance model. Part II: Representation of the efficacy of deep-ocean heat uptake and validation for CMIP5 AOGCMs. Journal of Climate, 26(6), 1859–1876. https://doi.org/10.1175/JCLI-D-12-00196.1

Goodwin, P. (2018). On the Time Evolution of Climate Sensitivity and Future Warming. Earth's Future, 6(9), 1336–1348. https://doi.org/10.1029/2018EF000889

Armour, K. C., Bitz, C. M., & Roe, G. H. (2013). Time-Varying Climate Sensitivity from Regional Feedbacks. Journal of Climate, 26(13), 4518–4534. https://doi.org/10.1175/JCLI-D-12-00544.1

Minor comments:
79: Replace 'to designate future' with 'for the': future and scenarios are redundant
101: 'land-use change': check hyphens throughout the entire paper
131: remove 'of climate'
132: remove 'because', start new sentence at 'this'
142: consider removing 'Bony et al.' sentence, I don't see the use
150: due to this update, our model is
186: which update
202-205: long sentence
209: 'that is our primary data source', maybe replace with: 'which we use as default'
220: rung→panel
235-237: unnecessary sentence
240: reword: for this simulation, kappa =1.28, W/m^2/C fits the OHC data best
242: remove 'the' before 'IOD'
243: remove 'temporal variations in'
245: slight -> small
343: remove 'consequently'
347: remove 'multiplicative': factor is by definition multiplicative
348: split sentence after '2015'
354: remove 'thus'
367: remove sentence, already clear
379: remove 'scientific': what else?
408: consider replacing 'upon' with 'on' throughout: make it easy for your reviewers and readers to read your text

419: consider using the improved HadSST4, which removes biases in the ship measurements.

420: remove 'variations in the strength'?

421: I'm not sure whether it's appropriate to detrend using RF. Temperature lags RF quite a bit, especially in oceans.

433: remove everything between brackets

435: surely the numbers are altered. I cannot imagine that the feedback parameter isn't dependant on AMOC in the fit.

438: consider using 'use' throughout instead of 'utilising'

453: is this old factor still valid?

455: remove sentence 'since … whole atmosphere', redundant.

459: remove 'temporal'

481: remove 'however'

505: equal to→of

510: upon consideration of→by including

534: colouring seems to be off in figure S10

539: remove 'the computation of'

552: remove sentence, redundant

Section 3.1: move methodology to methodology section 2.2.1 (the bit about blending)

Figure 8: what interval is plotted for each study?

772: changed word order, it seems like we're coupling a two-box model to 2.6

793: Cox et al. based on CMIP5

834: remove 'indicated on each plot', redundant

834-835: remove sentence, the reader will know how to do a global average

858: I don't think bimodality is clear here. There seems to be outliers, but not two roughly equal-sized groups of models. With so few models, passing any statistical test on bimodality would be tough. Drop it?

863: remove 'apparent in figure 9', redundant

918: remove 'our', redundant

Figure 12: choose bigger bin size: CMIP models displayed weirdly

934: three significant digits not justified, two better

Table 1: same

991-1003: you seem to be repeating the table, making the prose difficult to read, condense to half the size?

1015: since -> from / from … onwards

1023: I don't think either of them studied the entire climate system. Instead, those studies were about the atmosphere.

---

## Author Response (AR2)

Reviewer comments in black, author response in blue

The major comments from my previous review were about the data used, the assumption of a constant feedback parameter and the length/prose of the paper. While the authors have tried to address all, I am not yet satisfied with the solutions and I recommend another round of major revisions. I still believe the manuscript will become a valuable piece in the discussion of climate sensitivity, as it provides a comprehensive overview of modes of internal variability.

Major comments:
1. The authors still use HadCRUT4 as if it were a global average. They justified this by citing other papers that have done the same. The Nicholls paper seems to make the same mistake as the authors. However, the other paper does not. Liang et al. (2020) use HadCRUT4, but they do take into account that it is not a global average and that there is missing data, especially around the poles. They use a mask of the CMIP output so that the spatial coverage of the datasets is the same. This takes some work to implement, so I suggest the authors choose any of the four datasets with global coverage. Instead of comparing HadCRUT4 with CW14 (f.i. in Table 1, Figure 12), the authors can compare CW14 with NOAAGlobalTemp v5. Dropping HadCRUT4 from the manuscript completely also helps making the paper shorter.

HadCRUT4 uses HadSST3 for sea surface temperatures, which further shows slower warming due to biases in ship measurements in comparison with HadSST4. As I understand it, even the incomplete (not infilled) provisional version of HadCRUT5 shows more warming than HadCRUT4. CW14 uses HadSST3 as well, potentially explaining why it warms more slowly compared to some other global averages.

We thank the reviewer for suggesting we change our primary data set in our manuscript. We have decided to use the infilled, global version of HadCRUT5 throughout as our main data set, which was released in late 2020. Switching from HadCRUT4 to HadCRUT5 changes the quantitative details of our results in an important manner, as HadCRUT5 exhibits more warming over the past few decades than HadCRUT4. However, we still find that many of the CMIP6 GCMs warm too quickly over the past few decades compared to empirical determination of the attributable anthropogenic warming rate (AAWR), and that many of the GCMs exhibit higher values of ECS compared to our empirical determination. Upon use of the HadCRUT5 data record, we now find that 7 of the 28 CMIP6 GCMs for which we can compute AAWR and ECS lie within the minimum and maximum values of our empirical determination. This overlap is an incredibly important new result that follows from our adoption of the HadCRUT5 data record and is illustrated in New Fig. S17. We added the following text to the main manuscript on lines 854-862 to explain this new result: **We conclude this section by commenting on the relationship between ECS and AAWR in our model framework. Eight of the CMIP6 GCMs (GFDL-ESM4, GISS-E2-1-G, INM-CM5-0, INM-CM4-8, MIROC6, MIROC-ES2L, NorESM2-LM, and NorESM2-MM) exhibit values of ECS and AAWR consistent with the minimum and maximum estimates based on our EM-GC constrained by the HadCRUT5 GMST record (Table S5 and Fig. S17). An analysis of the relationship between AAWR and ECS from the CMIP6 GCMs illustrates that 78% of the variance in ECS among the 28 CMIP6 GCMs that provide both quantities is explained by AAWR (see Fig. S17). This result indicates CMIP6 GCMs that accurately simulate the rise in observed ΔT over the past few decades exhibit values of ECS that are in line with our empirically based estimate.**

[Figure]

**New Figure S17.** Values of ECS versus AAWR for the CMIP6 multi-model ensemble. The EM-GC estimates of AAWR and ECS based on training to the HadCRUT5 GMST record are plotted as a box and whisker. The box shows the average 25th, 50th, and 75th percentiles for the five OHC records shown for HadCRUT5 in Fig. 6 and Fig. 7. The whiskers represent the average 5th and 95th percentiles. The stars denote the average minimum and maximum values of AAWR or ECS. The eight CMIP6 GCMs that obtain values of AAWR and ECS that are both within the minimum and maximum estimates provided by the EM-GC are identified on the figure. Values of AAWR explain about 78% of the variance in ECS among the CMIP6 GCMs.

Upon switching to HadCRUT5, we noticed the uncertainties for the GMST anomaly decreased dramatically compared to those provided with the HadCRUT4 record (New Figure S4c versus S4e). The difference is especially noticeable in the beginning of the temperature record. If we use the HadCRUT5 uncertainties in our current model framework, the EM-GC is not able to achieve a value less than or equal to 2 for $\chi^2_{ATM}$ or $\chi^2_{RECENT}$. The fits between the observed and modeled GMST are visually good fits, but the values of $\chi^2$ that are calculated never fall below 2 due to the incredibly small uncertainties of the HadCRUT5 record. We could either increase our constraint limit to admit simulations with values of $\chi^2$ less than or equal to 4 or 8, adjust the HadCRUT5 uncertainties in some manner to be more "realistic", or use the HadCRUT4 uncertainties for the HadCRUT5 temperature record. After much internal debate and considerable analysis not shown in this reply, we decided to adopt the HadCRUT4 uncertainties for all of the GMST records. Our deep dive into the GMST records, highlighted in New Figs. S4, S5 and New Table 1, have led us to conclude the uncertainties for GMST provided by HadCRUT4 are more realistic than those provided by HadCRUT5. As explained below, the comparisons are complicated by the provision of GMST anomalies for different baselines by various data centers. Finally, our adoption of HadCRUT4 has a great benefit of allowing the uncertainty in all of the quantities found using data from various centers, such as AAWR, ECS, and the probability of achieving certain global warming limits, to now be treated in a consistent manner.

We have added the following text to the supplement beginning on line 66 to explain our use of the HadCRUT4 uncertainties for all of the GMST data records. **Figure S4 shows values of ΔT for the seven individual GMST records (GISTEMP, BEG, HadCRUT4, CW14, HadCRUT5,**

NOAAGT, and JMA) with their corresponding 1σ and 2σ uncertainties. A horizontal line at zero denotes the time period of the baseline for each record. The multi-record mean, excluding the data set that is plotted, is also shown. Since the multi-record mean and individual ΔT record are plotted on the same baseline, the data sets closely match over this time period. Panels (a), (b), (e), and (f) illustrate that the uncertainties for these GMST records are not large enough to encompass the multi-record mean over 1850-2019. The multi-record mean in panel (a) is below the GISTEMP 1σ uncertainty range between 1880 and 1900, and again between 1980 to 2019. In panel (b), the multi-record mean is above the BEG 1σ range from 1850 until 1865, 1880 to 1895, and below the 1σ uncertainty range from 2000 to 2019. The multi-record mean in panel (e) is below the HadCRUT5 1σ uncertainty range from 1990 until 2019. In panel (f), the multi-record mean is above the NOAAGT 1σ uncertainty range from 1920 until 1955. The JMA GMST record does not provide an uncertainty estimate. We therefore use the HadCRUT4 uncertainty estimate for JMA in panel (g). The multi-record mean lies at the edge of the 1σ uncertainty range from 1891 until 2000. After 2000, the multi-record mean falls above the 1σ and 2σ HadCRUT4 uncertainty range. The HadCRUT4 uncertainty time series shown in panel (c) is the only uncertainty estimate large enough to cover the spread in the various GMST records.

Figure S5 shows ΔT based on all seven GMST records and the multi-record mean relative to three baseline periods. The 1σ and 2σ uncertainties from HadCRUT4 are plotted about the multi-record mean. Panels (a) and (d) show the GMST records relative to 1891-1920, which are the first 30 years all of the data sets have in common. Between 1850-1970, all of the data sets fall within the 1σ uncertainty. After 1970, the GMST records start to deviate and some fall outside of the 1σ uncertainty and remain within the 2σ uncertainty, except for JMA which falls outside of the 2σ uncertainty. Panels (b) and (e) show the GMST records relative to the HadCRUT baseline period of 1961-1990. We see similar behavior as in panels (a) and (d), where the GMST records largely fall within the 1σ uncertainty until about 1970. Panels (c) and (f) show the GMST records forced to match HadCRUT5 from 2010-2019, which is baselined to 1961-1990. In these two panels, we see a large spread between the GMST records from the beginning of the time period until 2005.

Table S1 shows the percentage of ΔT annual values since 1940 from all seven data records that lie within the 1σ and 2σ of the multi-record mean, found using the HadCRUT4 uncertainties. Year 1940 is used to be consistent with the definition of our $\chi^2_{RECENT}$ parameter. Depending on the choice of baseline period, the number of data points within the uncertainty range varies. For a baseline of 1891-1920, 80% of the data points for all seven records are within the 1σ uncertainty and 95% of the data points are within the 2σ limit. For a baseline of 1961-1990, 88% and 93% of data points are within the 1σ and 2σ HadCRUT4 uncertainties, respectively. If the ΔT records are forced to match the average value of the HadCRUT5 data set over the last decade, 72% of the data points are within the 1σ uncertainty and 88% are within the 2σ uncertainty. This analysis shows that depending on which baseline is used, the percentage of points within the 1σ or 2σ uncertainty ranges varies. Overall, these comparisons support the utility of the HadCRUT4 uncertainty for GMST, since the 1σ and 2σ uncertainty ranges capture a percentage of points approximately correct for a pure Gaussian distribution. Therefore, we have adopted the HadCRUT4 uncertainties in GMST for all of the analyses in the main paper. The uncertainties published by other data centers tend to be smaller than the HadCRUT4 uncertainties. Since only the HadCRUT4 uncertainties span the range of values for ΔT from the seven data records in a somewhat realistic fashion, we have decided to use these uncertainties uniformly throughout the analysis.

[Figure]

**New Figure S4.** Annual GMST (ΔT) anomaly for seven data records relative to their individual baseline and the multi-record mean. The multi-record mean does not include the data set that is being shown. The 1σ and 2σ uncertainties for each GMST record are shown, and the horizontal like for ΔT=0 spans the baseline used for the specific panel. (a) GISTEMP. (b) BEG. (c) HadCRUT4. (d) CW14. (e) HadCRUT5. (f) NOAAGT. (g) JMA. Since the JMA data provider does not provide an uncertainty time series, the HadCRUT4 uncertainty is used.

[Figure]

**New Figure S5.** Annual GMST anomaly relative to several baseline periods for seven data records. The 1σ (shaded grey) and 2σ (dotted grey) HadCRUT4 uncertainties are plotted about the multi-model record mean (black). (a) Baseline of 1891-1920 plotted from 1850-2019. (b) Same as (a) using a baseline of 1961-1990. (c) Same as (a) except all of the ΔT records are forced to match the average ΔT anomaly over 2010-2019 given by HadCRUT5 that is relative to 1961-1990. (d) – (f) Same as (a) – (c) except plotted from 1940-2019.

**New Table S1.** Percentage of annual values between 1940-2019 of the GMST record within the 1σ or 2σ HadCRUT4 uncertainties about the multi-record mean for each baseline period.

| **Baseline: 1891-1920** | 1σ | | 2σ | |
|---|---|---|---|---|
| | $N_{WITHIN}$ | $N_{TOTAL}$ | $N_{WITHIN}$ | $N_{TOTAL}$ |
| HadCRUT4 | 77 = 96% | 80 | 80 = 100% | 80 |
| HadCRUT5 | 42 = 53% | 80 | 80 = 100% | 80 |
| CW14 | 80 = 100% | 80 | 80 = 100% | 80 |
| BEG | 71 = 89% | 80 | 80 = 100% | 80 |
| GISTEMP | 73 = 91% | 80 | 80 = 100% | 80 |
| NOAAGT | 76 = 95% | 80 | 80 = 100% | 80 |
| JMA | 29 = 36% | 80 | 54 = 68% | 80 |
| **AVERAGE** | **80%** | | **95%** | |

| Baseline: 1961-1990 | | | | |
|---|---|---|---|---|
| HadCRUT4 | 80 = 100% | 80 | 80 = 100% | 80 |
| HadCRUT5 | 68 = 85% | 80 | 80 = 100% | 80 |
| CW14 | 80 = 100% | 80 | 80 = 100% | 80 |
| BEG | 80 = 100% | 80 | 80 = 100% | 80 |
| GISTEMP | 75 = 94% | 80 | 80 = 100% | 80 |
| NOAAGT | 80 = 100% | 80 | 80 = 100% | 80 |
| JMA | 27 = 34% | 80 | 48 = 60% | 80 |
| **AVERAGE** | **88%** | | **93%** | |

| Match 2010-2019 | | | | |
|---|---|---|---|---|
| HadCRUT4 | 68 = 86% | 80 | 80 = 100% | 80 |
| HadCRUT5 | 47 = 59% | 80 | 86 = 95% | 80 |
| CW14 | 78 = 98% | 80 | 80 = 100% | 80 |
| BEG | 77 = 96% | 80 | 80 = 100% | 80 |
| GISTEMP | 47 = 59% | 80 | 79 = 99% | 80 |
| NOAAGT | 73 = 91% | 80 | 80 = 100% | 80 |
| JMA | 11 = 14% | 80 | 18 = 23% | 80 |
| **AVERAGE** | **72%** | | **88%** | |

We have included New Figs. S4, S5, and New Table S1 in the supplement to provide an explanation for why we use the HadCRUT4 uncertainties for all seven GMST data records. We added the following text on lines 270-273 of the main manuscript to direct the reader to the supplement for the explanation for using the HadCRUT4 uncertainties: **We use the uncertainty time series from HadCRUT4 for all GMST records because the HadCRUT4 uncertainty provides a realistic description of the variation in GMST among the seven records (see the supplement, Figs. S4 and S5, and Table S1 for more information).**

2. The authors have developed an application of EM-GC with blended observations, but temperature output of CMIP6 to test whether EM-GC has predictive power for future temperatures. This is not quite what I intended with my comment, but I admit I wasn't clear before. I had hoped the authors would develop a pure model-based test of predictive power. The outcome of the blended result shows that EM-CG often underestimates ECS, but the authors claim in the body of the text that it is a very good predictor.

We thank the reviewer for elaborating on their previous comment. Since our application of using the EM-GC with the CMIP6 multi-model output was not what the reviewer was referring to, we have removed this analysis from the supplement and the corresponding text from the main manuscript, which has made this revised paper shorter than the prior version.

In response to the prior review from the other reviewer, we created a model-based test of predictive power by altering the training period of the GMST to forecast future ΔT. We had, in the prior revision, included Fig. S2 of the supplement to make the point that altering the training period of our model has a small effect on our results

For this revision, we have modified Fig. S2 to show results for training of our model for various periods of time and included New Fig. S3 to address the reviewer's suggestion that we "develop a pure model-based test of predictive power". Both figures are included below, for convenience.

We project relatively similar results for end of century warming for training periods that start in 1850 and end in either 2009 or 2019. The "shape" of our model parameter space is similar for training periods ending in 1999, 2009, or 2019. The training period that ends in 1989 (Fig. S2a) yields a different "shape" of model parameter space for which good fits to the climate record can be obtained, compared to the other training periods. The different shape for this shorter training period is due to the formulation of the ocean component of our model. In training to 1989, we are only considering 35 years of the observed OHC record. We are able to calculate good fits to the OHC record over this shorter time period that diverge from the OHC record after 1989. Also, for this shorter time period, aerosol radiative forcing of climate cools in a manner that nearly mirrors the warming due to rising GHGs, resulting in a wider range of model parameters that lead to a "good fit" of the climate record, compared to model simulations constrained by data that extend closer to present-day. The highest values of $\Delta T_{2100}$ in Fig. S2a are associated with the largest values of $\lambda_\Sigma$, which in our model corresponds to excessively high values of $\kappa$ that we can rule out, based on OHC data collected during 1990 to 2019.

We added New Fig. S3 and the following text on lines 41-52 of the supplement to illustrate the predictive power of the EM-GC: **Figure S3 shows the observed (HadCRUT5) and modeled $\Delta T$ anomaly from 1945-2060 for the four different training periods described above. Each panel contains three projections of future $\Delta T$ for SSP4-3.4: projection using the value of climate feedback that provides the best fit to the historical climate record for a value of AER RF$_{2011}$ = −0.9 W m$^{-2}$, the lowest value of climate feedback that provides a good fit to the observed $\Delta T$ record for a value of AER RF$_{2011}$ = −0.1 W m$^{-2}$, and the highest value of climate feedback that provides a good fit to the historical climate record (the associated value of AER RF$_{2011}$ varies depending on the training period). As more years are added to the training period, the range of projection for future temperature decreases (Fig. S3a vs S3d). All of the best fit projections (solid line) and highest value of climate feedback (upper dashed line) closely follow the mid-point of the data, regardless of the training period. Given the nature of this test (i.e., predicting GMST out to 2019 for a series of trainings that stop in either 1989, 1999, or 2009), Figure S3 supports the quantitative accuracy of our approach for simulating and projecting future $\Delta T$.**

[Figure]

**Figure S2.** $\Delta T_{2100}$ as a function of climate feedback parameter and tropospheric aerosol radiative forcing in 2011 using the EM-GC trained with the HadCRUT5 $\Delta T$ record for SSP4-3.4. (a) Training period of 1850-1989. The region outside of the AER $RF_{2011}$ range provided by IPCC 2013 is shaded (grey). Colors denote the GMST change in year 2100 relative to pre-industrial. The color bar is the same across all four panels for comparison. (b) Training period of 1850-1999. (c) Training period of 1850-2009. (d) Training period of 1850-2019, which is the normal training period used in our analysis.

[Figure]

**New Figure S3.** Observed and modeled GMST anomaly relative to 1850-1900 from 1945-2060 for four training periods. (a) Observations from HadCRUT5 (black), the EM-GC GMST anomaly simulation for a training period of 1850-1989 (orange) of HadCRUT5, and the EM-GC projections for SSP4-3.4 out to 2060. Three EM-GC projections are shown in red: The best estimate of climate feedback for AER $RF_{2011}$ = −0.9 W m$^{-2}$, the lowest value of climate feedback that satisfies the $\chi^2$ constraints for AER $RF_{2011}$ = -0.1 W m$^{-2}$, and the highest value of climate feedback that satisfies the $\chi^2$ constraints (value of AER $RF_{2011}$ varies depending on training period). The IPCC 2013 likely range of warming is denoted as the black trapezoid. (b) Training period of 1850-1999. (c) Training period of 1850-2009. (d) Training period of 1850-2019.

3. The authors now examine a time-varying feedback parameter, which varies with radiative forcing. They do not give justification for why they integrate a time-varying feedback like that. Global feedback is thought to change because of cloud feedbacks above a slow- changing ocean. A delay of a couple of decades between radiative forcing and the change of feedback is therefore expected. Disregarding the physics lead to a biased outcome, as the model is trying to fit the rise in the feedbacks too early, and it is only natural that would fail. Scaling with RF would mean that there is barely any feedback in the first half of the twentieth century, which is also unphysical.

Different formulations for time-varying global feedback exist for simple models, such as (Armour et al., 2013; Geoffroy et al., 2013; Goodwin, 2018). All of these formulations have in common that the feedback only changes some time after radiative forcing, with different lags. I think the Goodwin approach is most suitable for adjustment into EM-GC. Goodwin, also using a data-driven model, shows that the upper range of climate sensitivity is extremely sensitive to the time-scale. I further believe that getting an optimal global constant by fitting, and then adjusting the model to include time-varying feedbacks will tend to favour the former. Ideally, the fitting is done simultaneously.

> We thank the reviewer for this suggestion on how to improve our analysis of a time-varying climate feedback. We examined the Goodwin (2018) study as suggested. In the Goodwin (2018) analysis, he separated the radiative forcing due to greenhouse gases into several terms, such as the radiative forcing due to $CO_2$, $CH_4$, and $N_2O$. Goodwin (2018) also partitioned the climate feedback into several terms based on the length of time it would take for the various feedbacks to respond to a change in radiative forcing.

> We have applied the idea from Goodwin (2018) that there is a delay in the response of the climate feedback due to a change in radiative forcing within our model framework. In the revised paper, we now incorporate a 20-year delay between the change in radiative forcing and our new calculation of the time variant lambda time series. We chose the 20-year delay because this delay is included in Table 1 of Goodwin (2018) as an estimate of the adjustment timescale between a change in radiative forcing and the cloud-spatial SST adjustment feedback. This feedback is the longest delay between the radiative forcing and climate feedback. The other forms of climate feedback such as water and lapse rate, fast cloud feedback, and snow and sea ice albedo feedback occur with response delays of several days to several months (Goodwin, 2018).

> New Figure 14 shows the change in observed and modeled GMST under four assumptions regarding $\lambda^{-1}$. First, the value of $\lambda^{-1}$ is constant over time (New Figs. 14a, e). Second, the value of $\lambda^{-1}$ varies by 50% between 1850-2100 (New Figs. 14b, f). The third assumption involves $\lambda^{-1}$ varying over time while $\chi^2_{RECENT}$ is always less than or equal to two (New Figs 14c, g). Fourth, $\lambda^{-1}$ varies over time while $\chi^2_{ATM}$ is always less than or equal to two (New Figs. 14d, h). The 20-year delay results in better fits to the observed change in GMST for a 50% increase in lambda (New Fig. 14b) than if the instantaneous response is used (New Fig. S23b). The 20-year delay allows larger variations in lambda over time that still lie below our reduced chi squared constraints (New Fig. 14c, d versus New Fig. 23 c, d). If there truly is time varying climate feedback that responds to a change in radiative forcing with a 20-year delay, then our projections of future GMST may increase up to 1.5°C above the estimates obtained assuming time invariant feedback. If climate feedback varies with a 20-year delay due to the change in radiative forcing and rises over time as suggested by some of the CMIP6 (Rugenstein et al., 2020) and CMIP5 GCMs (Marvel et al., 2018), our projections of global warming would be a few tenths of a degree warmer than our current best estimate, as indicated by the difference between the red line and black circle in 2100 for New Fig. 14b. If we changed the 20 year delay to the shorter delays used in Goodwin (2018), than our results would be between those from the instantaneous response of climate feedback to a change in RF to the 20-year delay in response of climate feedback to a change in RF.

[Figure]

**New Figure 14.** Change in GMST from 1850-1900 for observations from HadCRUT5 (black) and 1850-2100 for modeled (red) using SSP4-3.4 and the residual between modeled and observations incorporating a 20 year delay between $\lambda^{-1}$ and a change in RF. The black circles denote the amount of warming when $\lambda^{-1}$ is time invariant. (a) Rise in GMST assuming a constant value of $\lambda^{-1}$. (b) Rise in GMST allowing $\lambda^{-1}$ to increase by 50%. (c) Rise in GMST allowing $\lambda^{-1}$ to vary while the value of $\chi^2_{RECENT}$ is kept below 2. (d) Rise in GMST allowing $\lambda^{-1}$ to vary while the value of $\chi^2_{ATM}$ is kept below 2. (e) Residual between modeled and observed rise in GMST from 1850-2019 for constant $\lambda^{-1}$. (f) Same as (e) but for increasing $\lambda^{-1}$ by 50%. (g) Same as (f) but for varying $\lambda^{-1}$ while the value of $\chi^2_{RECENT}$ is kept below 2. (h) same as (g) but for varying $\lambda^{-1}$ while the value of $\chi^2_{ATM}$ is kept below 2.

[Figure]

**New Figure S23.** Change in $\Delta T$ from 1850-2019 for observations from HadCRUT5 (black) and 1850-2100 for modeled (red) using SSP4-3.4 and the residual between modeled and observations using an instantaneous time variant $\lambda^{-1}$. (a) Rise in GMST assuming a constant value of $\lambda^{-1}$. (b) Rise in GMST allowing $\lambda^{-1}$ to increase by 50%. (c) Rise in GMST allowing $\lambda^{-1}$ to vary while the value of $\chi^2_{RECENT}$ is kept below 2. (d) Rise in GMST allowing $\lambda^{-1}$ to vary while the value of $\chi^2_{ATM}$ is kept below 2. (e) Residual between modeled and observed rise in GMST from 1850-2019 for constant $\lambda^{-1}$. (f) Same as (e) but for increasing $\lambda^{-1}$ by 50%. (g) Same as (f) but for varying $\lambda^{-1}$ while the value of $\chi^2_{RECENT}$ is kept below 2. (h) same as (g)but for varying $\lambda^{-1}$ while the value of $\chi^2_{ATM}$ is kept below 2.

We have updated Fig. 14 in our revised manuscript to include the 20-year delay between the change in radiative forcing and the time varying climate feedback. We added the following text starting at line 1160 to the main manuscript to explain our new method: **In all cases for time varying feedback, we also assume the value of $\lambda^{-1}$ has the same shape as the SSP4-3.4 RF time series along with a lag of 20 years and that the new time series for $\lambda^{-1}$ maintains an average value over the observational record identical to the constant value for $\lambda^{-1}$ of 0.63 °C / W m$^{-2}$. We chose a lag of 20 years to represent the longest delay in response of climate feedback to a change in RF suggested by Goodwin (2018). If we use the shorter delays represented in Goodwin (2018), then our results would be between those from the instantaneous response of climate feedback to a change in RF (Fig. S23) and the 20-year delay. Finally, in the simulations described below, the value of $\lambda^{-1}$ is assumed to continue to rise into the future at the same proportionality to $\Delta T_{ATM,HUMAN}$ as the prior increase.**

We included New Fig. S23 to show how our results differ if we use an instantaneous response of λ⁻¹ to the change in RF.

4. I don't see how the authors determined the uncertainty around the carbon cycle. I cannot find a mention of 10% of Friedlingstein (which concluded that emission-driven simulations warm a tad more than concentration-driven simulations in CMIP5). Ten percent seems low, but this is not my expertise.

Thank you for this comment and alerting us that we need to better describe the uncertainty around the carbon cycle. The uncertainty we consider for our estimates of transient climate response to cumulative emissions (TCRE) is much larger than 10%. The largest variation in our estimates of TCRE is driven by the uncertainty in AER RF. This uncertainty is incorporated into the probability of achieving the Paris Agreement target and upper limit through the aerosol weighting method. New Figure S21 shows the rise in ΔT from pre-industrial for SSP5-8.5 versus the cumulative emissions of $CO_2$, in Gt C, since 1870. The colored lines denote the probability of reaching at least that temperature by the end of century. The large spread in projections of future ΔT is driven by the uncertainty in AER RF. The computed probabilities are based on the aerosol weighting method, so the uncertainty in AER RF is considered when determining the likelihood of achieving the Paris Agreement target of 1.5°C and upper limit of 2.0°C.

We also incorporate the uncertainty in how atmospheric $CO_2$ will respond to the prescribed carbon emissions, for the overall uncertainty of TCRE. We examined Fig. 2 and Table 3 (both included below) from Friedlingstein et al. 2014 and determined that the multi-model average of $CO_2$ concentrations in 2100 from emission driven runs of CMIP5 coupled carbon cycle models was 985 ppm with a standard deviation of 97 ppm, which is about 10% of the average. We used this 10% value to represent the 1σ uncertainty in the response of atmospheric $CO_2$ to prescribed carbon emissions, which is a component of the overall uncertainty in TCRE. We also examined Fig. 9b from Murphy et al., 2014 (below) to calculate an estimate in the uncertainty of emissions driven runs of the coupled carbon climate models from CMIP5. We also estimated an uncertainty of about 10%.

Finally, based on our desire to be sure 10% was realistic for this portion of the overall uncertainty in TCRE, we examined the very highly cited, albeit older study by Friedlingstein et al. 2006. Their Figure 1a (below) shows the rise in atmospheric $CO_2$ over time simulated by 11 coupled atmospheric / carbon cycle models. The mean estimate of atmospheric $CO_2$ in 2100 determined by reading values from the figure is 850 ppm with a 1σ sigma uncertainty of 89 ppm. Consequently, we again find a value close to 10%. Even though an uncertainty of 10% may seem low, this numerical value has been determined by three independent studies.

The 10% uncertainty is included in our determination of the carbon budgets for each probability of achieving the Paris Agreement target and upper limit. We have updated the main text of the manuscript and the supplement to better describe the uncertainties in the carbon cycle within the EM-GC framework. We have updated the following text on lines 1047-1049 of the main manuscript to better describe the uncertainties in the carbon cycle within the EM-GC framework: **The largest variation in our carbon budget estimates is driven by the uncertainty in AER RF, which is incorporated into the probability of achieving the Paris Agreement target and upper limit (see Fig. S21 and the supplement).**

We have added the following text beginning on line 277 of the supplement to further explain the uncertainties in our carbon budget estimates: **Figure S21 shows the rise in ΔT from pre-industrial for SSP5-8.5 versus the cumulative emissions of $CO_2$, in Gt C, since 1870. The colored lines denote the probability of reaching at least that temperature by the end of century. The large spread in projections of future ΔT is driven by the uncertainty in AER RF. The computed probabilities are based on the aerosol weighting method, so the uncertainty in AER RF is considered when determining the likelihood of achieving the Paris Agreement target of 1.5°C and upper limit of 2.0°C for the cumulative carbon emissions.**

**We use the uncertainty suggested by coupled atmospheric / carbon cycle models in how atmospheric $CO_2$ will respond to the prescribed carbon emissions. Examination of Fig. 2 and Table 3 from Friedlingstein et al. (2014) and Fig. 9b from Murphy et al. (2014) led to our determination that the uncertainty in estimates of atmospheric $CO_2$ from emissions driven runs of CMIP5 coupled atmospheric / carbon cycle models is about 10% (1σ). We include this 10% uncertainty in our determination of the carbon budgets for each probability of achieving the Paris Agreement target and upper limit shown in Table 2.**

[Figure]

**New Figure S21.** Transient climate response to cumulative $CO_2$ emissions for SSP5-8.5 using the EM-GC. Simulations of the rise in ΔT versus cumulative $CO_2$ emissions in units of Gt C. The orange line is observations of ΔT from HadCRUT5 plotted against cumulative carbon emissions from the Global Carbon Budget project (Friedlingstein et al., 2019). The dotted and dashed lines denote the Paris Agreement target and upper limit, respectively. The EM-GC projections represent the probability that the future value of ΔT will rise to the indicated level, considering only acceptable fits to the climate record. The probabilities were determined using the aerosol weighting method. The light grey, dark grey, and black curves denote the 95, 66, and 50% probabilities of either the Paris target (intersection of dotted horizontal lines) or upper limit (intersection of dashed lines with curves) being achieved.

[Figure]

FIG. 2. Range of (a) simulated atmospheric $CO_2$ (ppm) and (b) global surface temperature change (K) from the 11 ESMs E-driven (blue lines) and C-driven (red lines) simulations. Also shown is the full range of (c) simulated atmospheric $CO_2$ (ppm) and (d) global surface temperature change (K) simulated by MAGICC6 when emulating all 19 CMIP3 climate models and 10 $C^4$MIP climate–carbon cycle models. The red-line curve in (a) and (c) is the baseline estimate from MAGICC6.

TABLE 3. Twenty-first-century atmospheric $CO_2$ (2100), global surface warming (2081–99 relative to 1986–2005), cumulative land and ocean uptake (1850–2100) for the E-driven simulations and global surface warming (2081–99 relative to 1986–2005) for the C-driven simulations (where atmospheric $CO_2$ reaches 941 ppm by 2100). Also shown are the multimodel mean and range (1$\sigma$) as well as the same quantities simulated by MAGICC6 in its reference setting.

| | E-driven $CO_2$ (ppm) | E-driven delta $T$ (°C) | E-driven cumulative land C uptake (PgC) | E-driven cumulative ocean C uptake (PgC) | C-driven delta $T$ (°C) |
|---|---|---|---|---|---|
| CanESM2 | 1048 | 5.0 | 161 | 455 | 4.5 |
| GFDL-ESM2G | 997 | 2.9 | 167 | 550 | 2.8 |
| HadGEM2-ES | 998 | 4.3 | 352 | 543 | 4.7 |
| IPSL-CM5A-LR | 926 | 4.5 | 300 | 555 | 4.5 |
| MIROC-ESM | 1149 | 5.6 | −165 | 544 | 4.7 |
| MPI-ESM-LR | 969 | 3.7 | 231 | 412 | 3.6 |
| CESM1-BGC | 1142 | 4.1 | −145 | 541 | 3.6 |
| NorESM1-M | 934 | 3.8 | −173 | 649 | 3.4 |
| BCC-CSM-1* | 967 | 3.5 | 471* | 490 | 3.3 |
| INM-CM4.0* | 914 | 2.5 | 201 | 861 | 2.6 |
| MRI-ESM1* | 794 | 2.9 | 758 | 528 | 3.3 |
| Models average | 985 ± 97 | 3.9 ± 0.9 | 91 ± 218** | 557 ± 112 | 3.7 ± 0.7 |
| MAGICC6 | 941 | 4.0 | 204 | 617 | 4.0 |

\* $F_{Ln}$ estimated as no simulated LUC carbon flux in these ESMs.

\*\* Multimodel average for land carbon is only based on the eight ESMs simulating $F_L$. HadGEM2-ES and GFDL-ESM2G simulations start in 1860 and 1861, respectively. Note that BCC-CSM-1 simulations end in 2099; the 2099 − 2098 atmospheric $CO_2$ difference was used to infer atmospheric $CO_2$ by 2100.

Murphy et al., 2014

[Figure]

**Fig. 9 a** Future global mean surface air temperature (SAT) changes for 2080–2099 relative to 1980–1999 for members of the ESPPE and CMIP3 ensembles, plotted against corresponding historical changes for 1980–1999 relative to 1900–1949. Future forcing is from the A1B scenario, with $CO_2$ changes prescribed as emissions in the ESPPE, and atmospheric concentrations in CMIP3. *Grey shading* indicates the range for the median plus or minus two standard deviations of the observed historical change in SAT, obtained using 100 alternative realisations from the HadCRUT4 dataset. ESPPE model variants are colour-coded by simulated historical change in SAT. **b** Average historical and future $CO_2$ concentrations corresponding to the ESPPE simulations in **a**. The average observed global mean $CO_2$ concentration for the period 1980–1999 is indicated by the *dotted line* (Masarie and Tans 1995). **c** The average fraction of emitted $CO_2$ remaining in the atmosphere for the periods 1980–1999 and 2080–2099 for the ESPPE simulations in **a**. *Grey shading* corresponds shows the median plus or minus two standard deviation uncertainty range for average airborne $CO_2$ fraction for 1980–1999, derived from the observational estimates of Sabine et al. (2004)

Friedlingstein et al., 2006

5. The paper is still quite long. In the minor comments I will make another set of suggestions to make the paper easier to understand. This will not be an exhaustive list. There are good guides on the internet for writing concisely, that have helped me become a better writer. For instance: https://writingcenter.gmu.edu/guides/writing-concisely.

> We thank the reviewer for pointing us to some guides on writing concisely. We agree that the paper is long and have attempted to shorten the manuscript. We moved the description of the Gregory et al. (2004) method for computing ECS from the CMIP6 GCMs in Sect. 2.4 to the supplement. We added a sentence on lines 593-594 of the supplement: **For the estimate of ECS from the CMIP6**

**multi-model ensemble, we use the method described by Gregory et al. (2004) (See the supplement and Fig. 15 for more information).**

We removed some of the discussion of Table 1 and Table 2. We also moved Fig. S13 and the description of Fig. 13 to the supplement. In the main paper, we added a sentence on lines 1026-1027 to refer the reader to the supplement for Fig. 13 (which is now Fig. S21): **We use the probabilistic forecasts in Fig. S21 to determine the carbon budgets in Table 2.**

6. The EM-GC model does not model the carbon cycle explicitly, and discussion of the carbon cycle may also be an option to remove. I don't see the value of showing all SSPs in f.i. Figure 9. Consider dropping those with few CMIP6 models.

Discussion of the carbon cycle in lines 987-992 of the previous version of the revised paper (included here for reference) was added upon request by the reviewer: **Examination of (Friedlingstein et al., (2014); and Murphy et al., (2014) led to our determination that the uncertainty in estimates of atmospheric CO2 from emissions driven runs of CMIP5 coupled atmospheric / carbon cycle models is about 10% (1-sigma). We therefore use 10% as the uncertainty in how atmospheric CO2 will respond to the prescribed carbon emissions. We apply the 10% uncertainty estimate to the future remaining carbon budget.**

We have shortened the text and added more information into supplement. We have added the following text on lines 1049-1053 to the main manuscript: **We include a 10% uncertainty, determined from examination of CMIP5 coupled atmospheric / carbon cycle models from Friedlingstein et al. (2014) and Murphy et al. (2014) (see the supplement for more information), within each probability of attaining the Paris goals to represent how atmospheric CO$_2$ will respond to the prescribed carbon emissions.**

See our response to the previous comment on the information we added to the supplement to address the uncertainties in our carbon budget estimates.

We would like to keep all four panels of Fig. 9, because they illustrate the CMIP6 GMST projections for the four SSP scenarios analyzed in the main part of the manuscript. The multi-model mean, minimum, and maximum displayed in this figure for the four SSPs are shown again in Fig. 11. We would like to retain all panels of Fig. 9 so that the reader can see how we derived the CMIP6 multi-model mean, minimum, and maximum shown in Fig. 11.

Geoffroy, O., Saint-Martin, D., Bellon, G., Voldoire, A., Olivié, D. J. L., & Tytéca, S. (2013). Transient climate response in a two-layer energy-balance model. Part II: Representation of the efficacy of deep-ocean heat uptake and validation for CMIP5 AOGCMs. Journal of Climate, 26(6), 1859–1876. https://doi.org/10.1175/JCLI-D-12-00196.1

Goodwin, P. (2018). On the Time Evolution of Climate Sensitivity and Future Warming. Earth's Future, 6(9), 1336–1348. https://doi.org/10.1029/2018EF000889

Armour, K. C., Bitz, C. M., & Roe, G. H. (2013). Time-Varying Climate Sensitivity from Regional Feedbacks. Journal of Climate, 26(13), 4518–4534. https://doi.org/10.1175/JCLI-D-12-00544.1

Minor comments:

79: Replace 'to designate future' with 'for the': future and scenarios are redundant

Change made.

101: 'land-use change': check hyphens throughout the entire paper

We have ensured all instances of land-use change include a hyphen.

131: remove 'of climate'

Change made.

132: remove 'because', start new sentence at 'this'

Change made.

142: consider removing 'Bony et al.' sentence, I don't see the use

The sentence has been removed.

150: due to this update, our model is

Change made.

186: which update

We have modified this paragraph based on our update to use HadCRUT5 as the main data set, so the corresponding sentence has been removed.

202-205: long sentence

We have split this sentence into two separate sentences. The new sentences are as follows: **The equation for all three formulations of $\chi^2$ is based on annual averages, rather than monthly time series. We calculate $\chi^2$ with annual values because the autocorrelation functions of $\Delta T_{OBS}$ and $\Delta T_{MDL}$ display similar shapes using annual averages, and do not match utilizing monthly averages (see supplement of Canty et al. (2013) for further explanation).**

209: 'that is our primary data source', maybe replace with: 'which we use as default'

We have changed the sentence to be: **The average of five OHC data sets, which we use as our primary OHC series, extends from 1955-2017, a total of 63 years.**

220: rung→panel

Change made.

235-237: unnecessary sentence

This sentence does seem unnecessary as written, but it is referring to the dotted black line on Fig. 1e. We have changed the sentence to make it clear what we are referring to. The new sentence is: **Furthermore, the contribution of AMOC to the rise in GMST over 1975-2014 (the same time period used to define AAWR) is also specified on Fig. 1e (dotted black line).**

240: reword: for this simulation, kappa =1.28, W/m^2/C fits the OHC data best

Change made.

242: remove 'the' before 'IOD'

Change made.

243: remove 'temporal variations in'

Change made.

245: slight -> small

Change made.

343: remove 'consequently'

Change made.

347: remove 'multiplicative': factor is by definition multiplicative

Change made.

348: split sentence after '2015'

Change made.

354: remove 'thus'

Change made.

367: remove sentence, already clear

Change made.

379: remove 'scientific': what else?

We have removed the words "slight" and "scientific".

408: consider replacing 'upon' with 'on' throughout: make it easy for your reviewers and readers to read your text

Change made.

419: consider using the improved HadSST4, which removes biases in the ship measurements.

We will replace HadSST3 with HadSST4 to derive the AMOC time series used in the regression, since we are using HadCRUT5 as our primary data set.

420: remove 'variations in the strength'?

Change made.

421: I'm not sure whether it's appropriate to detrend using RF. Temperature lags RF quite a bit, especially in oceans.

Thank you for this inquiry. The Atlantic multidecadal variability (AMV) that we use as a proxy for AMOC is the change in SST between the equator and 60° N. The SSTs represent the upper ocean, or top 100 m. The response of the upper ocean to a change in radiative forcing is almost instantaneous, so a time delay is not needed.

Several studies (Mann and Emanuel, 2006; Ting et al., 2009; Trenberth and Shea, 2006) are critical of using linear detrending to remove the forced trend in the AMV index. This method assumes that the forced trend is linear over time, which may not be correct. Ting et al. (2009) say that the use of the linear detrending method of the AMV index may result in including a global warming signal into the index. All three studies suggest using a detrending method of regressing SST against the AMV index to remove the forced trend over time. We derived an anthropogenic detrending method to remove the influence of anthropogenic activities on SST in the North Atlantic. We achieve similar results if we use the anthropogenic or SST detrending methods (Response Fig. 1) but get different results if we use the linear detrending method.

[Figure]

**Response Figure 1.** GMST anomaly in 2100 from pre-industrial as a function of climate feedback parameter and AER RF$_{2011}$ for SSP4-3.4. (a) AMV index was detrended using the anthropogenic detrending option. (b) AMV index was detrended using the SST detrending method. (c) AMV index was detrended using the linear detrending method.

Response Figure 2 shows that some of the global warming signal is probably being aliased into the AMOC signal when using the linear detrending method. The AMOC contribution of the rise in GMST from 1975-2014 is 0.025°C/decade when using the anthropogenic detrending method but is 0.037°C when using the linear detrending method. The value of AAWR decreases from 0.167 to 0.160°C/decade upon switching from the anthropogenic to linear detrending method.

[Figure]

**Response Figure 2.** Measured and modeled GMST anomaly (ΔT) relative to a pre-industrial (1850-1900) baseline. (a) Observed (black) and modeled (red) ΔT from 1850-2019. (b) Contributions from total human activity. This panel also denotes the best estimate value of the attributable anthropogenic warming rate from 1975-2014 (black dashed) as well as the 2σ uncertainty in the slope for a model run that uses the best estimate of AER $RF_{2011}$ of −0.9 W m$^{-2}$. (c) TSI (purple) and SAOD (light blue). (d) Influences from ENSO on ΔT. (e) Contributions from AMOC to ΔT and to observed warming from 1975-2014 using the Linear detrending option. (f) Influences from PDO (blue) and IOD (pink) on ΔT. (g) Measured (black) and modeled (red) ocean heat content (OHC) as a function of time for the average of five data sets (see text), the value of $\chi^2_{OCEAN}$ for this run, as well as the ocean heat uptake efficiency, κ, needed to provide the best-fit to the OHC record.

433: remove everything between brackets

    Change made.

435: surely the numbers are altered. I cannot imagine that the feedback parameter isn't dependent on AMOC in the fit.

    Thank you for prompting us to make this sentence clearer. Our major scientific conclusions are not altered if we neglect AMV as a regression variable. As explained in the first author response, our value of $\lambda_\Sigma$ only changes slightly when we neglect AMV, IOD, and PDO as regression variables. The value of $\chi^2_{ATM}$ substantially increases. The increase in the value of $\chi^2_{ATM}$ indicates there will be less combinations of $\lambda_\Sigma$ and AER $RF_{2011}$ that provide a good fit to the historical climate record. This

would narrow our range of parameter space (Fig. 10), and slightly change our future temperature projections, AAWR, and ECS. However, we will still arrive at the same conclusions, that the CMIP6 GCMs warm too quickly, and the EM-GC provides more optimistic probabilities of achieving the Paris Agreement target and upper limit.

We have added some clarifying text to this sentence, Sect. 2.3 where we mention the impact of the inclusion of AMV on AAWR, and New Figure S11 to supplement (shown below) to show that AAWR does not change if AMOC is or is not included in the regression. The new text is: **We stress, as explained in Sect. 2.3, none of our major scientific conclusions are altered if we neglect AMV as a regression variable.**

[Figure]

**New Figure S11.** Measured (HadCRUT5) and modeled GMST anomaly ($\Delta T$) relative to a pre-industrial (1850-1900) baseline without AMOC, PDO, and IOD. (a) Observed (black) and modeled (red) $\Delta T$ from 1850-2019. This panel also displays the values of $\lambda_\Sigma$ and $\chi^2_{ATM}$ (see text) for this best-fit simulation. (b) Contributions from total human activity. This panel also denotes the numerical value of the attributable anthropogenic warming rate from 1975-2014 (black dashed) as well as the $2\sigma$ uncertainty in the slope. The estimates of AAWR show similar results if AMOC is or is not included (see Fig. 1). (c) Solar irradiance (light blue) and major volcanoes (purple). (d) Influences from ENSO on $\Delta T$. (e-f) Contributions from AMOC, PDO, and IOD to $\Delta T$ are set to zero (g) Measured (black) and modeled (red) ocean

heat content (OHC) as a function of time for the average of five data sets (see text), the value of $\chi^2_{OCEAN}$ for this run, as well as the ocean heat uptake efficiency, $\kappa$, needed to provide the best-fit to the OHC record. The error bars (blue) denote the uncertainty in OHC used in this analysis (see Sect. 2.2.8).

438: consider using 'use' throughout instead of 'utilising'

> Thank you for this suggestion. We have decided to retain both "use" and "utilize" in the manuscript, so the text is not repetitive with "use" repeated multiple times in one sentence.

453: is this old factor still valid?

> Thank you for this question. An article posted on climate.gov website published in August of 2020 explains that warming in the ocean below 700 m accounted for 30% of the total increase in OHC from 1971 – 2010 (https://www.climate.gov/news-features/understanding-climate/climate-change-ocean-heat-content). This estimate of 30% is from chapter 3 of IPCC 2013. If the ocean below 700 m accounts for 30% of the heat, then we can infer that the upper 700 m of the ocean holds 70% of the heat. We have verified this estimate ourselves by comparing the OHC in the upper 700 m to the OHC below 700 m from the Cheng et al. (2017) OHC record we use in our analysis. Response Figure 3 shows the change in OHC from 1955-2017 from the Cheng et al. (2017) OHC record for the upper 700 m and above 2000 m. We can divide the OHC in the upper 700 m by the OHC above 2000 m to obtain the ratio between the two time series, and average these values to determine the mean difference between the value of OHC in the upper 700 m and above 2000 m. If we exclude the baseline period (1991-2005) from the calculation, we determine that the ratio is 0.68, indicating the upper 700 m holds about 68% of the heat in the world's oceans. This result supports the assumption of the upper 700 m of the world's oceans holding 70% of the heat in our analysis.

[Figure]

**Response Figure 3.** Change in OHC ($10^{22}$ J) from 1955-2017 relative to 1991-2005 for the upper 700 m and above 2000 m from Cheng et al. (2017).

455: remove sentence 'since ... whole atmosphere', redundant.

Thank you for this suggestion. We would like to retain this sentence for clarity, to ensure the reader understands our method.

459: remove 'temporal'

Change made.

481: remove 'however'

Change made.

505: equal to→of

Thank you for this suggestion. We would like to retain the sentence as currently written to avoid using several instances of the word "of" in one sentence.

510: upon consideration of→by including

Change made.

534: colouring seems to be off in figure S10

We are not exactly sure what the reviewer is referring to here. We have reviewed Fig. S10 and the displayed colors seem fine on our monitors.

539: remove 'the computation of'

Change made.

552: remove sentence, redundant

Change made.

Section 3.1: move methodology to methodology section 2.2.1 (the bit about blending)

We moved the methodology on the blending effect to Sect. 2.3 where we discuss our method to calculate AAWR.

Figure 8: what interval is plotted for each study?

Thank you for bringing this to our attention. The confidence intervals/percentiles for each of the studies that are plotted are as follows: Lewis and Grunwald 2018 – 5th to 95th percentile, Skeie et al. (2018) – 90% confidence interval, Otto et al. (2013) – 5th to 95th confidence interval, Nijsse et al. (2020) – 5th to 95th % confidence interval,  Cox et al. (2018) – 95% confidence limits, Dessler et al. (2018) – minimum and maximum, Armour (2018) – 90% confidence interval, Sherwood et al. (2020) – 5 to 95% confidence intervals, Rugenstein et al. (2020) – minimum and maximum,

Tokarska et al. (2020) – 5th and 95th percentiles, IPCC 2013 – 66% confidence interval, Proistosescu and Huybers (2017) – 5 to 95% confidence interval, and Zelinka et al. (2020) – minimum and maximum. We have decided to include this information in supplement, so the interested reader can find this information without making the main manuscript longer. There is a reference in the Fig. 8 caption to point the reader to the supplement that reads: **See the supplement for the confidence intervals shown for each study.**

772: changed word order, it seems like we're coupling a two-box model to 2.6

Change made.

793: Cox et al. based on CMIP5

Change made.

834: remove 'indicated on each plot', redundant

Change made.

834-835: remove sentence, the reader will know how to do a global average

We would like to maintain this information for anyone trying to reproduce our results. We have moved the sentence to the Fig. 9 caption to help shorten the manuscript.

858: I don't think bimodality is clear here. There seems to be outliers, but not two roughly equal- sized groups of models. With so few models, passing any statistical test on bimodality would be tough. Drop it?

We have replaced "bimodality" with "two groups" so the sentence reads: **Figure 9 illustrates there are two groups of CMIP6 multi-model projections of ΔT, with a few GCMs having future values of ΔT that are considerably higher than others.**

863: remove 'apparent in figure 9', redundant

Change made

918: remove 'our', redundant

Change made: "our" was replaced with "the".

Figure 12: choose bigger bin size: CMIP models displayed weirdly

The bin size in Fig. 12 is consistent between the PDF for the EM-GC and PDF for the CMIP6 GCMs. The CMIP6 PDF looks different from the EM-GC because results are available for only 6 to 33 GCMs depending on the scenario. Conversely, thousands of simulations for the EM-GC allow rigorous sampling of the parameter space.

We prefer to keep the bin size the same between the EM-GC and CMIP6 PDFs. Response Fig. 4 shows the PDF for SSP4-3.4 with a bigger bin size for the CMIP6 GCMs. The bigger bin size looks awkward and makes the figure look like there are only 3 GCMs for SSP4-3.4, whereas there are actually 6.

[Figure]

**Response Figure 4.** Probability density functions (PDF) for $\Delta T_{2100}$ found using the EM-GC with the CW14 temperature record (dark blue) and CMIP6 multi-model results (red).

934: three significant digits not justified, two better

Change made.

Table 1: same

Change made.

991-1003: you seem to be repeating the table, making the prose difficult to read, condense to half the size?

We have eliminated some of the discussion of the table, so we are not repeating the same information.

1015: since -> from / from ... onwards

We have changed "since" to "after" so the sentence reads: **Their analysis indicates only 228 Gt C can be released after 2010 to have a 66% probability of achieving the Paris Agreement target of limiting the rise in $\Delta T$ below 1.5°C in 2100.**

1023: I don't think either of them studied the entire climate system. Instead, those studies were about the atmosphere.

Upon the changes from switching to HadCRUT5 for our primary GMST data record, this sentence was deleted.

In our revised supplement, Fig. S1 has slightly changed. The figure shows output from the EM-GC that is trained using the HadCRUT4 GMST record but is adjusted to be on the HadCRUT5 pre-industrial baseline, to be consistent with other figures in the manuscript. The new baseline causes the values of $\Delta T_{2100}$ shown on the left panel to increase a negligible amount. We fixed a small indexing error in the plotting code used to make Fig. S1, which causes the maximum value of $\Delta T_{2100}$ shown on the right panel to decrease slightly.

Reviewer comments are in black, author responses in blue

After a careful reading of the manuscript, I found the authors have addressed most of my comments and questions, and the revised manuscript has been improved. However, there is still one issue that deserves more attention. In the previous review, I asked about the comparison of the AAWRs that are obtained from EM-GC and CMIP6 models. My concern was whether the AAWR from the EM-GC and the AAWR from the CMIP6 can be compared fairly, as the AAWRs were calculated by different methods (For EM-GC, using Eq. (9); For CMIP6 models, using REG method). The authors have revised this part, but if I understand correctly, they compared the AAWR from REG method with the AAWR from LIN method. There is no direct comparison between the REG method and the method used in EM-GC. The confidence in using REG method comes from the "close agreement of AAWR" found using both the REG and the LIN methods, which I find not very convincing. I would suggest that, if possible, the authors may apply the REG method to the EM-GC simulations (Note, do not use the coefficient C1 from Eq. (2). Use the new coefficient obtained from the REG method). Then, compare the AAWR from the REG method and the AAWR calculated from Eq. (9).

We thank the reviewer for taking the time to read through our changes and are delighted to read that the reviewer finds the manuscript improved.

To address the remaining concern, we applied the REG method to the EM-GC simulations as suggested by the reviewer. We regressed the modeled GMST time series output from the EM-GC against SAOD (after applying a 6-month lag) and a linear function used to represent the anthropogenic effect on temperature from 1975-2014. New Fig. S13 below shows the resulting simulations of ΔT.

The value of AAWR from the EM-GC determined using the REG method is 0.188°C/decade, compared to 0.167°C/decade using Eq. (9) in the main paper (New Fig. S13c and Fig. 1). There is a 0.021°C/decade difference between these two estimates of AAWR. This difference arises because the REG method, when applied to the EM-GC modeled ΔT time series, includes the contribution of AMOC in the value of AAWR (New Fig. S13c). Figure 1 of our paper shows that AMOC contributes about 0.025°C/decade to the rise in ΔT from 1975-2014. If the effect of AMOC is not removed before applying the REG method to the output from the EM-GC, then the influence of AMOC will be erroneously included in the value of AAWR. These results shown on the left hand panels of Fig. S13 are similar to the approach by Foster and Rahmstorf (2011), who included the effect of AMOC by taking the linear fit of a residual. If we include AMOC as a regressor variable to the REG method, we obtain a value of AAWR of 0.161°C/decade (Fig. S13g). This new value of AAWR is within the uncertainty estimate of AAWR using Eq. 9, which is 0.167 ± 0.007°C/decade.

The close agreement of values of AAWR from the REG method once we account for AMOC and that found using Eq. (9) supports the validity of the REG method to determine AAWR from CMIP6 output. We do not explicitly use AMOC as a regressor variable when applying the REG method to CMIP6 GCMs for two reasons. The first

reason is that GCMs have been shown to underestimate key aspects of the Atlantic multidecadal oscillation and are unable to simulate the many oceanic and atmospheric footprints of AMOC (Kavvada et al., (2013). The second reason is that CMIP6 GCM historical runs do not use prescribed SSTs. If the CMIP6 GCMs are representing AMOC, it is a random signal that is averaged out when we analyze the 50 GCMs in order to calculate AAWR.

We have included New Fig. S13 in the supplement and added the following text on lines 541-543 to Sect. 2.3 of the main paper in reference to this figure:

**Figure S13 and the supplement compare values of AAWR found using the REG method applied to EM-GC output with values of AAWR found using Eq. (9), as support for the validity of using the REG method to determine AAWR from CMIP6 output.**

[Figure]

**New Figure S13**. The change in GMST relative to 1961-1990 from observations and modeled output. (a) Change in GMST from HadCRUT5 and EM-GC simulation. (b) The residual in the change of GMST from the EM-GC simulation after subtracting the contribution of SAOD determined by the REG method (grey) and the change in GMST due to humans from the REG method (orange). (c) The change in GMST due to humans from the REG method (orange) and from the EM-GC (blue). The values of AAWR determined using the REG method and Eq. (9) are shown. (d) The contribution of SAOD to GMST. (e) Same as (a). (f) Same as (b) but also subtracting the contribution of AMOC determined by the REG method. (g) Same as (c) but using AMOC as a regressor variable. (h) Same as (d) and also including the contribution of AMOC to ΔT determined by the REG method.

We have added the following text beginning on line 203 to the supplement to explain our justification of the use of the REG method for the determination of AAWR from the GCMs:

**We applied the REG method to the EM-GC simulations to check the validity of the REG method. We regressed the modeled ΔT time series from the EM-GC for an AER RF$_{2011}$ = −0.9 W m$^{-2}$ simulation with SAOD and applied a 6 month lag. A linear function is used to represent the anthropogenic effect on temperature from 1975-2014. Fig. S13 shows the results of using the REG method on output of the EM-GC.**

**The value of AAWR from the EM-GC determined using the REG method is 0.188°C/decade, compared to 0.167°C/decade using Eq. (9) (Fig. S13c and Fig. 1). There is a 0.021°C/decade difference between the two methods. This difference arises because the REG method, when applied to the EM-GC modeled ΔT time series, includes the contribution of AMOC in the value of AAWR (Fig. S13c). Figure 1 of our paper illustrates that AMOC contributes about 0.025°C/decade to the rise in ΔT. If we include AMOC as a regressor variable to the REG method, we obtain a value of AAWR of 0.161°C/decade from the output of the EM-GC (Fig. S13g).**

**The close agreement of values of AAWR from the REG method once we account for AMOC and Eq. (9) supports the validity of the REG method to determine AAWR from CMIP6 output. We do not explicitly use AMOC as a regressor variable when applying the REG method to CMIP6 GCMs for two reasons. The first reason is that GCMs have been shown to underestimate key aspects of the Atlantic multidecadal oscillation and are unable to simulate the many oceanic and atmospheric footprints of AMOC (Kavvada et al., 2013). The second reason is that CMIP6 GCM historical runs do not use prescribed SSTs. If the CMIP6 GCMs are representing AMOC, it is a random signal that is averaged out when we analyze the 50 GCMs in order to calculate AAWR.**

In our revised supplement, Fig. S1 has slightly changed. The figure shows output from the EM-GC that is trained using the HadCRUT4 GMST record but is adjusted to be on the HadCRUT5 pre-industrial baseline, to be consistent with other figures in the manuscript. The new baseline causes the values of $\Delta T_{2100}$ shown on the left panel to increase a negligible amount. We fixed a small indexing error in the plotting code used to make Fig. S1, which causes the maximum value of $\Delta T_{2100}$ shown on the right panel to decrease slightly.

[revised manuscript text omitted]
)". Figure S1 illustrates the impact of updating Eq. (2) in our model to be comparable to the formulation in Bony et al. (2006) and Schwartz (2012). This figure displays the change in GMST anomaly in 2100 relative to pre-industrial ($\Delta T_{2100}$) as a function of $\lambda_\Sigma$ and AER $RF_{2011}$ for the two formulations of Eq. (2). Figure S1a uses the previous version of the EM-GC, where $Q_{OCEAN}$ was subtracted outside of the climate feedback multiplicative term, and Fig. 1b uses the new version of the EM-GC where $Q_{OCEAN}$ is subtracted within the climate feedback multiplicative term.

In the EM-GC framework, we calculate our value of $Q_{OCEAN}$ by finding the $\kappa$ needed to multiply the temperature difference between the atmosphere and the ocean to fit the observed OHC record. The model iterates over the ocean module, specifically the value of $\Delta T_{OCEAN,HUMAN}$ in Eq. (4), until the EM-GC converges on an estimate of $\kappa$ for a single OHC record and value of AER $RF_{2011}$. Figure S1 illustrates that the effect of changing Eq. (2) in the EM-GC impacts our estimates of the rise in $\Delta T_{2100}$ at high values of AER $RF_{2011}$. Strong aerosol cooling results in the ocean taking up more heat from the atmosphere than in the previous version of the EM-GC. The larger value of $Q_{OCEAN}$ results in a higher value of climate feedback needed to fit the historical climate record, because both AER $RF_{2011}$ and $Q_{OCEAN}$ are acting to cool the climate system. The higher values of climate feedback increase our maximum value of $\Delta T_{2100}$. This change brings some of the projections of $\Delta T_{2100}$ from the EM-GC closer to values of $\Delta T_{2100}$ from the CMIP6 multi-model ensemble.

Section 2.1 states "Altering the training period of our model has a slight effect on our results (see Fig. S2, S3, and the supplement for information on various training periods)." Figure S2 shows the end of century projected warming as a function of $\lambda_\Sigma$ and AER $RF_{2011}$, for four different training periods: 1850-1989 (Fig. S2a), 1850-1999 (Fig. S2b), 1850-2009 (Fig. S2c) and 1850-2019 (Fig. S2d), which is the normal training period used in our analysis. Values of $\Delta T_{2100}$ are shown only for combinations of $\lambda_\Sigma$ and AER $RF_{2011}$ that lead to good fits $(\chi^2 \leq 2)$ to the climate record. We project relatively similar results for end of century warming for the training periods that end in 2019 and 2009. Our results using the training period from 1850-1999 are similar to observations and other reduced complexity models (Nicholls et al., 2020). The training period that ends in 1989 (Fig. S2a) yields a different "shape" of model parameter space for which good fits to the climate record can be obtained, compared to the other training periods. The different shape for this shorter training period is due to two factors. First, the formulation of the ocean component of our model for the training period that stops in 1989 uses  35 years

of the observed OHC record. We are able to calculate good fits to the OHC record over this shorter time period that diverge from the OHC record after 1989. Also, for this shorter time period, aerosol radiative forcing of climate cools in a manner that nearly mirrors the warming due to rising GHGs, resulting in a wider range of model parameters that lead to a "good fit" of the climate record, compared to model simulations constrained by data that extend closer to present-day. The highest values of $\Delta T_{2100}$ in Fig. S2a are associated with the largest values of $\lambda_\Sigma$, which in our model corresponds to excessively high values of $\kappa$ that we can rule out, based on OHC data collected during 1990 to 2019.

Figure S3 shows the observed (HadCRUT5) and modeled $\Delta T$ anomaly from 1945-2060 for the four different training periods described above. Each panel contains three projections of future $\Delta T$ for SSP4-3.4: projection using the value of climate feedback that provides the best fit to the historical climate record for a value of AER $RF_{2011} = -0.9$ W m$^{-2}$, the lowest value of climate feedback that provides a good fit to the observed $\Delta T$ record for a value of AER $RF_{2011} = -0.1$ W m$^{-2}$, and the highest value of climate feedback that provides a good fit to the historical climate record (the associated value of AER $RF_{2011}$ varies depending on the training period). As more years are added to the training period, the range of projection for future temperature decreases (Fig. S3a vs S3d). All of the best fit projections (solid line) and highest value of climate feedback (upper dashed line) closely follow the mid-point of the data, regardless of the training period. Given the nature of this test (i.e., predicting GMST out to 2019 for a series of trainings that stop in either 1989, 1999, or 2009), Figure S3 supports the quantitative accuracy of our approach for simulating and projecting future $\Delta T$.

~~Section 2.2.1 states "see Fig. S4 and the supplement for information on CW14 GMST record". Figure S4 shows the GMST anomaly relative to pre-industrial over time for (a) HadCRUT record with the HadCRUT uncertainties, (b) CW14 record with the CW14 uncertainties, (c) BEG record with the BEG uncertainties, and (d) CW14 record with HadCRUT uncertainties. The uncertainties for CW14 are much smaller than those for the HadCRUT and BEG records. The small values of CW14 uncertainties, especially from 1850-1900, cause the EM-GC to not be able to achieve good fits to this temperature record. We have two choices for use of the CW14 GMST record; either relax the constraint for $\chi^2_{ATM}$ (i.e., run with $\chi^2_{ATM} \leq$ 4), or modify the CW14 uncertainties. We chose to combine the uncertainties from HadCRUT with the values of GMST from the CW14 record since CW14 is based upon the HadCRUT GMST record. Upon use of this combination of data and uncertainty, we are able to find good fits to the CW14 temperature record that look similar to the fits obtained using the other GMST data sets.~~

Section 2.2.1 states "We use the uncertainty time series from HadCRUT4 for all GMST records (see the supplement, Figs. S4 and S5, and Table S1 for more information)". Figure S4 shows values of $\Delta T$ based on

the seven individual GMST records (GISTEMP, BEG, HadCRUT4, CW14, HadCRUT5, NOAAGT, and JMA) with their corresponding 1σ and 2σ uncertainties. A horizontal line at zero denotes the time period of the baseline for each ΔT record. The multi-record mean, excluding the data set that is plotted, is also shown. Since the multi-record mean and individual ΔT record are plotted on the same baseline, these two quantities closely match over this time period. Panels (a), (b), (e), and (f) illustrate that the uncertainties for these GMST records are not large enough to encompass the multi-record mean over 1850-2019. The multi-record mean in panel (a) is below the GISTEMP 1σ uncertainty range between 1880 and 1900, and again between 1980 to 2019. In panel (b), the multi-record mean is above the BEG 1σ range from 1850 until 1865, 1880 to 1895, and below the 1σ uncertainty range from 2000 to 2019. The multi-record mean in panel (e) is below the HadCRUT5 1σ uncertainty range from 1990 until 2019. In panel (f), the multi-record mean is above the NOAAGT 1σ uncertainty range from 1920 until 1955. The JMA GMST record does not provide an uncertainty estimate. We therefore use the HadCRUT4 combined uncertainty (measurement, sampling, bias, and coverage uncertainties (Morice et al., 2012)) estimate for JMA in panel (g). The multi-record mean of ΔT for all data sets other than JMA lies at the edge of the 1σ uncertainty range from 1891 until 2000. After 2000, the multi-record mean falls above both the 1σ and 2σ HadCRUT4 uncertainty range. The HadCRUT4 uncertainty time series shown in panel (c) is the only uncertainty estimate large enough to cover the spread in the various GMST records.

Figure S5 shows ΔT based on all seven GMST records and the multi-record mean relative to three baseline periods. The 1σ and 2σ uncertainties from HadCRUT4 are plotted about the multi-record mean. Panels (a) and (d) show the GMST records relative to 1891-1920, which are the first 30 years all of the data sets have in common. Between 1850-1970, all of the data sets fall within the 1σ HadCRUT4 uncertainty. After 1970, the GMST records start to deviate and some fall outside of the 1σ uncertainty but within the 2σ uncertainty, and JMA falls outside of the 2σ uncertainty. Panels (b) and (e) show the GMST records relative to the HadCRUT baseline period of 1961-1990. We see similar behavior as in panels (a) and (d), where the GMST records largely fall within the HadCRUT4 1σ uncertainty until about 1970. Panels (c) and (f) show the GMST records forced to match HadCRUT5 from 2010-2019, which is baselined to 1961-1990. In these two panels, we see a large spread between the GMST records from the beginning of the time period until 2005.

Table S1 shows the percentage of ΔT data points that lie within the 1σ or 2σ HadCRUT4 uncertainty about the multi-record mean for all seven data records since 1940. Year 1940 is used to be consistent with the definition of our $\chi^2_{RECENT}$ parameter. Depending on the choice of baseline period, the number of data points within the uncertainty range varies. For a baseline of 1891-1920, 80% of the data points for all seven records are within the 1σ uncertainty and 95% of the data points are within the 2σ. For a baseline of 1961-1990, 88% and 93% of data points are within the 1σ and 2σ HadCRUT4 uncertainties, respectively. If the

ΔT records are forced to match the average value of the HadCRUT5 data set over the last decade, 72% of the data points are within the 1σ uncertainty and 88% are within the 2σ uncertainty. This analysis shows that depending on which baseline is used, the percentage of points within the 1σ or 2σ uncertainty ranges varies. Overall, these comparisons support the utility of the HadCRUT4 uncertainty for the GMST, since the 1σ and 2σ uncertainty ranges capture a percentage of points approximately correct for a pure Gaussian distribution. Therefore, we have adopted the HadCRUT4 uncertainties in GMST for all of the analyses in the main paper. The uncertainties published by other data centers tend to be smaller than the HadCRUT4 uncertainties. Since only the HadCRUT4 uncertainties span the range of values for ΔT from the seven data records in a somewhat realistic fashion, we have decided to use these uncertainties uniformly throughout the analysis.

Section 2.2.1 also says "We then adjust each data set to the HadCRUT5 pre-industrial baseline as described in the supplement". The mean of the HadCRUT5 GMST record from 1850-1900 is −0.3589°C. We add 0.3589°C to each value of the HadCRUT5 record to adjust the data set onto the pre-industrial baseline. We use this same offset for all of the other data sets. We add 0.3589°C to each value of ΔT from the six data sets to match the HadCRUT5 1850-1900 baseline.

Section 2.2.3 states "Figure  S6 shows the ozone RF time series used in this analysis and the supplement provides more information about the creation of the time series for the RF due to $O_3^{TROP}$". Figure  S6 displays the time series of tropospheric ozone RF used in our analysis for the various SSPs. Tropospheric ozone is an important GHG that rivals nitrous oxide as the third most important anthropogenic GHG. We include the RF due to tropospheric ozone ($O_3^{TROP}$) in our model for completion, even though the SSP database does not provide RF estimates for the various SSPs. We use values from the RCP scenarios provided by the Potsdam Institute for Climate Impact Research (Meinshausen et al., 2011). The values of the RF due to $O_3^{TROP}$ for SSP1-1.9 and SSP1-2.6 are from the RCP2.6 pathway. The RCP4.5 time series of $O_3^{TROP}$ is used for SSP2-4.5, the RCP6.0 time series is used for SSP4-6.0, and the RCP8.5 time series is used for SSP5-8.5. We create linear combinations of RCP2.6 and RCP8.5 to generate two new time series of the RF due to $O_3^{TROP}$ for SSP4-3.4 and SSP3-7.0. There is a large gap between the time series of the RF due to $O_3^{TROP}$ for RCP6.0 (shown as SSP4-6.0) and RCP8.5 (shown as SSP5-8.5) in Fig. S6. We created a time series that would split the difference between the two RCPs to represent the RF due to $O_3^{TROP}$ for SSP3-7.0. The SSP4-3.4 time series of the RF due to $O_3^{TROP}$ that was created lies in between the RCP2.6 (shown as SSP1-2.6) and RCP4.5 (shown as SSP2-4.5) time series in Fig. S6.

Section 2.2.8 states "Figure  S9 shows the five OHC records as well as the multi-measurement average". Figure  S9 displays the five OHC content data sets, as well as the multi-measurement average, plotted as a function of time and normalized to year 1986. This figure illustrates how the shapes of the different OHC records compare. Each of the time series represents the amount of heat stored in the top 700 m of the world's oceans for that specific data set. Carton et al. (2018) is the shortest data set, and only spans 36 years (1982-2017). The second shortest record is Balmaseda et al. (2013a), which spans 52 years (1958.5-2009.5). Ishii et al. (2017) is the record in the middle with a range of 63 years (1955-2017). Both Cheng et al. (2017) and Levitus et al. (2012) have records that span 65 years (1955-2019). The length of the data set and the shape of the curve affect the estimate of ocean heat export (OHE), because we calculate OHE by taking a linear fit to the full OHC time series. Balmaseda et al. (2013a) has the lowest estimate of OHE because the slope of the curve is relatively shallow, due to the fact that it slightly rises, then decreases at the start of the record. Carton et al. (2018) has the highest estimate of OHE because the slope of the curve is the steepest of the five records.

Section 2.2.8 also says "For these five OHC data sets, uncertainty estimates are not always provided. Furthermore, some studies that do provide uncertainties give estimates that seem unreasonably small (see Fig  S10 and the supplement)" and "Figure  S10 and the supplement provide more detail on the creation of this time dependent uncertainty estimate for OHC". Figure  S10 shows the multi-measurement average as well as the five OHC data records as a function of time, the uncertainty for each corresponding data set, and the combined uncertainty used in this analysis. Panel (a) shows the multi-measurement OHC average with the standard deviation of the mean plotted around the average time series. The standard deviation is large at the beginning of the time series, due to the spread in the estimates of OHC between the different records (illustrated in Fig.  S9). The standard deviation decreases as the various OHC records converge near a similar estimate. The standard deviation is zero in 1986 because we normalized all of the time series to zero in this year to create the multi-measurement average. Because of this normalization, the standard deviation of the mean is not a realistic measure of uncertainty for the five OHC time series.

Panels (b), (c), (d), (e), and (f) display the uncertainty estimates for the five OHC data records. We use the standard deviation of the mean of five ensemble members of the European Centre for Medium-Range Weather Forecasts Ocean ReAnalysis System 4 (ORSA) (Balmaseda et al., 2013b) for the Balmaseda et al. (2013a) record. The standard deviation is plotted in panel (b) as the dotted blue line. The standard deviation is small at the beginning of the record, because the five ensemble members started at similar values of OHC in 1958 and diverged over time. The combined uncertainty of the standard deviation of the average of the five OHC records and the Cheng et al. (2017) estimate is plotted as a dashed blue line. Panel (c) shows the Levitus et al. (2012) time series for the top 700 m updated to the end of 2019. The Levitus time series utilizes

the standard error over the whole ocean for their uncertainty estimate and is plotted as the dotted light blue line. The standard error is a very small uncertainty estimate compared to the other OHC data records, which is unreasonable considering the large variations in OHC between the different records. We use the standard deviation of eight reanalysis experiments to represent the uncertainty associated with the Carton et al. (2018) OHC record and is plotted as a dotted orange line in panel (e). The standard deviation of the eight reanalysis experiments is rather small, which also is unrealistic. Panel (f) displays the Cheng et al. (2017) OHC record updated through the end of 2019 with the 1$\sigma$  uncertainty. This uncertainty does not vary much throughout the data record, making it more realistic as an estimate for such an uncertain quantity as OHC. We created the combined uncertainty estimate of the standard deviation of the average of the five OHC records and the Cheng et al. (2017) 1$\sigma$  uncertainty to have the largest uncertainty possible due to the fact that OHC varies between the different records. The EM-GC cannot achieve $\chi^2_{OCEAN} \leq 2$ for Balmaseda et al. (2013a), Levitus et al (2012), and Carton et al. (2018) using their own respective estimates of uncertainty. Creating one uncertainty estimate to be used for all of the OHC records provides consistency and allows the EM-GC to achieve good fits between the observed and modeled OHC.

Section 2.3 states "Figure  S12 illustrates the REG method used to determine AAWR from the CMIP6 GCMs" . Figure  S12 shows the change in $\Delta T$ from 1975-2014 from the CMIP6 GCMs and the contribution of SAOD from 1975-2014. There was about a 6 month lag between the response of $\Delta T$ and enhancements of SAOD following the eruption of Mount Pinatubo in June 1991 (Douglass and Knox, 2005; Thompson et al., 2009); a 6 month delay for the response of $\Delta T$ to SAOD is commonly used in regression analyses of the actual temperature record (Foster and Rahmstorf, 2011; Lean and Rind, 2008). The time needed for $\Delta T$ to respond to a change in the aerosol loading in the stratosphere due to a volcanic eruption in each GCM can exhibit a significant difference compared to this empirically determined response time. Therefore, a lag was determined for each GCM by calculating the value of the monthly delay that resulted in the largest regression coefficient for SAOD (versus $\Delta T$). Due to the difference in model physics between the various GCMs, the value of the delay between the volcanic forcing and surface temperature response ranged from 0 to 11 months. The effect of SAOD on $\Delta T$ for the 50 GCMs is shown in Fig. S12d. Figure  S12b shows the residual in $\Delta T$ after removing the influence of SOAD, and the median value of AAWR from the CMIP6 multi-model ensemble is plotted as a linear line. Figure  S12c shows the human component of global warming, $\Delta T_{ATM,HUMAN}$, from the EM-GC. A linear fit and quadratic fit are shown to illustrate that $\Delta T_{ATM,HUMAN}$ is almost nearly linear from 1975-2014, supporting the approximation of $\Delta T_{ATM,HUMAN}$ as a linear function from 1975-2014 for the REG calculation.

Section 2.3 also states "Figure S13 and the supplement compare values of AAWR found using the REG method applied to EM-GC output with values of AAWR found using Eq. (9), as support for the validity of using the REG method to determine AAWR from CMIP6 output". We applied the REG method to the EM-GC simulations to check the validity of the REG method. We regressed the modeled $\Delta T$ time series from the EM-GC for an AER $RF_{2011} = -0.9$ W m$^{-2}$ simulation with SAOD and applied a 6 month lag. A linear function is used to represent the anthropogenic effect on temperature from 1975-2014. Fig. S13 shows the results of using the REG method on output of the EM-GC.

The value of AAWR from the EM-GC determined using the REG method is 0.188°C/decade, compared to 0.167°C/decade using Eq. (9) (Fig. S13c and Fig. 1). There is a 0.021°C/decade difference between the two methods. This difference arises because the REG method, when applied to the EM-GC modeled $\Delta T$ time series, includes the contribution of AMOC in the value of AAWR (Fig. S13c). Figure 1 of our paper illustrates that AMOC contributes about 0.025°C/decade to the rise in $\Delta T$. If we include AMOC as a regressor variable to the REG method, we obtain a value of AAWR of 0.161°C/decade from the output of the EM-GC (Fig. S13g).

The close agreement of values of AAWR from the REG method once we account for AMOC and Eq. (9) supports the validity of the REG method to determine AAWR from CMIP6 output. We do not explicitly use AMOC as a regressor variable when applying the REG method to CMIP6 GCMs for two reasons. The first reason is that GCMs have been shown to underestimate key aspects of the Atlantic multidecadal oscillation and are unable to simulate the many oceanic and atmospheric footprints of AMOC (Kavvada et al., 2013). The second reason is that CMIP6 GCM historical runs do not use prescribed SSTs. If the CMIP6 GCMs are representing AMOC, it is a random signal that is averaged out when we analyze the 50 GCMs in order to calculate AAWR.

Section 2.3 also says "Analysis of AAWR for these 50 GCMs of LIN versus REG (see Fig. S14)…". Figure S11 S14 shows the similarity between the values of AAWR determined using the LIN and REG methods. The ratio between the values of AAWR determined utilizing LIN and REG is 1.009, indicating there is only about a 0.9% difference in the values of AAWR using the two methods. Figure S11 S14 also shows the values of AAWR that are below the maximum value of AAWR determined by the EM-GC utilizing the HadCRUT5 temperature record (blue) and the values that are above the maximum (red). About Less than half of the GCMs result in values of AAWR less than the maximum value from the EM-GC and half more than half of the GCMs result in values of AAWR greater than the maximum value from the EM-GC utilizing the HadCRUT5 GMST record.

Section 2.4 states "For the estimate of ECS from the CMIP6 multi-model ensemble, we use the method described by Gregory et al. (2004) (See the supplement and Fig. S15 for more information)". To use the Gregory method, near surface air temperature output from the Abrupt 4×$CO_2$ and piControl simulations, as well as net downward radiative flux output from the Abrupt 4×$CO_2$ simulation is used to calculate ECS. The near surface air temperature and net downward radiative flux was converted from monthly gridded output to annual global averages. We calculate the temperature change for the Abrupt 4×$CO_2$ simulation by subtracting the piControl near surface air temperature (Chen et al., 2019) (Fig. S15). This computed temperature anomaly is then regressed against the net downward radiative flux, with the x-intercept yielding the equilibrium response of ΔT to a quadrupling of $CO_2$. This equilibrium response is then divided by two (Jones et al., 2019) to arrive at the equilibrium climate sensitivity (Fig. S15).

Section 2.5 states "See Fig.  S16 for unweighted ECS values and Section 3.2 states "See Fig  S16 for results without aerosol weighting". Figure  S16 displays the values of ECS using the EM-GC and the CMIP6 multi-model ensemble. The EM-GC box contains the 25$^{th}$, 50$^{th}$, and 75$^{th}$ percentiles, the whiskers denote the 5$^{th}$ and 95$^{th}$ percentiles, and the stars represent the minimum and maximum values of ECS. The box labeled CMIP6 is unchanged from Fig. 7. The values of ECS are not treated with the aerosol weighting described in Sect. 2.5. This figure shows that most of the estimates of ECS found using the EM-GC are concentrated towards small values of ECS, due to the fact that the majority of the EM-GC model runs with good fits to the climate record ($\chi^2_{ATM}$, $\chi^2_{RECENT}$, and $\chi^2_{OCEAN}$) have weak aerosol cooling and low values of $\lambda_\Sigma$ (Fig. 5b). We use the aerosol weighting method to assign the same weights for the IPCC 2013 "likely" range limits of AER RF$_{2011}$ of −0.4 and −1.5 W m$^{-2}$ at the one sigma values of a Gaussian, and the −0.1 and −1.9 W m$^{-2}$ are at the two sigma values of a Gaussian. Using the aerosol weighting method adjusts our estimates of ECS so that the calculated percentiles occur at higher values.

~~Section 3.2 states "We tested our approach of calculating ECS utilizing the EM-GC to CMIP6 GCMs by altering the EM-GC framework to include CMIP6 output (see the supplement for details)". To use the EM-GC framework with the CMIP6 output, we calculated the CMIP6 multi-model mean change in GMST from 1850-2100 using the SSP2-4.5 scenario. We used the standard deviation of the CMIP6 multi-model mean to represent the uncertainty in the rise in GMST for our reduced chi-squared calculations. We trained the EM-GC from 1850-2100, included the CMIP6 prescribed values of SAOD and TSI, and did not include any natural variability, since effects on GMST due to factors such as ENSO should be randomly distributed within the various CMIP6 GCM runs that constitute the CMIP6 multi-model mean. We used the average of the five (Fig. S13a) and the Cheng OHC (Fig. S13b) records to calculate the amount of heat exported to the world's oceans. Our results in Fig. S13a,b show ΔT$_{2100}$ for the values of AER RF$_{2011}$ and $\lambda_\Sigma$ that the EM-GC~~

finds good fits to the CMIP6 multi-model GMST output for SSP2-4.5. Figure S13c shows the 5th, 25th, 50th, 75th, and 95th percentile values of ECS for the Gregory method and the altered EM-GC framework with the Cheng OHC record and the average of the five OHC records. The comparison of ECS found using quite different approaches, illustrated in Figure S13, provides strong support for the veracity of ECS inferred from GCMs and from the climate record throughout our analysis.

Section 3.2 in the Fig. 8 caption says, "See supplement for the confidence intervals plotted for each study". All of the studies except Dessler et al. (2018), Rugenstein et al. (2020), IPCC 2013, and Zelinka et al. (2020) have the 5 to 95% confidence intervals shown. The 66% confidence intervals are shown for IPCC 2013, and the minimum and maximum are shown for Dessler et al. (2020), Rugenstein et al. (2020) and Zelinka et al. (2020).

Section 3.3.4 states "see Fig. S21 and the supplement" and "see the supplement for more information". Figure S21 shows the rise in $\Delta T$ from pre-industrial for SSP5-8.5 versus the cumulative emissions of $CO_2$, in Gt C, since 1870. The colored lines denote the probability of reaching at least that temperature by the end of century. The large spread in projections of future $\Delta T$ is driven by the uncertainty in AER RF. The computed probabilities are based on the aerosol weighting method, so the uncertainty in AER RF is considered when determining the likelihood of achieving the Paris Agreement target of 1.5°C and upper limit of 2.0°C for the cumulative carbon emissions.

We use the uncertainty suggested by coupled atmospheric / carbon cycle models in how atmospheric $CO_2$ will respond to the prescribed carbon emissions. Examination of Fig. 2 and Table 3 from Friedlingstein et al. (2014) and Fig. 9b from Murphy et al. (2014) led to our determination that the uncertainty in estimates of atmospheric $CO_2$ from emissions driven runs of CMIP5 coupled atmospheric / carbon cycle models is about 10% ($1\sigma$). We include this 10% uncertainty in our determination of the carbon budgets for each probability of achieving the Paris Agreement target and upper limit shown in Table 2.

Section 3.3.6 states "see Fig. S23 and the supplement for results without the time delay". Figure S23 shows the effect of time variant $\lambda^{-1}$ with an instantaneous response between $\lambda^{-1}$ and a change in radiative forcing. The instantaneous response causes the modeled $\Delta T$ to deviate more from the observed temperature than the results using the 20 year delay in the response (Fig. S23g, h versus Fig. 14g, h). The deviation between the modeled and observed $\Delta T$ does not allow for a large change in $\lambda^{-1}$ over time to still achieve the $\chi^2_{ATM}$ and $\chi^2_{RECENT}$ constraints. The deviation between modeled and observed $\Delta T$ in Fig. 23d resembles the behavior of some CMIP6 GCMs (see Fig. 9 and Tokarska et al. (2020)).

300

[Figure]

**Figure S1.** GMST anomaly in 2100 relative to pre-industrial ($\Delta T_{2100}$) as a function of climate feedback parameter and AER RF$_{2011}$ for two versions of the EM-GC trained with the HadCRUT4 $\Delta T$ record. (a) The change in $\Delta T_{2100}$ for SSP4-3.4 using the original formulation of Eq. (2) where $Q_{OCEAN}$ is subtracted outside of the feedback multiplicative term. (b) The change in $\Delta T_{2100}$ for SSP4-3.4 using the updated formulation of Eq. (2) where $Q_{OCEAN}$ is subtracted within the feedback multiplicative term similar to Bony et al. (2006) and Schwartz (2012). The EM-GC is able to fit higher values of $\lambda_\Sigma$ at strong aerosol cooling (around $-1.5$ W m$^{-2}$) for the new Eq. (2) compared to the original formulation in Canty et al. (2013) and Hope et al. (2017). The maximum value of future warming has increased due to the higher $\lambda_\Sigma$ values.

[Figure]

**Figure S2.** $\Delta T_{2100}$ as a function of climate feedback parameter and tropospheric aerosol radiative forcing in 2011 using the EM-GC trained with the HadCRUT5 $\Delta T$ record for SSP4-3.4. (a) Training period of 1850-1989. The region outside of the AER $RF_{2011}$ range provided by IPCC 2013 is shaded (grey). Colors denote the GMST change in year 2100 relative to pre-industrial. The color bar is the same across all four panels for comparison. (b) Training period of 1850-1999. (c) Training period of 1850-2009. (d) Training period of 1850-2019, which is the normal training period used in our analysis.

[Figure]

**Figure S3.** Observed and modeled GMST anomaly relative to 1850-1900 from 1945-2060 for four training periods. (a) Observations from HadCRUT5 (black), the EM-GC $\Delta T$ simulation for a training period of 1850-1989 (orange) of HadCRUT5, and the EM-GC projections for SSP4-3.4 out to 2060. Three EM-GC projections are shown in red: The best estimate of climate feedback for AER $RF_{2011} = -0.9$ W m$^{-2}$, the lowest value of climate feedback that satisfies the $\chi^2$ constraints for AER $RF_{2011} = -0.1$ W m$^{-2}$, and the highest value of climate feedback that satisfies the $\chi^2$ constraints (the value of AER $RF_{2011}$ varies for each training period). The IPCC 2013 likely range of warming is denoted as the black trapezoid. (b) Training period of 1850-1999. (c) Training period of 1850-2009. (d) Training period of 1850-2019.

320

[Figure]

**Figure S4.** Annual GMST (ΔT) anomaly for seven data records relative to their individual baseline and the multi-record mean. The multi-record mean does not include the data set that is being shown. The 1σ and 2σ uncertainties for each GMST record are shown, and the horizontal line for ΔT=0 spans the baseline used for the specific panel. (a) GISTEMP. (b) BEG. (c) HadCRUT4. (d) CW14. (e) HadCRUT5. (f) NOAAGT. (g) JMA. Since the JMA data provider does not provide an uncertainty time series, the HadCRUT4 uncertainty is used.

[Figure]

330

**Figure S5.** Annual GMST (ΔT) anomaly relative to several baseline periods -for seven data records. The 1σ (shaded grey) and 2σ (dotted grey) HadCRUT4 uncertainties are plotted about the multi-model record mean (black). (a) Baseline of 1891-1920 plotted from 1850-2019. (b) Same as (a) using a baseline of 1961-1990. (c) Same as (a) except all of the ΔT records are forced to match the average ΔT anomaly over 2010-2019 given by HadCRUT5 that is relative to 1961-1990. (d) – (f) Same as (a) – (c) except plotted from 1940-2019.

335

**Table S1.** Percentage of annual values between 1940-2019 of the GMST record within the 1 sigma or 2 sigma HadCRUT4 uncertainties about the multi-record mean for each baseline period.

| Baseline: 1891-1920 | 1σ | | | 2σ | | |
|---|---|---|---|---|---|---|
| | N$_{WITHIN}$ | N$_{TOTAL}$ | % | N$_{WITHIN}$ | N$_{TOTAL}$ | % |
| HadCRUT4 | 77 | 80 | 96 | 80 | 80 | 100 |
| HadCRUT5 | 42 | 80 | 53 | 80 | 80 | 100 |
| CW14 | 80 | 80 | 100 | 80 | 80 | 100 |
| BEG | 71 | 80 | 89 | 80 | 80 | 100 |
| GISTEMP | 73 | 80 | 91 | 80 | 80 | 100 |
| NOAAGT | 76 | 80 | 95 | 80 | 80 | 100 |
| JMA | 29 | 80 | 36 | 54 | 80 | 68 |
| **AVERAGE** | | | **80%** | | | **95%** |
| | | | | | | |
| **Baseline: 1961-1990** | | | | | | |
| HadCRUT4 | 80 | 80 | 100 | 80 | 80 | 100 |
| HadCRUT5 | 68 | 80 | 85 | 80 | 80 | 100 |
| CW14 | 80 | 80 | 100 | 80 | 80 | 100 |
| BEG | 80 | 80 | 100 | 80 | 80 | 100 |
| GISTEMP | 75 | 80 | 94 | 80 | 80 | 100 |
| NOAAGT | 80 | 80 | 100 | 80 | 80 | 100 |
| JMA | 27 | 80 | 34 | 48 | 80 | 60 |
| **AVERAGE** | | | **88%** | | | **93%** |
| | | | | | | |
| **Match 2010-2019** | | | | | | |
| HadCRUT4 | 68 | 80 | 86 | 80 | 80 | 100 |
| HadCRUT5 | 47 | 80 | 59 | 76 | 80 | 95 |
| CW14 | 78 | 80 | 98 | 80 | 80 | 100 |
| BEG | 77 | 80 | 96 | 80 | 80 | 100 |
| GISTEMP | 47 | 80 | 59 | 79 | 80 | 99 |
| NOAAGT | 73 | 80 | 61 | 80 | 80 | 100 |
| JMA | 11 | 80 | 14 | 18 | 80 | 23 |
| **AVERAGE** | | | **72%** | | | **88%** |

340

[Figure]

**Figure** S6. Radiative forcing of tropospheric ozone for the various SSPs analyzed in our study. The time series labeled SSP1-2.6, SSP2-4.5, SSP4-6.0, and SSP5-8.5 are from the corresponding RCP scenarios. We created the time series from SSP4-3.4 and SSP3-7.0 using linear combinations of the SSP1-2.6 and SSP5-8.5 time series.

345

[Figure]

**Figure S7.** Radiative forcing time series due to tropospheric aerosols. (a) The RF time series due to tropospheric aerosols for SSP1-2.6. The solid grey circle denotes the value of AER $RF_{2011}$ given by the SSP database. The solid grey lined labeled the $-1.0$ W m$^{-2}$ time series is the AER RF time series given by the SSP database for SSP1-2.6. We appended a historical AER RF time series from the RCP scenarios and created five additional AER RF time series as described in Sect. 2.2.4. (b) Anthropogenic aerosol radiative forcing time series for SSP4-3.4.

350

[Figure]

**Figure S6̶8̲.** Measured (HadCRUT5) and modeled GMST anomaly (ΔT) relative to a pre-industrial (1850-1900) baseline for an AER RF$_{2011}$ = −0.1 W m$^{-2}$ and −1.5 W m$^{-2}$. (a) Observed (black) and modeled (red) ΔT from 1850-2019. This panel also displays the values of λ$_\Sigma$ and χ$^2_{ATM}$ (see text) for this best-fit simulation. (b) Contributions from total human activity. This panel also denotes the numerical value of the attributable anthropogenic warming rate from 1975-2014 (black dashed) as well as the 2σ uncertainty in the slope. (c) Solar irradiance (light blue) and major

360    volcanoes (purple). (d) Influences from ENSO on ΔT. (e) Contributions from AMOC to ΔT and to observed warming from 1975-2014. (f) Influences from PDO (blue) and IOD (pink) on ΔT. (g) Measured (black) and modeled (red) ocean heat content (OHC) as a function of time for the average of five data sets (see text), the value of χ$^2_{OCEAN}$ for this run, as well as the ocean heat uptake efficiency, κ, needed to provide the best-fit to the OHC record. The error bars (blue) denote the uncertainty in OHC used in this analysis (see Sect. 2.2.8). (h)-(n) Same as (a)-(g), except for AER RF$_{2011}$ =

365    −1.5 W m$^{-2}$.

[Figure]

**Figure S9.** Ocean heat content time series. The five ocean heat content data records used in this analysis, normalized
to the year 1986 because this year is in the middle of the average time series. The grey shaded region is the combined
uncertainty estimate used in this analysis, centered around the average of the five data sets. The average of the ocean
heat content records (1955 – 2017) is computed when there are three or more data sets available for a given year.

[Figure]

375 **Figure S9S10.** The ocean heat content records and uncertainty estimates analyzed in this study. (a) The average OHC record along with the standard deviation of the mean represented by the dotted black line, and the combined uncertainty of the 1σ sigma standard deviation of the average of the five OHC records and the Cheng et al. (2017) estimates shown as the dashed black line. (b) Balmaseda OHC record with the standard deviation of the five ORSA ensemble members as the dotted line, and the combined uncertainty as the dashed line. (c) Levitus OHC record with the standard error as 380 the native uncertainty, and the combined uncertainty. (d) Carton OHC record with the standard deviation of the mean of multiple ensemble members, and the combined uncertainty. (e) Cheng OHC record with the 1σ native uncertainty and the combined uncertainty. (f) Ishii OHC record with the combined uncertainty as the dashed line.

[Figure]

**Figure S11.** Measured (HadCRUT5) and modeled GMST anomaly (ΔT) relative to a pre-industrial (1850-1900) baseline without AMOC, PDO, and IOD. (a) Observed (black) and modeled (red) ΔT from 1850-2019. This panel also displays the values of $\lambda_\Sigma$ and $\chi^2_{ATM}$ (see text) for this best-fit simulation. (b) Contributions from total human activity. This panel also denotes the numerical value of the attributable anthropogenic warming rate from 1975-2014 (black dashed) as well as the 2σ uncertainty in the slope. The estimates of AAWR show similar results if AMOC is or is not included (see Fig. 1b). (c) Solar irradiance (light blue) and major volcanoes (purple). (d) Influences from ENSO on ΔT. (e-f) Contributions from AMOC, PDO, and IOD to ΔT are set to zero (g) Measured (black) and modeled (red) ocean heat content (OHC) as a function of time for the average of five data sets (see text), the value of $\chi^2_{OCEAN}$ for this run, as well as the ocean heat uptake efficiency, κ, needed to provide the best-fit to the OHC record. The error bars (blue) denote the uncertainty in OHC used in this analysis (see Sect. 2.2.8).

[Figure]

**Figure S12.** The change in GMST (ΔT) relative to 1961-1990 from the CMIP6 GCMs and the contribution from SAOD from 1975-2014. (a) ΔT from the 50 CMIP6 GCMs. (b) The residual in the change of GMST from the 50 CMIP6 GCMs after subtracting the contribution of SAOD determined by the updated REG method. The median value of AAWR is written on this panel and plotted in red. (c) The human component of global warming, $\Delta T_{ATM,HUMAN}$, from the EM-GC. A linear fit (black) and quadratic fit (red) are plotted on top to show that $\Delta T_{ATM,HUMAN}$ is almost exactly linear. (d) The contribution of SAOD in the 50 CMIP6 GCMs using a lag month calculated for each model.

400

[Figure]

**Figure S13.** The change in GMST (ΔT) relative to 1961-1990 from observations and modeled output. (a) ΔT from HadCRUT5 and EM-GC simulation. (b) The residual in ΔT from the EM-GC simulation after subtracting the contribution of SAOD determined by the REG method (grey) and ΔT due to humans from the REG method (orange). (c) ΔT due to humans from the REG method (orange) and from the EM-GC (blue). The values of AAWR determined using the REG method and Eq. (9) are shown. (d) The contribution of SAOD to ΔT. (e) Same as (a). (f) Same as (b) but also subtracting the contribution of AMOC determined by the REG method. (g) Same as (c) but using AMOC as a regressor. (h) Same as (d) also showing the contribution of AMOC to ΔT found using the REG method.

[Figure]

**Figure** S14. Values of AAWR for 50 CMIP6 GCMs using the LIN and REG methods. The solid black line is the
1:1 line and the vertical and horizontal dashed lines are the maximum value of AAWR determined using the EM-GC
and the HadCRUT temperature record. The CMIP6 GCMs that have values of AAWR less than the maximum value
from the EM-GC are blue, and the CMIP6 GCMs that have values of AAWR greater than the maximum value from
the EM-GC are red. The slope, $1\sigma$ standard deviation, and $R^2$ of the values of AAWR from the CMIP6 GCMs are
shown.

Table S2. Values of AAWR calculated using the EM-GC as a function of start and end year. The value of AAWR from 1975-2014 used in the main manuscript is shown in red. Each model run uses the best estimate of AER RF$_{2011}$ ($-0.9$ W m$^{-2}$), the average of five OHC records, and the HadCRUT GMST record. The impact on varying the start and end year on AAWR is slight, except when a short record is used (i.e. 1984-2004, a 21 year span). A two-decade time span is not long enough to calculate an accurate estimate of AAWR. The value of AAWR is more sensitive to the choice of OHC or temperature record used than the chosen time span.

**Start Year**

| AAWR (°C/decade) | 1970 | 1973 | 1975 | 1979 | 1982 | 1984 |
|---|---|---|---|---|---|---|
| **2004** | 0.181 ± 0.007 | 0.180 ± 0.009 | 0.180 ± 0.010 | 0.169 ± 0.011 | 0.159 ± 0.013 | 0.149 ± 0.012 |
| **2006** | 0.177 ± 0.008 | 0.175 ± 0.009 | 0.174 ± 0.010 | 0.163 ± 0.011 | 0.153 ± 0.012 | 0.143 ± 0.011 |
| **2008** | 0.173 ± 0.007 | 0.171 ± 0.008 | 0.169 ± 0.009 | 0.159 ± 0.010 | 0.150 ± 0.010 | 0.141 ± 0.009 |
| **2010** | 0.172 ± 0.007 | 0.169 ± 0.008 | 0.167 ± 0.008 | 0.158 ± 0.008 | 0.150 ± 0.009 | 0.143 ± 0.008 |
| **2012** | 0.171 ± 0.006 | 0.168 ± 0.007 | 0.167 ± 0.008 | 0.158 ± 0.008 | 0.152 ± 0.008 | 0.145 ± 0.007 |
| **2014** | 0.171 ± 0.005 | 0.168 ± 0.006 | 0.167 ± 0.007 | 0.160 ± 0.007 | 0.154 ± 0.007 | 0.149 ± 0.007 |
| **2016** | 0.171 ± 0.005 | 0.169 ± 0.006 | 0.168 ± 0.006 | 0.161 ± 0.006 | 0.157 ± 0.007 | 0.153 ± 0.007 |
| **2018** | 0.171 ± 0.005 | 0.170 ± 0.005 | 0.169 ± 0.006 | 0.163 ± 0.006 | 0.159 ± 0.006 | 0.156 ± 0.006 |

(Row label group: **End Year**)

**Table S3.** Average values of AAWR calculated from the CMIP6 multi-model results using the regression method as a function of start and end year. The uncertainty corresponds to the 1σ standard deviation of AAWR found from the 50 GCMs. The value of AAWR from 1975-2014 used in the main manuscript is shown in red. The values of AAWR from the CMIP6 multi-model ensemble is more sensitive to the choice of start and end year than the EM-GC due to the small number of models. We use the same start and end year, 1975-2014, for the determination of AAWR for both the EM-GC and the CMIP6 multi-model ensemble for consistency.

430

|  | AAWR (°C/decade) | Start Year | | | | | |
|---|---|---|---|---|---|---|---|
|  |  | 1970 | 1973 | 1975 | 1979 | 1982 | 1984 |
| **End Year** | **2004** | 0.185 | 0.196 | 0.200 | 0.208 | 0.224 | 0.230 |
|  | **2006** | 0.192 | 0.203 | 0.207 | 0.216 | 0.232 | 0.238 |
|  | **2008** | 0.196 | 0.207 | 0.211 | 0.220 | 0.234 | 0.241 |
|  | **2010** | 0.200 | 0.209 | 0.214 | 0.222 | 0.236 | 0.241 |
|  | **2012** | 0.204 | 0.213 | 0.218 | 0.226 | 0.239 | 0.244 |
|  | **2014** | 0.208 | 0.217 | 0.222 | 0.230 | 0.242 | 0.247 |

435

[Figure]

**Figure** S15. Steps for the calculation of ECS using the Gregory et al. (2004) method, using GISS-E2-1-H (Kelley et al., 2020) as an example. (a) The change in Abrupt $4\times CO_2$ GMST (variable: tas) from the piControl experiment for 150 years. (b) Abrupt $4\times CO_2$ net downward radiative flux (variable: rtmt) versus the Abrupt $4\times CO_2$ GMST change from the piControl experiment for 150 years. The x-intercept of the orthogonal linear least squares fit of the GCM output shown in panel (b), divided by two yields the equilibrium climate sensitivity, which in this case is 3.09°C.

440

**Table S43.** Values of AAWR from 1975-2014 for the 50 CMIP6 multi-model Historical simulations available at time of the analysis (April 2020) for both the REG and LIN methods. The asterisk symbol (*) indicates there is only one run used to compute the value of AAWR for that GCM. No asterisk indicates the AAWR value shown in the table is the average of the values of AAWR for all runs of that model. The average ratio of LIN to REG for all 50 models is $1.009 \pm 0.015$, shown at the bottom of the table and in Fig. S14. The correlation coefficient ($r^2$) of 0.995 is also shown. We conclude our determination of AAWR from the CMIP6 multi-model ensemble is accurate to $\pm 1\%$, which is much smaller than the difference between the CMIP6 multi-model ensemble values of AAWR and those found using the EM-GC framework.

| Model | AAWR, REG (°C/decade) | AAWR, LIN (°C/decade) | Model | AAWR, REG (°C/decade) | AAWR, LIN (°C/decade) |
|---|---|---|---|---|---|
| ACCESS-CM2 | 0.211 | 0.216 | GFDL-CM4* | 0.243 | 0.250 |
| ACCESS-ESM1-5 | 0.238 | 0.246 | GFDL-ESM4 | 0.203 | 0.224 |
| AWI-CM-1-1-MR | 0.215 | 0.220 | GISS-E2-1-G | 0.194 | 0.198 |
| BCC-CSM2-MR | 0.217 | 0.228 | GISS-E2-1-G-CC | 0.204 | 0.213 |
| BCC-ESM1 | 0.241 | 0.249 | GISS-E2-1-H | 0.237 | 0.244 |
| CAMS-CSM1-0 | 0.131 | 0.138 | HadGEM3-GC31-LL | 0.283 | 0.292 |
| CanESM5 | 0.354 | 0.361 | HadGEM3-GC31-MM | 0.227 | 0.234 |
| CanESM5-CanOE | 0.323 | 0.334 | INM-CM4-8* | 0.173 | 0.181 |
| CAS-ESM2-0 | 0.196 | 0.204 | INM-CM5-0 | 0.146 | 0.156 |
| CESM2 | 0.240 | 0.243 | IPSL-CM6A-LR | 0.230 | 0.236 |
| CESM2-FV2 | 0.221 | 0.224 | KACE-1-0-G | 0.254 | 0.260 |
| CESM2-WACCM | 0.273 | 0.291 | MCM-UA-1-0 | 0.225 | 0.231 |
| CESM2-WACCM-FV2 | 0.231 | 0.235 | MIROC6 | 0.157 | 0.168 |
| CIESM | 0.245 | 0.251 | MIROC-ES2L | 0.163 | 0.167 |
| CNRM-CM6-1 | 0.202 | 0.196 | MPI-ESM1-2-HAM | 0.180 | 0.186 |
| CNRM-CM6-1-HR* | 0.172 | 0.178 | MPI-ESM1-2-HR | 0.195 | 0.203 |
| CNRM-ESM2-1 | 0.170 | 0.172 | MPI-ESM1-2-LR | 0.192 | 0.197 |
| E3SM-1-0 | 0.267 | 0.278 | MRI-ESM2-0 | 0.203 | 0.210 |
| E3SM-1-1* | 0.283 | 0.285 | NESM3 | 0.242 | 0.253 |
| E3SM-1-1-ECA* | 0.275 | 0.274 | NorCPM1 | 0.180 | 0.185 |
| EC-Earth3* | 0.299 | 0.310 | NorESM2-LM | 0.167 | 0.182 |
| EC-Earth3-Veg* | 0.214 | 0.223 | NorESM2-MM* | 0.151 | 0.154 |
| FGOALS-f3-L | 0.218 | 0.226 | SAM0-UNICON* | 0.245 | 0.250 |
| FGOALS-g3 | 0.176 | 0.191 | TaiESM1* | 0.273 | 0.283 |
| FIO-ESM-2-0 | 0.229 | 0.237 | UKESM1-0-LL | 0.299 | 0.312 |
| Ratio = $1.009 \pm 0.015$ | | | $R^2 = 0.995$ | | |

[Figure]

**Figure S16.** Values of ECS found using the EM-GC and the CMIP6 multi-model ensemble without the aerosol weighting method. Values of ECS utilizing the EM-GC are calculated using seven temperature data sets and five ocean heat content records (as indicated). The box represents the 25th, 50th, and 75th percentiles of the values of ECS and the whiskers denote the 5th and 95th percentiles for the different OHC records and each temperature record without using the aerosol weighting method (unweighted). The stars indicate the minimum and maximum values of ECS. The circles are the values of ECS associated with the best estimate of AER $RF_{2011}$ of −0.9 W m$^{-2}$. The box labeled CMIP6 is the 25th, 50th, and 75th percentiles of the values of ECS from the CMIP6 multi-model ensemble, the whiskers indicate the 5th and 95th percentiles, and the stars represent the minimum and maximum values of ECS from the CMIP6 multi-model ensemble.

**Table S54.** Equilibrium climate sensitivity (ECS) from 28 CMIP6 GCMs. We can only calculate ECS for GCMs that provide Abrupt 4×$CO_2$ near surface air temperature (output variable: tas), net downward radiative flux (output variable: rtmt), and piControl near surface air temperature (output variable: tas) to the CMIP6 archive at time of the analysis (April 2020). All estimates are for one model run except for CanESM5, which is the average of two runs.

| Model | ECS (K) |
|---|---|
| ACCESS-CM2 | 4.93 |
| ACCESS-ESM1-5 | 3.63 |
| BCC-CSM2-MR | 3.16 |
| BCC-ESM1 | 3.74 |
| CanESM5 | 5.70 |
| CESM2 | 5.32 |
| CESM2-FV2 | 5.06 |
| CESM2-WACCM | 4.73 |
| CESM2-WACCM-FV2 | 4.56 |
| E3SM-1-0 | 5.28 |
| EC-Earth3-Veg | 4.34 |
| GFDL-CM4 | 3.78 |
| GFDL-ESM4 | 2.61 |
| GISS-E2-1-G | 2.71 |
| GISS-E2-2-G | 2.25 |
| GISS-E2-1-H | 3.09 |
| HadGEM3-GC31-LL | 5.65 |
| INM-CM4-8 | 2.32 |
| INM-CM5-0 | 2.39 |
| IPSL-CM6A-LR | 4.97 |
| MCM-UA-1-0 | 3.68 |
| MIROC6 | 2.84 |
| MIROC-ES2L | 2.83 |
| NorESM2-LM | 2.19 |
| NorESM2-MM | 2.15 |
| SAM0-UNICON | 3.53 |
| TaiESM1 | 4.33 |
| UKESM1-0-LL | 5.40 |

[Figure]

**Figure S17.** Values of ECS versus AAWR for the CMIP6 multi-model ensemble. The EM-GC estimates of AAWR and ECS based on training to the HadCRUT5 GMST record are plotted as a box and whisker. The box shows the average 25th, 50th, and 75th percentiles for the five OHC records shown for HadCRUT5 in Fig. 6 and Fig. 7. The whiskers represent the average 5th and 95th percentiles. The stars denote the average minimum and maximum values of AAWR or ECS. The eight CMIP6 GCMs that obtain values of AAWR and ECS that are both within the minimum and maximum estimates provided by the EM-GC are identified on the figure. Values of AAWR explain about 78% of the variance in ECS among the CMIP6 GCMs.

[Figure]

480 **Figure** S18. GMST anomaly in 2100 from pre-industrial ($\Delta T_{2100}$) as a function of climate feedback parameter and AER RF$_{2011}$ found using the EM-GC trained with $\Delta T$ from HadCRUT5. (a) $\Delta T_{2100}$ for SSP4-6.0. The region outside of the tropospheric aerosol radiative forcing rage provided by IPCC 2013 (Myhre et al., 2013) is shaded grey. Colors denote the change in $\Delta T_{2100}$. (b) $\Delta T_{2100}$ for SSP3-7.0. (c) $\Delta T_{2100}$ for SSP5-8.5.

[Figure]

**Figure S19.** Probabilistic forecasts of future projections of ΔT using the EM-GC trained with ΔT from HadCRUT5 for the SSP4-6.0, SSP3-7.0, and SSP5-8.5 scenarios. (a) Future projections of ΔT for SSP4-6.0. Observations (orange) are from HadCRUT5. The IPCC 2013 likely range of warming (black) is from Figure 11.25b of chapter 11 of the IPCC 2013 report. The Paris Agreement target and upper limit (yellow) are shown for comparison to projections of ΔT using the EM-GC. The CMIP6 minimum, multi-model mean, and maximum values of the rise in ΔT are shown to compare to projections from the EM-GC. Colors denote the probability of reaching at least that temperature by the end of the century and are computed using the aerosol weighting method (see Sect. 2.5). (b) Future projections of ΔT for SSP3-7.0. (c) Future projections of ΔT for SSP5-8.5.

495

[Figure]

**Figure** S17S20. Probability density functions (PDF) for the increase in $\Delta T_{2100}$ using the EM-GC and the CMIP6 multi-model ensemble. (a) PDF for EM-GC (blue) results trained with $\Delta T$ from HadCRUT5 and CMIP6 multi-model results (red) for SSP4-6.0. The left-hand y-axis is for EM-GC probabilities and the righthand y-axis is for GCM probabilities. (b) PDF for SSP3-7.0. (c) PDF for SSP5-8.5.

500

**Table S6.** Probabilities of achieving the Paris Agreement target and upper limit for the various SSP scenarios based on the EM-GC using the HadCRUT4 or HadCRUT5 GMST data set and the CMIP6 multi-model ensemble. The probabilities using the EM-GC are computed using the aerosol weighting method. The probabilities using the CMIP6 GCMs are computed by calculating how many of the models for that scenario are below the temperature limits compared to the total number of models.

| | Probability of Staying at or Below 1.5°C | | | Probability of Staying at or Below 2.0°C | | |
|---|---|---|---|---|---|---|
| | HadCRUT4 | HadCRUT5 | CMIP6 | HadCRUT4 | HadCRUT5 | CMIP6 |
| **SSP1-1.9** | 84% | 81% | 50% | 99% | 98% | 80% |
| **SSP1-2.6** | 64% | 53% | 18% | 90% | 86% | 47% |
| **SSP4-3.4** | 35% | 19% | 0% | 74% | 64% | 17% |
| **SSP2-4.5** | 9% | 0% | 0% | 52% | 33% | 3% |
| **SSP4-6.0** | 0% | 0% | 0% | 26% | 8% | 0% |
| **SSP3-7.0** | 0% | 0% | 0% | 1% | 0% | 0% |
| **SSP5-8.5** | 0% | 0% | 0% | 0% | 0% | 0% |

[Figure]

**Figure S21.** Transient climate response to cumulative $CO_2$ emissions for SSP5-8.5 using the EM-GC trained with the HadCRUT5 $\Delta T$ record. Simulations of the rise in $\Delta T$ versus cumulative $CO_2$ emissions in units of Gt C. The orange line is observations of $\Delta T$ from HadCRUT5 plotted against cumulative carbon emissions from the Global Carbon Budget project (Friedlingstein et al., 2019). The dotted and dashed lines denote the Paris Agreement target and upper limit, respectively. The EM-GC projections represent the probability that the future value of $\Delta T$ will rise to the indicated level, considering only acceptable fits to the climate record. The probabilities were determined using the aerosol weighting method. The light grey, dark grey, and black curves denote the 95, 66, and 50% probabilities of either the Paris target (intersection of dotted horizontal lines) or upper limit (intersection of dashed lines with curves) being achieved.

[Figure]

**Figure** S22. Blended methane mixing ratios. The dotted lines are linear combinations of the time series of methane abundances using SSP1-2.6 and SSP3-7.0 to span the range of values of future methane. The solid lines are the SSP1-2.6 and SSP3-7.0 methane mixing ratio time series.

520

[Figure]

**Figure S23.** Change in GMST ($\Delta T$) from 1850-2019 for observations from HadCRUT5 (black) and 1850-2100 for modeled (red) using SSP4-3.4 and the residual between modeled and observations using an instantaneous time variant $\lambda^{-1}$. (a) $\Delta T$ assuming a constant value of $\lambda^{-1}$. (b) $\Delta T$ allowing $\lambda^{-1}$ to increase by 50%. (c) $\Delta T$ allowing $\lambda^{-1}$ to vary while the value of $\chi^2_{RECENT}$ is kept below 2. (d) $\Delta T$ allowing $\lambda^{-1}$ to vary while the value of $\chi^2_{ATM}$ is kept below 2. (e) Residual between modeled and observed $\Delta T$ from 1850-2019 for constant $\lambda^{-1}$. (f) Same as (e) but for increasing $\lambda^{-1}$ by 50%. (g) Same as (f) but for varying $\lambda^{-1}$ while the value of $\chi^2_{RECENT}$ is kept below 2. (h) same as (g)but for varying $\lambda^{-1}$ while the value of $\chi^2_{ATM}$ is kept below 2.

**Table** S7**.** Details of the CMIP6 GCMs used in this study.

| Institution | Model | Model Output |
|---|---|---|
| AS-RCEC | TaiESM1 | No reference provided |
| AWI | AWI-CM-1-1-MR | (Semmler et al., 2018a, 2018b, 2018c, 2019a, 2019b) |
| BCC | BCC-CSM2-MR | (Wu et al., 2018a, 2018b, 2018c; Xin et al., 2019a, 2019b, 2019c, 2019d) |
| | BCC-ESM1 | (Zhang et al., 2018a, 2018b, 2019) |
| CAMS | CAMS-CSM1-0 | (Rong, 2019a, 2019b, 2019c, 2019d, 2019e, 2019f) |
| CAS | CAS-ESM2-0 | (Chai, 2019) |
| | FGOALS-f3-L | (YU, 2019a, 2019b, 2019c, 2019d, 2019e) |
| | FGOALS-g3 | (Li, 2019a, 2019b, 2019c, 2019d, 2019e) |
| CCCma | CanESM5 | (Swart et al., 2019f, 2019g, 2019h, 2019i, 2019j, 2019k, 2019l, 2019m, 2019n, 2019o) |
| | CanESM5-CanOE | (Swart et al., 2019a, 2019b, 2019c, 2019d, 2019e) |
| CNRM-CERFACS | CNRM-CM6-1 | (Voldoire, 2018, 2019c, 2019d, 2019e, 2019f) |
| | CNRM-CM6-1-HR | (Voldoire, 2019a, 2019b, 2020a, 2020b) |
| | CNRM-ESM2-1 | (Seferian, 2018; Voldoire, 2019g, 2019h, 2019i, 2019j, 2019k, 2019l) |
| CSIRO | ACCESS-ESM1-5 | (Ziehn et al., 2019a, 2019b, 2019c, 2019d, 2019e, 2019f, 2019g) |
| CSIRO-ARCCSS | ACCESS-CM2 | (Dix et al., 2019a, 2019b, 2019c, 2019d, 2019e, 2019f, 2019g) |
| E3SM-Project | E3SM-1-0 | (Bader et al., 2018, 2019a, 2019b) |
| | E3SM-1-1-ECA | (Bader et al., 2020) |
| E3SM-Project RUBISCO | E3SM-1-1 | (Bader et al., 2019c) |
| EC-Earth-Consortium | EC-Earth3 | (EC-Earth Consortium (EC-Earth), 2019i, 2019j, 2019k, 2019l, 2019m) |
| | EC-Earth3-Veg | (EC-Earth Consortium (EC-Earth), 2019a, 2019b, 2019c, 2019d, 2019e, 2019f, 2019g, 2019h) |

| | | |
|---|---|---|
| FIO-QLNM | FIO-ESM-2-0 | (Song et al., 2019a, 2019b, 2019c, 2019d) |
| HAMMOZ-Consortium | MPI-ESM1-2-HAM | (Neubauer et al., 2019) |
| INM | INM-CM4-8 | (Volodin et al., 2019a, 2019b, 2019c, 2019d, 2019e, 2019f, 2019g) |
| | INM-CM5-0 | (Volodin et al., 2019m, 2019h, 2019n, 2019i, 2019j, 2019k, 2019l) |
| IPSL | IPSL-CM6A-LR | (Boucher et al., 2018a, 2018b, 2018c, 2019a, 2019b, 2019c, 2019d, 2019e, 2019f, 2019g) |
| MIROC | MIROC6 | (Shiogama et al., 2019a, 2019b, 2019c, 2019d, 2019e, 2019f, 2019g; Tatebe and Watanabe, 2018a, 2018b, 2018c) |
| | MIROC-ES2L | (Hajima et al., 2019; Tachiiri et al., 2019a, 2019b, 2019c, 2019d, 2019e) |
| MOHC | HadGEM3-GC31-MM | (Ridley et al., 2019c) |
| MOHC NERC | HadGEM3-GC31-LL | (Good, 2019, 2020a, 2020b; Ridley et al., 2018, 2019a, 2019b) |
| MOHC, NERC, NIMS-KMA, NIWA | UKESM1-0-LL | (Byun, 2020; Good et al., 2019a, 2019b, 2019c, 2019d, 2019e, 2019f; Tang et al., 2019a, 2019b, 2019c) |
| MPI-M AWI | MPI-ESM1-2-LR | (Wieners et al., 2019a, 2019b, 2019c, 2019d, 2019e) |
| MPI-M DWD DKRZ | MPI-ESM1-2-HR | (Jungclaus et al., 2019; Schupfner et al., 2019a, 2019b, 2019c, 2019d; Steger et al., 2019) |
| MRI | MRI-ESM2-0 | (Yukimoto et al., 2019a, 2019b, 2019c, 2019d, 2019e, 2019f, 2019g, 2019h) |
| NASA-GISS | GISS-E2-1-G | (NASA Goddard Institute for Space Studies (NASA/GISS), 2018a, 2018b, 2018c, 2020a, 2020b, 2020c, 2020d) |
| | GISS-E2-1-G-CC | No reference provided |
| | GISS-E2-2-G | (NASA Goddard Institute for Space Studies (NASA/GISS), 2019a) |
| | GISS-E2-1-H | (NASA Goddard Institute for Space Studies (NASA/GISS), 2018d, 2019b, 2019c) |
| NCAR | CESM2-WACCM-FV2 | (Danabasoglu, 2019d, 2019e, 2020a) |

| | | |
|---|---|---|
| | CESM2 | (Danabasoglu, 2019c, 2019d, 2019e, 2019f, 2019g, 2019h; Danabasoglu et al., 2019) |
| | CESM2-FV2 | (Danabasoglu, 2019b, 2019c, 2020b) |
| | CESM2-WACCM | (Danabasoglu, 2019f, 2019g, 2019h, 2019a, 2019i, 2019j, 2019k) |
| | NorCPM1 | (Bethke et al., 2019a, 2019b, 2019c) |
| NCC | NorESM2-LM | (Seland et al., 2019a, 2019b, 2019c, 2019d, 2019e, 2019f, 2019g) |
| | NorESM2-MM | (Bentsen et al., 2019a, 2019b, 2019c, 2019d, 2019e, 2019f, 2019g) |
| NIMS-KMA | KACE-1-0-G | (Byun et al., 2019a, 2019b, 2019c, 2019d, 2019e) |
| NOAA-GFDL | GFDL-CM4 | (Guo et al., 2018a, 2018b, 2018c, 2018d, 2018e) |
| | GFDL-ESM4 | (John et al., 2018a, 2018b, 2018c, 2018d, 2018e; Krasting et al., 2018a, 2018b, 2018c) |
| NUIST | NESM3 | (Cao, 2019a, 2019b, 2019c; Cao and Wang, 2019) |
| SNU | SAM0-UNICON | (Park and Shin, 2019a, 2019b, 2019c) |
| THU | CIESM | (Huang, 2019a, 2019b, 2020a, 2020b) |
| UA | MCM-UA-1-0 | (Stouffer, 2019a, 2019b, 2019c, 2019d, 2019e, 2019f, 2019g) |

535

540

---

## Author Response (AR3)

Reviewer comments are in black and author responses are in blue.

The authors have addressed two of my major comments almost to satisfaction (1 and 4), and for two others I'm willing to agree to disagree (5 and 6). However, the answers to major comment 3, and to lesser extent 2, remain unsatisfactory.

Major comment 1)
I see no reason to include JMA in the analysis. It has an even lower spatial coverage than HadCRUT4 and is therefore not a global temperature. Its uncritical inclusion may therefore lead to a biased understanding of the topic. Furthermore, the images are really crammed. Similarly, Cowtan can be dropped, now that HadCRUT5 is out, improving Figures 6 and 7.

> It is important for us to continue to show results for both JMA and Cowtan and Way (CW14) datasets. These two maps show the spatial coverage of the JMA and HadCRUT4 data sets in 2019. They have similar spatial coverage. Hence, the notion that the JMA data set does not reflect global mean surface temperature could be applied as well to the HadCRUT4 data set, which has long been used in studies similar to our analysis.

[Figure]

The circles indicate temperature anomalies from 1981-2010 baseline averaged in 5° x 5° grid boxes.

JMA map website (https://ds.data.jma.go.jp/tcc/tcc/products/gwp/temp/map/temp_map.html)

> We strongly prefer to retain both the JMA and CW14 data sets to represent the current state of knowledge based on the seven GMST data sets that are available. Including CW14 is important because our analysis displayed in Figs. 6 and 7 is now the first to show that the transformation of HadCRUT4 into a more complete global coverage data set by CW14 results in very similar values of AAWR and ECS obtained from HadCRUT5. This is an important result that will be of interest to the community. It would not be fair to Kevin Cowtan and Robert Way, creators of this data set, to relegate the result found using their time series of GMST to the supplement.

[Figure]

HadCRUT4 map website (https://crudata.uea.ac.uk/~timo/diag/tempdiag.htm)

Major comment 3)

In Figure 14, the authors show that their model gives a good fit for a wide variety of assumptions on lambda. They show that an 50% higher ECS is very well in line with observations, and that even a doubling is still consistent with their criterion of $X^2<2$. Despite this, the authors say choosing a point estimate is reasonable, by visual inspection of the graph (14e-h) that I find questionable and is not in line with the $X2 < 2$ criterion used in the rest of the manuscript.

> Thank you for this comment. We have heavily revised our discussion of effective climate sensitivity in Sect. 3.3.6, the Abstract, and the Conclusions to reflect values found using time varying climate feedback. The specific, detailed changes are given in the response to comments that appear below.

With the insistence of using the assumption that lamdba is constant, the authors do not compute ECS and do not give a 'comprehensive analysis of uncertainties in AAWR, ECS, and projections of delta T in our EM-GC framework,', as they claim in the conclusion. Instead, the authors compute what is sometimes called effective climate sensitivity (https://iopscience.iop.org/article/10.1088/1748-9326/ab738f), and their analysis of future temperatures should be described as a lower bound consistently throughout the entire manuscipt. By comparing effective ECS with ECS computed with the Gregory method, they compare apples with pears. In discussing other papers, the authors also do not make this important distinction.

> We have examined several other studies to determine the type of climate sensitivity we are estimating within the EM-GC framework and for the CMIP6 GCMs via the Gregory et al. (2004) method. As indicated by the paper you provided (Tokarska et al., 2020) that is now cited, we are indeed estimating effective climate sensitivity with the EM-GC. We have updated our abstract, Sect. 2.4, Sect. 3.2, and the conclusions to make it clear to the reader we are estimating effective climate sensitivity rather than equilibrium climate sensitivity. In our abstract on line 13, we define ECS as effective climate sensitivity. For

Sect. 2.4, we have renamed the section header **Effective climate sensitivity**. We also added a new sentence starting on line 564 that reads: **In our model framework, we infer the climate sensitivity based on an estimate of climate feedback from the historical record, resulting in the effective climate sensitivity (ECS) (Tokarska et al., 2020a).** In Sect. 3.2, we redefine ECS as **Effective climate sensitivity.**

Close inspection of Gregory et al. (2004), Zelinka et al. (2020) and Sherwood et al. (2020) indicate that the method devised by Gregory et al. (2004) also computes *effective climate sensitivity* from the GCMs rather than true equilibrium climate sensitivity, because the Gregory et al. (2004) method assumes the feedbacks inferred from the first 150 years following the abrupt $4\times CO_2$ simulation persist until equilibrium. We have added the following text starting on line 576: **We refer to the quantity in Eq. (10) as effective climate sensitivity, rather than equilibrium climate sensitivity, because for most of our analysis we assume a constant value of climate feedback inferred from prior observations.** and on line 581: **The Gregory et al. (2004) method also estimates effective climate sensitivity from the CMIP6 GCMs (Gregory et al., 2004; Sherwood et al., 2020; Zelinka et al., 2020) because it assumes the feedbacks inferred from the first 150 years of the abrupt $4\times CO_2$ CMIP6 GCM simulations persist until equilibrium**

The estimates of effective climate sensitivity based on Eq. (10) and the Gregory et al. (2004) method are not comparing the exact same type of effective climate sensitivity because the Gregory et al. (2004) method involves a large perturbation to RF that has not occurred in the historical climate record. We have added the following text starting on line 593 to make this point clear to the reader: **The estimates of climate sensitivity from Eq. (10) and those found using the Gregory et al. (2004) method are termed "effective" because they assume climate feedback inferred from either the historical climate record or the abrupt $4\times CO_2$ experiment persists until equilibrium. However, these estimates of ECS differ in that the perturbation to the RF of climate over the historical record is considerably smaller than the RF of climate that underlies the $4\times CO_2$ experiment of the Gregory et al. (2004) method. We quantify the impact of time variable climate feedback on climate sensitivity in Sect. 3.3.6.**

We have also added information to the Fig. 8 caption designating whether the estimate of climate sensitivity provided by other studies are effective climate sensitivity or equilibrium climate sensitivity. We use the definitions given by Gregory et al. (2004), Tokarska et al. (2020), and Zelinka et al. (2020) that effective climate sensitivity estimates assume values of climate feedback inferred from either the historical record or the first 150 years of the Gregory method will persist until equilibrium. Some of the studies have designated their estimates of climate sensitivity as effective, while others have demonstrated why theirs are true estimates of equilibrium climate sensitivity. A few studies have not specified if their estimates of climate sensitivity are truly at equilibrium, so we used information from the manuscripts to make a designation. We added the following text to the Fig. 8 caption designating which studies provide estimates of

effective or equilibrium climate sensitivity: **The studies estimating effective climate sensitivity are AO13, NL+PG18, RS18, FN20, SS20 KT20a, KT20b, and MZ20. The studies estimating equilibrium climate sensitivity are KA17, AD18, PC18, MR20, and CP+PH17. See the supplement for…more information about which studies are estimating effective and equilibrium climate sensitivity.**

We added new text starting on line 250 of the supplement to explain to the reader how we designate the studies providing effective or equilibrium climate sensitivity: **The Fig. 8 caption in Sect. 3.2 also refers to the supplement for information about which studies are estimating effective climate sensitivity or equilibrium climate sensitivity. We designate each study based on information found in their manuscripts if their analysis uses the Gregory et al. (2004) method or infers climate feedback from the historical climate record will persist until equilibrium. The use of either of these two factors results in our designation of effective climate sensitivity (Gregory et al., 2004; Sherwood et al., 2020; Tokarska et al., 2020a; Zelinka et al., 2020). Based on our examination of IPCC 2013, it seems their estimate is a combination of effective climate sensitivity and equilibrium climate sensitivity.**

I had very much hoped the researchers would extend their model so that they compute true ECS instead. A simultaneous evaluation of time variation in lambda and aerosol uncertainty would lead to interesting results, considering the authors are able to account for internal variability.

\* The authors state in the abstract that RF of aerosols is the main uncertainty but show in their results that the time-component of lambda is equally uncertain (I quote: Increasing $\lambda-1$ by 50% results in a similar value of $\Delta T2100$ as when utilizing a higher value of AER RF2011 (i.e., AER RF2011 less than $-0.9$ W m$-2$) in the EM-GC framework).

Thanks for these excellent comments that has led to an important improvement to our paper.

First, the uncertainty in aerosol radiative forcing (AER RF) is the main uncertainty for values of AAWR. The value of climate feedback, if it has varied over the historical climate record, would change by only a small amount from 1975-2014, the time period of the analysis for AAWR. The impact of uncertainty on AAWR of time variant lambda would be very small, so the dominant uncertainty for AAWR remains AER RF.

For effective climate sensitivity (ECS), the uncertainty in AER RF is still the dominant form of uncertainty. However, the variation in $\lambda^{-1}$ over time does introduce an additional, important uncertainty in our estimates of ECS that we now consider. New Figure 14 shows the change in $\Delta T$ for SSP2-4.5 using a simulation of the EM-GC that is trained to the HadCRUT5 GMST record, uses a value of AER RF$_{2011}$ = $-0.9$ W m$^{-2}$, and the average of the five data sets for the OHC record. The best estimate of ECS given in Sect. 3.2 of our manuscript is 2.33°C. If we allow $\lambda^{-1}$ to vary over time in a manner that mimics the behavior of GCMs, the best estimate of ECS, which we denote as ECS$_{\lambda(t)}$, is

3.08°C (range 2.23 to 5.53°C). We have added new text to the Abstract, Sect. 3.3.6, and the Conclusions pointing out these numerical values of $ECS_{\lambda(t)}$. We also added information to the supplement for estimates of $ECS_{\lambda(t)}$ if we allow $\lambda^{-1}$ to increase over time so values of $\chi^2_{RECENT}$ are less than or equal to two. Our estimate of $ECS_{\lambda(t)}$ rises to 3.52°C (range of 2.71 to 5.53°C)

We added New Fig. 14 to the main manuscript and New Figs. S24 and S25 to the supplement to address this comment and support our estimates of $ECS_{\lambda(t)}$. We also added text starting on line 1125 illustrating how our estimates of ECS will change if we allow $\lambda^{-1}$ to rise over time. We have also added the new estimate of $ECS_{\lambda(t)}$ to the abstract and the conclusions. We added the following text to the end of Sect. 3.3.6 starting on line 1143 to indicate that the change in $\lambda^{-1}$ is an additional uncertainty for ECS: **If $\lambda^{-1}$ is allowed to increase by 50%, our best estimate of ECS would rise from 2.33 to 3.08°C, which is a 32% increase. Time variant $\lambda^{-1}$ introduces additional uncertainty into our estimates of ECS; however, the largest uncertainty is still due to the imprecise knowledge of the RF due to tropospheric aerosols.**

To further address this reviewer's comment, we have extended our analysis to include a model simulation where only the RF due to $CO_2$ changes into the future. We denote this scenario as SSP2-4.5′. All other GHGs and the RF due to tropospheric aerosols are held constant at their December 2019 values into the future. We used SSP2-4.5 because $CO_2$ doubles from preindustrial concentrations in this scenario, approximating the definition of equilibrium climate sensitivity. The results from SSP2-4.5′ are shown as the dotted lines in Fig. 14, S23, S24, and S25.

We have added the following text starting on line 1077 of our main manuscript describing our SSP2-4.5′ simulation: **In Figs. 14 and S23 we also analyze a RF scenario termed SSP2-4.5′ that serves as a doubled $CO_2$ scenario (dotted lines). For SSP2-4.5′, the RFs due to all GHGs other than $CO_2$ as well as tropospheric aerosols from the start of 2020 onwards are held constant at end of 2019 values. The only component of RF allowed to vary after the start of 2020 is $CO_2$. The RF of climate due to all GHGs and tropospheric aerosols for SSP2-4.5′ is identical to that in SSP2-4.5 from the start of the simulation until the end of 2019. Since the mixing ratio of $CO_2$ at the end of century is 566 ppm, the warming found at the end of century for SSP2-4.5′ serves as the transient response of $\Delta T$ to rising $CO_2$ in our model framework. The fact that projections of $\Delta T$ found allowing only for future increases in $CO_2$ (dotted lines) agree so closely with those found assuming changes in RF due to all GHGs and tropospheric aerosols (solid lines) means that under the AER $RF_{2011} = -0.9$ W m$^{-2}$ scaling assumption, the future change in RF due to all agents other than $CO_2$ nearly cancel. Projections found using the original SSP2-4.5 scenario may serve as a useful surrogate for a double $CO_2$ simulation. Figures S24 and S25 are the same as Fig. 14, except for the use of AER $RF_{2011}$ values of −0.4 and −1.5 W m$^{-2}$, respectively. There are slight departures between the SSP2-4.5 and SSP2-4.5′ projections of $\Delta T$ for these alternate aerosol scaling assumptions.**

**Nonetheless, these projections are quite similar because the future decline in RF due to the assumption of declining CH$_4$ within SSP2-4.5 nearly balances the future increase in RF due to N$_2$O and all of the minor GHGs.**

We have also added the following text starting on line 281 of the supplement to further describe our SSP2-4.5′ simulation: **Section 3.3.6 also states "In Figs. 14 and S23 we also analyze a RF scenario termed SSP2-4.5′ that serves as a doubled CO$_2$ scenario (dotted lines)". In the SSP2-4.5′ simulation, only CO$_2$ is allowed to change after the end of 2019. All other GHGs and aerosols are kept constant at their December 2019 values. This simulation allows us to examine the effect of time variant $\lambda^{-1}$ on changes in $\Delta$T due only to the future rise in CO$_2$. In the SSP2-4.5 scenario, CH$_4$, tropospheric O$_3$, and ODSs decrease after 2019 leading to a future decline in RF, whereas N$_2$O and tropospheric aerosols result in a future increase in RF. When all of these RF are kept constant in the SSP2-4.5′ scenario for the AER RF$_{2011}$ = −0.9 W m$^{-2}$ scaling assumption, the terms result in a near balance out to 2100. For the weaker aerosol cooling scenario (AER RF$_{2011}$ = −0.4 W m$^{-2}$), the value of RF due to tropospheric aerosols is not large enough to completely offset the other GHGs that are held constant. Consequently, the SSP2-4.5′ simulation (dotted line) results in slightly larger total RF and associated warming than the SSP2-4.5 scenario (solid line) shown in Fig. S24. For the stronger aerosol cooling scenario (AER RF$_{2011}$ = −1.5 W m$^{-2}$), the value of RF due to tropospheric aerosols is larger than the RF due to the other GHGs that are held constant, resulting in the SSP2-4.5′ having a slightly lower total RF and associated warming than the SSP2-4.5 scenario (Fig. S25).**

[Figure]

**New Figure 14.** Change in GMST from 1850-2019 for observations from HadCRUT5 (black) and 1850-2100 for modeled (red) using SSP2-4.5 and a value of AER $RF_{2011}$ = −0.9 W m$^{-2}$ and the residual between modeled and observations incorporating a 32.5-year delay between $\lambda^{-1}$ and a change in RF. The solid line denotes a simulation for the original SSP2-4.5 scenario and the dashed line indicates the SSP2-4.5′ simulation (see text). (a) Rise in GMST assuming a constant value of $\lambda^{-1}$. (b) Rise in GMST allowing $\lambda^{-1}$ to increase by 50%. (c) Rise in GMST allowing $\lambda^{-1}$ to vary while the value of $\chi^2_{RECENT}$ is kept below 2. (d) Rise in GMST allowing $\lambda^{-1}$ to vary while the value of $\chi^2_{ATM}$ is kept below 2. (e) Residual between modeled and observed rise in GMST from 1850-2019 for constant $\lambda^{-1}$. (f) Same as (e) but for increasing $\lambda^{-1}$ by 50%. (g) Same as (f) but for varying $\lambda^{-1}$ while the value of $\chi^2_{RECENT}$ is kept below 2. (h) same as (g) but for varying $\lambda^{-1}$ while the value of $\chi^2_{ATM}$ is kept below 2.

[Figure]

**New Figure S23.** Change in GMST ($\Delta T$) from 1850-2019 for observations from HadCRUT5 (black) and 1850-2100 for modeled (red) using SSP2-4.5 and a value of AER $RF_{2011}$ = −0.9 W m$^{-2}$ and the residual between modeled and observations using an instantaneous time variant $\lambda^{-1}$. The solid line denotes a simulation for the original SSP2-4.5 scenario and the dashed line indicates the SSP2-4.5′ simulation (see Sect. 3.3.6). (a) $\Delta T$ assuming a constant value of $\lambda^{-1}$. (b) $\Delta T$ allowing $\lambda^{-1}$ to increase by 50%. (c) $\Delta T$ allowing $\lambda^{-1}$ to vary while the value of $\chi^2_{RECENT}$ is kept below 2. (d) $\Delta T$ allowing $\lambda^{-1}$ to vary while the value of $\chi^2_{ATM}$ is kept below 2. (e) Residual between modeled and observed $\Delta T$ from 1850-2019 for constant $\lambda^{-1}$. (f) Same as (e) but for increasing $\lambda^{-1}$ by 50%. (g) Same as (f) but for varying $\lambda^{-1}$ while the value of $\chi^2_{RECENT}$ is kept below 2. (h) same as (g) but for varying $\lambda^{-1}$ while the value of $\chi^2_{ATM}$ is kept below 2.

[Figure]

**New Figure S24.** Change in GMST ($\Delta T$) from 1850-2019 for observations from HadCRUT5 (black) and 1850-2100 for modeled (red) using SSP2-4.5 and a value of AER $RF_{2011}$ = −0.4 W m$^{-2}$ and the residual between modeled and observations incorporating a 32.5-year delay between $\lambda^{-1}$ and a change in RF. The solid line denotes a simulation for the original SSP2-4.5 scenario and the dashed line indicates the SSP2-4.5′ simulation (see Sect. 3.3.6). (a) $\Delta T$ assuming a constant value of $\lambda^{-1}$. (b) $\Delta T$ allowing $\lambda^{-1}$ to increase by 50%. (c) $\Delta T$ allowing $\lambda^{-1}$ to vary while the value of $\chi^2_{RECENT}$ is kept below 2. (d) $\Delta T$ allowing $\lambda^{-1}$ to vary while the value of $\chi^2_{ATM}$ is kept below 2. (e) Residual between modeled and observed $\Delta T$ from 1850-2019 for constant $\lambda^{-1}$. (f) Same as (e) but for increasing $\lambda^{-1}$ by 50%. (g) Same as (f) but for varying $\lambda^{-1}$ while the value of $\chi^2_{RECENT}$ is kept below 2. (h) same as (g) but for varying $\lambda^{-1}$ while the value of $\chi^2_{ATM}$ is kept below 2.

[Figure]

**New Figure S25.** Change in GMST ($\Delta T$) from 1850-2019 for observations from HadCRUT5 (black) and 1850-2100 for modeled (red) using SSP2-4.5 and a value of AER $RF_{2011} = -1.5$ W m$^{-2}$ and the residual between modeled and observations incorporating a 32.5-year delay between $\lambda^{-1}$ and a change in RF. The solid line denotes a simulation for the original SSP2-4.5 scenario and the dashed line indicates the SSP2-4.5′ simulation (see Sect. 3.3.6). (a) $\Delta T$ assuming a constant value of $\lambda^{-1}$. (b) $\Delta T$ allowing $\lambda^{-1}$ to increase by 50%. (c) $\Delta T$ allowing $\lambda^{-1}$ to vary while the value of $\chi^2_{RECENT}$ is kept below 2. (d) $\Delta T$ allowing $\lambda^{-1}$ to vary while the value of $\chi^2_{ATM}$ is kept below 2. (e) Residual between modeled and observed $\Delta T$ from 1850-2019 for constant $\lambda^{-1}$. (f) Same as (e) but for increasing $\lambda^{-1}$ by 50%. (g) Same as (f) but for varying $\lambda^{-1}$ while the value of $\chi^2_{RECENT}$ is kept below 2. (h) same as (g) but for varying $\lambda^{-1}$ while the value of $\chi^2_{ATM}$ is kept below 2.

\* The manuscript misrepresents the findings by Rugenstein. They did not study CMIP6 models (but mostly CMIP5, and some CMIP3), and they found that all models had an increasing feedback parameter over time, not just some

> Thank you for pointing out this oversight. We have updated the text such that it is now clear the Rugenstein et al. (2020) study examined output from CMIP5 GCMs (rather than CMIP6), and we have changed "some" to "many".

\* Similarly, Marvel et al show that estimates from historical simulations strongly underestimate true ECS in virtually all CMIP5 models. This is misrepresented by saying 'some' models. The

mean bias is 0.8 degrees. This difference would bring the manuscript in line with conventional estimates of ECS of around 3 degrees.

> We have changed "some" to "many".

\* In the authors want to include a reference for CMIP6, https://journals.ametsoc.org/view/journals/clim/33/18/jcliD191011.xml may work, shows that 26 out of 29 models show an increasing 1/lambda, also not 'some'.

> Thank you for pointing out this study by Dong et al. (2020), which we now cite. Dong et al. (2020) show that the median increase in $\lambda$ in the CMIP6 GCMs is +0.4 W m$^{-2}$K$^{-1}$. The median estimate of $\lambda$ from the CMIP6 GCMs is $-1.2$ W m$^{-2}$K$^{-1}$. This +0.4 W m$^{-2}$K$^{-1}$ increase in $\lambda$ corresponds to a 50% increase in $\lambda^{-1}$, which is similar to the findings by Marvel et al., 2018.

> We have added a citation to Dong et al., 2020 on lines 1054,1118 and 1120, and 1139. We have also added the following sentence on line 1115: **An analysis by Dong et al. (2020) estimates a median increase in $\lambda$ of +0.4 W m$^{-2}$ K$^{-1}$, which corresponds to a 50% increase in $\lambda^{-1}$ (Fig. 1c, d of Dong et al. (2020)).**

\* The manuscript misrepresents Goodwin et al (2018). That papers indicates that there are time lags up to a hundred years, and they model a time-scale lag of 20 to 45 years for the Cloud − spatial SST adjustment feedback. The manuscript claims they have a maximum time delay of 20 years.

> Thanks again for pointing out this oversight, which has also been addressed.

> We now use a 32.5-year delay (midpoint between 20 and 45 years) to represent the time delay between the response of climate feedback to a change in RF for Figs. 14, S24, and S25. The text in Sect. 3.3.6 has been changed to read: **We use a lag of 32.5 years to represent the mean value of the slowest response of the climate system to a RF perturbation reported by Goodwin (2018), which is associated with clouds and spatial adjustments of SST (32.5 years is the average of 20 and 45 years, the minimum and maximum values of the slowest response given in his Table 1).**

> The 32.5-year delay is the longest value of a time delay noted in Table 1 of Goodwin (2018), due to the response of feedbacks involving clouds and the spatial adjustment of SST to a RF perturbation.

> Goodwin (2018) does also estimate a value of ECS for a 100-year response timescale for climate sensitivity, although the specific mechanism (presumably the cryosphere) is not named. Their estimate of ECS for the 100-year response is 2.9°C (range of 2.3 to 3.6°C). Our best estimate of ECS$_{\lambda(t)}$ found using a 32.5-year delay is 3.08°C (2.23-5.53°C), which is quite similar to the value of ECS reported by Goodwin. We have added the following text starting on line 1132 of the main manuscript to highlight this similarity:

**Our best estimate of $ECS_{\lambda(t)}$ of 3.08°C (range of 2.23 to 5.53°C) for a 32.5-year delay is similar to the value of ECS reported by Goodwin (2018) for a 100-year response time (2.9°C; range of 2.3 to 3.6°C).**

We have explored the possibility of using time delays longer than 32.5 years to analyze the GMST record using our EM-GC, but quite simply:

a) The modern temperature record does not extend far enough in time for this constraint to be applied in a meaningful manner.
b) Since our projections extend only to 2100, a time delay of longer than the time it will take for the calendar to reach 2100 does not seem appropriate.
c) Our results for $ECS_{\lambda(t)}$ found using the 32.5-year delay are, as noted above and now mentioned in the paper, quite similar to the value given by Goodwin (2018) upon his consideration of a 100-year response.

Major comment 2)
I had wanted the authors to compare model effective ECS with model Gregory ECS. This would show whether empirically estimated effective ECS can be compared with model Gregory ECS. The authors have instead done a sensitivity analysis of what happens if less data is used. I don't think that exercise is insightful, and certainly does not answer my question.

We have updated the manuscript to indicate we are comparing effective climate sensitivity from the EM-GC to effective climate sensitivity from the Gregory et al. (2004) method. We have also added more information to Sect. 3.3.6 on how our estimates of ECS will change based on a rise in $\lambda^{-1}$ over time. We will retain the analysis of the various training periods because this analysis was specifically requested by the other reviewer.

[revised manuscript text omitted]
)". Figure S1 illustrates the impact of updating Eq. (2) in our model to be comparable to the formulation in Bony et al. (2006) and Schwartz (2012). This figure displays the change in GMST anomaly in 2100 relative to pre-industrial ($\Delta T_{2100}$) as a function of $\lambda_\Sigma$ and AER $RF_{2011}$ for the two formulations of Eq. (2). Figure S1a uses the previous version of the EM-GC, where $Q_{OCEAN}$ was subtracted outside of the climate feedback multiplicative term, and Fig. 1b uses the new version of the EM-GC where $Q_{OCEAN}$ is subtracted within the climate feedback multiplicative term.

In the EM-GC framework, we calculate our value of $Q_{OCEAN}$ by finding the $\kappa$ needed to multiply the temperature difference between the atmosphere and the ocean to fit the observed OHC record. The model iterates over the ocean module, specifically the value of $\Delta T_{OCEAN,HUMAN}$ in Eq. (4), until the EM-GC converges on an estimate of $\kappa$ for a single OHC record and value of AER $RF_{2011}$. Figure S1 illustrates that the effect of changing Eq. (2) in the EM-GC impacts our estimates of the rise in $\Delta T_{2100}$ at high values of AER $RF_{2011}$. Strong aerosol cooling results in the ocean taking up more heat from the atmosphere than in the previous version of the EM-GC. The larger value of $Q_{OCEAN}$ results in a higher value of climate feedback needed to fit the historical climate record, because both AER $RF_{2011}$ and $Q_{OCEAN}$ are acting to cool the climate system. The higher values of climate feedback increase our maximum value of $\Delta T_{2100}$. This change brings some of the projections of $\Delta T_{2100}$ from the EM-GC closer to values of $\Delta T_{2100}$ from the CMIP6 multi-model ensemble.

Section 2.1 states "Altering the training period of our model has a slight effect on our results (see Fig. S2, S3, and the supplement for information on various training periods)." Figure S2 shows the end of century projected warming as a function of $\lambda_\Sigma$ and AER $RF_{2011}$, for four different training periods: 1850-1989 (Fig. S2a), 1850-1999 (Fig. S2b), 1850-2009 (Fig. S2c) and 1850-2019 (Fig. S2d), which is the normal training period used in our analysis. Values of $\Delta T_{2100}$ are shown only for combinations of $\lambda_\Sigma$ and AER $RF_{2011}$ that lead to good fits ($\chi^2 \leq 2$) to the climate record. We project relatively similar results for end of century warming for the training periods that end in 2019 and 2009. Our results using the training period from 1850-1999 are similar to observations and other reduced complexity models (Nicholls et al., 2020). The training period that ends in 1989 (Fig. S2a) yields a different "shape" of model parameter space for which good fits to the climate record can be obtained, compared to the other training periods. The different shape for this shorter training period is due to two factors. First, the formulation of the ocean component of our model for the training period that stops in 1989 uses 35 years of the observed OHC record. We are able to calculate good fits to the

OHC record over this shorter time period that diverge from the OHC record after 1989. Also, for this shorter time period, aerosol radiative forcing of climate cools in a manner that nearly mirrors the warming due to rising GHGs, resulting in a wider range of model parameters that lead to a "good fit" of the climate record, compared to model simulations constrained by data that extend closer to present-day. The highest values of $\Delta T_{2100}$ in Fig. S2a are associated with the largest values of $\lambda_\Sigma$, which in our model corresponds to excessively high values of $\kappa$ that we can rule out, based on OHC data collected during 1990 to 2019.

Figure S3 shows the observed (HadCRUT5) and modeled $\Delta T$ anomaly from 1945-2060 for the four different training periods described above. Each panel contains three projections of future $\Delta T$ for SSP4-3.4: projection using the value of climate feedback that provides the best fit to the historical climate record for a value of AER $RF_{2011} = -0.9$ W m$^{-2}$, the lowest value of climate feedback that provides a good fit to the observed $\Delta T$ record for a value of AER $RF_{2011} = -0.1$ W m$^{-2}$, and the highest value of climate feedback that provides a good fit to the historical climate record (the associated value of AER $RF_{2011}$ varies depending on the training period). As more years are added to the training period, the range of projection for future temperature decreases (Fig. S3a vs S3d). All of the best fit projections (solid line) and highest value of climate feedback (upper dashed line) closely follow the mid-point of the data, regardless of the training period. Given the nature of this test (i.e., predicting GMST out to 2019 for a series of trainings that stop in either 1989, 1999, or 2009), Figure S3 supports the quantitative accuracy of our approach for simulating and projecting future $\Delta T$.

Section 2.2.1 states "We use the uncertainty time series from HadCRUT4 for all GMST records (see the supplement, Figs. S4 and S5, and Table S1 for more information)". Figure S4 shows values of $\Delta T$ based on the seven individual GMST records (GISTEMP, BEG, HadCRUT4, CW14, HadCRUT5, NOAAGT, and JMA) with their corresponding 1σ and 2σ uncertainties. A horizontal line at zero denotes the time period of the baseline for each $\Delta T$ record. The multi-record mean, excluding the data set that is plotted, is also shown. Since the multi-record mean and individual $\Delta T$ record are plotted on the same baseline, these two quantities closely match over this time period. Panels (a), (b), (e), and (f) illustrate that the uncertainties for these GMST records are not large enough to encompass the multi-record mean over 1850-2019. The multi-record mean in panel (a) is below the GISTEMP 1σ uncertainty range between 1880 and 1900, and again between 1980 to 2019. In panel (b), the multi-record mean is above the BEG 1σ range from 1850 until 1865, 1880 to 1895, and below the 1σ uncertainty range from 2000 to 2019. The multi-record mean in panel (e) is below the HadCRUT5 1σ uncertainty range from 1990 until 2019. In panel (f), the multi-record mean is above the NOAAGT 1σ uncertainty range from 1920 until 1955. The JMA GMST record does not provide an uncertainty estimate. We therefore use the HadCRUT4 combined uncertainty (measurement, sampling, bias,

and coverage uncertainties (Morice et al., 2012)) estimate for JMA in panel (g). The multi-record mean of $\Delta T$ for all data sets other than JMA lies at the edge of the $1\sigma$ uncertainty range from 1891 until 2000. After 2000, the multi-record mean falls above both the $1\sigma$ and $2\sigma$ HadCRUT4 uncertainty range. The HadCRUT4 uncertainty time series shown in panel (c) is the only uncertainty estimate large enough to cover the spread in the various GMST records.

Figure S5 shows $\Delta T$ based on all seven GMST records and the multi-record mean relative to three baseline periods. The $1\sigma$ and $2\sigma$ uncertainties from HadCRUT4 are plotted about the multi-record mean. Panels (a) and (d) show the GMST records relative to 1891-1920, which are the first 30 years all of the data sets have in common. Between 1850-1970, all of the data sets fall within the $1\sigma$ HadCRUT4 uncertainty. After 1970, the GMST records start to deviate and some fall outside of the $1\sigma$ uncertainty but within the $2\sigma$ uncertainty, and JMA falls outside of the $2\sigma$ uncertainty. Panels (b) and (e) show the GMST records relative to the HadCRUT baseline period of 1961-1990. We see similar behavior as in panels (a) and (d), where the GMST records largely fall within the HadCRUT4 $1\sigma$ uncertainty until about 1970. Panels (c) and (f) show the GMST records forced to match HadCRUT5 from 2010-2019, which is baselined to 1961-1990. In these two panels, we see a large spread between the GMST records from the beginning of the time period until 2005.

Table S1 shows the percentage of $\Delta T$ data points that lie within the $1\sigma$ or $2\sigma$ HadCRUT4 uncertainty about the multi-record mean for all seven data records since 1940. Year 1940 is used to be consistent with the definition of our $\chi^2_{\text{RECENT}}$ parameter. Depending on the choice of baseline period, the number of data points within the uncertainty range varies. For a baseline of 1891-1920, 80% of the data points for all seven records are within the $1\sigma$ uncertainty and 95% of the data points are within the $2\sigma$. For a baseline of 1961-1990, 88% and 93% of data points are within the $1\sigma$ and $2\sigma$ HadCRUT4 uncertainties, respectively. If the $\Delta T$ records are forced to match the average value of the HadCRUT5 data set over the last decade, 72% of the data points are within the $1\sigma$ uncertainty and 88% are within the $2\sigma$ uncertainty. This analysis shows that depending on which baseline is used, the percentage of points within the $1\sigma$ or $2\sigma$ uncertainty ranges varies. Overall, these comparisons support the utility of the HadCRUT4 uncertainty for the GMST, since the $1\sigma$ and $2\sigma$ uncertainty ranges capture a percentage of points approximately correct for a pure Gaussian distribution. Therefore, we have adopted the HadCRUT4 uncertainties in GMST for all of the analyses in the main paper. The uncertainties published by other data centers tend to be smaller than the HadCRUT4 uncertainties. Since only the HadCRUT4 uncertainties span the range of values for $\Delta T$ from the seven data records in a somewhat realistic fashion, we have decided to use these uncertainties uniformly throughout the analysis.

Section 2.2.1 also says "We then adjust each data set to the HadCRUT5 pre-industrial baseline as described in the supplement". The mean of the HadCRUT5 GMST record from 1850-1900 is −0.3589°C. We add 0.3589°C to each value of the HadCRUT5 record to adjust the data set onto the pre-industrial baseline. We use this same offset for all of the other data sets. We add 0.3589°C to each value of ΔT from the six data sets to match the HadCRUT5 1850-1900 baseline.

Section 2.2.3 states "Figure S6 shows the ozone RF time series used in this analysis and the supplement provides more information about the creation of the time series for the RF due to $O_3^{TROP}$". Figure S6 displays the time series of tropospheric ozone RF used in our analysis for the various SSPs. Tropospheric ozone is an important GHG that rivals nitrous oxide as the third most important anthropogenic GHG. We include the RF due to tropospheric ozone ($O_3^{TROP}$) in our model for completion, even though the SSP database does not provide RF estimates for the various SSPs. We use values from the RCP scenarios provided by the Potsdam Institute for Climate Impact Research (Meinshausen et al., 2011). The values of the RF due to $O_3^{TROP}$ for SSP1-1.9 and SSP1-2.6 are from the RCP2.6 pathway. The RCP4.5 time series of $O_3^{TROP}$ is used for SSP2-4.5, the RCP6.0 time series is used for SSP4-6.0, and the RCP8.5 time series is used for SSP5-8.5. We create linear combinations of RCP2.6 and RCP8.5 to generate two new time series of the RF due to $O_3^{TROP}$ for SSP4-3.4 and SSP3-7.0. There is a large gap between the time series of the RF due to $O_3^{TROP}$ for RCP6.0 (shown as SSP4-6.0) and RCP8.5 (shown as SSP5-8.5) in Fig. S6. We created a time series that would split the difference between the two RCPs to represent the RF due to $O_3^{TROP}$ for SSP3-7.0. The SSP4-3.4 time series of the RF due to $O_3^{TROP}$ that was created lies in between the RCP2.6 (shown as SSP1-2.6) and RCP4.5 (shown as SSP2-4.5) time series in Fig. S6.

Section 2.2.8 states "Figure S9 shows the five OHC records as well as the multi-measurement average". Figure S9 displays the five OHC content data sets, as well as the multi-measurement average, plotted as a function of time and normalized to year 1986. This figure illustrates how the shapes of the different OHC records compare. Each of the time series represents the amount of heat stored in the top 700 m of the world's oceans for that specific data set. Carton et al. (2018) is the shortest data set, and only spans 36 years (1982-2017). The second shortest record is Balmaseda et al. (2013a), which spans 52 years (1958.5-2009.5). Ishii et al. (2017) is the record in the middle with a range of 63 years (1955-2017). Both Cheng et al. (2017) and Levitus et al. (2012) have records that span 65 years (1955-2019). The length of the data set and the shape of the curve affect the estimate of ocean heat export (OHE), because we calculate OHE by taking a linear fit to the full OHC time series. Balmaseda et al. (2013a) has the lowest estimate of OHE because the slope of the curve is relatively shallow, due to the fact that it slightly rises, then decreases at the

start of the record. Carton et al. (2018) has the highest estimate of OHE because the slope of the curve is the steepest of the five records.

135    Section 2.2.8 also says "For these five OHC data sets, uncertainty estimates are not always provided. Furthermore, some studies that do provide uncertainties give estimates that seem unreasonably small (see Fig S10 and the supplement)" and "Figure S10 and the supplement provide more detail on the creation of this time dependent uncertainty estimate for OHC". Figure S10 shows the multi-measurement average as well as the five OHC data records as a function of time, the uncertainty for each corresponding data set, and the

140    combined uncertainty used in this analysis. Panel (a) shows the multi-measurement OHC average with the standard deviation of the mean plotted around the average time series. The standard deviation is large at the beginning of the time series, due to the spread in the estimates of OHC between the different records (illustrated in Fig. S9). The standard deviation decreases as the various OHC records converge near a similar estimate. The standard deviation is zero in 1986 because we normalized all of the time series to zero in this

145    year to create the multi-measurement average. Because of this normalization, the standard deviation of the mean is not a realistic measure of uncertainty for the five OHC time series.

       Panels (b), (c), (d), (e), and (f) display the uncertainty estimates for the five OHC data records. We use the standard deviation of the mean of five ensemble members of the European Centre for Medium-Range Weather Forecasts Ocean ReAnalysis System 4 (ORSA) (Balmaseda et al., 2013b) for the Balmaseda et al.

150    (2013a) record. The standard deviation is plotted in panel (b) as the dotted blue line. The standard deviation is small at the beginning of the record, because the five ensemble members started at similar values of OHC in 1958 and diverged over time. The combined uncertainty of the standard deviation of the average of the five OHC records and the Cheng et al. (2017) estimate is plotted as a dashed blue line. Panel (c) shows the Levitus et al. (2012) time series for the top 700 m updated to the end of 2019. The Levitus time series utilizes

155    the standard error over the whole ocean for their uncertainty estimate and is plotted as the dotted light blue line. The standard error is a very small uncertainty estimate compared to the other OHC data records, which is unreasonable considering the large variations in OHC between the different records. We use the standard deviation of eight reanalysis experiments to represent the uncertainty associated with the Carton et al. (2018) OHC record and is plotted as a dotted orange line in panel (e). The standard deviation of the eight reanalysis

160    experiments is rather small, which also is unrealistic. Panel (f) displays the Cheng et al. (2017) OHC record updated through the end of 2019 with the 1σ uncertainty. This uncertainty does not vary much throughout the data record, making it more realistic as an estimate for such an uncertain quantity as OHC. We created the combined uncertainty estimate of the standard deviation of the average of the five OHC records and the Cheng et al. (2017) 1σ uncertainty to have the largest uncertainty possible due to the fact that OHC varies

165    between the different records. The EM-GC cannot achieve $\chi^2_{OCEAN} \leq 2$ for Balmaseda et al. (2013a), Levitus

et al (2012), and Carton et al. (2018) using their own respective estimates of uncertainty. Creating one uncertainty estimate to be used for all of the OHC records provides consistency and allows the EM-GC to achieve good fits between the observed and modeled OHC.

170    Section 2.3 states "Figure S12 illustrates the REG method used to determine AAWR from the CMIP6 GCMs". Figure S12 shows the change in $\Delta T$ from 1975-2014 from the CMIP6 GCMs and the contribution of SAOD from 1975-2014. There was about a 6 month lag between the response of $\Delta T$ and enhancements of SAOD following the eruption of Mount Pinatubo in June 1991 (Douglass and Knox, 2005; Thompson et al., 2009); a 6 month delay for the response of $\Delta T$ to SAOD is commonly used in regression analyses of the
175    actual temperature record (Foster and Rahmstorf, 2011; Lean and Rind, 2008). The time needed for $\Delta T$ to respond to a change in the aerosol loading in the stratosphere due to a volcanic eruption in each GCM can exhibit a significant difference compared to this empirically determined response time. Therefore, a lag was determined for each GCM by calculating the value of the monthly delay that resulted in the largest regression coefficient for SAOD (versus $\Delta T$). Due to the difference in model physics between the various GCMs, the
180    value of the delay between the volcanic forcing and surface temperature response ranged from 0 to 11 months. The effect of SAOD on $\Delta T$ for the 50 GCMs is shown in Fig. S12d. Figure S12b shows the residual in $\Delta T$ after removing the influence of SOAD, and the median value of AAWR from the CMIP6 multi-model ensemble is plotted as a linear line. Figure S12c shows the human component of global warming, $\Delta T_{ATM,HUMAN}$, from the EM-GC. A linear fit and quadratic fit are shown to illustrate that $\Delta T_{ATM,HUMAN}$ is
185    almost nearly linear from 1975-2014, supporting the approximation of $\Delta T_{ATM,HUMAN}$ as a linear function from 1975-2014 for the REG calculation.

    Section 2.3 also states "Figure S13 and the supplement compare values of AAWR found using the REG method applied to EM-GC output with values of AAWR found using Eq. (9), as support for the validity
190    of using the REG method to determine AAWR from CMIP6 output". We applied the REG method to the EM-GC simulations to check the validity of the REG method. We regressed the modeled $\Delta T$ time series from the EM-GC for an AER $RF_{2011} = -0.9$ W m$^{-2}$ simulation with SAOD and applied a 6 month lag. A linear function is used to represent the anthropogenic effect on temperature from 1975-2014. Fig. S13 shows the results of using the REG method on output of the EM-GC.
195    The value of AAWR from the EM-GC determined using the REG method is 0.188°C decade$^{-1}$/decade, compared to 0.167°C decade$^{-1}$/decade using Eq. (9) (Fig. S13c and Fig. 1). There is a 0.021°C decade$^{-1}$/decade difference between the two methods. This difference arises because the REG method, when applied to the EM-GC modeled $\Delta T$ time series, includes the contribution of AMOC in the value of AAWR

(Fig. S13c). Figure 1 of our paper illustrates that AMOC contributes about 0.025°C decade$^{-1}$/ to the rise in $\Delta$T. If we include AMOC as a regressor variable to the REG method, we obtain a value of AAWR of 0.161°C decade$^{-1}$/ from the output of the EM-GC (Fig. S13g).

The close agreement of values of AAWR from the REG method once we account for AMOC and Eq. (9) supports the validity of the REG method to determine AAWR from CMIP6 output. We do not explicitly use AMOC as a regressor variable when applying the REG method to CMIP6 GCMs for two reasons. The first reason is that GCMs have been shown to underestimate key aspects of the Atlantic multidecadal oscillation and are unable to simulate the many oceanic and atmospheric footprints of AMOC (Kavvada et al., 2013). The second reason is that CMIP6 GCM historical runs do not use prescribed SSTs. If the CMIP6 GCMs are representing AMOC, it is a random signal that is averaged out when we analyze the 50 GCMs in order to calculate AAWR.

Section 2.3 also says "Analysis of AAWR for these 50 GCMs of LIN versus REG (see Fig. S14)…". Figure S14 shows the similarity between the values of AAWR determined using the LIN and REG methods. The ratio between the values of AAWR determined utilizing LIN and REG is 1.009, indicating there is only about a 0.9% difference in the values of AAWR using the two methods. Figure S14 also shows the values of AAWR that are below the maximum value of AAWR determined by the EM-GC utilizing the HadCRUT5 temperature record (blue) and the values that are above the maximum (red). Less than half of the GCMs result in values of AAWR less than the maximum value from the EM-GC and more than half of the GCMs result in values of AAWR greater than the maximum value from the EM-GC utilizing the HadCRUT5 GMST record.

Section 2.4 states "For the estimate of  climate sensitivity from the CMIP6 multi-model ensemble, we use the method described by Gregory et al. (2004) (See the supplement and Fig. S15 for more information)". To use the Gregory method, near surface air temperature output from the Abrupt 4×$CO_2$ and piControl simulations, as well as net downward radiative flux output from the Abrupt 4×$CO_2$ simulation is used to calculate ECS. The near surface air temperature and net downward radiative flux was converted from monthly gridded output to annual global averages. We calculate the temperature change for the Abrupt 4×$CO_2$ simulation by subtracting the piControl near surface air temperature (Chen et al., 2019) (Fig. S15). This computed temperature anomaly is then regressed against the net downward radiative flux, with the x-intercept yielding the  response of $\Delta$T to a quadrupling of $CO_2$. This  response is then divided by two (Jones et al., 2019) to arrive at the  effective climate sensitivity (Fig. S15).

Section 2.5 states "See Fig. S16 for unweighted ECS values and Section 3.2 states "See Fig S16 for results without aerosol weighting". Figure S16 displays the values of ECS using the EM-GC and the CMIP6 multi-model ensemble. The EM-GC box contains the $25^{th}$, $50^{th}$, and $75^{th}$ percentiles, the whiskers denote the $5^{th}$ and $95^{th}$ percentiles, and the stars represent the minimum and maximum values of ECS. The box labeled CMIP6 is unchanged from Fig. 7. The values of ECS are not treated with the aerosol weighting described in Sect. 2.5. This figure shows that most of the estimates of ECS found using the EM-GC are concentrated towards small values of ECS, due to the fact that the majority of the EM-GC model runs with good fits to the climate record ($\chi^2_{ATM}$, $\chi^2_{RECENT}$, and $\chi^2_{OCEAN}$) have weak aerosol cooling and low values of $\lambda_\Sigma$ (Fig. 5b). We use the aerosol weighting method to assign the same weights for the IPCC 2013 "likely" range limits of AER $RF_{2011}$ of $-0.4$ and $-1.5$ W m$^{-2}$ at the one sigma values of a Gaussian, and the $-0.1$ and $-1.9$ W m$^{-2}$ are at the two sigma values of a Gaussian. Using the aerosol weighting method adjusts our estimates of ECS so that the calculated percentiles occur at higher values.

Section 3.2 in the Fig. 8 caption says, "See supplement for the confidence intervals plotted for each study". All of the studies except Dessler et al. (2018), Rugenstein et al. (2020), IPCC 2013, and Zelinka et al. (2020) have the 5 to 95% confidence intervals shown. The 66% confidence intervals are shown for IPCC 2013, and the minimum and maximum are shown for Dessler et al. (2020), Rugenstein et al. (2020) and Zelinka et al. (2020).

The Fig. 8 caption in Sect. 3.2 also refers to the supplement for information about which studies are estimating effective climate sensitivity or equilibrium climate sensitivity. We designate each study based on information found in their manuscripts if their analysis uses the Gregory et al. (2004) method or infers climate feedback from the historical climate record will persist until equilibrium. The use of either of these two factors results in our designation of effective climate sensitivity (Gregory et al., 2004; Sherwood et al., 2020; Tokarska et al., 2020a; Zelinka et al., 2020). Based on our examination of IPCC 2013, it seems their estimate is a combination of effective climate sensitivity and equilibrium climate sensitivity.

Section 3.3.4 states "see Fig. S21 and the supplement" and "see the supplement for more information". Figure S21 shows the rise in $\Delta T$ from pre-industrial for SSP5-8.5 versus the cumulative emissions of $CO_2$, in Gt C, since 1870. The colored lines denote the probability of reaching at least that temperature by the end of century. The large spread in projections of future $\Delta T$ is driven by the uncertainty in AER RF. The computed probabilities are based on the aerosol weighting method, so the uncertainty in

AER RF is considered when determining the likelihood of achieving the Paris Agreement target of 1.5°C and
upper limit of 2.0°C for the cumulative carbon emissions.

We use the uncertainty suggested by coupled atmospheric / carbon cycle models in how atmospheric $CO_2$ will respond to the prescribed carbon emissions. Examination of Fig. 2 and Table 3 from Friedlingstein et al. (2014) and Fig. 9b from Murphy et al. (2014) led to our determination that the uncertainty in estimates of atmospheric $CO_2$ from emissions driven runs of CMIP5 coupled atmospheric / carbon cycle models is about 10% (1σ). We include this 10% uncertainty in our determination of the carbon budgets for each probability of achieving the Paris Agreement target and upper limit shown in Table 2.

Section 3.3.6 states "Figure S23 is identical to Fig. 14, except for the use of no delay between the RF perturbations and the response of climate feedback.". Figure S23 shows the effect of time variant $\lambda^{-1}$, assuming  an instantaneous response between $\lambda^{-1}$ and a change in radiative forcing for a simulation using a value of AER $RF_{2011}$ = −0.9 W m$^{-2}$. The instantaneous response causes the modeled ΔT to deviate more from the observed temperature than  results found using the 32.5-year delay in the response (Fig. S23g, h versus Fig. 14g, h). The deviation between the modeled and observed ΔT does not allow for a large change in $\lambda^{-1}$ over time to still achieve the $\chi^2_{ATM} \leq 2$ and $\chi^2_{RECENT} \leq 2$ constraints. The deviation between modeled and observed ΔT in Fig. S23d resembles the behavior of some CMIP6 GCMs (see Fig. 9 and Tokarska et al. (2020b)).

Section 3.3.6 also states "In Figs. 14 and S23 we also analyze a RF scenario termed SSP2-4.5′ that serves as a doubled $CO_2$ scenario (dotted lines)". In the SSP2-4.5′ simulation, only $CO_2$ is allowed to change after the end of 2019. All other GHGs and aerosols are kept constant at their December 2019 values. This simulation allows us to examine the effect of time variant $\lambda^{-1}$ on changes in ΔT due only to the future rise in $CO_2$. In the SSP2-4.5 scenario, $CH_4$, tropospheric $O_3$, and ODSs decrease after 2019 leading to a future decline in RF, whereas $N_2O$ and tropospheric aerosols result in a future increase in RF. When all of these RF are kept constant in the SSP2-4.5′ scenario for the AER $RF_{2011}$ = −0.9 W m$^{-2}$ scaling assumption, the terms result in a near balance out to 2100. For the weaker aerosol cooling scenario (AER $RF_{2011}$ = −0.4 W m$^{-2}$), the value of RF due to tropospheric aerosols is not large enough to completely offset the other GHGs that are held constant. Consequently, the SSP2-4.5′ simulation (dotted line) results in slightly larger total RF and associated warming than the SSP2-4.5 scenario (solid line) shown in Fig. S24. For the stronger aerosol cooling scenario (AER $RF_{2011}$ = −1.5 W m$^{-2}$), the value of RF due to tropospheric aerosols is larger than the RF due to the other GHGs that are held constant, resulting in the SSP2-4.5′ having a slightly lower total RF and associated warming than the SSP2-4.5 scenario (Fig. S25).

The effect of the uncertainty in AER $RF_{2011}$ on ECS found using time dependent climate feedback ($ECS_{\lambda(t)}$ in Main) is based on results shown in Figs. 14b, 24b, and 25c. If we apply the $\chi^2_{RECENT}$ constraint equally to the −0.4, −0.9, and −1.5 W m$^{-2}$ simulations, our new estimate of $ECS_{\lambda(t)}$ is 3.52°C (range of 2.71 to 5.53°C)

300

[Figure]

**Figure S1.** GMST anomaly in 2100 relative to pre-industrial ($\Delta T_{2100}$) as a function of climate feedback parameter and AER $RF_{2011}$ for two versions of the EM-GC trained with the HadCRUT4 $\Delta T$ record. (a) The change in $\Delta T_{2100}$ for SSP4-3.4 using the original formulation of Eq. (2) where $Q_{OCEAN}$ is subtracted outside of the feedback multiplicative term. (b) The change in $\Delta T_{2100}$ for SSP4-3.4 using the updated formulation of Eq. (2) where $Q_{OCEAN}$ is subtracted within the feedback multiplicative term similar to Bony et al. (2006) and Schwartz (2012). The EM-GC is able to fit higher values of $\lambda_\Sigma$ at strong aerosol cooling (around $-1.5$ W m$^{-2}$) for the new Eq. (2) compared to the original formulation in Canty et al. (2013) and Hope et al. (2017). The maximum value of future warming has increased due to the higher $\lambda_\Sigma$ values.

[Figure]

**Figure S2.** $\Delta T_{2100}$ as a function of climate feedback parameter and tropospheric aerosol radiative forcing in 2011 using the EM-GC trained with the HadCRUT5 $\Delta T$ record for SSP4-3.4. (a) Training period of 1850-1989. The region outside of the AER $RF_{2011}$ range provided by IPCC 2013 is shaded (grey). Colors denote the GMST change in year 2100 relative to pre-industrial. The color bar is the same across all four panels for comparison. (b) Training period of 1850-1999. (c) Training period of 1850-2009. (d) Training period of 1850-2019, which is the normal training period used in our analysis.

[Figure]

**Figure S3.** Observed and modeled GMST anomaly relative to 1850-1900 from 1945-2060 for four training periods.
(a) Observations from HadCRUT5 (black), the EM-GC $\Delta T$ simulation for a training period of 1850-1989 (orange) of
HadCRUT5, and the EM-GC projections for SSP4-3.4 out to 2060. Three EM-GC projections are shown in red: The
best estimate of climate feedback for AER $RF_{2011}$ = −0.9 W m$^{-2}$, the lowest value of climate feedback that satisfies the
$\chi^2$ constraints for AER $RF_{2011}$ = −0.1 W m$^{-2}$, and the highest value of climate feedback that satisfies the $\chi^2$ constraints
(the value of AER $RF_{2011}$ varies for each training period). The IPCC 2013 likely range of warming is denoted as the
black trapezoid. (b) Training period of 1850-1999. (c) Training period of 1850-2009. (d) Training period of 1850-2019.

[Figure]

**Figure S4.** Annual GMST (ΔT) anomaly for seven data records relative to their individual baseline and the multi-record mean. The multi-record mean does not include the data set that is being shown. The 1σ and 2σ uncertainties for each GMST record are shown, and the horizontal line for ΔT=0 spans the baseline used for the specific panel. (a) GISTEMP. (b) BEG. (c) HadCRUT4. (d) CW14. (e) HadCRUT5. (f) NOAAGT. (g) JMA. Since the JMA data provider does not provide an uncertainty time series, the HadCRUT4 uncertainty is used.

[Figure]

**Figure S5.** Annual GMST (ΔT) anomaly relative to several baseline periods for seven data records. The 1σ (shaded grey) and 2σ (dotted grey) HadCRUT4 uncertainties are plotted about the multi-model record mean (black). (a) Baseline of 1891-1920 plotted from 1850-2019. (b) Same as (a) using a baseline of 1961-1990. (c) Same as (a) except all of the ΔT records are forced to match the average ΔT anomaly over 2010-2019 given by HadCRUT5 that is relative to 1961-1990. (d) – (f) Same as (a) – (c) except plotted from 1940-2019.

335

**Table S1.** Percentage of annual values between 1940-2019 of the GMST record within the 1 sigma or 2 sigma HadCRUT4 uncertainties about the multi-record mean for each baseline period.

| Baseline: 1891-1920 | 1σ | | | 2σ | | |
|---|---|---|---|---|---|---|
| | $N_{WITHIN}$ | $N_{TOTAL}$ | % | $N_{WITHIN}$ | $N_{TOTAL}$ | % |
| HadCRUT4 | 77 | 80 | 96 | 80 | 80 | 100 |
| HadCRUT5 | 42 | 80 | 53 | 80 | 80 | 100 |
| CW14 | 80 | 80 | 100 | 80 | 80 | 100 |
| BEG | 71 | 80 | 89 | 80 | 80 | 100 |
| GISTEMP | 73 | 80 | 91 | 80 | 80 | 100 |
| NOAAGT | 76 | 80 | 95 | 80 | 80 | 100 |
| JMA | 29 | 80 | 36 | 54 | 80 | 68 |
| **AVERAGE** | | | **80%** | | | **95%** |
| | | | | | | |
| **Baseline: 1961-1990** | | | | | | |
| HadCRUT4 | 80 | 80 | 100 | 80 | 80 | 100 |
| HadCRUT5 | 68 | 80 | 85 | 80 | 80 | 100 |
| CW14 | 80 | 80 | 100 | 80 | 80 | 100 |
| BEG | 80 | 80 | 100 | 80 | 80 | 100 |
| GISTEMP | 75 | 80 | 94 | 80 | 80 | 100 |
| NOAAGT | 80 | 80 | 100 | 80 | 80 | 100 |
| JMA | 27 | 80 | 34 | 48 | 80 | 60 |
| **AVERAGE** | | | **88%** | | | **93%** |
| | | | | | | |
| **Match 2010-2019** | | | | | | |
| HadCRUT4 | 68 | 80 | 86 | 80 | 80 | 100 |
| HadCRUT5 | 47 | 80 | 59 | 76 | 80 | 95 |
| CW14 | 78 | 80 | 98 | 80 | 80 | 100 |
| BEG | 77 | 80 | 96 | 80 | 80 | 100 |
| GISTEMP | 47 | 80 | 59 | 79 | 80 | 99 |
| NOAAGT | 73 | 80 | 61 | 80 | 80 | 100 |
| JMA | 11 | 80 | 14 | 18 | 80 | 23 |
| **AVERAGE** | | | **72%** | | | **88%** |

340

[Figure]

**Figure S6.** Radiative forcing of tropospheric ozone for the various SSPs analyzed in our study. The time series labeled SSP1-2.6, SSP2-4.5, SSP4-6.0, and SSP5-8.5 are from the corresponding RCP scenarios. We created the time series from SSP4-3.4 and SSP3-7.0 using linear combinations of the SSP1-2.6 and SSP5-8.5 time series.

[Figure]

**Figure S7.** Radiative forcing time series due to tropospheric aerosols. (a) The RF time series due to tropospheric aerosols for SSP1-2.6. The solid grey circle denotes the value of AER $RF_{2011}$ given by the SSP database. The solid grey lined labeled the $-1.0$ W m$^{-2}$ time series is the AER RF time series given by the SSP database for SSP1-2.6. We appended a historical AER RF time series from the RCP scenarios and created five additional AER RF time series as described in Sect. 2.2.4. (b) Anthropogenic aerosol radiative forcing time series for SSP4-3.4.

350

[Figure]

**Figure S8.** Measured (HadCRUT5) and modeled GMST anomaly (ΔT) relative to a pre-industrial (1850-1900) baseline for an AER $RF_{2011}$ = −0.1 W m$^{-2}$ and −1.5 W m$^{-2}$. (a) Observed (black) and modeled (red) ΔT from 1850-2019. This panel also displays the values of $\lambda_\Sigma$ and $\chi^2_{ATM}$ (see text) for this best-fit simulation. (b) Contributions from total human activity. This panel also denotes the numerical value of the attributable anthropogenic warming rate from 1975-2014 (black dashed) as well as the 2σ uncertainty in the slope. (c) Solar irradiance (light blue) and major volcanoes (purple). (d) Influences from ENSO on ΔT. (e) Contributions from AMOC to ΔT and to observed warming from 1975-2014. (f) Influences from PDO (blue) and IOD (pink) on ΔT. (g) Measured (black) and modeled (red) ocean heat content (OHC) as a function of time for the average of five data sets (see text), the value of $\chi^2_{OCEAN}$ for this run, as well as the ocean heat uptake efficiency, κ, needed to provide the best-fit to the OHC record. The error bars (blue) denote the uncertainty in OHC used in this analysis (see Sect. 2.2.8). (h)-(n) Same as (a)-(g), except for AER $RF_{2011}$ = −1.5 W m$^{-2}$.

[Figure]

**Figure S9.** Ocean heat content time series. The five ocean heat content data records used in this analysis, normalized to the year 1986 because this year is in the middle of the average time series. The grey shaded region is the combined uncertainty estimate used in this analysis, centered around the average of the five data sets. The average of the ocean heat content records (1955 – 2017) is computed when there are three or more data sets available for a given year.

[Figure]

**Figure S10.** The ocean heat content records and uncertainty estimates analyzed in this study. (a) The average OHC record along with the standard deviation of the mean represented by the dotted black line, and the combined uncertainty of the 1σ standard deviation of the average of the five OHC records and the Cheng et al. (2017) estimates shown as the dashed black line. (b) Balmaseda OHC record with the standard deviation of the five ORSA ensemble members as the dotted line, and the combined uncertainty as the dashed line. (c) Levitus OHC record with the standard error as the native uncertainty, and the combined uncertainty. (d) Carton OHC record with the standard deviation of the mean of multiple ensemble members, and the combined uncertainty. (e) Cheng OHC record with the 1σ native uncertainty and the combined uncertainty. (f) Ishii OHC record with the combined uncertainty as the dashed line.

[Figure]

**Figure S11.** Measured (HadCRUT5) and modeled GMST anomaly (ΔT) relative to a pre-industrial (1850-1900) baseline without AMOC, PDO, and IOD. (a) Observed (black) and modeled (red) ΔT from 1850-2019. This panel also displays the values of $\lambda_\Sigma$ and $\chi^2_{ATM}$ (see text) for this best-fit simulation. (b) Contributions from total human activity. This panel also denotes the numerical value of the attributable anthropogenic warming rate from 1975-2014 (black dashed) as well as the 2σ uncertainty in the slope. The estimates of AAWR show similar results if AMOC is or is not included (see Fig. 1b). (c) Solar irradiance (light blue) and major volcanoes (purple). (d) Influences from ENSO on ΔT. (e-f) Contributions from AMOC, PDO, and IOD to ΔT are set to zero (g) Measured (black) and modeled (red) ocean heat content (OHC) as a function of time for the average of five data sets (see text), the value of $\chi^2_{OCEAN}$ for this run, as well as the ocean heat uptake efficiency, $\kappa$, needed to provide the best-fit to the OHC record. The error bars (blue) denote the uncertainty in OHC used in this analysis (see Sect. 2.2.8).

[Figure]

**Figure S12.** The change in GMST (ΔT) relative to 1961-1990 from the CMIP6 GCMs and the contribution from SAOD from 1975-2014. (a) ΔT from the 50 CMIP6 GCMs. (b) The residual in the change of GMST from the 50 CMIP6 GCMs after subtracting the contribution of SAOD determined by the updated REG method. The median value of AAWR is written on this panel and plotted in red. (c) The human component of global warming, $\Delta T_{ATM,HUMAN}$, from the EM-GC. A linear fit (black) and quadratic fit (red) are plotted on top to show that $\Delta T_{ATM,HUMAN}$ is almost exactly linear. (d) The contribution of SAOD in the 50 CMIP6 GCMs using a lag month calculated for each model.

[Figure]

**Figure S13**. The change in GMST (ΔT) relative to 1961-1990 from observations and modeled output. (a) ΔT from HadCRUT5 and EM-GC simulation. (b) The residual in ΔT from the EM-GC simulation after subtracting the contribution of SAOD determined by the REG method (grey) and ΔT due to humans from the REG method (orange). (c) ΔT due to humans from the REG method (orange) and from the EM-GC (blue). The values of AAWR determined using the REG method and Eq. (9) are shown. (d) The contribution of SAOD to ΔT. (e) Same as (a). (f) Same as (b) but also subtracting the contribution of AMOC determined by the REG method. (g) Same as (c) but using AMOC as a regressor. (h) Same as (d) also showing the contribution of AMOC to ΔT found using the REG method.

[Figure]

**Figure S14.** Values of AAWR for 50 CMIP6 GCMs using the LIN and REG methods. The solid black line is the 1:1 line and the vertical and horizontal dashed lines are the maximum value of AAWR determined using the EM-GC and the HadCRUT temperature record. The CMIP6 GCMs that have values of AAWR less than the maximum value from the EM-GC are blue, and the CMIP6 GCMs that have values of AAWR greater than the maximum value from the EM-GC are red. The slope, $1\sigma$ standard deviation, and $R^2$ of the values of AAWR from the CMIP6 GCMs are shown.

**Table S2.** Values of AAWR calculated using the EM-GC as a function of start and end year. The value of AAWR
from 1975-2014 used in the main manuscript is shown in red. Each model run uses the best estimate of AER RF$_{2011}$
($-0.9$ W m$^{-2}$), the average of five OHC records, and the HadCRUT5 GMST record. The impact on varying the start
and end year on AAWR is slight, except when a short record is used (i.e., 1984-2004, a 21-year span). A two-decade
time span is not long enough to calculate an accurate estimate of AAWR. The value of AAWR is more sensitive to the
choice of OHC or temperature record used than the chosen time span.

<table>
<tr><td></td><td></td><td colspan="6" align="center">Start Year</td></tr>
<tr><td></td><td>AAWR (°C decade$^{-1}$/decade)</td><td>1970</td><td>1973</td><td>1975</td><td>1979</td><td>1982</td><td>1984</td></tr>
<tr><td rowspan="8">End Year</td><td>2004</td><td>0.181 ± 0.007</td><td>0.180 ± 0.009</td><td>0.180 ± 0.010</td><td>0.169 ± 0.011</td><td>0.159 ± 0.013</td><td>0.149 ± 0.012</td></tr>
<tr><td>2006</td><td>0.177 ± 0.008</td><td>0.175 ± 0.009</td><td>0.174 ± 0.010</td><td>0.163 ± 0.011</td><td>0.153 ± 0.012</td><td>0.143 ± 0.011</td></tr>
<tr><td>2008</td><td>0.173 ± 0.007</td><td>0.171 ± 0.008</td><td>0.169 ± 0.009</td><td>0.159 ± 0.010</td><td>0.150 ± 0.010</td><td>0.141 ± 0.009</td></tr>
<tr><td>2010</td><td>0.172 ± 0.007</td><td>0.169 ± 0.008</td><td>0.167 ± 0.008</td><td>0.158 ± 0.008</td><td>0.150 ± 0.009</td><td>0.143 ± 0.008</td></tr>
<tr><td>2012</td><td>0.171 ± 0.006</td><td>0.168 ± 0.007</td><td>0.167 ± 0.008</td><td>0.158 ± 0.008</td><td>0.152 ± 0.008</td><td>0.145 ± 0.007</td></tr>
<tr><td>2014</td><td>0.171 ± 0.005</td><td>0.168 ± 0.006</td><td>0.167 ± 0.007</td><td>0.160 ± 0.007</td><td>0.154 ± 0.007</td><td>0.149 ± 0.007</td></tr>
<tr><td>2016</td><td>0.171 ± 0.005</td><td>0.169 ± 0.006</td><td>0.168 ± 0.006</td><td>0.161 ± 0.006</td><td>0.157 ± 0.007</td><td>0.153 ± 0.007</td></tr>
<tr><td>2018</td><td>0.171 ± 0.005</td><td>0.170 ± 0.005</td><td>0.169 ± 0.006</td><td>0.163 ± 0.006</td><td>0.159 ± 0.006</td><td>0.156 ± 0.006</td></tr>
</table>

**Table S3.** Average values of AAWR calculated from the CMIP6 multi-model results using the regression method as a function of start and end year. The uncertainty corresponds to the 1σ standard deviation of AAWR found from the 50 GCMs. The value of AAWR from 1975-2014 used in the main manuscript is shown in red. The values of AAWR from the CMIP6 multi-model ensemble is more sensitive to the choice of start and end year than the EM-GC due to the small number of models. We use the same start and end year, 1975-2014, for the determination of AAWR for both the EM-GC and the CMIP6 multi-model ensemble for consistency.

| | | Start Year | | | | | |
|---|---|---|---|---|---|---|---|
| | **AAWR (°C decade$^{-1}$/decade)** | **1970** | **1973** | **1975** | **1979** | **1982** | **1984** |
| **End Year** | **2004** | 0.185 | 0.196 | 0.200 | 0.208 | 0.224 | 0.230 |
| | **2006** | 0.192 | 0.203 | 0.207 | 0.216 | 0.232 | 0.238 |
| | **2008** | 0.196 | 0.207 | 0.211 | 0.220 | 0.234 | 0.241 |
| | **2010** | 0.200 | 0.209 | 0.214 | 0.222 | 0.236 | 0.241 |
| | **2012** | 0.204 | 0.213 | 0.218 | 0.226 | 0.239 | 0.244 |
| | **2014** | 0.208 | 0.217 | 0.222 | 0.230 | 0.242 | 0.247 |

[Figure]

**Figure S15.** Steps for the calculation of ECS using the Gregory et al. (2004) method, using GISS-E2-1-H (Kelley et al., 2020) as an example. (a) The change in Abrupt 4×CO₂ GMST (variable: tas) from the piControl experiment for 150 years. (b) Abrupt 4×CO₂ net downward radiative flux (variable: rtmt) versus the Abrupt 4×CO₂ GMST change from the piControl experiment for 150 years. The x-intercept of the orthogonal linear least squares fit of the GCM output shown in panel (b), divided by two yields the  effective climate sensitivity, which in this case is 3.09°C.

**Table S4.** Values of AAWR from 1975-2014 for the 50 CMIP6 multi-model Historical simulations available at time of the analysis (April 2020) for both the REG and LIN methods. The asterisk symbol (*) indicates there is only one run used to compute the value of AAWR for that GCM. No asterisk indicates the AAWR value shown in the table is the average of the values of AAWR for all runs of that model. The average ratio of LIN to REG for all 50 models is $1.009 \pm 0.015$, shown at the bottom of the table and in Fig. S14. The correlation coefficient ($r^2$) of 0.995 is also shown. We conclude our determination of AAWR from the CMIP6 multi-model ensemble is accurate to $\pm 1\%$, which is much smaller than the difference between the CMIP6 multi-model ensemble values of AAWR and those found using the EM-GC framework.

| Model | AAWR, REG (°C decade$^{-1}$/) | AAWR, LIN (°C decade$^{-1}$/) | Model | AAWR, REG (°C decade$^{-1}$/) | AAWR, LIN (°C decade$^{-1}$/) |
|---|---|---|---|---|---|
| ACCESS-CM2 | 0.211 | 0.216 | GFDL-CM4* | 0.243 | 0.250 |
| ACCESS-ESM1-5 | 0.238 | 0.246 | GFDL-ESM4 | 0.203 | 0.224 |
| AWI-CM-1-1-MR | 0.215 | 0.220 | GISS-E2-1-G | 0.194 | 0.198 |
| BCC-CSM2-MR | 0.217 | 0.228 | GISS-E2-1-G-CC | 0.204 | 0.213 |
| BCC-ESM1 | 0.241 | 0.249 | GISS-E2-1-H | 0.237 | 0.244 |
| CAMS-CSM1-0 | 0.131 | 0.138 | HadGEM3-GC31-LL | 0.283 | 0.292 |
| CanESM5 | 0.354 | 0.361 | HadGEM3-GC31-MM | 0.227 | 0.234 |
| CanESM5-CanOE | 0.323 | 0.334 | INM-CM4-8* | 0.173 | 0.181 |
| CAS-ESM2-0 | 0.196 | 0.204 | INM-CM5-0 | 0.146 | 0.156 |
| CESM2 | 0.240 | 0.243 | IPSL-CM6A-LR | 0.230 | 0.236 |
| CESM2-FV2 | 0.221 | 0.224 | KACE-1-0-G | 0.254 | 0.260 |
| CESM2-WACCM | 0.273 | 0.291 | MCM-UA-1-0 | 0.225 | 0.231 |
| CESM2-WACCM-FV2 | 0.231 | 0.235 | MIROC6 | 0.157 | 0.168 |
| CIESM | 0.245 | 0.251 | MIROC-ES2L | 0.163 | 0.167 |
| CNRM-CM6-1 | 0.202 | 0.196 | MPI-ESM1-2-HAM | 0.180 | 0.186 |
| CNRM-CM6-1-HR* | 0.172 | 0.178 | MPI-ESM1-2-HR | 0.195 | 0.203 |
| CNRM-ESM2-1 | 0.170 | 0.172 | MPI-ESM1-2-LR | 0.192 | 0.197 |
| E3SM-1-0 | 0.267 | 0.278 | MRI-ESM2-0 | 0.203 | 0.210 |
| E3SM-1-1* | 0.283 | 0.285 | NESM3 | 0.242 | 0.253 |
| E3SM-1-1-ECA* | 0.275 | 0.274 | NorCPM1 | 0.180 | 0.185 |
| EC-Earth3* | 0.299 | 0.310 | NorESM2-LM | 0.167 | 0.182 |
| EC-Earth3-Veg* | 0.214 | 0.223 | NorESM2-MM* | 0.151 | 0.154 |
| FGOALS-f3-L | 0.218 | 0.226 | SAM0-UNICON* | 0.245 | 0.250 |
| FGOALS-g3 | 0.176 | 0.191 | TaiESM1* | 0.273 | 0.283 |
| FIO-ESM-2-0 | 0.229 | 0.237 | UKESM1-0-LL | 0.299 | 0.312 |
| Ratio = $1.009 \pm 0.015$ | | | $R^2 = 0.995$ | | |

[Figure]

**Figure S16.** Values of ECS found using the EM-GC and the CMIP6 multi-model ensemble without the aerosol weighting method. Values of ECS utilizing the EM-GC are calculated using seven temperature data sets and five ocean heat content records (as indicated). The box represents the 25th, 50th, and 75th percentiles of the values of ECS and the whiskers denote the 5th and 95th percentiles for the different OHC records and each temperature record without using the aerosol weighting method (unweighted). The stars indicate the minimum and maximum values of ECS. The circles are the values of ECS associated with the best estimate of AER $RF_{2011}$ of $-0.9$ W m$^{-2}$. The box labeled CMIP6 is the 25th, 50th, and 75th percentiles of the values of ECS from the CMIP6 multi-model ensemble, the whiskers indicate the 5th and 95th percentiles, and the stars represent the minimum and maximum values of ECS from the CMIP6 multi-model ensemble.

**Table S5.**  Effective climate sensitivity (ECS) from 28 CMIP6 GCMs. We can only calculate ECS for
GCMs that provide Abrupt 4×$CO_2$ near surface air temperature (output variable: tas), net downward radiative flux
(output variable: rtmt), and piControl near surface air temperature (output variable: tas) to the CMIP6 archive at time
of the analysis (April 2020). All estimates are for one model run except for CanESM5, which is the average of two
runs.

| Model | ECS (K) |
|---|---|
| ACCESS-CM2 | 4.93 |
| ACCESS-ESM1-5 | 3.63 |
| BCC-CSM2-MR | 3.16 |
| BCC-ESM1 | 3.74 |
| CanESM5 | 5.70 |
| CESM2 | 5.32 |
| CESM2-FV2 | 5.06 |
| CESM2-WACCM | 4.73 |
| CESM2-WACCM-FV2 | 4.56 |
| E3SM-1-0 | 5.28 |
| EC-Earth3-Veg | 4.34 |
| GFDL-CM4 | 3.78 |
| GFDL-ESM4 | 2.61 |
| GISS-E2-1-G | 2.71 |
| GISS-E2-2-G | 2.25 |
| GISS-E2-1-H | 3.09 |
| HadGEM3-GC31-LL | 5.65 |
| INM-CM4-8 | 2.32 |
| INM-CM5-0 | 2.39 |
| IPSL-CM6A-LR | 4.97 |
| MCM-UA-1-0 | 3.68 |
| MIROC6 | 2.84 |
| MIROC-ES2L | 2.83 |
| NorESM2-LM | 2.19 |
| NorESM2-MM | 2.15 |
| SAM0-UNICON | 3.53 |
| TaiESM1 | 4.33 |
| UKESM1-0-LL | 5.40 |

[Figure]

470 **Figure S17.** Values of ECS versus AAWR for the CMIP6 multi-model ensemble. The EM-GC estimates of AAWR and ECS based on training to the HadCRUT5 GMST record are plotted as a box and whisker. The box shows the average 25[th], 50[th], and 75[th] percentiles for the five OHC records shown for HadCRUT5 in Fig. 6 and Fig. 7. The whiskers represent the average 5[th] and 95[th] percentiles. The stars denote the average minimum and maximum values of AAWR or ECS. The eight CMIP6 GCMs that obtain values of AAWR and ECS that are both within the minimum
475 and maximum estimates provided by the EM-GC are identified on the figure. Values of AAWR explain about 78% of the variance in ECS among the CMIP6 GCMs.

[Figure]

480     **Figure S18.** GMST anomaly in 2100 from pre-industrial ($\Delta T_{2100}$) as a function of climate feedback parameter and AER $RF_{2011}$ found using the EM-GC trained with $\Delta T$ from HadCRUT5. (a) $\Delta T_{2100}$ for SSP4-6.0. The region outside of the tropospheric aerosol radiative forcing rage provided by IPCC 2013 (Myhre et al., 2013) is shaded grey. Colors denote the change in $\Delta T_{2100}$. (b) $\Delta T_{2100}$ for SSP3-7.0. (c) $\Delta T_{2100}$ for SSP5-8.5.

485

[Figure]

**Figure S19.** Probabilistic forecasts of future projections of ΔT using the EM-GC trained with ΔT from HadCRUT5 for the SSP4-6.0, SSP3-7.0, and SSP5-8.5 scenarios. (a) Future projections of ΔT for SSP4-6.0. Observations (orange) are from HadCRUT5. The IPCC 2013 likely range of warming (black) is from Figure 11.25b of chapter 11 of the IPCC 2013 report. The Paris Agreement target and upper limit (yellow) are shown for comparison to projections of ΔT using the EM-GC. The CMIP6 minimum, multi-model mean, and maximum values of the rise in ΔT are shown to compare to projections from the EM-GC. Colors denote the probability of reaching at least that temperature by the end of the century and are computed using the aerosol weighting method (see Sect. 2.5). (b) Future projections of ΔT for SSP3-7.0. (c) Future projections of ΔT for SSP5-8.5.

490

495

[Figure]

**Figure S20**. Probability density functions (PDF) for the increase in $\Delta T_{2100}$ using the EM-GC and the CMIP6 multi-model ensemble. (a) PDF for EM-GC (blue) results trained with $\Delta T$ from HadCRUT5 and CMIP6 multi-model results (red) for SSP4-6.0. The left-hand y-axis is for EM-GC probabilities and the righthand y-axis is for GCM probabilities. (b) PDF for SSP3-7.0. (c) PDF for SSP5-8.5.

500

**Table S6.** Probabilities of achieving the Paris Agreement target and upper limit for the various SSP scenarios based on the EM-GC using the HadCRUT4 or HadCRUT5 GMST data set and the CMIP6 multi-model ensemble. The probabilities using the EM-GC are computed using the aerosol weighting method. The probabilities using the CMIP6 GCMs are computed by calculating how many of the models for that scenario are below the temperature limits compared to the total number of models.

| | Probability of Staying at or Below 1.5°C | | | Probability of Staying at or Below 2.0°C | | |
|---|---|---|---|---|---|---|
| | HadCRUT4 | HadCRUT5 | CMIP6 | HadCRUT4 | HadCRUT5 | CMIP6 |
| **SSP1-1.9** | 84% | 81% | 50% | 99% | 98% | 80% |
| **SSP1-2.6** | 64% | 53% | 18% | 90% | 86% | 47% |
| **SSP4-3.4** | 35% | 19% | 0% | 74% | 64% | 17% |
| **SSP2-4.5** | 9% | 0% | 0% | 52% | 33% | 3% |
| **SSP4-6.0** | 0% | 0% | 0% | 26% | 8% | 0% |
| **SSP3-7.0** | 0% | 0% | 0% | 1% | 0% | 0% |
| **SSP5-8.5** | 0% | 0% | 0% | 0% | 0% | 0% |

[Figure]

**Figure S21.** Transient climate response to cumulative $CO_2$ emissions for SSP5-8.5 using the EM-GC trained with the HadCRUT5 $\Delta T$ record. Simulations of the rise in $\Delta T$ versus cumulative $CO_2$ emissions in units of Gt C. The orange line is observations of $\Delta T$ from HadCRUT5 plotted against cumulative carbon emissions from the Global Carbon Budget project (Friedlingstein et al., 2019). The dotted and dashed lines denote the Paris Agreement target and upper limit, respectively. The EM-GC projections represent the probability that the future value of $\Delta T$ will rise to the indicated level, considering only acceptable fits to the climate record. The probabilities were determined using the aerosol weighting method. The light grey, dark grey, and black curves denote the 95, 66, and 50% probabilities of either the Paris target (intersection of dotted horizontal lines) or upper limit (intersection of dashed lines with curves) being achieved.

[Figure]

**Figure S22.** Blended methane mixing ratios. The dotted lines are linear combinations of the time series of methane abundances using SSP1-2.6 and SSP3-7.0 to span the range of values of future methane. The solid lines are the SSP1-2.6 and SSP3-7.0 methane mixing ratio time series.

520

[Figure]

**Figure S23.** Change in GMST (ΔT) from 1850-2019 for observations from HadCRUT5 (black) and 1850-2100 for modeled (red) using SSP 2-4.5 and a value of AER $RF_{2011} = -0.9$ W m$^{-2}$ and the residual between modeled and observations using an instantaneous time variant $\lambda^{-1}$. The solid line denotes a simulation for the original SSP2-4.5 scenario and the dashed line indicates the SSP2-4.5′ simulation (see Sect. 3.3.6). (a) ΔT assuming a constant value of $\lambda^{-1}$. (b) ΔT allowing $\lambda^{-1}$ to increase by 50%. (c) ΔT allowing $\lambda^{-1}$ to vary while the value of $\chi^2_{RECENT}$ is kept below 2. (d) ΔT allowing $\lambda^{-1}$ to vary while the value of $\chi^2_{ATM}$ is kept below 2. (e) Residual between modeled and observed ΔT from 1850-2019 for constant $\lambda^{-1}$. (f) Same as (e) but for increasing $\lambda^{-1}$ by 50%. (g) Same as (f) but for varying $\lambda^{-1}$ while the value of $\chi^2_{RECENT}$ is kept below 2. (h) same as (g) but for varying $\lambda^{-1}$ while the value of $\chi^2_{ATM}$ is kept below 2.

[Figure]

**Figure S24.** Change in GMST (ΔT) from 1850-2019 for observations from HadCRUT5 (black) and 1850-2100 for modeled (red) using SSP2-4.5 and a value of AER RF$_{2011}$ = −0.4 W m$^{-2}$ and the residual between modeled and observations incorporating a 32.5-year delay between λ$^{-1}$ and a change in RF. The solid line denotes a simulation for the original SSP2-4.5 scenario and the dashed line indicates the SSP2-4.5′ simulation (see Sect. 3.3.6). (a) ΔT assuming a constant value of λ$^{-1}$. (b) ΔT allowing λ$^{-1}$ to increase by 50%. (c) ΔT allowing λ$^{-1}$ to vary while the value of χ$^2_{RECENT}$ is kept below 2. (d) ΔT allowing λ$^{-1}$ to vary while the value of χ$^2_{ATM}$ is kept below 2. (e) Residual between modeled and observed ΔT from 1850-2019 for constant λ$^{-1}$. (f) Same as (e) but for increasing λ$^{-1}$ by 50%. (g) Same as (f) but for varying λ$^{-1}$ while the value of χ$^2_{RECENT}$ is kept below 2. (h) same as (g) but for varying λ$^{-1}$ while the value of χ$^2_{ATM}$ is kept below 2.

[Figure]

**Figure S25.** Change in GMST (ΔT) from 1850-2019 for observations from HadCRUT5 (black) and 1850-2100 for modeled (red) using SSP2-4.5 and a value of AER $RF_{2011} = -1.5$ W m$^{-2}$ and the residual between modeled and observations incorporating a 32.5-year delay between $\lambda^{-1}$ and a change in RF. The solid line denotes a simulation for the original SSP2-4.5 scenario and the dashed line indicates the SSP2-4.5′ simulation (see Sect. 3.3.6). (a) ΔT assuming a constant value of $\lambda^{-1}$. (b) ΔT allowing $\lambda^{-1}$ to increase by 50%. (c) ΔT allowing $\lambda^{-1}$ to vary while the value of $\chi^2_{RECENT}$ is kept below 2. (d) ΔT allowing $\lambda^{-1}$ to vary while the value of $\chi^2_{ATM}$ is kept below 2. (e) Residual between modeled and observed ΔT from 1850-2019 for constant $\lambda^{-1}$. (f) Same as (e) but for increasing $\lambda^{-1}$ by 50%. (g) Same as (f) but for varying $\lambda^{-1}$ while the value of $\chi^2_{RECENT}$ is kept below 2. (h) same as (g) but for varying $\lambda^{-1}$ while the value of $\chi^2_{ATM}$ is kept below 2.

**Table S7.** Details of the CMIP6 GCMs used in this study.

| Institution | Model | Model Output |
|---|---|---|
| AS-RCEC | TaiESM1 | No reference provided |
| AWI | AWI-CM-1-1-MR | (Semmler et al., 2018a, 2018b, 2018c, 2019a, 2019b) |
| BCC | BCC-CSM2-MR | (Wu et al., 2018a, 2018b, 2018c; Xin et al., 2019a, 2019b, 2019c, 2019d) |
| | BCC-ESM1 | (Zhang et al., 2018a, 2018b, 2019) |
| CAMS | CAMS-CSM1-0 | (Rong, 2019a, 2019b, 2019c, 2019d, 2019e, 2019f) |
| CAS | CAS-ESM2-0 | (Chai, 2019) |
| | FGOALS-f3-L | (YU, 2019a, 2019b, 2019c, 2019d, 2019e) |
| | FGOALS-g3 | (Li, 2019a, 2019b, 2019c, 2019d, 2019e) |
| CCCma | CanESM5 | (Swart et al., 2019f, 2019g, 2019h, 2019i, 2019j, 2019k, 2019l, 2019m, 2019n, 2019o) |
| | CanESM5-CanOE | (Swart et al., 2019a, 2019b, 2019c, 2019d, 2019e) |
| CNRM-CERFACS | CNRM-CM6-1 | (Voldoire, 2018, 2019c, 2019d, 2019e, 2019f) |
| | CNRM-CM6-1-HR | (Voldoire, 2019a, 2019b, 2020a, 2020b) |
| | CNRM-ESM2-1 | (Seferian, 2018; Voldoire, 2019g, 2019h, 2019i, 2019j, 2019k, 2019l) |
| CSIRO | ACCESS-ESM1-5 | (Ziehn et al., 2019a, 2019b, 2019c, 2019d, 2019e, 2019f, 2019g) |
| CSIRO-ARCCSS | ACCESS-CM2 | (Dix et al., 2019a, 2019b, 2019c, 2019d, 2019e, 2019f, 2019g) |
| E3SM-Project | E3SM-1-0 | (Bader et al., 2018, 2019a, 2019b) |
| | E3SM-1-1-ECA | (Bader et al., 2020) |
| E3SM-Project RUBISCO | E3SM-1-1 | (Bader et al., 2019c) |
| EC-Earth-Consortium | EC-Earth3 | (EC-Earth Consortium (EC-Earth), 2019i, 2019j, 2019k, 2019l, 2019m) |
| | EC-Earth3-Veg | (EC-Earth Consortium (EC-Earth), 2019a, 2019b, 2019c, 2019d, 2019e, 2019f, 2019g, 2019h) |

| | | |
|---|---|---|
| FIO-QLNM | FIO-ESM-2-0 | (Song et al., 2019a, 2019b, 2019c, 2019d) |
| HAMMOZ-Consortium | MPI-ESM1-2-HAM | (Neubauer et al., 2019) |
| INM | INM-CM4-8 | (Volodin et al., 2019a, 2019b, 2019c, 2019d, 2019e, 2019f, 2019g) |
| | INM-CM5-0 | (Volodin et al., 2019m, 2019h, 2019n, 2019i, 2019j, 2019k, 2019l) |
| IPSL | IPSL-CM6A-LR | (Boucher et al., 2018a, 2018b, 2018c, 2019a, 2019b, 2019c, 2019d, 2019e, 2019f, 2019g) |
| MIROC | MIROC6 | (Shiogama et al., 2019a, 2019b, 2019c, 2019d, 2019e, 2019f, 2019g; Tatebe and Watanabe, 2018a, 2018b, 2018c) |
| | MIROC-ES2L | (Hajima et al., 2019; Tachiiri et al., 2019a, 2019b, 2019c, 2019d, 2019e) |
| MOHC | HadGEM3-GC31-MM | (Ridley et al., 2019c) |
| MOHC NERC | HadGEM3-GC31-LL | (Good, 2019, 2020a, 2020b; Ridley et al., 2018, 2019a, 2019b) |
| MOHC, NERC, NIMS-KMA, NIWA | UKESM1-0-LL | (Byun, 2020; Good et al., 2019a, 2019b, 2019c, 2019d, 2019e, 2019f; Tang et al., 2019a, 2019b, 2019c) |
| MPI-M AWI | MPI-ESM1-2-LR | (Wieners et al., 2019a, 2019b, 2019c, 2019d, 2019e) |
| MPI-M DWD DKRZ | MPI-ESM1-2-HR | (Jungclaus et al., 2019; Schupfner et al., 2019a, 2019b, 2019c, 2019d; Steger et al., 2019) |
| MRI | MRI-ESM2-0 | (Yukimoto et al., 2019a, 2019b, 2019c, 2019d, 2019e, 2019f, 2019g, 2019h) |
| NASA-GISS | GISS-E2-1-G | (NASA Goddard Institute for Space Studies (NASA/GISS), 2018a, 2018b, 2018c, 2020a, 2020b, 2020c, 2020d) |
| | GISS-E2-1-G-CC | No reference provided |
| | GISS-E2-2-G | (NASA Goddard Institute for Space Studies (NASA/GISS), 2019a) |
| | GISS-E2-1-H | (NASA Goddard Institute for Space Studies (NASA/GISS), 2018d, 2019b, 2019c) |
| NCAR | CESM2-WACCM-FV2 | (Danabasoglu, 2019d, 2019e, 2020a) |

| | | |
|---|---|---|
| | CESM2 | (Danabasoglu, 2019c, 2019d, 2019e, 2019f, 2019g, 2019h; Danabasoglu et al., 2019) |
| | CESM2-FV2 | (Danabasoglu, 2019b, 2019c, 2020b) |
| | CESM2-WACCM | (Danabasoglu, 2019f, 2019g, 2019h, 2019a, 2019i, 2019j, 2019k) |
| | NorCPM1 | (Bethke et al., 2019a, 2019b, 2019c) |
| NCC | NorESM2-LM | (Seland et al., 2019a, 2019b, 2019c, 2019d, 2019e, 2019f, 2019g) |
| | NorESM2-MM | (Bentsen et al., 2019a, 2019b, 2019c, 2019d, 2019e, 2019f, 2019g) |
| NIMS-KMA | KACE-1-0-G | (Byun et al., 2019a, 2019b, 2019c, 2019d, 2019e) |
| NOAA-GFDL | GFDL-CM4 | (Guo et al., 2018a, 2018b, 2018c, 2018d, 2018e) |
| | GFDL-ESM4 | (John et al., 2018a, 2018b, 2018c, 2018d, 2018e; Krasting et al., 2018a, 2018b, 2018c) |
| NUIST | NESM3 | (Cao, 2019a, 2019b, 2019c; Cao and Wang, 2019) |
| SNU | SAM0-UNICON | (Park and Shin, 2019a, 2019b, 2019c) |
| THU | CIESM | (Huang, 2019a, 2019b, 2020a, 2020b) |
| UA | MCM-UA-1-0 | (Stouffer, 2019a, 2019b, 2019c, 2019d, 2019e, 2019f, 2019g) |

560